# Representation Learning for General-sum Low-rank Markov Games

**Chengzhuo Ni**
Princeton University
cn10@princeton.edu

**Yuda Song**
Carnegie Mellon University
yudas@andrew.cmu.edu

**Xuezhou Zhang**
Princeton University
xz7392@princeton.edu

**Zihan Ding**
Princeton University
zihand@princeton.edu

**Chi Jin**
Princeton University
chij@princeton.edu

**Mengdi Wang**
Princeton University
mengdiw@princeton.edu

## Abstract

We study multi-agent general-sum Markov games with nonlinear function approximation. We focus on low-rank Markov games whose transition matrix admits a hidden low-rank structure on top of an unknown non-linear representation. The goal is to design an algorithm that (1) finds an $\varepsilon$-equilibrium policy sample efficiently without prior knowledge of the environment or the representation, and (2) permits a deep-learning friendly implementation. We leverage representation learning and present a model-based and a model-free approach to construct an effective representation from collected data. For both approaches, the algorithm achieves a sample complexity of $\mathrm{poly}(H, d, A, 1/\varepsilon)$, where $H$ is the game horizon, $d$ is the dimension of the feature vector, $A$ is the size of the joint action space and $\varepsilon$ is the optimality gap. When the number of players is large, the above sample complexity can scale exponentially with the number of players in the worst case. To address this challenge, we consider Markov Games with a factorized transition structure and present an algorithm that escapes such exponential scaling. To our best knowledge, this is the first sample-efficient algorithm for multi-agent general-sum Markov games that incorporates (non-linear) function approximation. We accompany our theoretical result with a neural network-based implementation of our algorithm and evaluate it against the widely used deep RL baseline, DQN with fictitious play.

## 1 Introduction

Multi-agent reinforcement learning (MARL) studies the problem where multiple agents learn to make sequential decisions in an unknown environment to maximize their (own) cumulative rewards. Recently, MARL has achieved remarkable empirical success, such as in traditional games like GO (Silver et al., 2016, 2017) and Poker (Moravčík et al., 2017), real-time video games such as Starcraft and Dota 2(Vinyals et al., 2019; Berner et al., 2019), decentralized controls or multi-agent robotics systems (Brambilla et al., 2013) and autonomous driving (Shalev-Shwartz et al., 2016).

On the theoretical front, however, provably sample-efficient algorithms for Markov games have been largely restricted to either two-player zero-sum games (Bai et al., 2020; Xie et al., 2020; Chen et al., 2021; Jin et al., 2021c) or general-sum games with small and finite state and action spaces (Bai and Jin, 2020; Liu et al., 2021; Jin et al., 2021b). These algorithms typically do not permit a scalable implementation applicable to real-world games, due to either (1) they only work for tabular or linear Markov games which are too restrictive to model real-world games, or (2) the ones that do handle rich non-linear function approximation (Jin et al., 2021c) are not computationally efficient. This motivates us to ask the following question:

*Can we design an efficient algorithm that (1) provably learns multi-player general-sum Markov games with rich nonlinear function approximation and (2) permits scalable implementations?*

This paper presents the first positive answer to the above question. In particular, we make the following contributions:

1. We design a new centralized self-play meta algorithm for multi-agent low-rank Markov games: **Ge**neral **R**epresentation **L**earning for **M**ulti-player **G**eneral-sum **M**arkov **G**ame (**GeRL_MG2**). We present a model-based and a model-free instantiation of GeRL_MG2 which differ by the way function approximation is used, and a clean and unified analysis for both approaches.

2. We show that the model-based variant requires access to an MLE oracle and a NE/CE/CCE oracle for matrix games, and enjoys a $\tilde{O}\left(H^6 d^4 A^2 \log(|\Phi||\Psi|)/\varepsilon^2\right)$ sample complexity to learn an $\varepsilon$-NE/CE/CCE equilibrium policy, where $d$ is the dimension of the feature vector, $A$ is the size of the joint action space, $H$ is the game horizon, $\Phi$ and $\Psi$ are the function classes for the representation and emission process. The model-free variant replaces model-learning with solving a minimax optimization problem, and enjoys a sample complexity of $\tilde{O}\left(H^6 d^4 A^3 M \log(|\Phi|)/\varepsilon^2\right)$ for a slightly restricted class of Markov game with latent block structure.

3. Both of the above algorithms have sample complexities scaling with the joint action space size, which is exponential in the number of players. This unfavorable scaling is referred to as the *curse of multi-agent*, and is unavoidable in the worst case under general function approximation. We consider a spatial factorization structure where the transition of each player's local state is directly affected only by at most $L = O(1)$ players in its adjacency. Given this additional structure, we provide an algorithm that achieves $\tilde{O}(M^4 H^6 d^{2(L+1)^2} \tilde{A}^{2(L+1)}/\varepsilon^2)$ sample complexity, where $\tilde{A}$ is the size of a single player's action space, thus escaping the exponential scaling to the number of agents.

4. Finally, we provide an efficient implementation of our reward-free algorithm, and show that it achieves superior performance against traditional deep RL baselines without principled representation learning.

## 1.1 RELATED WORKS

**Markov games** Markov games (Littman, 1994; Shapley, 1953) is an extensively used framework introduced for game playing with sequential decision making. Previous works (Littman, 1994; Hu and Wellman, 2003; Hansen et al., 2013) studied how to find the Nash equilibrium of a Markov game when the transition matrix and reward function are known. When the dynamic of the Markov game is unknown, recent works provide a line of finite-sample guarantees for learning Nash equilibrium in two-player zero-sum Markov games (Bai and Jin, 2020; Xie et al., 2020; Bai et al., 2020; Zhang et al., 2020; Liu et al., 2021; Jin et al., 2021c; Huang et al., 2021) and learning various equilibriums (including NE,CE,CCE, which are standard solution notions in games (Roughgarden, 2010)) in general-sum Markov games (Liu et al., 2021; Bai et al., 2021; Jin et al., 2021b). Some of the analyses in these works are based on the techniques for learning single-agent Markov Decision Processes (MDPs) (Azar et al., 2017; Jin et al., 2018, 2020).

**RL with Function Approximation** Function approximation in reinforcement learning has been extensively studied in recent years. For the single-agent Markov decision process, function approximation is adopted to achieve a better sample complexity that depends on the complexity of function approximators rather than the size of the state-action space. For example, (Yang and Wang, 2019; Jin et al., 2020; Zanette et al., 2020) considered the linear MDP model, where the transition probability function and reward function are linear in some feature mapping over state-action pairs. Another line of works (see, e.g., Jiang et al., 2017; Jin et al., 2021a; Du et al., 2021; Foster et al., 2021) studied the MDPs with general nonlinear function approximations.

When it comes to Markov game, (Chen et al., 2021; Xie et al., 2020; Jia et al., 2019) studied the Markov games with linear function approximations. Recently, (Huang et al., 2021) and (Jin et al., 2021c) proposed the first algorithms for two-player zero-sum Markov games with general function approximation, and provided a sample complexity governed by the minimax Eluder dimension. However, technical difficulties prevent extending these results to multi-player general-sum Markov games with nonlinear function approximation. The results for linear function approximation assume a known state-action feature, and are unable to solve the Markov games with a more general non-linear approximation where both the feature and function parameters are unknown. For the general function class works, their approaches rely heavily on the two-player nature, and it's not clear how to apply their methods to the general multi-player setting.

**Representation Learning in RL** Our work is closely related to representation learning in single-agent RL, where the study mainly focuses on the low-rank MDPs. A low-rank MDP is strictly more

general than a linear MDP which assumes the representation is known a priori. Several related works studied low-rank MDPs with provable sample complexities. (Agarwal et al., 2020b; Ren et al., 2021) and (Uehara et al., 2021) consider the model-based setting, where the algorithm learns the representation with the model class of the transition probability given. (Modi et al., 2021) provided a representation learning algorithm under the model-free setting and proved its sample efficiency when the MDP satisfies the minimal reachability assumption. (Zhang et al., 2022) proposed a model-free method for the more restricted MDP class called Block MDP, but does not rely on the reachability assumption, which is also studied in papers including (Du et al., 2019) and (Misra et al., 2020). A concurrent work (Qiu et al., 2022) studies representation learning in RL with contrastive learning and extends their algorithm to the Markov game setting. However, their method requires strong data assumption and does not provide any practical implementation in the Markov game setting.

## 2 PROBLEM SETTINGS

A general-sum Markov game with $M$ players is defined by a tuple $(\mathcal{S}, \{\mathcal{A}_i\}_{i=1}^M, P^\star, \{r_i\}_{i=1}^M, H, d_1)$. Here $\mathcal{S}$ is the state space, $\mathcal{A}_i$ is the action space for player $i$, $H$ is the time horizon of each episode and $d_1$ is the initial state distribution. We let $\mathcal{A} = \mathcal{A}_1 \times \ldots \times \mathcal{A}_M$ and use $\boldsymbol{a} = (a_1, a_2, \ldots, a_M)$ to denote the joint actions by all $M$ players. Denote $\tilde{A} = \max_i |\mathcal{A}_i|$ and $A = |\mathcal{A}|$. $P^\star = \{P_h^\star\}_{h=1}^H$ is a collection of transition probabilities, so that $P_h^\star(\cdot|s, \boldsymbol{a})$ gives the distribution of the next state if actions $\boldsymbol{a}$ are taken at state $s$ and step $h$. And $r_i = \{r_{h,i}\}_{h=1}^H$ is a collection of reward functions, so that $r_{h,i}(s, \boldsymbol{a})$ gives the reward received by player $i$ when actions $\boldsymbol{a}$ are taken at state $s$ and step $h$.

### 2.1 SOLUTION CONCEPTS

The policy of player $i$ is denoted as $\pi_i := \{\pi_{h,i} : \mathcal{S} \to \Delta_{\mathcal{A}_i}\}_{h \in [H]}$. We denote the product policy of all the players as $\pi := \pi_1 \times \ldots \times \pi_M$, here "product" means that conditioned on the same state, the action of each player is sampled independently according to their own policy. We denote the policy of all the players except the $i$th player as $\pi_{-i}$. We define $V_{h,i}^\pi(s)$ as the expected cumulative reward that will be received by the $i$th player if starting at state $s$ at step $h$ and all players follow policy $\pi$. For any strategy $\pi_{-i}$, there exists a best response of the $i$th player, which is a policy $\mu^\dagger(\pi_{-i})$ satisfying $V_{h,i}^{\mu^\dagger(\pi_{-i}), \pi_{-i}}(s) = \max_{\pi_i} V_{h,i}^{\pi_i, \pi_{-i}}(s)$ for any $(s, h) \in \mathcal{S} \times [H]$. We denote $V_{h,i}^{\dagger, \pi_{-i}} := V_{h,i}^{\mu^\dagger(\pi_{-i}), \pi_{-i}}$. Let $v_i^{\dagger, \pi_{-i}} := \mathbb{E}_{s \sim d_1}\left[V_{1,i}^{\dagger, \pi_{-i}}(s)\right], v_i^\pi := \mathbb{E}_{s \sim d_1}\left[V_{1,i}^\pi(s)\right]$.

**Definition 2.1** (NE). *A product policy $\pi$ is a Nash equilibrium (NE) if $v_i^\pi = v_i^{\dagger, \pi_{-i}}, \forall i \in [M]$. And we call $\pi$ an $\varepsilon$-approximate NE if $\max_{i \in [M]}\{v_i^{\dagger, \pi_{-i}} - v_i^\pi\} < \varepsilon$.*

The coarse correlated equilibrium (CCE) is a relaxed version of Nash equilibrium in which we consider general correlated policies instead of product policies.

**Definition 2.2** (CCE). *A correlated policy $\pi$ is a CCE if $V_{h,i}^{\dagger, \pi_{-i}}(s) \le V_{h,i}^\pi(s)$ for all $s \in \mathcal{S}, h \in [H], i \in [M]$. And we call $\pi$ an $\varepsilon$-approximate CCE if $\max_{i \in [M]}\{v_i^{\dagger, \pi_{-i}} - v_i^\pi\} < \varepsilon$.*

The correlated equilibrium (CE) is another relaxation of the Nash equilibrium. To define CE, we first introduce the concept of strategy modification: A strategy modification $\omega_i := \{\omega_{h,i}\}_{h \in [H]}$ for player $i$ is a set of $H$ functions from $\mathcal{S} \times \mathcal{A}_i$ to $\mathcal{A}_i$. Let $\Omega_i := \{\Omega_{h,i}\}_{h \in [H]}$ denote the set of all possible strategy modifications for player $i$. One can compose a strategy modification $\omega_i$ with any Markov policy $\pi$ and obtain a new policy $\omega_i \circ \pi$ such that when policy $\pi$ chooses to play $\boldsymbol{a} := (a_1, \ldots, a_M)$ at state $s$ and step $h$, policy $\omega_i \circ \pi$ will play $(a_1, \ldots, a_{i-1}, \omega_{h,i}(s, a_i), a_{i+1}, \ldots, a_M)$ instead.

**Definition 2.3** (CE). *A correlated policy $\pi$ is a CE if $\max_{i \in [M]} \max_{\omega_i \in \Omega_i} V_{h,i}^{\omega_i \circ \pi}(s) \le V_{h,i}^\pi(s)$ for all $(s, h) \in \mathcal{S} \times [H]$. And we call $\pi$ an $\varepsilon$-approximate CE if $\max_{i \in [M]}\{\max_{\omega_i \in \Omega_i} v_i^{\omega_i \circ \pi} - v_i^\pi\} < \varepsilon$.*

**Remark 2.1.** *For general-sum Markov Games, we have $\{NE\} \subseteq \{CE\} \subseteq \{CCE\}$, so that they form a nested set of notions of equilibria (Roughgarden, 2010). While there exist algorithms to approximately compute the Nash equilibrium (Berg and Sandholm, 2017), the computation of NE for general-sum games in the worst case is still PPAD-hard (Daskalakis, 2013). On the other hand, CCE and CE can be solved in polynomial time using linear programming (Examples include Papadimitriou and Roughgarden (2008); Blum et al. (2008)). Therefore, in this paper we study both NE and these weaker equilibrium concepts that permit more computationally efficient solutions.*

## 2.2 LOW-RANK MARKOV GAMES

In this paper, we consider the class of low-rank Markov games. A Markov game is called a low-rank Markov game if the transition probability at any time step $h$ has a latent low-rank structure.

**Definition 2.4** (Low-Rank Markov Game). *We call a Markov game a low-rank Markov game if for any $s, s' \in \mathcal{S}, \boldsymbol{a} \in \mathcal{A}, h \in [H], i \in [M]$, we have $P_h^\star(s'|s, \boldsymbol{a}) = \phi_h^\star(s, \boldsymbol{a})^\top w_h^\star(s')$, where $\|\phi_h^\star(s, \mathbf{a})\|_2 \le 1$ and $\|w_h^\star(s')\|_2 \le \sqrt{d}$ for all $(s, \boldsymbol{a}, s')$.*

A special case of low-rank Markov Game is the Block Markov game:

**Definition 2.5** (Block Markov Game). *Consider any $h \in [H]$. A Block Markov game has an emission distribution $o_h(\cdot|z) \in \Delta_\mathcal{S}$ and a latent state space transition $T_h(z'|z, \boldsymbol{a})$, such that for any $s \in \mathcal{S}, o_h(s|z) > 0$ for a unique latent state $z \in \mathcal{Z}$, denoted as $\psi_h^\star(s)$. Denote $Z = |\mathcal{Z}|$. Together with the ground truth decoder $\psi_h^\star$, it defines the transitions $P_h^\star(s'|s, a) = \sum_{z' \in \mathcal{Z}} o_h(s'|z') T_h(z'|\psi_h^\star(s), a)$.*

With the definition of the Block Markov game, one can naturally derive a feature vector that in addition takes the one-hot form: we just need to let the ground truth $\phi_h^\star(s, \boldsymbol{a})$ at step $h$ be a $Z \cdot A$-dimensional vector $e_{(\psi^\star(s), \boldsymbol{a})}$ where $e_i$ is the $i$-th basis vector. Correspondingly, for any $s \in \mathcal{S}, w_h^\star(s)$ is a $Z \cdot A$ dimensional vector such that the $(z, \boldsymbol{a})$-th entry is $\sum_{z' \in \mathcal{Z}} o_h(s|z') T_h(z'|z, \boldsymbol{a})$. Then $P_h^\star(s'|s, \boldsymbol{a}) = \phi_h^\star(s, \boldsymbol{a})^\top w_h^\star(s')$, so that the Block Markov game is a low-rank Markov game with rank $d = Z \cdot A$.

**Learning Objective** The goal of multi-agent reinforcement learning is to design algorithms for Markov games that find an $\varepsilon$-approximate equilibrium (NE, CCE, CE) from a small number of interactions with the environment. We focus on the low-rank Markov games whose feature vector $\phi^\star$ and transition probability $P^\star$ are both *unknown*, and the goal is to identify a $\varepsilon$-approximate equilibrium policy with a number of interactions scaling polynomially with $d, A, H, \frac{1}{\varepsilon}$ and the log-cardinality of the function class, without depending on the number of raw states which could be infinite.

## 3 ALGORITHM DESCRIPTION

In this section, we present our algorithm GERL_MG2 (see Alg. 1). The algorithm mainly consists of two modules: the representation learning module and the planning module. We develop the representation learning module base on the past works on the single agent MDP (e.g. Agarwal et al. (2020b); Uehara et al. (2021); Modi et al. (2021)), and but modify them to work with UCB-style planning module. Here we denote $d_{P,h}^\pi$ as the state distribution under transition probability $P$ and policy $\pi$ at step $h$, and $U(\mathcal{A})$ as the uniform distribution over the joint action space.

### 3.1 REPRESENTATION LEARNING

In the representation learning module, the main goal is to learn a representation function $\hat{\phi}$ to approximate $\phi^\star$, using the data collected so far. In each episode, the algorithm first collects some new data using the policy derived from the previous episode. Note that in our data collection scheme, for each time step $h$, we maintain two buffers $\mathcal{D}_h^{(n)}$ and $\tilde{\mathcal{D}}_h^{(n)}$ of transition tuples $(s, a, s')$ (line 7 of Alg. 1) which draw the state $s$ from slightly different distributions. Based on the data collected in history, the representation learning module estimates the feature $\hat{\phi}^{(n)}$ and transition probability $\hat{P}^{(n)}$. We propose two versions of the representation learning algorithm (model-based, Alg. 2; model-free, Alg. 3) based on whether we are given the full model class $\mathcal{M}_h$ of the transition probability, or only the function class of the state-action features $\Phi_h$.

**Model-based Representation Learning** In the model-based setting, we assume the access to a *realizable* model class $\mathcal{M}_h = \{(w_h, \phi_h) : w_h \in \Psi_h, \phi_h \in \Phi_h\}, h \in [H]$ such that the true model is included in this class, i.e., $w_h^\star \in \Psi_h, \phi_h^\star \in \Phi_h, \forall h \in [H]$. Following the norm bounds on $\phi_h^\star, w_h^\star$, we assume that the same norm bounds hold for our function approximator, i.e., for any $\phi_h \in \Phi_h, w_h \in \Psi_h$, we have $\|\phi_h(s, \boldsymbol{a})\|_2 \le 1$ and $\|w_h(s')\|_2 \le \sqrt{d}$ for all $(s, \boldsymbol{a}, s')$, and $\int \phi_h(s, \boldsymbol{a})^\top w_h(s') \mathrm{d}s' = 1$. Given the dataset $\mathcal{D} := \mathcal{D}_h^{(n)} \cup \tilde{\mathcal{D}}_h^{(n)}$, MBREPLEARN learns the features and transition probability using maximum likelihood estimation (MLE):

$$\left(\hat{w}_h^{(n)}, \hat{\phi}_h^{(n)}\right) = \arg\max_{(w,\phi) \in \mathcal{M}_h} \mathbb{E}_\mathcal{D}\left[\log\left(\phi(s, \boldsymbol{a})^\top w(s')\right)\right], \quad \hat{P}_h^{(n)}(s'|s, \boldsymbol{a}) = \hat{\phi}_h^{(n)}(s, \boldsymbol{a})^\top \hat{w}_h^{(n)}(s').$$

---

**Algorithm 1** General Representation Learning for Multi-player General-sum Low-Rank Markov Game with UCB-driven Exploration (GERL_MG2)

---

1: **Input:** Regularizer $\lambda$, iteration $N$, parameter $\{\alpha^{(n)}\}_{n=1}^N$, $\{\zeta^{(n)}\}_{n=1}^N$.
2: Initialize $\pi^{(0)}$ to be uniform; set $\mathcal{D}_h^{(0)} = \emptyset$, $\tilde{\mathcal{D}}_h^{(0)} = \emptyset$, $\forall h \in [H]$.
3: **for** episode $n = 1, 2, \cdots, N$ **do**
4:     Set $\overline{V}_{H+1,i}^{(n)} \leftarrow 0$, $\underline{V}_{H+1,i}^{(n)} \leftarrow 0$
5:     **for** step $h = H, H-1\ldots, 1$ **do**
6:         Collect two triples $(s, \boldsymbol{a}, s')$, $(\tilde{s}', \tilde{\boldsymbol{a}}', \tilde{s}'')$ with
        $s \sim d_{P^\star, h}^{\pi^{(n-1)}}$, $\boldsymbol{a} \sim U(\mathcal{A})$, $s' \sim P_h^\star(s, \boldsymbol{a})$,
        $\tilde{s} \sim d_{P^\star, h-1}^{\pi^{(n-1)}}$, $\tilde{\boldsymbol{a}} \sim U(\mathcal{A})$, $\tilde{s}' \sim P_{h-1}^\star(\tilde{s}, \tilde{\boldsymbol{a}})$, $\tilde{\boldsymbol{a}}' \sim U(\mathcal{A})$, $\tilde{s}'' \sim P_h^\star(\tilde{s}', \tilde{\boldsymbol{a}}')$.
7:         Update datasets: $\mathcal{D}_h^{(n)} = \mathcal{D}_h^{(n-1)} \cup \{(s, \boldsymbol{a}, s')\}$, $\tilde{\mathcal{D}}_h^{(n)} = \tilde{\mathcal{D}}_h^{(n-1)} \cup \{(\tilde{s}', \tilde{\boldsymbol{a}}', \tilde{s}'')\}$.
8:         Learn representation via model-based or model-free methods:
        $\phi_h^{(n)}, \hat{P}_h^{(n)} = \text{MBREPLEARN}\left(\mathcal{D}_h^{(n)} \cup \tilde{\mathcal{D}}_h^{(n)}, h\right)$ or $\text{MFREPLEARN}\left(\mathcal{D}_h^{(n)} \cup \tilde{\mathcal{D}}_h^{(n)}, h, \lambda\right)$
9:         Compute $\hat{\beta}_h^{(n)}$ from equation 5, for each $(s, \boldsymbol{a}) \in \mathcal{S} \times \mathcal{A}, i \in [M]$, set

$$\overline{Q}_{h,i}^{(n)}(s, \boldsymbol{a}) \leftarrow r_{h,i}(s, \boldsymbol{a}) + \left(\hat{P}_h^{(n)}\overline{V}_{h+1,i}^{(n)}\right)(s, \boldsymbol{a}) + \hat{\beta}_h^{(n)}(s, \boldsymbol{a})$$

$$\underline{Q}_{h,i}^{(n)}(s, \boldsymbol{a}) \leftarrow r_{h,i}(s, \boldsymbol{a}) + \left(\hat{P}_h^{(n)}\underline{V}_{h+1,i}^{(n)}\right)(s, \boldsymbol{a}) - \hat{\beta}_h^{(n)}(s, \boldsymbol{a}).$$

10:         Compute $\pi_h^{(n)}$ from equation 2 or equation 3 or equation 4. For each $s \in \mathcal{S}, i \in [M]$, set

$$\overline{V}_{h,i}^{(n)}(s) \leftarrow \left(\mathbb{D}_{\pi_h^{(n)}}\overline{Q}_{h,i}^{(n)}\right)(s), \quad \underline{V}_{h,i}^{(n)}(s) \leftarrow \left(\mathbb{D}_{\pi_h^{(n)}}\underline{Q}_{h,i}^{(n)}\right)(s), \quad \forall s \in \mathcal{S}.$$

11:     **end for**
12:     Let $\Delta^{(n)} = \max_{i \in [M]}\left\{\overline{v}_i^{(n)} - \underline{v}_i^{(n)}\right\} + 2H\sqrt{A\zeta^{(n)}}$, where $\overline{v}_i^{(n)} = \int_{\mathcal{S}}\overline{V}_{1,i}^{(n)}(s)d_1(s)\mathrm{d}s$, and
    $\underline{v}_i^{(n)} = \int_{\mathcal{S}}\underline{V}_{1,i}^{(n)}(s)d_1(s)\mathrm{d}s$.
13: **end for**
14: **Return** $\hat{\pi} = \pi^{(n^\star)}$ where $n^\star = \arg\min_{n \in [N]}\Delta^{(n)}$.

---

**Model-free Representation Learning** In the model-free setting, we are only given the function class of the feature vectors, $\Phi_h$, which we assume also includes the true feature $\phi_h^\star$. Given the dataset $\mathcal{D} := \mathcal{D}_h^{(n)} \cup \tilde{\mathcal{D}}_h^{(n)}$, MFREPLEARN aims to learn a feature vector that is able to linearly fit the Bellman backup of any function $f(s)$ in an appropriately chosen discriminator function class $\mathcal{F}_h$. To be precise, we aim to optimize the following objective:

$$\min_{\phi \in \Phi_h} \max_{f \in \mathcal{F}_h}\left[\min_\theta \mathbb{E}_{\mathcal{D}}\left[\left(\phi(s, \boldsymbol{a})^\top \theta - f(s')\right)^2\right] - \min_{\tilde{\theta}, \tilde{\phi} \in \Phi_h} \mathbb{E}_{\mathcal{D}}^{(n)}\left[\left(\tilde{\phi}(s, \boldsymbol{a})^\top \tilde{\theta} - f(s')\right)^2\right]\right],$$

where the first term is the empirical squared loss and the second term is the conditional expectation of $f(s')$ given $(s, a)$, subtracted for the purpose of bias reduction. Once we obtain an estimation $\hat{\phi}_h^{(n)}$, we can construct a *non-parametric* transition model defined as:

$$\hat{P}_h^{(n)}(s'|s, \boldsymbol{a}) = \hat{\phi}_h^{(n)}(s, \boldsymbol{a})^\top \left(\sum_{(\tilde{s}, \tilde{\boldsymbol{a}}) \in \mathcal{D}} \hat{\phi}_h^{(n)}(\tilde{s}, \tilde{\boldsymbol{a}})\hat{\phi}_h^{(n)}(\tilde{s}, \tilde{\boldsymbol{a}})^\top + \lambda I_d\right)^{-1} \sum_{(\tilde{s}, \tilde{\boldsymbol{a}}, \tilde{s}') \in \mathcal{D}} \hat{\phi}_h^{(n)}(\tilde{s}, \tilde{\boldsymbol{a}})\mathbf{1}_{\tilde{s}'=s'}. \quad (1)$$

We show that doing Least-square Value Iteration (LSVI) is equivalent to doing model-based planning inside $\hat{P}_h^{(n)}$ (line 10 of Alg. 1), and thus the model-free algorithm can be analyzed in the same way as the model-based algorithm. In practice, for applications where the raw observation states are high-dimensional, e.g. images, estimating the transition is often much harder than estimating the one-directional feature function. In such cases, we expect the $\Psi$ class to be much larger than the $\Phi$ class and the model-free approach to be more efficient.

## 3.2 PLANNING

Based on the feature vector and transition probability computed from the representation learning phase, a new policy $\pi^{(n+1)}$ is computed using the planning module. The planning phase is conducted

---

**Algorithm 2** Model-based Representation Learning, MBRepLearn

---

1: **Input:** Dataset $\mathcal{D}$, step $h$.
2: Compute $(\hat{w}, \hat{\phi}) := \arg\max_{(w,\phi)\in\mathcal{M}_h} \mathbb{E}_{\mathcal{D}} \left[\log w(s')^\top \phi(s, \boldsymbol{a})\right]$.
3: **Return** $\hat{\phi}, \hat{P} : \hat{P}(s'|s, \boldsymbol{a}) = \hat{w}(s')^\top \hat{\phi}(s, \boldsymbol{a})$.

---

**Algorithm 3** Model-free Representation Learning, MFRepLearn

---

1: **Input:** Dataset $\mathcal{D}$, step $h$, regularization $\lambda$.
2: Denote least squares loss: $\mathcal{L}_{\lambda,\mathcal{D}}(\phi, \theta, f) := \mathbb{E}_{\mathcal{D}} \left[\left(\phi(s, \boldsymbol{a})^\top \theta - f(s')\right)^2\right] + \lambda \|\theta\|_2^2$.
3: Compute $\hat{\phi} = \arg\min_{\phi\in\Phi_h} \max_{f\in\mathcal{F}_h} [\min_\theta \mathcal{L}_{\lambda,\mathcal{D}}(\phi, \theta, f) - \min_{\tilde{\phi}\in\Phi_h,\tilde{\theta}} \mathcal{L}_{\lambda,\mathcal{D}}(\tilde{\phi}, \tilde{\theta}, f)]$
4: **Return** $\hat{\phi}, \hat{P}$ where $\hat{P}$ is calculated from equation 1.

---

with a Upper-Confidence-Bound (UCB) style approach, and we maintain both an optimistic and a pessimistic estimation of the value functions and the Q-value functions $\overline{V}_{h,i}^{(n)}, \underline{V}_{h,i}^{(n)}, \overline{Q}_{h,i}^{(n)}, \underline{Q}_{h,i}^{(n)}$, which are computed recursively through the Bellman's equation with the bonus function $\hat{\beta}_h^{(n)}$ (Line 9 and 10 of Alg. 1). Here the operator $\mathbb{D}$ is defined by $(\mathbb{D}_\pi f)(s) := \mathbb{E}_{\boldsymbol{a}\sim\pi(s)}[f(s, \boldsymbol{a})], \forall f : \mathcal{S} \times \mathcal{A} \to \mathbb{R}$, and $\pi_h^{(n)}$ is the policy computed from $M$ induced Q-value functions $\tilde{Q}_{h,i}^{(n)}$. For the model-based setting, we simply let $\tilde{Q}_{h,i}^{(n)}$ be the optimistic estimator $\overline{Q}_{h,i}^{(n)}$. For the model-free setting, for technical reasons, we instead let $\tilde{Q}_{h,i}^{(n)}$ be the nearest neighbor of $\overline{Q}_{h,i}^{(n)}$ in $\mathcal{N}_h$ with respect to the $\|\cdot\|_\infty$ metric, where $\mathcal{N}_h \subseteq \mathbb{R}^{\mathcal{S}\times\mathcal{A}}$ is a properly designed set of functions, whose construction is deferred to the appendix.

Depending on the problem settings, the policy $\pi_h^{(n)}$ takes either one of the following formulations:

- For the NE, we compute $\pi_h^{(n)} = \left(\pi_{h,1}^{(n)}, \pi_{h,2}^{(n)}, \ldots, \pi_{h,M}^{(n)}\right)$ such that $\forall s \in \mathcal{S}, i \in [M]$,
$$\pi_{h,i}^{(n)}(\cdot|s) = \arg\max_{\pi_{h,i}} \left(\mathbb{D}_{\pi_{h,i},\pi_{h,-i}^{(n)}} \tilde{Q}_{h,i}^{(n)}\right)(s). \tag{2}$$

- For the CCE, we compute $\pi_h^{(n)}$ such that $\forall s \in \mathcal{S}, i \in [M]$,
$$\max_{\pi_{h,i}} \left(\mathbb{D}_{\pi_{h,i},\pi_{h,-i}^{(n)}} \tilde{Q}_{h,i}^{(n)}\right)(s) \leq \left(\mathbb{D}_{\pi^{(n)}} \tilde{Q}_{h,i}^{(n)}\right)(s). \tag{3}$$

- For CE, we compute $\pi_h^{(n)}$ such that $\forall s \in \mathcal{S}, i \in [M]$,
$$\max_{\omega_{h,i}\in\Omega_{h,i}} \left(\mathbb{D}_{\omega_{h,i}\circ\pi_h^{(n)}} \tilde{Q}_{h,i}^{(n)}\right)(s) \leq \left(\mathbb{D}_{\pi^{(n)}} \tilde{Q}_{h,i}^{(n)}\right)(s). \tag{4}$$

Without loss of generality we assume the solution to the above formulations is unique, if there are multiple solutions, one can always adopt a deterministic selection rule such that it always outputs the same policy given the same inputs.

Note that although the policy is computed using only the optimistic estimations, we still maintain a pessimistic estimator, which is used to estimate the optimality gap $\Delta^{(n)}$ of the current policy. The algorithm's output policy $\hat{\pi}$ is chosen to be the one with the minimum estimated optimality gap.

The bonus term $\hat{\beta}_h^{(n)}$ is a linear bandit style bonus computed using the learned feature $\hat{\phi}$:
$$\hat{\beta}_h^{(n)}(s, \boldsymbol{a}) := \min\{\alpha^{(n)} \|\hat{\phi}_h^{(n)}(s, \boldsymbol{a})\|_{\left(\hat{\Sigma}_h^{(n)}\right)^{-1}}, H\}. \tag{5}$$

where $\hat{\Sigma}_h^{(n)} := \sum_{(s,\boldsymbol{a})\in\mathcal{D}_h^{(n)}} \hat{\phi}_h^{(n)}(s, \boldsymbol{a})\hat{\phi}_h^{(n)}(s, \boldsymbol{a})^\top + \lambda I_d$ is the empirical covariance matrix.

## 4 THEORETICAL RESULTS

In this section, we provide the theoretical guarantees of the proposed algorithm for both the model-based and model-free approaches. We denote $|\mathcal{M}| := \max_{h\in[H]} |\mathcal{M}_h|$ and $|\Phi| := \max_{h\in[H]} |\Phi_h|$. The first theorem provides a guarantee of the sample complexity for the model-based method.

---

**Algorithm 4** Model-based Representation Learning for Factored MG (MBREPLEARN_FACTOR)

1: **Input:** Dataset $\mathcal{D}$, step $h$.
2: Compute $(\hat{w}_i, \hat{\phi}_i) := \arg\max_{(w,\phi) \in \mathcal{M}_{h,i}} \mathbb{E}_{\mathcal{D}} \left[ \log w(s_i')^\top \phi(s[Z_i], \boldsymbol{a}_i) \right]$, for each $i \in [M]$.
3: **Return** $\{\hat{\phi}_i\}_{i=1}^M$, $\hat{P} : \hat{P}(s'|s, \boldsymbol{a}) = \prod_{i=1}^M \left( \hat{w}_i(s_i')^\top \hat{\phi}_i(s[Z_i], \boldsymbol{a}_i) \right)$.

---

**Theorem 4.1** (PAC guarantee of Algorithm 1 (model-based))**.** *When Alg. 1 is applied with model-based representation learning algorithm Alg. 2, with parameters $\lambda = \Theta\left(d \log(NH|\Phi|/\delta)\right), \alpha^{(n)} = \Theta\left(Hd\sqrt{A \log(|\mathcal{M}|HN/\delta)}\right), \zeta^{(n)} = \Theta\left(n^{-1} \log(|\mathcal{M}|HN/\delta)\right)$, by setting the number of episodes $N$ to be at most*

$$O\left(H^6 d^4 A^2 \varepsilon^{-2} \log^2\left(HdA|\mathcal{M}|/\delta\varepsilon\right)\right),$$

*with probability $1 - \delta$, the output policy $\hat{\pi}$ is an $\varepsilon$-approximate $\{\text{NE}, \text{CCE}, \text{CE}\}$.*

Theorem 4.1 shows that GERL_MG2 can find an $\varepsilon$-approximate $\{\text{NE}, \text{CCE}, \text{CE}\}$ by running the algorithm for at most $\tilde{O}\left(H^6 d^4 A^2 \varepsilon^{-2}\right)$ episodes, which depends polynomially on the parameters $H, d, A, \varepsilon^{-1}$ and only has a logarithmic dependency on the cardinality of the model class $|\mathcal{M}|$. In particular, when reducing the Markov game to the single-agent MDP setting, the sample complexity of the model-based approach matches the result provided in (Uehara et al., 2021), which is known to have the best sample complexity among all oracle efficient algorithms for low-rank MDPs.

For model-free representation learning, we have the following guarantee:

**Theorem 4.2** (PAC guarantee of Algorithm 1 (model-free))**.** *When Alg. 1 is applied with model-free representation learning algorithm Alg. 3, and $\lambda = \Theta\left(d \log(NH|\Phi|/\delta)\right), \alpha^{(n)} = \Theta\left(HAd\sqrt{M \log(dNHAM|\Phi|/\delta)}\right), \zeta^{(n)} = \Theta\left(d^2 A n^{-1} \log(dNHAM|\Phi|/\delta)\right)$, and the Markov game is a Block Markov game. When we set the number of episodes $N$ to be at most*

$$O\left(H^6 d^4 A^3 M \varepsilon^{-2} \log^2\left(HdAM|\Phi|/\delta\varepsilon\right)\right),$$

*for an appropriately designed function class $\{\mathcal{N}_h\}_{h=1}^H$ and discriminator class $\{\mathcal{F}_h\}_{h=1}^H$, with probability $1 - \delta$, the output policy $\hat{\pi}$ is an $\varepsilon$-approximate $\{\text{NE}, \text{CCE}, \text{CE}\}$.*

For the model-free block Markov game setting, the number of episodes required to find an $\varepsilon$-approximate $\{\text{NE}, \text{CCE}, \text{CE}\}$ becomes $\tilde{O}\left(H^6 d^4 A^3 M \varepsilon^{-2}\right)$. While it has a worse dependency compared with the model-based approach, the advantage of the model-free approach is it doesn't require the full model class of the transition probability but only the model class of the feature vector, which applies to a wider range of RL problems.

The proofs of Theorem 4.1 and Theorem 4.2 are deferred to Appendix B and C. Theorem 4.1 and Theorem 4.2 show that GERL_MG2 learns low-rank Markov games in a statistically efficient and oracle-efficient manner. We also remark that our modular analysis can be of independent theoretical interest. Unlike prior works that make heavy distinctions between model-based and model-free approaches, e.g. (Liu et al., 2021), we show that both approaches can be analyzed in a unified manner.

## 5 FACTORED MARKOV GAMES

The result in Theorem 4.1 is tractable in games with a moderate number of players. However, in applications with a large number of players, such as the scenario of autonomous traffic control, the total number of players in the game can be so large that the joint action space size $A = \tilde{A}^M$ dominates all other factors in the sample complexity bound. This exponential scaling with the number of players is sometimes referred to as the *curse of multi-player*. The only known class of algorithms that overcomes this challenge in Markov games is V-learning (see, e.g., Bai et al., 2020; Jin et al., 2021b), a value-based method that fits the V-function rather than the Q-function, thus removing the dependency on the action space size. However, V-learning only works for tabular Markov games with finite state and action spaces. Extending V-learning to the function approximation setting is extremely

non-trivial, because even in the single agent setting, no known algorithm can achieve sample efficient learning in MDPs while only performing function approximation on the V-function.

In this section we take a different approach that relies on the following observation. In a setting where the number of agents is large, there is often a spatial correlation among the agents, such that each agent's local state is only immediately affected by the agent's own action and the states of agents in its adjacency. For example, in smart traffic control, a vehicle's local environment is only immediately affected by the states of the vehicles around it. On the other hand, it takes time for the course of actions of a vehicle from afar to propagate its influence on the vehicle of reference. Such spatial structure motivates the definition of a *factored* Markov Game.

In a factored Markov Game, each agent $i$ has its local state $s_i$, whose transition is affected by agent $i$'s action $\boldsymbol{a}_i$ and the state of the agents in its neighborhood $Z_i$. We remark that the factored Markov Game structure still allows an agent to be affected by all other agents in the long run, as long as the directed graph defined by the neighborhood sets $Z_i$ is connected. In particular, we have

**Definition 5.1** (Low-Rank Factored Markov Game). *We call a Markov game a low-rank factored Markov game if for any $s, s' \in \mathcal{S}, \boldsymbol{a} \in \mathcal{A}, h \in [H], i \in [M]$, we have*

$$P_h^\star(s'|s, \boldsymbol{a}) = \prod_{i=1}^M \left[ \phi_{h,i}^\star(s[Z_i], \boldsymbol{a}_i)^\top w_{h,i}^\star(s_i') \right].$$

*where $Z_i \subseteq [M]$, $\phi_{h,i}^\star(s[Z_i], \boldsymbol{a}_i), w_{h,i}^\star(s_i') \in \mathbb{R}^d$, $\|\phi_{h,i}^\star(s[Z_i], \boldsymbol{a}_i)\|_2 \le 1$ and $\|w_{h,i}^\star(s_i')\|_2 \le \sqrt{d}$ for all $(s[Z_i], \boldsymbol{a}_i, s_i')$. We assume $|Z_i| \le L, \forall i \in [M]$. And we are given a group of model classes $\mathcal{M}_{h,i}, h \in [H], i \in [M]$ such that $(\phi_{h,i}^\star, w_{h,i}^\star) \in \mathcal{M}_{h,i}$.*

We are now ready to present our algorithm and result in the low-rank factored Markov Game setting. Surprisingly, the same algorithm GERL_MG2 works in this setting, with the representation learning module Alg. 2 replaced by Alg. 4, and a few changes of variables. For simplicity, we focus on the model-based version. Define $\bar{\phi}_{h,i}^{(n)}(s, \boldsymbol{a}) = \bigotimes_{j \in Z_i} \hat{\phi}_{h,j}^{(n)}(s[Z_j], \boldsymbol{a}_j) \in \mathbb{R}^{d^{|Z_i|}}$ where $\otimes$ means the Kronecker product. Let

$$\hat{\beta}_h^{(n)}(s, \boldsymbol{a}) := \sum_{i=1}^M \min\{\alpha^{(n)} \|\bar{\phi}_{h,i}^{(n)}(s, \boldsymbol{a})\|_{\left(\bar{\Sigma}_{h,i}^{(n)}\right)^{-1}}, H\}, \Delta^{(n)} := \max_{i \in [M]}\{\overline{v}_i^{(n)} - \underline{v}_i^{(n)}\} + 2HM\sqrt{\tilde{A}\zeta^{(n)}}$$

where $\bar{\Sigma}_{h,i}^{(n)} := \sum_{(s, \boldsymbol{a}) \in \mathcal{D}_h^{(n)}} \bar{\phi}_{h,i}(s, \boldsymbol{a}) \bar{\phi}_{h,i}(s, \boldsymbol{a})^\top + \lambda I_{d^{|Z_i|}}$. Then, GERL_MG2 with $\bar{\phi}$ and the newly defined $\hat{\beta}^{(n)}, \Delta^{(n)}$ achieves the following guarantee:

**Theorem 5.1** (PAC guarantee of GERL_MG2 in Low-Rank Factored Markov Game). *When Alg. 1 is applied with model-based representation learning algorithm Alg. 4, with $L = O(1)$ and parameters $\lambda = \Theta\left(Ld^L \log(NHM|\Phi|/\delta)\right), \alpha^{(n)} = \Theta\left(H\tilde{A}d^L\sqrt{L \log(|\mathcal{M}|HNM/\delta)}\right), \zeta^{(n)} = \Theta\left(n^{-1} \log(|\mathcal{M}|HNM/\delta)\right)$, by setting the number of episodes $N$ to be at most*

$$O\left(M^4 H^6 d^{2(L+1)^2} \tilde{A}^{2(L+1)} \varepsilon^{-2} \log^2\left(HdALM|\mathcal{M}|/\delta\varepsilon\right)\right),$$

*with probability $1 - \delta$, the output policy $\hat{\pi}$ is an $\varepsilon$-approximate $\{NE, CCE, CE\}$.*

**Remark 5.1.** *This sample complexity only scales with $\exp(L)$ where $L$ is the degree of the connection graph, which is assumed to be $O(1)$ in Definition 5.1 and in general much smaller than the total number of agents in practice. We remark that the factored structure is also previously studied in single-agent tabular MDPs (examples include Chen et al. (2020); Kearns and Koller (1999); Guestrin et al. (2002, 2003); Strehl et al. (2007)). Chen et al. (2020) provided a lower-bound showing that the exponential dependency on $L$ is unimprovable in the worst case. Therefore, our bound here is also nearly tight, upto polynomial factors.*

## 6 EXPERIMENT

In this section we investigate our algorithm with proof-of-concept empirical studies. We design our testing bed using rich observation Markov game with arbitrary latent transitions and rewards. To

Table 1: **Top:** Short Horizon (H=3) exploitability of the final policy of DQN and GERL_MG2. **Bottom:** Long Horizon (H=10) exploitability of the final policy of DQN and GERL_MG2. Note that lower exploitability implies that the policy is closer to the NE policy.

|  | H=3 Environment 1 | H=3 Environment 2 | H=3 Environment 3 |
|---|---|---|---|
| DQN | 0.0851 (0.1152) | 0.0877 (0.1961) | 0.0090 (0.0200) |
| GERL_MG2 | 0.0013 (0.0018) | 0.0032 (0.0032) | 0.0004 (0.0009) |
|  | H=10 Environment 1 | H=10 Environment 2 | H=10 Environment 3 |
| DQN | 0.2730 (0.3270) | 0.0340 (0.0760) | 0.0320 (0.0170) |
| GERL_MG2 | 0.0780 (0.1560) | 0.0070 (0.0160) | 0.0060 (0.0130) |

solve the rich observation Markov game, an algorithm must correctly decode the latent structure (thus learning the dynamics) as well as solve the latent Markov game to find the NE/CE/CCE strategies concurrently. Below, we first introduce the setup of the experiments and then make comparisons with prior baselines in the two-player zero-sum setting. We then follow by showing the efficiency of GERL_MG2 in the general-sum setting. All further experiment details can be found in Appendix. F. Here we focus on the model-free version of GERL_MG2. Specifically, we implement Algorithm. 3 with deep learning libraries (Paszke et al., 2017). We defer more details to Appendix. F.2.

**Block Markov game** Block Markov game is a multi-agent extension of single agent Block MDP, as defined in Def. 2.5. We design our Block Markov game by first randomly generating a tabular Markov game with horizon $H$, 3 states, 2 players each with 3 actions, and random reward matrix $R_h \in (0,1)^{3 \times 3^2 \times H}$ and random transition matrix $T_h(s_h, a_h) \in \Delta_{S_{h+1}}$. We provide more details (e.g., generation of rich observation) in Appendix F.1.

**Zero-sum Markov game** In this section we first show the empirical evaluations under the two-player zero-sum Markov game setting. For an environment with horizon $H$, the randomly generated matrix $R$ denotes the reward for player 1 and $-R^\top$ denotes the reward for player 2, respectively. For the zero-sum game setting, we designed two variants of Block Markov games: one with short horizon ($H = 3$) and one with long horizon ($H = 10$). We show in the following that GERL_MG2 works in both settings where the other baseline could only work in the short horizon setting.

**Baseline** We adopt one open-sourced implementation of DQN (Silver et al., 2016) with fictitious self-play (Heinrich et al., 2015).

We keep track of the *exploitability of the returned strategy* to evaluate the practical performances of the baselines. In the zero-sum setting, we only need to fix one agent (e.g., agent 2), train the other single agent (the exploiter) to maximize its corresponding return until convergence, and report the difference between the returns of the exploiter and the final return of the final policies. We include the exploitability in Table. 1. We provide training curves in Appendix. F.3 for completeness. We note that compared with the Deep RL baseline, GERL_MG2 shows a faster and more stable convergence in both environments, where the baseline is unstable during training and has a much larger exploitability.

**General-sum Markov game.** In this section we move on to the general-sum setting. To our best knowledge, our algorithm is the only principled algorithm that can be implemented on scale under the general-sum setting. For the general sum setting, we can not just compare our returned value to the oracle NE values, because multiple NE/CCE values may exist. Instead, we keep track of the exploitability of the policy and plot the training curve on the exploitability in Fig. 2 (deferred to Appendix. F). Note that in this case we need to test both policies since their reward matrices are independently sampled.

## 7 DISCUSSION AND FUTURE WORKS

In this paper, we present the first algorithm that solves general-sum Markov games under function approximation. We provide both a model-based and a model-free variant of the algorithm and present a unified analysis. Empirically, we show that our algorithm outperforms existing deep RL baselines in a general benchmark with rich observation. Future work includes evaluating more challenging benchmarks and extending beyond the low-rank Markov game structure.

REPRODUCIBILITY STATEMENT

For theory, we provide proof and additional results in the Appendix. For empirical results, we provide implementation and environment details and hyperparameters in the Appendix. We also submit anonymous code in the supplemental materials.

ACKNOWLEDGMENTS

Mengdi Wang acknowledges the support by NSF grants DMS-1953686, IIS-2107304, CMMI1653435, ONR grant 1006977, and http://C3.AI.
Chi Jin gratefully acknowledges the support by Office of Naval Research Grant N00014-22-1-2253.

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

## A  ADDITIONAL NOTATIONS

Given a (possibly not normalized) transition probability $P : \mathcal{S} \times \mathcal{A} \times \mathcal{S} \times [H] \to [0,1]$ and a policy $\pi : \mathcal{S} \times [H] \to \Delta_{\mathcal{A}}$, we define the the density function of the state-action pair $(s, \boldsymbol{a})$ at step $h$ under transition $P$ and $\pi$ by

$$d_{P,1}^{\pi}(s, \boldsymbol{a}) := d_1(s)\pi_1(\boldsymbol{a}|s), \quad d_{P,h+1}^{\pi}(s, \boldsymbol{a}) := \sum_{\tilde{s} \in \mathcal{S}, \tilde{\boldsymbol{a}} \in \mathcal{A}} d_{P,h}^{\pi}(\tilde{s}, \tilde{\boldsymbol{a}}) P_h(s|\tilde{s}, \tilde{\boldsymbol{a}})\pi_{h+1}(\boldsymbol{a}|s), \forall h \geq 1.$$

We abuse the notations a bit and denote $d_{P,h}^{\pi}(s)$ as the marginalized state distribution, i.e., $d_{P,h}^{\pi}(s) = \sum_{\boldsymbol{a} \in \mathcal{A}} d_{P,h}^{\pi}(s, \boldsymbol{a})$. For any $n \in [N], h \in [H]$, define

$$\rho_h^{(n)}(s, \boldsymbol{a}) = \frac{1}{n} \sum_{i=1}^{n} d_{P^\star,h}^{\pi^{(i)}}(s) u_{\mathcal{A}}(\boldsymbol{a}),$$

$$\tilde{\rho}_h^{(n)}(s, \boldsymbol{a}) = \frac{1}{n} \sum_{i=1}^{n} \mathbb{E}_{\tilde{s} \sim d_{P^\star,h-1}^{\pi^{(i)}}, \tilde{\boldsymbol{a}} \sim U(\mathcal{A})} \left[ P^\star(s|\tilde{s}, \tilde{\boldsymbol{a}}) u_{\mathcal{A}}(\boldsymbol{a}) \right],$$

$$\gamma_h^{(n)}(s, \boldsymbol{a}) = \frac{1}{n} \sum_{i=1}^{n} d_{P^\star,h}^{\pi^{(i)}}(s, \boldsymbol{a}).$$

When we use the expectation $\mathbb{E}_{(s,\boldsymbol{a})\sim\rho}[f(s, \boldsymbol{a})]$ (or $\mathbb{E}_{s\sim\rho}[f(s)]$) for some (possibly not normalized) distribution $\rho$ and function $f$, we simply mean $\sum_{s \in \mathcal{S}, \boldsymbol{a} \in \mathcal{A}} \rho(s, \boldsymbol{a}) f(s, \boldsymbol{a})$ (or $\sum_{s \in \mathcal{S}} \rho(s) f(s)$) so that the expectation can be naturally extended to the unnormalized distributions. For an iteration $n$, a distribution $\rho$ and a feature $\phi$, we denote the expected feature covariance as

$$\Sigma_{n,\rho,\phi} = n\mathbb{E}_{(s,\boldsymbol{a})\sim\rho}\left[ \phi(s, \boldsymbol{a})\phi(s, \boldsymbol{a})^{\top} \right] + \lambda I_d.$$

Meanwhile, define the empirical covariance by

$$\hat{\Sigma}_{h,\phi}^{(n)} := \sum_{(s,\boldsymbol{a}) \in \mathcal{D}_h^{(n)}} \phi(s, \boldsymbol{a})\phi(s, \boldsymbol{a})^{\top} + \lambda I_d.$$

## B  ANALYSIS OF THE MODEL-BASED METHOD

### B.1  HIGH PROBABILITY EVENTS

We define the following event

$$\mathcal{E}_1 : \ \forall n \in [N], h \in [H], \rho \in \left\{ \rho_h^{(n)}, \tilde{\rho}_h^{(n)} \right\}, \quad \mathbb{E}_{(s,\boldsymbol{a})\sim\rho}\left[ \left\| \hat{P}_h^{(n)}(\cdot|s, \boldsymbol{a}) - P_h^\star(\cdot|s, \boldsymbol{a}) \right\|_1^2 \right] \leq \zeta^{(n)},$$

$$\mathcal{E}_2 : \ \forall n \in [N], h \in [H], \phi_h \in \Phi_h, s \in \mathcal{S}, \boldsymbol{a} \in \mathcal{A}, \quad \| \phi_h(s, \boldsymbol{a}) \|_{\left( \hat{\Sigma}_{h,\phi_h}^{(n)} \right)^{-1}} = \Theta\left( \| \phi_h(s, \boldsymbol{a}) \|_{\Sigma_{n,\rho_h^{(n)},\phi_h}^{-1}} \right)$$

$$\mathcal{E} := \mathcal{E}_1 \cap \mathcal{E}_2.$$

To prove $\mathcal{E}$ holds with a high probability, we first introduce the following MLE guarantee, whose original version can be found in (Agarwal et al., 2020b):

**Lemma B.1** (MLE guarantee). *For a fixed episode $n$ and any step $h$, with probability $1 - \delta$,*

$$\mathbb{E}_{(s,\boldsymbol{a})\sim\{0.5\rho_h^{(n)} + 0.5\tilde{\rho}_h^{(n)}\}} \left[ \left\| \hat{P}_h^{(n)}(\cdot|s, \boldsymbol{a}) - P_h^\star(\cdot|s, \boldsymbol{a}) \right\|_1^2 \right] \lesssim \frac{1}{n} \log \frac{|\mathcal{M}|}{\delta}.$$

*As a straightforward corollary, with probability $1 - \delta$,*

$$\forall n \in \mathbb{N}^+, \forall h \in [H], \quad \mathbb{E}_{(s,\boldsymbol{a})\sim\{0.5\rho_h^{(n)} + 0.5\tilde{\rho}_h^{(n)}\}} \left[ \left\| \hat{P}_h^{(n)}(\cdot|s, \boldsymbol{a}) - P_h^\star(\cdot|s, \boldsymbol{a}) \right\|_1^2 \right] \lesssim \frac{1}{n} \log \frac{nH|\mathcal{M}|}{\delta}.$$

(6)

*Proof.* See Agarwal et al.(Agarwal et al., 2020b) (Theorem 21). $\qquad\square$

Based on Lemma B.1 and Lemma E.1 in Appendix E, we directly get the following guarantee:

**Lemma B.2.** *When $\hat{P}_h^{(n)}$ is computed using Alg. 2, if we set*

$$\lambda = \Theta\left(d \log \frac{NH|\Phi|}{\delta}\right), \ \zeta^{(n)} = \Theta\left(\frac{1}{n} \log \frac{|\mathcal{M}|HN}{\delta}\right),$$

*then $\mathcal{E}$ holds with probability at least $1 - \delta$.*

### B.2 STATISTICAL GUARANTEES

**Lemma B.3** (One-step back inequality for the learned model). *Suppose the event $\mathcal{E}$ holds. Consider a set of functions $\{g_h\}_{h=1}^H$ that satisfies $g_h \in \mathcal{S} \times \mathcal{A} \to \mathbb{R}_+$, s.t. $\|g_h\|_\infty \leq B$. For any given policy $\pi$, we have*

$$\mathbb{E}_{(s,\boldsymbol{a})\sim d_{\hat{P}^{(n)},h}^\pi}\left[g_h(s,\boldsymbol{a})\right]$$
$$\leq \begin{cases} \sqrt{A\mathbb{E}_{(s,\boldsymbol{a})\sim\rho_1^{(n)}}\left[g_1^2(s,\boldsymbol{a})\right]}, & h = 1 \\ \mathbb{E}_{(\tilde{s},\tilde{\boldsymbol{a}})\sim d_{\hat{P}^{(n)},h-1}^\pi}\left[\min\left\{\left\|\hat{\phi}_{h-1}^{(n)}(\tilde{s},\tilde{\boldsymbol{a}})\right\|_{\Sigma_{n,\rho_{h-1}^{(n)},\hat{\phi}_{h-1}^{(n)}}^{-1}} \sqrt{nA\mathbb{E}_{(s,\boldsymbol{a})\sim\tilde{\rho}_h^{(n)}}\left[g_h^2(s,\boldsymbol{a})\right] + B^2\lambda d + B^2 n\zeta^{(n)}}, B\right\}\right], & h \geq 2 \end{cases}$$

Recall $\Sigma_{n,\rho_h^{(n)},\hat{\phi}_h^{(n)}} = n\mathbb{E}_{(s,\boldsymbol{a})\sim\rho_h^{(n)}}\left[\hat{\phi}_h^{(n)}(s,\boldsymbol{a})\hat{\phi}_h^{(n)}(s,\boldsymbol{a})^\top\right] + \lambda I_d$.

*Proof.* For step $h = 1$, we have

$$\mathbb{E}_{(s,\boldsymbol{a})\sim d_{\hat{P}^{(n)},1}^\pi}\left[g_1(s,\boldsymbol{a})\right] = \mathbb{E}_{s\sim d_1,\boldsymbol{a}\sim\pi_1(s)}\left[g_1(s,\boldsymbol{a})\right]$$
$$\leq \sqrt{\max_{(s,\boldsymbol{a})} \frac{d_1(s)\pi_1(\boldsymbol{a}|s)}{\rho_1^{(n)}(s,\boldsymbol{a})} \mathbb{E}_{(s',\boldsymbol{a}')\sim\rho_1^{(n)}}\left[g_1^2(s',\boldsymbol{a}')\right]}$$
$$= \sqrt{\max_{(s,\boldsymbol{a})} \frac{d_1(s)\pi_1(\boldsymbol{a}|s)}{d_1(s)u_\mathcal{A}(\boldsymbol{a})} \mathbb{E}_{(s',\boldsymbol{a}')\sim\rho_1^{(n)}}\left[g_1^2(s',\boldsymbol{a}')\right]}$$
$$\leq \sqrt{A\mathbb{E}_{(s,\boldsymbol{a})\sim\rho_1^{(n)}}\left[g_1^2(s,\boldsymbol{a})\right]}.$$

For step $h = 2, \dots, H - 1$, we observe the following one-step-back decomposition:

$$\mathbb{E}_{(s,\boldsymbol{a})\sim d_{\hat{P}^{(n)},h}^\pi}\left[g_h(s,\boldsymbol{a})\right]$$
$$= \mathbb{E}_{(\tilde{s},\tilde{\boldsymbol{a}})\sim d_{\hat{P}^{(n)},h-1}^\pi, s\sim\hat{P}_{h-1}^{(n)}(\tilde{s},\tilde{\boldsymbol{a}}),\boldsymbol{a}\sim\pi_h(s)}\left[g_h(s,\boldsymbol{a})\right]$$
$$= \mathbb{E}_{(\tilde{s},\tilde{\boldsymbol{a}})\sim d_{\hat{P}^{(n)},h-1}^\pi}\left[\hat{\phi}_{h-1}^{(n)}(\tilde{s},\tilde{\boldsymbol{a}})^\top \int_\mathcal{S} \sum_{\boldsymbol{a}\in\mathcal{A}} \hat{w}_{h-1}^{(n)}(s)\pi_h(\boldsymbol{a}|s)g_h(s,\boldsymbol{a})\mathrm{d}s\right]$$
$$= \mathbb{E}_{(\tilde{s},\tilde{\boldsymbol{a}})\sim d_{\hat{P}^{(n)},h-1}^\pi}\left[\min\left\{\hat{\phi}_{h-1}^{(n)}(\tilde{s},\tilde{\boldsymbol{a}})^\top \int_\mathcal{S} \sum_{\boldsymbol{a}\in\mathcal{A}} \hat{w}_{h-1}^{(n)}(s)\pi_h(\boldsymbol{a}|s)g_h(s,\boldsymbol{a})\mathrm{d}s, B\right\}\right]$$
$$\leq \mathbb{E}_{(\tilde{s},\tilde{\boldsymbol{a}})\sim d_{\hat{P}^{(n)},h-1}^\pi}\left[\min\left\{\left\|\hat{\phi}_{h-1}^{(n)}(\tilde{s},\tilde{\boldsymbol{a}})\right\|_{\Sigma_{n,\rho_{h-1}^{(n)},\hat{\phi}_{h-1}^{(n)}}^{-1}} \left\|\int_\mathcal{S} \sum_{\boldsymbol{a}\in\mathcal{A}} \hat{w}_{h-1}^{(n)}(s)\pi_h(\boldsymbol{a}|s)g_h(s,\boldsymbol{a})\mathrm{d}s\right\|_{\Sigma_{n,\rho_{h-1}^{(n)},\hat{\phi}_{h-1}^{(n)}}}, B\right\}\right].$$

where we use the fact that $g_h$ is bounded by $B$. Then,

$$\left\|\int_\mathcal{S} \sum_{\boldsymbol{a}\in\mathcal{A}} \hat{w}_{h-1}^{(n)}(s)\pi_h(\boldsymbol{a}|s)g_h(s,\boldsymbol{a})\mathrm{d}s\right\|_{\Sigma_{n,\rho_{h-1}^{(n)},\hat{\phi}_{h-1}^{(n)}}}^2$$

$$\leq \left( \int_{\mathcal{S}} \sum_{\boldsymbol{a} \in \mathcal{A}} \hat{w}_{h-1}^{(n)}(s) \pi_h(\boldsymbol{a}|s) g_h(s,\boldsymbol{a}) \mathrm{d}s \right)^{\top} \left( n \mathbb{E}_{(s,\boldsymbol{a}) \sim \rho_{h-1}^{(n)}} \left[ \hat{\phi}_{h-1}^{(n)}(s,\boldsymbol{a}) \hat{\phi}_{h-1}^{(n)}(s,\boldsymbol{a})^{\top} \right] + \lambda I_d \right) \left( \int_{\mathcal{S}} \sum_{\boldsymbol{a} \in \mathcal{A}} \hat{w}_{h-1}^{(n)}(s) \pi_h(\boldsymbol{a}|s) g_h(s,\boldsymbol{a}) \mathrm{d}s \right)$$

$$\leq n \mathbb{E}_{(\tilde{s},\tilde{\boldsymbol{a}}) \sim \rho_{h-1}^{(n)}} \left[ \left( \int_{\mathcal{S}} \sum_{\boldsymbol{a} \in \mathcal{A}} \hat{w}_{h-1}^{(n)}(s)^{\top} \hat{\phi}_{h-1}^{(n)}(\tilde{s},\tilde{\boldsymbol{a}}) \pi_h(\boldsymbol{a}|s) g_h(s,\boldsymbol{a}) \mathrm{d}s \right)^2 \right] + B^2 \lambda d$$

$$\left( \left\| \sum_{a \in \mathcal{A}} \pi_h(\boldsymbol{a}|s) g_h(s,\boldsymbol{a}) \right\|_{\infty} \leq B \text{ and by assumption } \left\| \hat{w}_{h-1}^{(n)}(s) \right\|_2 \leq \sqrt{d}. \right)$$

$$= n \mathbb{E}_{(\tilde{s},\tilde{\boldsymbol{a}}) \sim \rho_{h-1}^{(n)}} \left[ \left( \mathbb{E}_{s \sim \hat{P}_{h-1}^{(n)}(\tilde{s},\tilde{\boldsymbol{a}}), \boldsymbol{a} \sim \pi_h(s)} [g_h(s,\boldsymbol{a})] \right)^2 \right] + B^2 \lambda d$$

$$\leq n \mathbb{E}_{(\tilde{s},\tilde{\boldsymbol{a}}) \sim \rho_{h-1}^{(n)}} \left[ \left( \mathbb{E}_{s \sim P_{h-1}^{\star}(\tilde{s},\tilde{\boldsymbol{a}}), \boldsymbol{a} \sim \pi_h(s)} [g_h(s,\boldsymbol{a})] \right)^2 \right] + B^2 \lambda d + n B^2 \xi^{(n)} \qquad \text{(Event } \mathcal{E}\text{)}$$

$$\leq n \mathbb{E}_{(\tilde{s},\tilde{\boldsymbol{a}}) \sim \rho_{h-1}^{(n)}, s \sim P_{h-1}^{\star}(\tilde{s},\tilde{\boldsymbol{a}}), \boldsymbol{a} \sim \pi_h(s)} [g_h^2(s,\boldsymbol{a})] + B^2 \lambda d + B^2 n \xi^{(n)}. \qquad \text{(Jensen)}$$

$$\leq n A \mathbb{E}_{(\tilde{s},\tilde{\boldsymbol{a}}) \sim \rho_{h-1}^{(n)}, s \sim P_{h-1}^{\star}(\tilde{s},\tilde{\boldsymbol{a}}), \boldsymbol{a} \sim U(\mathcal{A})} [g_h^2(s,\boldsymbol{a})] + B^2 \lambda d + B^2 n \zeta^{(n)} \qquad \text{(Importance sampling)}$$

$$\leq n A \mathbb{E}_{(s,\boldsymbol{a}) \sim \tilde{\rho}_h^{(n)}} [g_h^2(s,\boldsymbol{a})] + B^2 \lambda d + B^2 n \zeta^{(n)}. \qquad \text{(Definition of } \tilde{\rho}_h^{(n)}\text{)}$$

Combing the above results together, we get

$$\mathbb{E}_{(s,\boldsymbol{a}) \sim d_{\hat{P}^{(n)},h}^{\pi}} [g_h(s,\boldsymbol{a})]$$

$$\leq \mathbb{E}_{(\tilde{s},\tilde{\boldsymbol{a}}) \sim d_{\hat{P}^{(n)},h-1}^{\pi}} \left[ \min \left\{ \left\| \hat{\phi}_{h-1}^{(n)}(\tilde{s},\tilde{\boldsymbol{a}}) \right\|_{\Sigma_{n,\rho_{h-1}^{(n)},\hat{\phi}_{h-1}}^{-1}} \left\| \int_{\mathcal{S}} \sum_{\boldsymbol{a} \in \mathcal{A}} \hat{w}_{h-1}^{(n)}(s) \pi_h(\boldsymbol{a}|s) g_h(s,\boldsymbol{a}) \mathrm{d}s \right\|_{\Sigma_{n,\rho_{h-1}^{(n)},\hat{\phi}_{h-1}}}, B \right\} \right]$$

$$\leq \mathbb{E}_{(\tilde{s},\tilde{\boldsymbol{a}}) \sim d_{\hat{P}^{(n)},h-1}^{\pi}} \left[ \min \left\{ \left\| \hat{\phi}_{h-1}^{(n)}(\tilde{s},\tilde{\boldsymbol{a}}) \right\|_{\Sigma_{n,\rho_{h-1}^{(n)},\hat{\phi}_{h-1}}^{-1}} \sqrt{n A \mathbb{E}_{(s,\boldsymbol{a}) \sim \tilde{\rho}_h^{(n)}} [g_h^2(s,\boldsymbol{a})] + B^2 \lambda d + B^2 n \zeta^{(n)}}, B \right\} \right],$$

which has finished the proof. $\qquad \square$

**Lemma B.4** (One-step back inequality for the true model). *Consider a set of functions $\{g_h\}_{h=1}^H$ that satisfies $g_h \in \mathcal{S} \times \mathcal{A} \to \mathbb{R}_+$, s.t. $\|g_h\|_{\infty} \leq B$. Then for any given policy $\pi$, we have*

$$\mathbb{E}_{(s,\boldsymbol{a}) \sim d_{P^{\star},h}^{\pi}} [g_h(s,\boldsymbol{a})]$$

$$\leq \begin{cases} \sqrt{A \mathbb{E}_{(s,\boldsymbol{a}) \sim \rho_1^{(n)}} [g_1^2(s,\boldsymbol{a})]}, & h = 1 \\ \mathbb{E}_{(\tilde{s},\tilde{\boldsymbol{a}}) \sim d_{P^{\star},h-1}^{\pi}} \left[ \left\| \phi_{h-1}^{\star}(\tilde{s},\tilde{\boldsymbol{a}}) \right\|_{\Sigma_{n,\gamma_{h-1}^{(n)},\phi_{h-1}^{\star}}^{-1}} \right] \sqrt{n A \mathbb{E}_{(s,\boldsymbol{a}) \sim \rho_h^{(n)}} [g_h^2(s,\boldsymbol{a})] + B^2 \lambda d}, & h \geq 2 \end{cases}$$

Recall $\Sigma_{n,\gamma_h^{(n)},\phi_h^{\star}} = n \mathbb{E}_{(s,\boldsymbol{a}) \sim \gamma_h^{(n)}} \left[ \phi_h^{\star}(s,\boldsymbol{a}) \phi_h^{\star}(s,\boldsymbol{a})^{\top} \right] + \lambda I_d$.

*Proof.* For step $h = 1$, we have

$$\mathbb{E}_{(s,\boldsymbol{a}) \sim d_{P^{\star},1}^{\pi}} [g_1(s,\boldsymbol{a})] = \mathbb{E}_{s \sim d_1, \boldsymbol{a} \sim \pi_1(s)} [g_1(s,\boldsymbol{a})]$$

$$\leq \sqrt{\max_{(s,\boldsymbol{a})} \frac{d_1(s) \pi_1(\boldsymbol{a}|s)}{\rho_1^{(n)}(s,\boldsymbol{a})} \mathbb{E}_{(s',\boldsymbol{a}') \sim \rho_1^{(n)}} [g_1^2(s',\boldsymbol{a}')]}$$

$$= \sqrt{\max_{(s,\boldsymbol{a})} \frac{d_1(s) \pi_1(\boldsymbol{a}|s)}{d_1(s) u_{\mathcal{A}}(\boldsymbol{a})} \mathbb{E}_{(s',\boldsymbol{a}') \sim \rho_1^{(n)}} [g_1^2(s',\boldsymbol{a}')]}$$

$$\leq \sqrt{A \mathbb{E}_{(s,\boldsymbol{a}) \sim \rho_1^{(n)}} [g_1^2(s,\boldsymbol{a})]}.$$

For step $h = 2, \ldots, H - 1$, we observe the following one-step-back decomposition:

$$\mathbb{E}_{(s,\boldsymbol{a}) \sim d_{P^{\star},h}^{\pi}} [g_h(s,\boldsymbol{a})]$$

$$=\mathbb{E}_{(\tilde{s},\tilde{\boldsymbol{a}})\sim d^{\pi}_{P^{\star},h-1},s\sim P^{\star}_{h-1}(\tilde{s},\tilde{\boldsymbol{a}}),\boldsymbol{a}\sim\pi_h(s)}\left[g_h(s,\boldsymbol{a})\right]$$

$$=\mathbb{E}_{(\tilde{s},\tilde{\boldsymbol{a}})\sim d^{\pi}_{P^{\star},h-1}}\left[\phi^{\star}_{h-1}(\tilde{s},\tilde{\boldsymbol{a}})^{\top}\int_{\mathcal{S}}\sum_{\boldsymbol{a}\in\mathcal{A}}w^{\star}_{h-1}(s)\pi_h(\boldsymbol{a}|s)g_h(s,\boldsymbol{a})\mathrm{d}s\right]$$

$$\leq\mathbb{E}_{(\tilde{s},\tilde{\boldsymbol{a}})\sim d^{\pi}_{P^{\star},h-1}}\left[\left\|\phi^{\star}_{h-1}(\tilde{s},\tilde{\boldsymbol{a}})\right\|_{\Sigma^{-1}_{n,\gamma^{(n)}_{h-1},\phi^{\star}_{h-1}}}\right]\left\|\int_{\mathcal{S}}\sum_{\boldsymbol{a}\in\mathcal{A}}w^{\star}_{h-1}(s)\pi_h(\boldsymbol{a}|s)g_h(s,\boldsymbol{a})\mathrm{d}s\right\|_{\Sigma_{n,\gamma^{(n)}_{h-1},\phi^{\star}_{h-1}}}.$$

Then,

$$\left\|\int_{\mathcal{S}}\sum_{\boldsymbol{a}\in\mathcal{A}}w^{\star}_{h-1}(s)\pi_h(\boldsymbol{a}|s)g_h(s,\boldsymbol{a})\mathrm{d}s\right\|^2_{\Sigma_{n,\gamma^{(n)}_{h-1},\phi^{\star}_{h-1}}}$$

$$\leq\left(\int_{\mathcal{S}}\sum_{\boldsymbol{a}\in\mathcal{A}}w^{\star}_{h-1}(s)\pi_h(\boldsymbol{a}|s)g_h(s,\boldsymbol{a})\mathrm{d}s\right)^{\top}\left(n\mathbb{E}_{(s,\boldsymbol{a})\sim\gamma^{(n)}_{h-1}}\left[\phi^{\star}_{h-1}(s,\boldsymbol{a})\phi^{\star}_{h-1}(s,\boldsymbol{a})^{\top}\right]+\lambda I_d\right)\left(\int_{\mathcal{S}}\sum_{\boldsymbol{a}\in\mathcal{A}}w^{\star}_{h-1}(s)\pi_h(\boldsymbol{a}|s)g_h(s,\boldsymbol{a})\mathrm{d}s\right)$$

$$\leq n\mathbb{E}_{(\tilde{s},\tilde{\boldsymbol{a}})\sim\gamma^{(n)}_{h-1}}\left[\left(\int_{\mathcal{S}}\sum_{\boldsymbol{a}\in\mathcal{A}}w^{\star}_{h-1}(s)^{\top}\phi^{\star}_{h-1}(\tilde{s},\tilde{\boldsymbol{a}})\pi_h(\boldsymbol{a}|s)g_h(s,\boldsymbol{a})\mathrm{d}s\right)^2\right]+B^2\lambda d$$

$$\text{(Use the assumption }\left\|\sum_{\boldsymbol{a}\in\mathcal{A}}\pi_h(\boldsymbol{a}|s)g_h(s,\boldsymbol{a})\right\|_{\infty}\leq B\text{ and }\|w^{\star}_{h-1}(s)\|_2\leq\sqrt{d}.)$$

$$=n\mathbb{E}_{(\tilde{s},\tilde{\boldsymbol{a}})\sim\gamma^{(n)}_{h-1}}\left[\left(\mathbb{E}_{s\sim P^{\star}_{h-1}(\tilde{s},\tilde{\boldsymbol{a}}),\boldsymbol{a}\sim\pi_h(s)}\left[g_h(s,\boldsymbol{a})\right]\right)^2\right]+B^2\lambda d$$

$$\leq n\mathbb{E}_{(\tilde{s},\tilde{\boldsymbol{a}})\sim\gamma^{(n)}_{h-1},s\sim P^{\star}_{h-1}(\tilde{s},\tilde{\boldsymbol{a}}),\boldsymbol{a}\sim\pi_h(s)}\left[g^2_h(s,\boldsymbol{a})\right]+B^2\lambda d \qquad\text{(Jensen)}$$

$$\leq nA\mathbb{E}_{(\tilde{s},\tilde{\boldsymbol{a}})\sim\gamma^{(n)}_{h-1},s\sim P^{\star}_{h-1}(\tilde{s},\tilde{\boldsymbol{a}}),\boldsymbol{a}\sim U(\mathcal{A})}\left[g^2_h(s,\boldsymbol{a})\right]+B^2\lambda d \qquad\text{(Importance sampling)}$$

$$\leq nA\mathbb{E}_{(s,\boldsymbol{a})\sim\rho^{(n)}_h}\left[g^2_h(s,\boldsymbol{a})\right]+B^2\lambda d, \qquad\text{(Definition of }\rho^{(n)}_h)$$

Combing the above results together, we get

$$\mathbb{E}_{(s,\boldsymbol{a})\sim d^{\pi}_{P^{\star},h}}\left[g_h(s,\boldsymbol{a})\right]$$

$$=\mathbb{E}_{(\tilde{s},\tilde{\boldsymbol{a}})\sim d^{\pi}_{P^{\star},h-1},s\sim P^{\star}_{h-1}(\tilde{s},\tilde{\boldsymbol{a}}),\boldsymbol{a}\sim\pi_h(s)}\left[g_h(s,\boldsymbol{a})\right]$$

$$\leq\mathbb{E}_{(\tilde{s},\tilde{\boldsymbol{a}})\sim d^{\pi}_{P^{\star},h-1}}\left[\left\|\phi^{\star}_{h-1}(\tilde{s},\tilde{\boldsymbol{a}})\right\|_{\Sigma^{-1}_{n,\gamma^{(n)}_{h-1},\phi^{\star}_{h-1}}}\right]\left\|\int_{\mathcal{S}}\sum_{\boldsymbol{a}\in\mathcal{A}}w^{\star}_{h-1}(s)\pi_h(\boldsymbol{a}|s)g_h(s,\boldsymbol{a})\mathrm{d}s\right\|_{\Sigma_{n,\gamma^{(n)}_{h-1},\phi^{\star}_{h-1}}}$$

$$\leq\mathbb{E}_{(\tilde{s},\tilde{\boldsymbol{a}})\sim d^{\pi}_{P^{\star},h-1}}\left[\left\|\phi^{\star}_{h-1}(\tilde{s},\tilde{\boldsymbol{a}})\right\|_{\Sigma^{-1}_{n,\gamma^{(n)}_{h-1},\phi^{\star}_{h-1}}}\right]\sqrt{nA\mathbb{E}_{(s,\boldsymbol{a})\sim\rho^{(n)}_h}\left[g^2_h(s,\boldsymbol{a})\right]+B^2\lambda d},$$

which has finished the proof. $\qquad\square$

**Lemma B.5** (Optimism for NE and CCE). *Consider an episode $n\in[N]$ and set $\alpha^{(n)}=\Theta\left(H\sqrt{nA\zeta^{(n)}+d\lambda}\right)$. When the event $\mathcal{E}$ holds and the policy $\pi^{(n)}$ is computed by solving NE or CCE, we have*

$$\overline{v}^{(n)}_i(s)-v^{\dagger,\pi^{(n)}_{-i}}_i(s)\geq -H\sqrt{A\zeta^{(n)}},\quad\forall n\in[N],i\in[M].$$

*Proof.* Define $\tilde{\mu}^{(n)}_{h,i}(\cdot|s):=\arg\max_{\mu}\left(\mathbb{D}_{\mu,\pi^{(n)}_{h,-i}}Q^{\dagger,\pi^{(n)}_{-i}}_{h,i}\right)(s)$ as the best response policy for player $i$ at step $h$, and let $\tilde{\pi}^{(n)}_h=\tilde{\mu}^{(n)}_{h,i}\times\pi^{(n)}_{h,-i}$. Let $f^{(n)}_h(s,\boldsymbol{a})=\left\|\hat{P}^{(n)}_h(\cdot|s,\boldsymbol{a})-P^{\star}_h(\cdot|s,\boldsymbol{a})\right\|_1$, then according to the event $\mathcal{E}$, we have

$$\mathbb{E}_{(s,\boldsymbol{a})\sim\rho^{(n)}_h}\left[\left(f^{(n)}_h(s,\boldsymbol{a})\right)^2\right]\leq\zeta^{(n)},\quad\mathbb{E}_{(s,\boldsymbol{a})\sim\tilde{\rho}^{(n)}_h}\left[\left(f^{(n)}_h(s,\boldsymbol{a})\right)^2\right]\leq\zeta^{(n)},\quad\forall n\in[N],h\in[H],$$

$$\|\phi_h(s,\boldsymbol{a})\|_{\left(\hat{\Sigma}_{h,\phi_h}^{(n)}\right)^{-1}} = \Theta\left(\|\phi_h(s,\boldsymbol{a})\|_{\Sigma_{n,\rho_h^{(n)},\phi_h}^{-1}}\right), \quad \forall n \in [N], h \in [H], \phi_h \in \Phi_h.$$

A direct conclusion of the event $\mathcal{E}$ is we can find an absolute constant $c$, such that

$$\beta_h^{(n)}(s,\boldsymbol{a}) = \min\left\{\alpha^{(n)}\left\|\hat{\phi}_h^{(n)}(\tilde{s},\tilde{\boldsymbol{a}})\right\|_{\left(\Sigma_{h,\hat{\phi}_h^{(n)}}^{(n)}\right)^{-1}}, H\right\}$$

$$\geq \min\left\{c\alpha^{(n)}\left\|\hat{\phi}_h^{(n)}(\tilde{s},\tilde{\boldsymbol{a}})\right\|_{\Sigma_{n,\rho_h^{(n)},\hat{\phi}_h^{(n)}}^{-1}}, H\right\}, \quad \forall n \in [N], h \in [H].$$

Next, we prove by induction that

$$\mathbb{E}_{s\sim d_{\hat{P}^{(n)},h}^{\tilde{\pi}^{(n)}}}\left[\overline{V}_{h,i}^{(n)}(s) - V_{h,i}^{\dagger,\pi_{-i}^{(n)}}(s)\right] \geq \sum_{h'=h}^{H} \mathbb{E}_{(s,\boldsymbol{a})\sim d_{\hat{P}^{(n)},h'}^{\tilde{\pi}^{(n)}}}\left[\hat{\beta}_{h'}^{(n)}(s,\boldsymbol{a}) - H\min\{f_{h'}^{(n)}(s,\boldsymbol{a}),1\}\right], \quad \forall h \in [H]. \tag{7}$$

First, notice that $\forall h \in [H]$,

$$\mathbb{E}_{s\sim d_{\hat{P}^{(n)},h}^{\tilde{\pi}^{(n)}}}\left[\overline{V}_{h,i}^{(n)}(s) - V_{h,i}^{\dagger,\pi_{-i}^{(n)}}(s)\right] = \mathbb{E}_{s\sim d_{\hat{P}^{(n)},h}^{\tilde{\pi}^{(n)}}}\left[\left(\mathbb{D}_{\pi_h^{(n)}}\overline{Q}_{h,i}^{(n)}\right)(s) - \left(\mathbb{D}_{\tilde{\pi}_h^{(n)}}Q_{h,i}^{\dagger,\pi_{-i}^{(n)}}\right)(s)\right]$$

$$\geq \mathbb{E}_{s\sim d_{\hat{P}^{(n)},h}^{\tilde{\pi}^{(n)}}}\left[\left(\mathbb{D}_{\tilde{\pi}_h^{(n)}}\overline{Q}_{h,i}^{(n)}\right)(s) - \left(\mathbb{D}_{\tilde{\pi}_h^{(n)}}Q_{h,i}^{\dagger,\pi_{-i}^{(n)}}\right)(s)\right]$$

$$= \mathbb{E}_{(s,\boldsymbol{a})\sim d_{\hat{P}^{(n)},h}^{\tilde{\pi}^{(n)}}}\left[\overline{Q}_{h,i}^{(n)}(s,\boldsymbol{a}) - Q_{h,i}^{\dagger,\pi_{-i}^{(n)}}(s,\boldsymbol{a})\right],$$

where the inequality uses the fact that $\pi_h^{(n)}$ is the NE (or CCE) solution for $\left\{\overline{Q}_{h,i}^{(n)}\right\}_{i=1}^{M}$. Now we are ready to prove equation 7:

- When $h = H$, we have

$$\mathbb{E}_{s\sim d_{\hat{P}^{(n)},H}^{\tilde{\pi}^{(n)}}}\left[\overline{V}_{H,i}^{(n)}(s) - V_{H,i}^{\dagger,\pi_{-i}^{(n)}}(s)\right] \geq \mathbb{E}_{(s,\boldsymbol{a})\sim d_{\hat{P}^{(n)},H}^{\tilde{\pi}^{(n)}}}\left[\overline{Q}_{H,i}^{(n)}(s,\boldsymbol{a}) - Q_{H,i}^{\dagger,\pi_{-i}^{(n)}}(s,\boldsymbol{a})\right]$$

$$= \mathbb{E}_{(s,\boldsymbol{a})\sim d_{\hat{P}^{(n)},H}^{\tilde{\pi}^{(n)}}}\left[\hat{\beta}_h^{(n)}(s,\boldsymbol{a})\right]$$

$$\geq \mathbb{E}_{(s,\boldsymbol{a})\sim d_{\hat{P}^{(n)},H}^{\tilde{\pi}^{(n)}}}\left[\hat{\beta}_h^{(n)}(s,\boldsymbol{a}) - H\min\left\{f_H^{(n)}(s,\boldsymbol{a}),1\right\}\right].$$

- Suppose the statement is true for step $h+1$, then for step $h$, we have

$$\mathbb{E}_{s\sim d_{\hat{P}^{(n)},h}^{\tilde{\pi}^{(n)}}}\left[\overline{V}_{h,i}^{(n)}(s) - V_{h,i}^{\dagger,\pi_{-i}^{(n)}}(s)\right]$$

$$\geq \mathbb{E}_{(s,\boldsymbol{a})\sim d_{\hat{P}^{(n)},h}^{\tilde{\pi}^{(n)}}}\left[\overline{Q}_{h,i}^{(n)}(s,\boldsymbol{a}) - Q_{h,i}^{\dagger,\pi_{-i}^{(n)}}(s,\boldsymbol{a})\right]$$

$$= \mathbb{E}_{(s,\boldsymbol{a})\sim d_{\hat{P}^{(n)},h}^{\tilde{\pi}^{(n)}}}\left[\hat{\beta}_h^{(n)}(s,\boldsymbol{a}) + \left(\hat{P}_h^{(n)}\overline{V}_{h+1,i}^{(n)}\right)(s,\boldsymbol{a}) - \left(P_h^{\star}V_{h+1,i}^{\dagger,\pi_{-i}^{(n)}}\right)(s,\boldsymbol{a})\right]$$

$$= \mathbb{E}_{(s,\boldsymbol{a})\sim d_{\hat{P}^{(n)},h}^{\tilde{\pi}^{(n)}}}\left[\hat{\beta}_h^{(n)}(s,\boldsymbol{a}) + \left(\hat{P}_h^{(n)}\left(\overline{V}_{h+1,i}^{(n)} - V_{h+1,i}^{\dagger,\pi_{-i}^{(n)}}\right)\right)(s,\boldsymbol{a}) + \left(\left(\hat{P}_h^{(n)} - P_h^{\star}\right)V_{h+1,i}^{\dagger,\pi_{-i}^{(n)}}\right)(s,\boldsymbol{a})\right]$$

$$= \mathbb{E}_{(s,\boldsymbol{a})\sim d_{\hat{P}^{(n)},h}^{\tilde{\pi}^{(n)}}}\left[\hat{\beta}_h^{(n)}(s,\boldsymbol{a}) + \left(\left(\hat{P}_h^{(n)} - P_h^{\star}\right)V_{h+1,i}^{\dagger,\pi_{-i}^{(n)}}\right)(s,\boldsymbol{a})\right] + \mathbb{E}_{s\sim d_{\hat{P}^{(n)},h+1}^{\tilde{\pi}^{(n)}}}\left[\overline{V}_{h+1,i}^{(n)}(s) - V_{h+1,i}^{\dagger,\pi_{-i}^{(n)}}(s)\right]$$

$$\geq \mathbb{E}_{(s,\boldsymbol{a})\sim d_{\hat{P}^{(n)},h}^{\tilde{\pi}^{(n)}}}\left[\hat{\beta}_h^{(n)}(s,\boldsymbol{a}) - H\min\left\{f_h^{(n)}(s,\boldsymbol{a}),1\right\}\right] + \mathbb{E}_{s\sim d_{\hat{P}^{(n)},h+1}^{\tilde{\pi}^{(n)}}}\left[\overline{V}_{h+1,i}^{(n)}(s) - V_{h+1,i}^{\dagger,\pi_{-i}^{(n)}}(s)\right]$$

$$\geq \sum_{h'=h}^{H} \mathbb{E}_{(s,\boldsymbol{a}) \sim d_{\hat{P}^{(n)},h'}^{\tilde{\pi}^{(n)}}} \left[ \hat{\beta}_{h'}^{(n)}(s,\boldsymbol{a}) - H \min \left\{ f_{h'}^{(n)}(s,\boldsymbol{a}), 1 \right\} \right],$$

where we use the fact

$$\left| \left( \hat{P}_h^{(n)} - P_h^\star \right) V_{h+1,i}^{\dagger,\pi_{-i}^{(n)}} \right| (s,\boldsymbol{a}) \leq \min \left\{ H, \left\| \hat{P}_h^{(n)}(\cdot|s,\boldsymbol{a}) - P_h^\star(\cdot|s,\boldsymbol{a}) \right\|_1 \left\| V_{h+1,i}^{\dagger,\pi_{-i}^{(n)}} \right\|_\infty \right\}$$

$$\leq H \min \left\{ 1, \left\| \hat{P}_h^{(n)}(\cdot|s,\boldsymbol{a}) - P_h^\star(\cdot|s,\boldsymbol{a}) \right\|_1 \right\}$$

$$= H \min \left\{ 1, f_{h'}^{(n)}(s,\boldsymbol{a}) \right\}$$

and the last row uses the induction assumption.

Therefore, we have proved equation 7. We then apply $h = 1$ to equation 7, and get

$$\mathbb{E}_{s \sim d_1} \left[ \overline{V}_{1,i}^{(n)}(s) - V_{1,i}^{\dagger,\pi_{-i}^{(n)}}(s) \right] = \mathbb{E}_{s \sim d_{\hat{P}^{(n)},1}^{\tilde{\pi}^{(n)}}} \left[ \overline{V}_{1,i}^{(n)}(s) - V_{1,i}^{\dagger,\pi_{-i}^{(n)}}(s) \right]$$

$$\geq \sum_{h=1}^{H} \mathbb{E}_{(s,\boldsymbol{a}) \sim d_{\hat{P}^{(n)},h}^{\tilde{\pi}^{(n)}}} \left[ \hat{\beta}_h^{(n)}(s,\boldsymbol{a}) - H \min \left\{ f_h^{(n)}(s,\boldsymbol{a}), 1 \right\} \right]$$

$$= \sum_{h=1}^{H} \mathbb{E}_{(s,\boldsymbol{a}) \sim d_{\hat{P}^{(n)},h}^{\tilde{\pi}^{(n)}}} \left[ \hat{\beta}_h^{(n)}(s,\boldsymbol{a}) \right] - H \sum_{h=1}^{H} \mathbb{E}_{(s,\boldsymbol{a}) \sim d_{\hat{P}^{(n)},h}^{\tilde{\pi}^{(n)}}} \left[ \min \left\{ f_h^{(n)}(s,\boldsymbol{a}), 1 \right\} \right].$$

Next we are going to bound the second term, let $g_h(s,\boldsymbol{a}) = \min\{f_h^{(n)}(s,\boldsymbol{a}), 1\}$ and apply Lemma B.3 to $g_h$, we have for $h = 1$,

$$\mathbb{E}_{(s,\boldsymbol{a}) \sim d_{\hat{P}^{(n)},1}^{\tilde{\pi}^{(n)}}} \left[ \min \left\{ f_1^{(n)}(s,\boldsymbol{a}), 1 \right\} \right] \leq \sqrt{A \mathbb{E}_{(s,\boldsymbol{a}) \sim \rho_1^{(n)}} \left[ \left( f_1^{(n)}(s,\boldsymbol{a}) \right)^2 \right]} \leq \sqrt{A \zeta^{(n)}}.$$

And $\forall h \geq 2$, we have

$$\mathbb{E}_{(s,\boldsymbol{a}) \sim d_{\hat{P}^{(n)},h}^{\tilde{\pi}^{(n)}}} \left[ \min \left\{ f_h^{(n)}(s,\boldsymbol{a}), 1 \right\} \right]$$

$$\leq \mathbb{E}_{(\tilde{s},\tilde{\boldsymbol{a}}) \sim d_{\hat{P}^{(n)},h-1}^{\tilde{\pi}^{(n)}}} \left[ \min \left\{ \left\| \hat{\phi}_{h-1}^{(n)}(\tilde{s},\tilde{\boldsymbol{a}}) \right\|_{\Sigma_{n,\rho_{h-1}^{(n)},\hat{\phi}_{h-1}}^{-1}} \sqrt{n A \mathbb{E}_{(s,\boldsymbol{a}) \sim \tilde{\rho}_h^{(n)}} \left[ \left( f_h^{(n)}(s,\boldsymbol{a}) \right)^2 \right] + d\lambda + n\zeta^{(n)}}, 1 \right\} \right]$$

$$\lesssim \mathbb{E}_{(\tilde{s},\tilde{\boldsymbol{a}}) \sim d_{\hat{P}^{(n)},h-1}^{\tilde{\pi}^{(n)}}} \left[ \min \left\{ \left\| \hat{\phi}_{h-1}^{(n)}(\tilde{s},\tilde{\boldsymbol{a}}) \right\|_{\Sigma_{n,\rho_{h-1}^{(n)},\hat{\phi}_{h-1}}^{-1}} \sqrt{n A \zeta^{(n)} + d\lambda + n\zeta^{(n)}}, 1 \right\} \right].$$

Note that we here use the fact $\min\{f_h^{(n)}(s,\boldsymbol{a}), 1\} \leq 1$, $\mathbb{E}_{(s,\boldsymbol{a}) \sim \rho_h^{(n)}} \left[ \left( f_h^{(n)}(s,\boldsymbol{a}) \right)^2 \right] \leq \zeta^{(n)}$ and

$\mathbb{E}_{(s,\boldsymbol{a}) \sim \tilde{\rho}_h^{(n)}} \left[ \left( f_h^{(n)}(s,\boldsymbol{a}) \right)^2 \right] \leq \zeta^{(n)}$. Then according to our choice of $\alpha^{(n)}$, we get

$$\mathbb{E}_{(s,\boldsymbol{a}) \sim d_{\hat{P}^{(n)},h}^{\tilde{\pi}^{(n)}}} \left[ \min \left\{ f_h^{(n)}(s,\boldsymbol{a}), 1 \right\} \right] \leq \mathbb{E}_{(\tilde{s},\tilde{\boldsymbol{a}}) \sim d_{\hat{P}^{(n)},h-1}^{\tilde{\pi}^{(n)}}} \left[ \min \left\{ \frac{c\alpha^{(n)}}{H} \left\| \hat{\phi}_{h-1}^{(n)}(\tilde{s},\tilde{\boldsymbol{a}}) \right\|_{\Sigma_{n,\rho_{h-1}^{(n)},\hat{\phi}_{h-1}}^{-1}}, 1 \right\} \right].$$

Combining all things together,

$$\overline{v}_i^{(n)} - v_i^{\dagger,\pi_{-i}^{(n)}} = \mathbb{E}_{s \sim d_1} \left[ \overline{V}_{1,i}^{(n)}(s) - V_{1,i}^{\dagger,\pi_{-i}^{(n)}}(s) \right]$$

$$\geq \sum_{h=1}^{H} \mathbb{E}_{(s,\boldsymbol{a}) \sim d_{\hat{P}^{(n)},h}^{\tilde{\pi}^{(n)}}} \left[ \hat{\beta}_h^{(n)}(s,\boldsymbol{a}) \right] - H \sum_{h=1}^{H} \mathbb{E}_{(s,\boldsymbol{a}) \sim d_{\hat{P}^{(n)},h}^{\tilde{\pi}^{(n)}}} \left[ \min \left\{ f_h^{(n)}(s,\boldsymbol{a}), 1 \right\} \right]$$

$$\geq \sum_{h=1}^{H-1} \mathbb{E}_{(\tilde{s},\tilde{a})\sim d_{\hat{P}^{(n)},h}^{\tilde{\pi}^{(n)}}} \left[ \hat{\beta}_h^{(n)}(s,\boldsymbol{a}) - \min\left\{ c\alpha^{(n)} \left\| \hat{\phi}_h^{(n)}(\tilde{s},\tilde{\boldsymbol{a}}) \right\|_{\Sigma_{n,\rho_h^{(n)},\hat{\phi}_h^{(n)}}^{-1}}, H \right\} \right] - H\sqrt{A\zeta^{(n)}}$$

$$\geq -H\sqrt{A\zeta^{(n)}},$$

which proves the inequality. $\qquad\square$

**Lemma B.6** (Optimism for CE). *Consider an episode $n \in [N]$ and set $\alpha^{(n)} = \Theta\left(H\sqrt{nA\zeta^{(n)} + d\lambda}\right)$. When the event $\mathcal{E}$ holds, we have*

$$\bar{v}_i^{(n)}(s) - \max_{\omega\in\Omega_i} v_i^{\omega\circ\pi^{(n)}}(s) \geq -H\sqrt{A\zeta^{(n)}}, \quad \forall n\in[N], i\in[M].$$

*Proof.* Denote $\tilde{\omega}_{h,i}^{(n)} = \arg\max_{\omega_h\in\Omega_{h,i}}\left(\mathbb{D}_{\omega_h\circ\pi_h^{(n)}}\max_{\omega\in\Omega_i}Q_{h,i}^{\omega\circ\pi^{(n)}}\right)(s)$ and let $\tilde{\pi}_h^{(n)} = \tilde{\omega}_{h,i}\circ\pi_h^{(n)}$.
Let $f_h^{(n)}(s,\boldsymbol{a}) = \left\|\hat{P}_h^{(n)}(\cdot|s,\boldsymbol{a}) - P_h^\star(\cdot|s,\boldsymbol{a})\right\|_1$, then according to the event $\mathcal{E}$, we have

$$\mathbb{E}_{(s,\boldsymbol{a})\sim\rho_h^{(n)}}\left[\left(f_h^{(n)}(s,\boldsymbol{a})\right)^2\right] \leq \zeta^{(n)}, \quad \mathbb{E}_{(s,\boldsymbol{a})\sim\tilde{\rho}_h^{(n)}}\left[\left(f_h^{(n)}(s,\boldsymbol{a})\right)^2\right] \leq \zeta^{(n)}, \quad \forall n\in[N], h\in[H],$$

$$\|\phi_h(s,\boldsymbol{a})\|_{\left(\hat{\Sigma}_{h,\phi_h}^{(n)}\right)^{-1}} = \Theta\left(\|\phi_h(s,\boldsymbol{a})\|_{\Sigma_{n,\rho_h^{(n)},\phi_h}^{-1}}\right), \quad \forall n\in[N], h\in[H], \phi_h\in\Phi_h.$$

A direct conclusion of the event $\mathcal{E}$ is we can find an absolute constant $c$, such that

$$\beta_h^{(n)}(s,\boldsymbol{a}) = \min\left\{\alpha^{(n)}\left\|\hat{\phi}_h^{(n)}(\tilde{s},\tilde{\boldsymbol{a}})\right\|_{\left(\Sigma_{h,\hat{\phi}_h^{(n)}}^{(n)}\right)^{-1}}, H\right\}$$

$$\geq \min\left\{c\alpha^{(n)}\left\|\hat{\phi}_h^{(n)}(\tilde{s},\tilde{\boldsymbol{a}})\right\|_{\Sigma_{n,\rho_h^{(n)},\hat{\phi}_h^{(n)}}^{-1}}, H\right\}, \quad \forall n\in[N], h\in[H].$$

Next, we prove by induction that

$$\mathbb{E}_{s\sim d_{\hat{P}^{(n)},h}^{\tilde{\pi}^{(n)}}}\left[\overline{V}_{h,i}^{(n)}(s) - \max_{\omega\in\Omega_i}V_{h,i}^{\omega\circ\pi^{(n)}}(s)\right] \geq \sum_{h'=h}^{H}\mathbb{E}_{(s,\boldsymbol{a})\sim d_{\hat{P}^{(n)},h'}^{\tilde{\pi}^{(n)}}}\left[\hat{\beta}_{h'}^{(n)}(s,\boldsymbol{a}) - H\min\left\{f_{h'}^{(n)}(s,\boldsymbol{a}),1\right\}\right], \quad \forall h\in[H]. \tag{8}$$

First, notice that $\forall h\in[H]$,

$$\mathbb{E}_{s\sim d_{\hat{P}^{(n)},h}^{\tilde{\pi}^{(n)}}}\left[\overline{V}_{h,i}^{(n)}(s) - \max_{\omega\in\Omega_i}V_{h,i}^{\omega\circ\pi^{(n)}}(s)\right] = \mathbb{E}_{s\sim d_{\hat{P}^{(n)},h}^{\tilde{\pi}^{(n)}}}\left[\left(\mathbb{D}_{\pi_h^{(n)}}\overline{Q}_{h,i}^{(n)}\right)(s) - \left(\mathbb{D}_{\tilde{\pi}_h^{(n)}}\max_{\omega\in\Omega_i}Q_{h,i}^{\omega\circ\pi^{(n)}}\right)(s)\right]$$

$$\geq \mathbb{E}_{s\sim d_{\hat{P}^{(n)},h}^{\tilde{\pi}^{(n)}}}\left[\left(\mathbb{D}_{\tilde{\pi}_h^{(n)}}\overline{Q}_{h,i}^{(n)}\right)(s) - \left(\mathbb{D}_{\tilde{\pi}_h^{(n)}}\max_{\omega\in\Omega_i}Q_{h,i}^{\omega\circ\pi^{(n)}}\right)(s)\right]$$

$$= \mathbb{E}_{(s,\boldsymbol{a})\sim d_{\hat{P}^{(n)},h}^{\tilde{\pi}^{(n)}}}\left[\overline{Q}_{h,i}^{(n)}(s,\boldsymbol{a}) - \max_{\omega\in\Omega_i}Q_{h,i}^{\omega\circ\pi^{(n)}}(s,\boldsymbol{a})\right].$$

where the inequality uses the fact that $\pi_h^{(n)}$ is the CE solution for $\left\{\overline{Q}_{h,i}^{(n)}\right\}_{i=1}^{M}$. Now we are ready to prove equation 8:

- When $h = H$, we have

$$\mathbb{E}_{s\sim d_{\hat{P}^{(n)},H}^{\tilde{\pi}^{(n)}}}\left[\overline{V}_{H,i}^{(n)}(s) - \max_{\omega\in\Omega_i}V_{H,i}^{\omega\circ\pi^{(n)}}(s)\right] \geq \mathbb{E}_{(s,\boldsymbol{a})\sim d_{\hat{P}^{(n)},H}^{\tilde{\pi}^{(n)}}}\left[\overline{Q}_{H,i}^{(n)}(s,\boldsymbol{a}) - \max_{\omega\in\Omega_i}Q_{H,i}^{\omega\circ\pi^{(n)}}(s,\boldsymbol{a})\right]$$

$$= \mathbb{E}_{(s,\boldsymbol{a})\sim d_{\hat{P}^{(n)},H}^{\tilde{\pi}^{(n)}}}\left[\hat{\beta}_h^{(n)}(s,\boldsymbol{a})\right]$$

$$\geq \mathbb{E}_{(s,\boldsymbol{a})\sim d_{\hat{P}^{(n)},H}^{\tilde{\pi}^{(n)}}}\left[\hat{\beta}_h^{(n)}(s,\boldsymbol{a}) - H\min\left\{f_H^{(n)}(s,\boldsymbol{a}),1\right\}\right].$$

- Suppose the statement is true for $h + 1$, then for step $h$, we have

$$\mathbb{E}_{s \sim d_{\hat{P}^{(n)},h}^{\tilde{\pi}^{(n)}}} \left[ \overline{V}_{h,i}^{(n)}(s) - \max_{\omega \in \Omega_i} V_{h,i}^{\omega \circ \pi^{(n)}}(s) \right]$$

$$\geq \mathbb{E}_{(s,\boldsymbol{a}) \sim d_{\hat{P}^{(n)},h}^{\tilde{\pi}^{(n)}}} \left[ \overline{Q}_{h,i}^{(n)}(s,\boldsymbol{a}) - \max_{\omega \in \Omega_i} Q_{h,i}^{\omega \circ \pi^{(n)}}(s,\boldsymbol{a}) \right]$$

$$= \mathbb{E}_{(s,\boldsymbol{a}) \sim d_{\hat{P}^{(n)},h}^{\tilde{\pi}^{(n)}}} \left[ \hat{\beta}_h^{(n)}(s,\boldsymbol{a}) + \left( \hat{P}_h^{(n)} \overline{V}_{h+1,i}^{(n)} \right)(s,\boldsymbol{a}) - \left( P_h^{\star} \max_{\omega \in \Omega_i} V_{h+1,i}^{\omega \circ \pi^{(n)}} \right)(s,\boldsymbol{a}) \right]$$

$$= \mathbb{E}_{(s,\boldsymbol{a}) \sim d_{\hat{P}^{(n)},h}^{\tilde{\pi}^{(n)}}} \left[ \hat{\beta}_h^{(n)}(s,\boldsymbol{a}) + \left( \hat{P}_h^{(n)} \left( \overline{V}_{h+1,i}^{(n)} - \max_{\omega \in \Omega_i} V_{h+1,i}^{\omega \circ \pi^{(n)}} \right) \right)(s,\boldsymbol{a}) + \left( \left( \hat{P}_h^{(n)} - P_h^{\star} \right) \max_{\omega \in \Omega_i} V_{h+1,i}^{\omega \circ \pi^{(n)}} \right)(s,\boldsymbol{a}) \right]$$

$$= \mathbb{E}_{(s,\boldsymbol{a}) \sim d_{\hat{P}^{(n)},h}^{\tilde{\pi}^{(n)}}} \left[ \hat{\beta}_h^{(n)}(s,\boldsymbol{a}) + \left( \left( \hat{P}_h^{(n)} - P_h^{\star} \right) \max_{\omega \in \Omega_i} V_{h+1,i}^{\omega \circ \pi^{(n)}} \right)(s,\boldsymbol{a}) \right]$$

$$+ \mathbb{E}_{s \sim d_{\hat{P}^{(n)},h+1}^{\tilde{\pi}^{(n)}}} \left[ \overline{V}_{h+1,i}^{(n)}(s) - \max_{\omega \in \Omega_i} V_{h+1,i}^{\omega \circ \pi^{(n)}}(s) \right]$$

$$\geq \mathbb{E}_{(s,\boldsymbol{a}) \sim d_{\hat{P}^{(n)},h}^{\tilde{\pi}^{(n)}}} \left[ \hat{\beta}_h^{(n)}(s,\boldsymbol{a}) - H \min \left\{ f_h^{(n)}(s,\boldsymbol{a}), 1 \right\} \right] + \mathbb{E}_{s \sim d_{\hat{P}^{(n)},h+1}^{\tilde{\pi}^{(n)}}} \left[ \overline{V}_{h+1,i}^{(n)}(s) - \max_{\omega \in \Omega_i} V_{h+1,i}^{\omega \circ \pi^{(n)}}(s) \right]$$

$$\geq \sum_{h'=h}^{H} \mathbb{E}_{(s,\boldsymbol{a}) \sim d_{\hat{P}^{(n)},h'}^{\tilde{\pi}^{(n)}}} \left[ \hat{\beta}_{h'}^{(n)}(s,\boldsymbol{a}) - H \min \left\{ f_{h'}^{(n)}(s,\boldsymbol{a}), 1 \right\} \right],$$

where we use the fact

$$\left| \left( \hat{P}_h^{(n)} - P_h^{\star} \right) \max_{\omega \in \Omega_i} V_{h+1,i}^{\omega \circ \pi^{(n)}} \right| (s,\boldsymbol{a}) \leq \min \left\{ H, \left\| \hat{P}_h^{(n)}(\cdot|s,\boldsymbol{a}) - P_h^{\star}(\cdot|s,\boldsymbol{a}) \right\|_1 \left\| \max_{\omega \in \Omega_i} V_{h+1,i}^{\omega \circ \pi^{(n)}} \right\|_{\infty} \right\}$$

$$\leq H \min \left\{ 1, \left\| \hat{P}_h^{(n)}(\cdot|s,\boldsymbol{a}) - P_h^{\star}(\cdot|s,\boldsymbol{a}) \right\|_1 \right\}$$

$$= H \min \left\{ 1, f_{h'}^{(n)}(s,\boldsymbol{a}) \right\}$$

and the last row uses the induction assumption.

Therefore, we have proved equation 8. We then apply $h = 1$ to equation 8, and get

$$\mathbb{E}_{s \sim d_1} \left[ \overline{V}_{1,i}^{(n)}(s) - \max_{\omega \in \Omega_i} V_{1,i}^{\omega \circ \pi^{(n)}}(s) \right] = \mathbb{E}_{s \sim d_{\hat{P}^{(n)},1}^{\tilde{\pi}^{(n)}}} \left[ \overline{V}_{1,i}^{(n)}(s) - \max_{\omega \in \Omega_i} V_{1,i}^{\omega \circ \pi^{(n)}}(s) \right]$$

$$\geq \sum_{h=1}^{H} \mathbb{E}_{(s,\boldsymbol{a}) \sim d_{\hat{P}^{(n)},h}^{\tilde{\pi}^{(n)}}} \left[ \hat{\beta}_h^{(n)}(s,\boldsymbol{a}) - H \min \left\{ f_h^{(n)}(s,\boldsymbol{a}), 1 \right\} \right]$$

$$= \sum_{h=1}^{H} \mathbb{E}_{(s,\boldsymbol{a}) \sim d_{\hat{P}^{(n)},h}^{\tilde{\pi}^{(n)}}} \left[ \hat{\beta}_h^{(n)}(s,\boldsymbol{a}) \right] - H \sum_{h=1}^{H} \mathbb{E}_{(s,\boldsymbol{a}) \sim d_{\hat{P}^{(n)},h}^{\tilde{\pi}^{(n)}}} \left[ \min \left\{ f_h^{(n)}(s,\boldsymbol{a}), 1 \right\} \right].$$

Next we are going to bound the second term, let $g_h(s,\boldsymbol{a}) = \min\{ f_h^{(n)}(s,\boldsymbol{a}), 1 \}$ and apply Lemma B.3 to $g_h$, we have for $h = 1$,

$$\mathbb{E}_{(s,\boldsymbol{a}) \sim d_{\hat{P}^{(n)},1}^{\tilde{\pi}^{(n)}}} \left[ \min \left\{ f_1^{(n)}(s,\boldsymbol{a}), 1 \right\} \right] \leq \sqrt{A \mathbb{E}_{(s,\boldsymbol{a}) \sim \rho_1^{(n)}} \left[ \left( f_1^{(n)}(s,\boldsymbol{a}) \right)^2 \right]} \leq \sqrt{A \zeta^{(n)}}.$$

And $\forall h \geq 2$, we have

$$\mathbb{E}_{(s,\boldsymbol{a}) \sim d_{\hat{P}^{(n)},h}^{\tilde{\pi}^{(n)}}} \left[ \min \left\{ f_h^{(n)}(s,\boldsymbol{a}), 1 \right\} \right]$$

$$\leq \mathbb{E}_{(\tilde{s},\tilde{\boldsymbol{a}}) \sim d_{\hat{P}^{(n)},h-1}^{\tilde{\pi}^{(n)}}} \left[ \min \left\{ \left\| \hat{\phi}_{h-1}^{(n)}(\tilde{s},\tilde{\boldsymbol{a}}) \right\|_{\Sigma_{n,\rho_{h-1}^{(n)},\hat{\phi}_{h-1}^{(n)}}^{-1}} \sqrt{n A \mathbb{E}_{(s,\boldsymbol{a}) \sim \tilde{\rho}_h^{(n)}} \left[ \left( f_h^{(n)}(s,\boldsymbol{a}) \right)^2 \right] + d\lambda + n\zeta^{(n)}}, 1 \right\} \right]$$

$$\lesssim \mathbb{E}_{(\tilde{s},\tilde{\boldsymbol{a}})\sim d_{\hat{P}^{(n)},h-1}^{\tilde{\pi}^{(n)}}} \left[ \min\left\{ \left\| \hat{\phi}_{h-1}^{(n)}(\tilde{s},\tilde{\boldsymbol{a}}) \right\|_{\Sigma_{n,\rho_{h-1}^{(n)},\hat{\phi}_{h-1}^{(n)}}^{-1}} \sqrt{nA\zeta^{(n)}+d\lambda+n\zeta^{(n)}},1 \right\} \right].$$

Note that we here use the fact $\min\{f_h^{(n)}(s,\boldsymbol{a}),1\} \le 1$, $\mathbb{E}_{(s,\boldsymbol{a})\sim\rho_h^{(n)}}\left[\left(f_h^{(n)}(s,\boldsymbol{a})\right)^2\right] \le \zeta^{(n)}$ and

$\mathbb{E}_{(s,\boldsymbol{a})\sim\tilde{\rho}_h^{(n)}}\left[\left(f_h^{(n)}(s,\boldsymbol{a})\right)^2\right] \le \zeta^{(n)}$. Then according to our choice of $\alpha^{(n)}$, we get

$$\mathbb{E}_{(s,\boldsymbol{a})\sim d_{\hat{P}^{(n)},h}^{\tilde{\pi}^{(n)}}}\left[f_h^{(n)}(s,\boldsymbol{a})\right] \le \mathbb{E}_{(\tilde{s},\tilde{\boldsymbol{a}})\sim d_{\hat{P}^{(n)},h-1}^{\tilde{\pi}^{(n)}}}\left[\min\left\{\frac{c\alpha^{(n)}}{H}\left\|\hat{\phi}_{h-1}^{(n)}(\tilde{s},\tilde{\boldsymbol{a}})\right\|_{\Sigma_{n,\rho_{h-1}^{(n)},\hat{\phi}_{h-1}^{(n)}}^{-1}},1\right\}\right].$$

Combining all things together,

$$\overline{v}_i^{(n)} - \max_{\omega\in\Omega_i} v_i^{\omega\circ\pi^{(n)}} = \mathbb{E}_{s\sim d_1}\left[\overline{V}_{1,i}^{(n)}(s) - \max_{\omega\in\Omega_i} V_{1,i}^{\omega\circ\pi^{(n)}}(s)\right]$$

$$\ge \sum_{h=1}^{H}\mathbb{E}_{(s,\boldsymbol{a})\sim d_{\hat{P}^{(n)},h}^{\tilde{\pi}^{(n)}}}\left[\hat{\beta}_h^{(n)}(s,\boldsymbol{a})\right] - H\sum_{h=1}^{H}\mathbb{E}_{(s,\boldsymbol{a})\sim d_{\hat{P}^{(n)},h}^{\tilde{\pi}^{(n)}}}\left[\min\left\{f_h^{(n)}(s,\boldsymbol{a}),1\right\}\right]$$

$$\ge \sum_{h=1}^{H-1}\mathbb{E}_{(\tilde{s},\tilde{\boldsymbol{a}})\sim d_{\hat{P}^{(n)},h}^{\tilde{\pi}^{(n)}}}\left[\hat{\beta}_h^{(n)}(s,\boldsymbol{a}) - \min\left\{c\alpha^{(n)}\left\|\hat{\phi}_h^{(n)}(\tilde{s},\tilde{\boldsymbol{a}})\right\|_{\Sigma_{n,\rho_h^{(n)},\hat{\phi}_h^{(n)}}^{-1}},H\right\}\right] - H\sqrt{A\zeta^{(n)}}$$

$$\ge -H\sqrt{A\zeta^{(n)}},$$

which proves the inequality. $\qquad\square$

**Lemma B.7** (Pessimism). *Consider an episode $n\in[N]$ and set $\alpha^{(n)} = \Theta\left(H\sqrt{nA\zeta^{(n)}+d\lambda}\right)$. When the event $\mathcal{E}$ holds, we have*

$$\underline{v}_i^{(n)}(s) - v_i^{\pi^{(n)}}(s) \le H\sqrt{A\zeta^{(n)}}, \quad \forall n\in[N], i\in[M].$$

*Proof.* Let $f_h^{(n)}(s,\boldsymbol{a}) = \left\|\hat{P}_h^{(n)}(\cdot|s,\boldsymbol{a}) - P_h^{\star}(\cdot|s,\boldsymbol{a})\right\|_1$, then according to the event $\mathcal{E}$, we have

$$\mathbb{E}_{(s,\boldsymbol{a})\sim\rho_h^{(n)}}\left[\left(f_h^{(n)}(s,\boldsymbol{a})\right)^2\right] \le \zeta^{(n)}, \quad \mathbb{E}_{(s,\boldsymbol{a})\sim\tilde{\rho}_h^{(n)}}\left[\left(f_h^{(n)}(s,\boldsymbol{a})\right)^2\right] \le \zeta^{(n)}, \quad \forall n\in[N], h\in[H],$$

$$\|\phi_h(s,\boldsymbol{a})\|_{\left(\hat{\Sigma}_{h,\phi_h}^{(n)}\right)^{-1}} = \Theta\left(\|\phi_h(s,\boldsymbol{a})\|_{\Sigma_{n,\rho_h^{(n)},\phi_h}^{-1}}\right), \quad \forall n\in[N], h\in[H], \phi_h\in\Phi_h.$$

A direct conclusion of the event $\mathcal{E}$ is we can find an absolute constant $c$, such that

$$\beta_h^{(n)}(s,\boldsymbol{a}) = \min\left\{\alpha^{(n)}\left\|\hat{\phi}_h^{(n)}(\tilde{s},\tilde{\boldsymbol{a}})\right\|_{\left(\Sigma_{h,\hat{\phi}_h^{(n)}}^{(n)}\right)^{-1}},H\right\}$$

$$\ge \min\left\{c\alpha^{(n)}\left\|\hat{\phi}_h^{(n)}(\tilde{s},\tilde{\boldsymbol{a}})\right\|_{\Sigma_{n,\rho_h^{(n)},\hat{\phi}_h^{(n)}}^{-1}},H\right\}, \quad \forall n\in[N], h\in[H].$$

Again, we prove the following inequality by induction:

$$\mathbb{E}_{s\sim d_{\hat{P}^{(n)},h}^{\pi^{(n)}}}\left[\underline{V}_{h,i}^{(n)}(s) - V_{h,i}^{\pi^{(n)}}(s)\right] \le \sum_{h'=h}^{H}\mathbb{E}_{(s,\boldsymbol{a})\sim d_{\hat{P}^{(n)},h'}^{\pi^{(n)}}}\left[-\hat{\beta}_{h'}^{(n)}(s,\boldsymbol{a}) + H\min\left\{f_{h'}^{(n)}(s,\boldsymbol{a}),1\right\}\right], \quad \forall h\in[H].$$

$$\tag{9}$$

- When $h = H$, we have

$$
\begin{aligned}
\mathbb{E}_{s \sim d^{\pi^{(n)}}_{\hat{P}^{(n)}, H}} \left[ \underline{V}^{(n)}_{H,i}(s) - V^{\pi^{(n)}}_{H,i}(s) \right] &= \mathbb{E}_{(s,\boldsymbol{a}) \sim d^{\pi^{(n)}}_{\hat{P}^{(n)}, H}} \left[ \underline{Q}^{(n)}_{H,i}(s,\boldsymbol{a}) - Q^{\pi^{(n)}}_{H,i}(s,\boldsymbol{a}) \right] \\
&= \mathbb{E}_{(s,\boldsymbol{a}) \sim d^{\pi^{(n)}}_{\hat{P}^{(n)}, H}} \left[ -\hat{\beta}^{(n)}_H(s,\boldsymbol{a}) \right] \\
&\leq \mathbb{E}_{(s,\boldsymbol{a}) \sim d^{\pi^{(n)}}_{\hat{P}^{(n)}, H}} \left[ -\hat{\beta}^{(n)}_H(s,\boldsymbol{a}) + H \min \left\{ f^{(n)}_H(s,\boldsymbol{a}), 1 \right\} \right]
\end{aligned}
$$

- Suppose the statement is true for $h+1$, then for step $h$, we have

$$
\begin{aligned}
&\mathbb{E}_{s \sim d^{\pi^{(n)}}_{\hat{P}^{(n)}, h}} \left[ \underline{V}^{(n)}_{h,i}(s) - V^{\pi^{(n)}}_{h,i}(s) \right] \\
&= \mathbb{E}_{(s,\boldsymbol{a}) \sim d^{\pi^{(n)}}_{\hat{P}^{(n)}, h}} \left[ \underline{Q}^{(n)}_{h,i}(s,\boldsymbol{a}) - Q^{\pi^{(n)}}_{h,i}(s,\boldsymbol{a}) \right] \\
&= \mathbb{E}_{(s,\boldsymbol{a}) \sim d^{\pi^{(n)}}_{\hat{P}^{(n)}, h}} \left[ -\hat{\beta}^{(n)}_h(s,\boldsymbol{a}) + \left( \hat{P}^{(n)}_h \underline{V}^{(n)}_{h+1,i} \right)(s,\boldsymbol{a}) - \left( P^\star_h V^{\pi^{(n)}}_{h+1,i} \right)(s,\boldsymbol{a}) \right] \\
&= \mathbb{E}_{(s,\boldsymbol{a}) \sim d^{\pi^{(n)}}_{\hat{P}^{(n)}, h}} \left[ -\hat{\beta}^{(n)}_h(s,\boldsymbol{a}) + \left( \hat{P}^{(n)}_h \left( \underline{V}^{(n)}_{h+1,i} - V^{\pi^{(n)}}_{h+1,i} \right) \right)(s,\boldsymbol{a}) + \left( \left( \hat{P}^{(n)}_h - P^\star_h \right) V^{\pi^{(n)}}_{h+1,i} \right)(s,\boldsymbol{a}) \right] \\
&= \mathbb{E}_{(s,\boldsymbol{a}) \sim d^{\pi^{(n)}}_{\hat{P}^{(n)}, h}} \left[ -\hat{\beta}^{(n)}_h(s,\boldsymbol{a}) + \left( \left( \hat{P}^{(n)}_h - P^\star_h \right) V^{\pi^{(n)}}_{h+1,i} \right)(s,\boldsymbol{a}) \right] + \mathbb{E}_{s \sim d^{\pi^{(n)}}_{\hat{P}^{(n)}, h+1}} \left[ \left( \underline{V}^{(n)}_{h+1,i} - V^{\pi^{(n)}}_{h+1,i} \right)(s) \right] \\
&\leq \mathbb{E}_{(s,\boldsymbol{a}) \sim d^{\pi^{(n)}}_{\hat{P}^{(n)}, h}} \left[ -\hat{\beta}^{(n)}_h(s,\boldsymbol{a}) + H \min \left\{ f^{(n)}_h(s,\boldsymbol{a}), 1 \right\} \right] + \mathbb{E}_{s \sim d^{\pi^{(n)}}_{\hat{P}^{(n)}, h+1}} \left[ \left( \underline{V}^{(n)}_{h+1,i} - V^{\pi^{(n)}}_{h+1,i} \right)(s) \right] \\
&\leq \sum_{h'=h}^{H} \mathbb{E}_{(s,\boldsymbol{a}) \sim d^{\pi^{(n)}}_{\hat{P}^{(n)}, h'}} \left[ -\hat{\beta}^{(n)}_{h'}(s,\boldsymbol{a}) + H \min \left\{ f^{(n)}_{h'}(s,\boldsymbol{a}), 1 \right\} \right].
\end{aligned}
$$

where we use the fact

$$
\begin{aligned}
\left| \left( \hat{P}^{(n)}_h - P^\star_h \right) V^{\pi^{(n)}}_{h+1,i} \right| (s,\boldsymbol{a}) &\leq \min \left\{ H, \left\| \hat{P}^{(n)}_h(\cdot|s,\boldsymbol{a}) - P^\star_h(\cdot|s,\boldsymbol{a}) \right\|_1 \left\| V^{\pi^{(n)}}_{h+1,i} \right\|_\infty \right\} \\
&\leq H \min \left\{ 1, \left\| \hat{P}^{(n)}_h(\cdot|s,\boldsymbol{a}) - P^\star_h(\cdot|s,\boldsymbol{a}) \right\|_1 \right\} \\
&= H \min \left\{ 1, f^{(n)}_{h'}(s,\boldsymbol{a}) \right\}
\end{aligned}
$$

and the last row uses the induction assumption.

The remaining steps are exactly the same as the proof in Lemma B.5 or Lemma B.6, we may prove

$$
\mathbb{E}_{(s,\boldsymbol{a}) \sim d^{\pi^{(n)}}_{\hat{P}^{(n)}, 1}} \left[ \min \left\{ f^{(n)}_1(s,\boldsymbol{a}), 1 \right\} \right] \leq \sqrt{A \zeta^{(n)}},
$$

and

$$
\mathbb{E}_{(s,\boldsymbol{a}) \sim d^{\pi^{(n)}}_{\hat{P}^{(n)}, h}} \left[ f^{(n)}_h(s,\boldsymbol{a}) \right] \leq \mathbb{E}_{(\tilde{s},\tilde{\boldsymbol{a}}) \sim d^{\pi^{(n)}}_{\hat{P}^{(n)}, h-1}} \left[ \min \left\{ \frac{c \alpha^{(n)}}{H} \left\| \hat{\phi}^{(n)}_{h-1}(\tilde{s},\tilde{\boldsymbol{a}}) \right\|_{\Sigma^{-1}_{n, \rho^{(n)}_{h-1}, \hat{\phi}^{(n)}_{h-1}}}, 1 \right\} \right], \quad \forall h \geq 2.
$$

Combining all things together, we get

$$
\begin{aligned}
\underline{v}^{(n)}_i - v^{\pi^{(n)}}_i &= \mathbb{E}_{s \sim d_1} \left[ \underline{V}^{(n)}_{1,i}(s) - V^{\pi^{(n)}}_{1,i}(s) \right] \\
&\leq \sum_{h=1}^{H} \mathbb{E}_{(s,\boldsymbol{a}) \sim d^{\pi^{(n)}}_{\hat{P}^{(n)}, h}} \left[ -\hat{\beta}^{(n)}_h(s,\boldsymbol{a}) + H \min \left\{ f^{(n)}_h(s,\boldsymbol{a}), 1 \right\} \right] \\
&\leq \sum_{h=1}^{H-1} \mathbb{E}_{(s,\boldsymbol{a}) \sim d^{\pi^{(n)}}_{\hat{P}^{(n)}, h}} \left[ -\hat{\beta}^{(n)}_h(s,\boldsymbol{a}) + \min \left\{ c \alpha^{(n)} \left\| \hat{\phi}^{(n)}_h(\tilde{s},\tilde{\boldsymbol{a}}) \right\|_{\Sigma^{-1}_{n, \rho^{(n)}_h, \hat{\phi}^{(n)}_h}}, H \right\} \right] + H \sqrt{A \zeta^{(n)}} \\
&\leq H \sqrt{A \zeta^{(n)}},
\end{aligned}
$$

which has finished the proof. $\qquad\square$

**Lemma B.8.** *For the model-based algorithm, when we pick* $\lambda = \Theta\left(d \log \frac{NH|\Phi|}{\delta}\right)$, $\zeta^{(n)} = \Theta\left(\frac{1}{n} \log \frac{|\mathcal{M}|HN}{\delta}\right)$ *and* $\alpha^{(n)} = \Theta\left(H\sqrt{nA\zeta^{(n)} + d\lambda}\right)$, *with probability* $1 - \delta$, *we have*

$$\sum_{n=1}^{N} \Delta^{(n)} \lesssim H^3 d^2 A N^{\frac{1}{2}} \log \frac{|\mathcal{M}|HN}{\delta}.$$

*Proof.* With our choice of $\lambda$ and $\zeta^{(n)}$, according to Lemma B.2, we know $\mathcal{E}$ holds with probability $1 - \delta$. Furthermore, we have

$$\alpha^{(n)} = \Theta\left(H\sqrt{A \log \frac{|\mathcal{M}|HN}{\delta} + d^2 \log \frac{NH|\Phi|}{\delta}}\right) = O\left(dH\sqrt{A \log \frac{|\mathcal{M}|HN}{\delta}}\right)$$

Let $f_h^{(n)}(s, \boldsymbol{a}) = \left\|\hat{P}_h^{(n)}(\cdot|s, \boldsymbol{a}) - P_h^\star(\cdot|s, \boldsymbol{a})\right\|_1$. According to the definition of the event $\mathcal{E}$, we have

$$\mathbb{E}_{s\sim\rho_h^{(n)}}\left[\left(f_h^{(n)}(s, \boldsymbol{a})\right)^2\right] \leq \zeta^{(n)}, \|\phi_h(s, \boldsymbol{a})\|_{\left(\hat{\Sigma}_{h,\phi_h}^{(n)}\right)^{-1}} = \Theta\left(\|\phi_h(s, \boldsymbol{a})\|_{\Sigma_{n,\rho_h^{(n)},\phi_h}^{-1}}\right), \quad \forall n \in [N], h \in [H], \phi_h \in \Phi_h.$$

(10)

By definition, we have

$$\Delta^{(n)} = \max_{i\in[M]} \left\{\overline{v}_i^{(n)} - \underline{v}_i^{(n)}\right\} + 2H\sqrt{A\zeta^{(n)}}.$$

For each fixed $i \in [M], h \in [H]$ and $n \in [N]$, we have

$$\mathbb{E}_{s\sim d_{P^\star,h}^{\pi^{(n)}}}\left[\overline{V}_{h,i}^{(n)}(s) - \underline{V}_{h,i}^{(n)}(s)\right]$$
$$=\mathbb{E}_{s\sim d_{P^\star,h}^{\pi^{(n)}}}\left[\left(\mathbb{D}_{\pi_h^{(n)}}\overline{Q}_{h,i}^{(n)}\right)(s) - \left(\mathbb{D}_{\pi_h^{(n)}}\underline{Q}_{h,i}^{(n)}\right)(s)\right]$$
$$=\mathbb{E}_{(s,\boldsymbol{a})\sim d_{P^\star,h}^{\pi^{(n)}}}\left[\overline{Q}_{h,i}^{(n)}(s, \boldsymbol{a}) - \underline{Q}_{h,i}^{(n)}(s, \boldsymbol{a})\right]$$
$$=\mathbb{E}_{(s,\boldsymbol{a})\sim d_{P^\star,h}^{\pi^{(n)}}}\left[2\hat{\beta}_h^{(n)}(s, \boldsymbol{a}) + \left(\hat{P}_h^{(n)}\left(\overline{V}_{h+1,i}^{(n)} - \underline{V}_{h+1,i}^{(n)}\right)\right)(s, \boldsymbol{a})\right]$$
$$=\mathbb{E}_{(s,\boldsymbol{a})\sim d_{P^\star,h}^{\pi^{(n)}}}\left[2\hat{\beta}_h^{(n)}(s, \boldsymbol{a}) + \left(\left(\hat{P}_h^{(n)} - P_h^\star\right)\left(\overline{V}_{h+1,i}^{(n)} - \underline{V}_{h+1,i}^{(n)}\right)\right)(s, \boldsymbol{a})\right] + \mathbb{E}_{s\sim d_{P^\star,h+1}^{\pi^{(n)}}}\left[\overline{V}_{h+1,i}^{(n)}(s) - \underline{V}_{h+1,i}^{(n)}(s)\right]$$
$$\leq\mathbb{E}_{(s,\boldsymbol{a})\sim d_{P^\star,h}^{\pi^{(n)}}}\left[2\hat{\beta}_h^{(n)}(s, \boldsymbol{a}) + 2H^2 f_h^{(n)}(s, \boldsymbol{a})\right] + \mathbb{E}_{s\sim d_{P^\star,h+1}^{\pi^{(n)}}}\left[\overline{V}_{h+1,i}^{(n)}(s) - \underline{V}_{h+1,i}^{(n)}(s)\right].$$

Note that we use the fact $\overline{V}_{h+1,i}^{(n)}(s) - \underline{V}_{h+1,i}^{(n)}(s)$ is upper bounded by $2H^2$, which can be proved easily using induction using the fact that $\hat{\beta}_h^{(n)}(s, \boldsymbol{a}) \leq H$. Applying the above formula recursively to $\mathbb{E}_{s\sim d_{P^\star,h+1}^{\pi^{(n)}}}\left[\overline{V}_{h+1,i}^{(n)}(s) - \underline{V}_{h+1,i}^{(n)}(s)\right]$, one gets the following result (or more formally, one can prove by induction, just like what we did in Lemma B.5, Lemma B.6 and Lemma B.7):

$$\mathbb{E}_{s\sim d_{P^\star,1}^{\pi^{(n)}}}\left[\overline{V}_{1,i}^{(n)}(s) - \underline{V}_{1,i}^{(n)}(s)\right] \leq 2\underbrace{\sum_{h=1}^{H} \mathbb{E}_{(s,\boldsymbol{a})\sim d_{P^\star,h}^{\pi^{(n)}}}\left[\hat{\beta}_h^{(n)}(s, \boldsymbol{a})\right]}_{(a)} + 2H^2\underbrace{\sum_{h=1}^{H} \mathbb{E}_{(s,\boldsymbol{a})\sim d_{P^\star,h}^{\pi^{(n)}}}\left[f_h^{(n)}(s, \boldsymbol{a})\right]}_{(b)}.$$

(11)

First, we calculate the first term (a) in Inequality equation 11. Following Lemma B.4 and noting the bonus $\hat{\beta}_h^{(n)}$ is $O(H)$, we have

$$\sum_{h=1}^{H} \mathbb{E}_{(s,\boldsymbol{a})\sim d_{P^\star,h}^{\pi^{(n)}}}\left[\hat{\beta}_h^{(n)}(s, \boldsymbol{a})\right]$$

$$\lesssim \sum_{h=1}^{H} \mathbb{E}_{(s,\boldsymbol{a})\sim d_{P^\star,h}^{\pi^{(n)}}} \left[ \min\left\{ \alpha^{(n)} \left\| \hat{\phi}_h^{(n)}(s,\boldsymbol{a}) \right\|_{\Sigma_{n,\rho_h^{(n)},\hat{\phi}_h^{(n)}}^{-1}}, H \right\} \right] \qquad \text{(From equation 10)}$$

$$\lesssim \sum_{h=1}^{H-1} \mathbb{E}_{(\tilde{s},\tilde{\boldsymbol{a}})\sim d_{P^\star,h}^{\pi^{(n)}}} \left[ \|\phi_h^\star(\tilde{s},\tilde{\boldsymbol{a}})\|_{\Sigma_{n,\gamma_h^{(n)},\phi_h^\star}^{-1}} \right] \sqrt{ nA\left(\alpha^{(n)}\right)^2 \mathbb{E}_{(s,\boldsymbol{a})\sim\rho_h^{(n)}} \left[ \left\| \hat{\phi}_h^{(n)}(s,\boldsymbol{a}) \right\|_{\Sigma_{n,\rho_h^{(n)},\hat{\phi}_h^{(n)}}^{-1}}^2 \right] + H^2 d\lambda }$$

$$+ \sqrt{ A\left(\alpha^{(n)}\right)^2 \mathbb{E}_{(s,\boldsymbol{a})\sim\rho_1^{(n)}} \left[ \left\| \hat{\phi}_1^{(n)}(s,\boldsymbol{a}) \right\|_{\Sigma_{n,\rho_1^{(n)},\hat{\phi}_1^{(n)}}^{-1}}^2 \right] }.$$

Note that we use the fact that $B = H$ when applying Lemma B.4. In addition, we have

$$n\mathbb{E}_{(s,\boldsymbol{a})\sim\rho_h^{(n)}} \left[ \left\| \hat{\phi}_h^{(n)}(s,\boldsymbol{a}) \right\|_{\Sigma_{n,\rho_h^{(n)},\hat{\phi}_h^{(n)}}^{-1}}^2 \right]$$

$$= n\text{Tr}\left( \mathbb{E}_{(s,\boldsymbol{a})\sim\rho_h^{(n)}} \left[ \hat{\phi}_h^{(n)}(s,\boldsymbol{a})\hat{\phi}_h^{(n)}(s,\boldsymbol{a})^\top \right] \left( n\mathbb{E}_{(s,\boldsymbol{a})\sim\rho_h^{(n)}} \left[ \hat{\phi}_h^{(n)}(s,\boldsymbol{a})\hat{\phi}_h^{(n)}(s,\boldsymbol{a})^\top \right] + \lambda I_d \right)^{-1} \right)$$

$$\leq d.$$

Then,

$$\sum_{h=1}^{H} \mathbb{E}_{(s,\boldsymbol{a})\sim d_{P^\star,h}^{\pi^{(n)}}} \left[ \hat{\beta}_h^{(n)}(s,\boldsymbol{a}) \right] \leq \mathbb{E}_{(\tilde{s},\tilde{\boldsymbol{a}})\sim d_{P^\star,h}^{\pi^{(n)}}} \left[ \|\phi_h^\star(\tilde{s},\tilde{\boldsymbol{a}})\|_{\Sigma_{n,\gamma_h^{(n)},\phi_h^\star}^{-1}} \right] \sqrt{ dA\left(\alpha^{(n)}\right)^2 + H^2 d\lambda } + \sqrt{ dA\left(\alpha^{(n)}\right)^2/n }.$$

Second, we calculate the term (b) in inequality equation 11. Following Lemma B.4 and noting that $f_h^{(n)}(s,\boldsymbol{a}$ is upper-bounded by 2 (i.e., $B = 2$ in Lemma B.4), we have

$$\sum_{h=1}^{H} \mathbb{E}_{(s,\boldsymbol{a})\sim d_{P^\star,h}^{\pi^{(n)}}} [f_h^{(n)}(s,\boldsymbol{a})]$$

$$\leq \sum_{h=1}^{H-1} \mathbb{E}_{(\tilde{s},\tilde{\boldsymbol{a}})\sim d_{P^\star,h}^{\pi^{(n)}}} \left[ \|\phi_h^\star(\tilde{s},\tilde{\boldsymbol{a}})\|_{\Sigma_{n,\gamma_h^{(n)},\phi_h^\star}^{-1}} \right] \sqrt{ nA\mathbb{E}_{(s,\boldsymbol{a})\sim\rho_h^{(n)}} \left[ \left(f_h^{(n)}(s,\boldsymbol{a})\right)^2 \right] + d\lambda } + \sqrt{ A\mathbb{E}_{(s,\boldsymbol{a})\sim\rho_h^{(n)}} \left[ \left(f_1^{(n)}(s,\boldsymbol{a})\right)^2 \right] }$$

$$\leq \sum_{h=1}^{H-1} \mathbb{E}_{(\tilde{s},\tilde{\boldsymbol{a}})\sim d_{P^\star,h}^{\pi^{(n)}}} \left[ \|\phi_h^\star(\tilde{s},\tilde{\boldsymbol{a}})\|_{\Sigma_{n,\gamma_h^{(n)},\phi_h^\star}^{-1}} \right] \sqrt{ nA\zeta^{(n)} + d\lambda } + \sqrt{ A\zeta^{(n)} }$$

$$\lesssim \frac{\alpha^{(n)}}{H} \sum_{h=1}^{H-1} \mathbb{E}_{(\tilde{s},\tilde{\boldsymbol{a}})\sim d_{P^\star,h}^{\pi^n}} \left[ \|\phi_h^\star(\tilde{s},\tilde{\boldsymbol{a}})\|_{\Sigma_{n,\gamma_h^{(n)},\phi_h^\star}^{-1}} \right] + \sqrt{ A\zeta^{(n)} },$$

where in the second inequality, we use $\mathbb{E}_{(s,\boldsymbol{a})\sim\rho_h^{(n)}} \left[ \left(f_h^{(n)}(s,\boldsymbol{a})\right)^2 \right] \leq \zeta^{(n)}$, and in the last line, recall $\sqrt{nA\zeta^{(n)} + d\lambda} \lesssim \alpha^{(n)}/H$. Then, by combining the above calculation of the term (a) and term (b) in inequality equation 11, we have:

$$\overline{v}_i^{(n)} - \underline{v}_i^{(n)} = \mathbb{E}_{s\sim d_{P^\star,1}^{\pi^{(n)}}} \left[ \overline{V}_{1,i}^{(n)}(s) - \underline{V}_{1,i}^{(n)}(s) \right]$$

$$\lesssim \sum_{h=1}^{H-1} \left( \mathbb{E}_{(\tilde{s},\tilde{\boldsymbol{a}})\sim d_{P^\star,h}^{\pi^{(n)}}} \left[ \|\phi_h^\star(\tilde{s},\tilde{\boldsymbol{a}})\|_{\Sigma_{n,\gamma_h^{(n)},\phi_h^\star}^{-1}} \right] \sqrt{ dA\left(\alpha^{(n)}\right)^2 + H^2 d\lambda } + \sqrt{ \frac{dA\left(\alpha^{(n)}\right)^2}{n} } \right)$$

$$+ H^2 \sum_{h=1}^{H-1} \left( \frac{\alpha^{(n)}}{H} \mathbb{E}_{(\tilde{s},\tilde{\boldsymbol{a}})\sim d_{P^\star,h}^{\pi^{(n)}}} \left[ \|\phi_h^\star(\tilde{s},\tilde{\boldsymbol{a}})\|_{\Sigma_{n,\gamma_h^{(n)},\phi_h^\star}^{-1}} \right] + \sqrt{ A\zeta^{(n)} } \right).$$

Taking maximum over $i$ on both sides and using the definition of $\Delta^{(n)}$, we get

$$\Delta^{(n)} = \max_{i\in[M]} \left\{ \overline{v}_i^{(n)} - \underline{v}_i^{(n)} \right\} + 2H\sqrt{A\zeta^{(n)}}$$

$$\lesssim \sum_{h=1}^{H-1} \left( \mathbb{E}_{(\tilde{s},\tilde{\boldsymbol{a}})\sim d_{P^\star,h}^{\pi^{(n)}}} \left[ \|\phi_h^\star(\tilde{s},\tilde{\boldsymbol{a}})\|_{\Sigma_{n,\gamma_h^{(n)},\phi_h^\star}^{-1}} \right] \sqrt{dA\left(\alpha^{(n)}\right)^2 + H^2 d\lambda} + \sqrt{\frac{dA\left(\alpha^{(n)}\right)^2}{n}} \right)$$

$$+ H^2 \sum_{h=1}^{H-1} \left( \frac{\alpha^{(n)}}{H} \mathbb{E}_{(\tilde{s},\tilde{\boldsymbol{a}})\sim d_{P^\star,h}^{\pi^{(n)}}} \left[ \|\phi_h^\star(\tilde{s},\tilde{\boldsymbol{a}})\|_{\Sigma_{n,\gamma_h^{(n)},\phi_h^\star}^{-1}} \right] + \sqrt{A\zeta^{(n)}} \right).$$

Hereafter, we take the dominating term out. Note that

$$\sum_{n=1}^N \mathbb{E}_{(\tilde{s},\tilde{\boldsymbol{a}})\sim d_{P^\star,h}^{\pi^{(n)}}} \left[ \|\phi_h^\star(\tilde{s},\tilde{\boldsymbol{a}})\|_{\Sigma_{n,\gamma_h^{(n)},\phi_h^\star}^{-1}} \right] \leq \sqrt{N \sum_{n=1}^N \mathbb{E}_{(\tilde{s},\tilde{\boldsymbol{a}})\sim d_{P^\star,h}^{\pi^{(n)}}} \left[ \phi_h^\star(\tilde{s},\tilde{\boldsymbol{a}})^\top \Sigma_{n,\gamma_h^{(n)},\phi_h^\star}^{-1} \phi_h^\star(\tilde{s},\tilde{\boldsymbol{a}}) \right]}$$

(CS inequality)

$$\lesssim \sqrt{N \left( \log\det\left( \sum_{n=1}^N \mathbb{E}_{(\tilde{s},\tilde{\boldsymbol{a}})\sim d_{P^\star,h}^{\pi^{(n)}}} [\phi_h^\star(\tilde{s},\tilde{\boldsymbol{a}})\phi_h^\star(\tilde{s},\tilde{\boldsymbol{a}})^\top] \right) - \log\det(\lambda I_d) \right)}$$  (Lemma E.2)

$$\leq \sqrt{dN \log\left( 1 + \frac{N}{d\lambda} \right)}.$$

(Potential function bound, Lemma E.3 noting $\|\phi_h^\star(s,\boldsymbol{a})\|_2 \leq 1$ for any $(s,\boldsymbol{a})$.)

Finally,

$$\sum_{n=1}^N \Delta^{(n)} \lesssim H \left( \sqrt{dN \log\left( 1 + \frac{N}{d} \right)} \sqrt{dA\left(\alpha^{(N)}\right)^2 + H^2 d\lambda} + \sum_{n=1}^N \sqrt{\frac{dA\left(\alpha^{(n)}\right)^2}{n}} \right)$$

$$+ H^3 \left( \frac{1}{H} \sqrt{dN \log\left( 1 + \frac{N}{d\lambda} \right)} \alpha^{(N)} + \sum_{n=1}^N \sqrt{A\zeta^{(n)}} \right)$$

$$\lesssim H^2 d \sqrt{NA \log\left( 1 + \frac{N}{d\lambda} \right)} \alpha^{(N)}$$

(Some algebra. We take the dominating term out. Note that $\alpha^{(n)}$ is increasing in $n$)

$$\lesssim H^3 d^2 A N^{\frac{1}{2}} \log\frac{|\mathcal{M}|HN}{\delta}.$$

This concludes the proof. $\qquad\square$

**Proof of Theorem 4.1**

*Proof.* For any fixed episode $n$ and agent $i$, by Lemma B.5, Lemma B.6 and Lemma B.7, we have

$$v_i^{\dagger,\pi_{-i}^{(n)}} - v_i^{\pi^{(n)}} \left( \text{or} \max_{\omega\in\Omega_i} v_i^{\omega\circ\pi^{(n)}} - v_i^{\pi^{(n)}} \right) \leq \overline{v}_i^{(n)} - \underline{v}_i^{(n)} + 2H\sqrt{A\zeta^{(n)}} \leq \Delta^{(n)}.$$

Taking maximum over $i$ on both sides, we have

$$\max_{i\in[M]} \left\{ v_i^{\dagger,\pi_{-i}^{(n)}} - v_i^{\pi^{(n)}} \right\} \left( \text{or} \max_{i\in[M]} \left\{ \max_{\omega\in\Omega_i} v_i^{\omega\circ\pi^{(n)}} - v_i^{\pi^{(n)}} \right\} \right) \leq \Delta^{(n)}. \qquad (12)$$

From Lemma B.8, with probability $1-\delta$, we can ensure

$$\sum_{n=1}^N \Delta^{(n)} \lesssim H^3 d^2 A N^{\frac{1}{2}} \log\frac{|\mathcal{M}|HN}{\delta}.$$

Therefore, according to Lemma E.4, when we pick $N$ to be

$$O\left( \frac{H^6 d^4 A^2}{\varepsilon^2} \log^2\left( \frac{HdA|\mathcal{M}|}{\delta\varepsilon} \right) \right),$$

we have

$$\frac{1}{N} \sum_{n=1}^{N} \Delta^{(n)} \leq \varepsilon.$$

On the other hand, from equation 12, we have

$$\max_{i \in [M]} \left\{ v_i^{\dagger, \hat{\pi}_{-i}} - v_i^{\hat{\pi}} \right\} \left( \text{or } \max_{i \in [M]} \left\{ \max_{\omega \in \Omega_i} v_i^{\omega \circ \hat{\pi}} - v_i^{\hat{\pi}} \right\} \right)$$

$$= \max_{i \in [M]} \left\{ v_i^{\dagger, \pi_{-i}^{(n^\star)}} - v_i^{\pi^{(n^\star)}} \right\} \left( \text{or } \max_{i \in [M]} \left\{ \max_{\omega \in \Omega_i} v_i^{\omega \circ \pi^{(n^\star)}} - v_i^{\pi^{(n^\star)}} \right\} \right)$$

$$\leq \Delta^{(n^\star)} = \min_{n \in [N]} \Delta^{(n)} \leq \frac{1}{N} \sum_{n=1}^{N} \Delta^{(n)} \leq \varepsilon,$$

which has finished the proof. $\qquad\square$

## C ANALYSIS OF THE MODEL-FREE METHOD

For the model-free method, throughout this section we assume the Markov game is a block Markov game.

### C.1 CONSTRUCTION OF $\mathcal{N}_h$ AND $\mathcal{F}_h$

Let $\mathcal{C}_h = \{ \Sigma_h : \Sigma_h = \lambda I_d + \sum_{k=1}^{l} \phi_h(s_k, \boldsymbol{a}_k) \phi_h(s_k, \boldsymbol{a}_k)^\top | \phi_h \in \Phi_h, l \in [N], s_k \in \mathcal{S}, \boldsymbol{a}_k \in \mathcal{A}, \forall k \in [l] \}$. Fix a variable $L$, for each $h \in [H]$, define a function class $\tilde{\mathcal{F}}_h \in \mathbb{R}^{\mathcal{S} \times \mathcal{A}}$ by

$$\tilde{\mathcal{F}}_h = \Big\{ f(s, \boldsymbol{a}) := r_{h,i}(s, \boldsymbol{a}) + \phi_h(s, \boldsymbol{a})^\top \theta + \min\left( c \| \phi_h(s, \boldsymbol{a}) \|_{\Sigma_h^{-1}}, H \right) \Big|$$

$$i \in [M], \phi_h \in \Phi_h, \|\theta\|_2 \leq 2H^2 \sqrt{d}, c \in [0, L], \Sigma_h \in \mathcal{C}_h \Big\}$$

For a given parameter $\tilde{\varepsilon}$, let $\mathcal{N}_h$ be a $\tilde{\varepsilon}$-net of $\tilde{\mathcal{F}}_h$ under the $\| \cdot \|_\infty$ metric. Define $\Pi_h$ as the set of all possible policies produced by equation 2 (or equation 3 or equation 4, according to the problem setting). We then define the discriminator function class $\mathcal{F}_h$ as followings:

$$\mathcal{F}_{1,h} := \Big\{ f(s) := \mathbb{E}_{\boldsymbol{a} \sim U(\mathcal{A})} \left[ \left| \phi_h(s, \boldsymbol{a})^\top \theta - \phi_h'(s, \boldsymbol{a})^\top \theta' \right| \right] \Big| \phi_h, \phi_h' \in \Phi_h, \max\{\|\theta\|_2, \|\theta'\|_2\} \leq \sqrt{d} \Big\},$$

$$\mathcal{F}_{2,h} := \Big\{ f(s) := \mathbb{E}_{\boldsymbol{a} \sim \pi_{h+1}(s)} \left[ \frac{r_{h+1,i}(s, \boldsymbol{a})}{H} + \phi_{h+1}(s, \boldsymbol{a})^\top \theta \right] \Big| i \in [M], \pi_{h+1} \in \Pi_{h+1}, \phi_{h+1} \in \Phi_{h+1}, \|\theta\|_2 \leq \sqrt{d} \Big\},$$

$$\mathcal{F}_{3,h} := \Big\{ f(s) := \max_{\tilde{\mu}_{h+1,i}} \mathbb{E}_{\boldsymbol{a} \sim (\tilde{\mu}_{h+1,i} \times \pi_{h+1,-i})(s)} \left[ \frac{r_{h+1,i}(s, \boldsymbol{a})}{H} + \phi_{h+1}(s, \boldsymbol{a})^\top \theta \right] \Big|$$

$$i \in [M], \pi_{h+1} \in \Pi_{h+1}, \phi_{h+1} \in \Phi_{h+1}, \|\theta\|_2 \leq \sqrt{d} \Big\}, \qquad\qquad \text{(For NE and CCE)}$$

$$\mathcal{F}_{3,h} := \Big\{ f(s) := \max_{\omega_{h+1,i} \in \Omega_{h+1,i}} \mathbb{E}_{\boldsymbol{a} \sim (\omega_{h+1,i} \circ \pi_{h+1})(s)} \left[ \frac{r_{h+1,i}(s, \boldsymbol{a})}{H} + \phi_{h+1}(s, \boldsymbol{a})^\top \theta \right] \Big|$$

$$i \in [M], \pi_{h+1} \in \Pi_{h+1}, \phi_{h+1} \in \Phi_{h+1}, \|\theta\|_2 \leq \sqrt{d} \Big\}, \qquad\qquad \text{(For CE)}$$

$$\mathcal{F}_{4,h} := \Big\{ f(s) := \mathbb{E}_{\boldsymbol{a} \sim \pi_{h+1}(s)} \left[ \frac{\min\left\{ c \| \phi_{h+1}(s, \boldsymbol{a}) \|_{\Sigma_{h+1}^{-1}}, H \right\}}{H^2} + \phi_{h+1}(s, \boldsymbol{a})^\top \theta \right] \Big|$$

$$c \in [0, L], \pi_{h+1} \in \Pi_{h+1}, \Sigma_{h+1} \in \mathcal{C}_{h+1}, \phi_{h+1} \in \Phi_{h+1}, \|\theta\|_2 \leq \sqrt{d} \Big\},$$

$$\mathcal{G} := \{ f : \mathcal{S} \to [0, 1] \},$$

$$\mathcal{F}_h := (\mathcal{F}_{1,h} \cup \mathcal{F}_{2,h} \cup \mathcal{F}_{3,h} \cup \mathcal{F}_{4,h}) \cap \mathcal{G}.$$

## C.2 HIGH PROBABILITY EVENTS

We define the following event

$$\mathcal{E}_1 : \forall n \in [N], h \in [H], \rho \in \left\{ \rho_h^{(n)}, \tilde{\rho}_h^{(n)} \right\}, f \in \mathcal{F}_h, \quad \mathbb{E}_\rho \left[ \left( \left( \left( \hat{P}_h^{(n)} - P_h^\star \right) f \right)(s, \boldsymbol{a}) \right)^2 \right] \leq \zeta^{(n)},$$

$$\mathcal{E}_2 : \forall n \in [N], h \in [H], \phi_h \in \Phi_h, \quad \|\phi_h(s, \boldsymbol{a})\|_{\left( \hat{\Sigma}_{h,\phi_h}^{(n)} \right)^{-1}} = \Theta \left( \|\phi_h(s, \boldsymbol{a})\|_{\Sigma_{n,\rho_h^{(n)},\phi_h}^{-1}} \right)$$

$$\mathcal{E} := \mathcal{E}_1 \cap \mathcal{E}_2.$$

Similar to the procedure of the model-based case, we first prove a few lemmas which lead to the conclusion that $\mathcal{E}$ holds with a high probability.

**Lemma C.1.** *For any $n \in [N], h \in [H]$, we have $\hat{P}_h^{(n)}(s'|s, \boldsymbol{a}) = \hat{\phi}_h^{(n)}(s, \boldsymbol{a})^\top \hat{w}_h^{(n)}(s')$ for some $\hat{w}_h^{(n)} : \mathcal{S} \rightarrow \mathbb{R}^d$. For any function $f : \mathcal{S} \rightarrow [0, 1]$ and $n \in [N], h \in [H]$, we have $\left\| \int_{\mathcal{S}} \hat{w}_h^{(n)}(s')f(s')\mathrm{d}s' \right\|_2 \leq \sqrt{d}$, and there exist $\theta, \tilde{\theta} \in \mathbb{R}^d$ such that $(P_h^\star f)(s, \boldsymbol{a}) = \phi_h^\star(s, \boldsymbol{a})^\top \theta$, $\left( \hat{P}_h^{(n)} f \right)(s, \boldsymbol{a}) = \hat{\phi}_h^{(n)}(s, \boldsymbol{a})^\top \tilde{\theta}$ and $\max\{\|\theta\|_2, \|\tilde{\theta}\|_2\} \leq \sqrt{d}$. Furthermore, we have $\|\tilde{\theta}\|_\infty \leq 1$.*

*Proof.* By definition, we have

$$(P_h^\star f)(s, \boldsymbol{a}) = \int_{\mathcal{S}} P_h^\star(s'|s, \boldsymbol{a})f(s')\mathrm{d}s'$$

$$= \phi_h^\star(s, \boldsymbol{a})^\top \int_{\mathcal{S}} w_h^\star(s')f(s')\mathrm{d}s'$$

$$= \phi_h^\star(s, \boldsymbol{a})^\top \theta,$$

where $\theta = \int_{\mathcal{S}} w_h^\star(s')f(s')\mathrm{d}s'$. Furthermore, note that $\|f\|_\infty \leq 1$, according to the assumption on $w_h^\star$, we have

$$\left\| \int_{\mathcal{S}} w_h^\star(s')f(s')\mathrm{d}s' \right\|_2 \leq \sqrt{d},$$

which implies $\|\theta\|_2 \leq \sqrt{d}$. For $\left( \hat{P}_h^{(n)} f \right)(s, \boldsymbol{a})$, let

$$\hat{w}_h^{(n)}(s') := \left( \sum_{(\tilde{s}, \tilde{\boldsymbol{a}}) \in \mathcal{D}_h^{(n)} \cup \tilde{\mathcal{D}}_h^{(n)}} \phi_h^{(n)}(\tilde{s}, \tilde{\boldsymbol{a}})\phi_h^{(n)}(\tilde{s}, \tilde{\boldsymbol{a}})^\top + \lambda I_d \right)^{-1} \sum_{(\tilde{s}, \tilde{\boldsymbol{a}}, \tilde{s}') \in \mathcal{D}_h^{(n)} \cup \tilde{\mathcal{D}}_h^{(n)}} \phi_h^{(n)}(\tilde{s}, \tilde{\boldsymbol{a}})\mathbf{1}_{\tilde{s}'=s'}.$$

Since $\phi_h^{(n)}(s, \boldsymbol{a})$ is an one-hot vector, one has $\left\| \hat{w}_h^{(n)}(s') \right\|_\infty \leq 1, \forall s' \in \mathcal{S}$. It follows that $\left\| \int_{\mathcal{S}} \hat{w}_h^{(n)}(s')f(s')\mathrm{d}s' \right\|_\infty \leq 1$, and therefore, $\left\| \int_{\mathcal{S}} \hat{w}_h^{(n)}(s')f(s')\mathrm{d}s' \right\|_2 \leq \sqrt{d}$. By definition, we have

$$\left( \hat{P}_h^{(n)} f \right)(s, \boldsymbol{a}) = \int_{\mathcal{S}} \hat{P}_h^{(n)}(s'|s, \boldsymbol{a})f(s')\mathrm{d}s'$$

$$= \phi_h^{(n)}(s, \boldsymbol{a})^\top \int_{\mathcal{S}} \hat{w}_h^{(n)}(s')f(s')\mathrm{d}s'$$

$$= \phi_h^{(n)}(s, \boldsymbol{a})^\top \tilde{\theta},$$

where $\tilde{\theta} = \int_{\mathcal{S}} \hat{w}_h^{(n)}(s')f(s')\mathrm{d}s'$. Due to the property we just derived for $\hat{w}_h^{(n)}$, similar to the proof of the true model, we also have $\|\theta\|_2 \leq \sqrt{d}$. Meanwhile, one can easily see that $\|\theta\|_\infty \leq 1$, using the fact $\left\| \int_{\mathcal{S}} \hat{w}_h^{(n)}(s')f(s')\mathrm{d}s' \right\|_\infty \leq 1$. $\square$

**Lemma C.2** (Covering Number of $\tilde{\mathcal{F}}_h$). *When $\Phi_h$ is the set of one-hot vectors and $\lambda \geq 1$, it's possible to construct the $\tilde{\varepsilon}$-net $\mathcal{N}_h$ such that $|\mathcal{N}_h| \leq M\left(\frac{12H^2L^2d}{\tilde{\varepsilon}}\right)^{3d}|\Phi|, \forall h \in [H]$. Furthermore, we have $|\Pi_h| \leq |\mathcal{N}_h|^M \leq M^M\left(\frac{12H^2L^2d}{\tilde{\varepsilon}}\right)^{3Md}|\Phi|^M$.*

*Proof.* Recall that

$$\tilde{\mathcal{F}}_h = \{ f(s, \boldsymbol{a}) := r_{h,i}(s, \boldsymbol{a}) + \phi_h(s, \boldsymbol{a})^\top \theta + \min\{c\|\phi_h(s, \boldsymbol{a})\|_{\Sigma_h^{-1}}, H\}\big|$$

$$i \in [M], \phi_h \in \Phi_h, \|\theta\|_2 \leq 2H^2\sqrt{d}, c \in [0, L], \Sigma \in \mathcal{C}_h\}.$$

Note that when $\Phi_h$ is the set of one-hot vectors, $\Sigma_h$ will be a diagonal matrix. In this case, $\tilde{\mathcal{F}}_h$ is the subset of the following function class:

$$\tilde{\mathcal{F}}'_h := \{ f(s, \boldsymbol{a}) := r_{h,i}(s, \boldsymbol{a}) + \min\{c\phi_h(s, \boldsymbol{a})^\top \theta', H\} + \phi_h(s, \boldsymbol{a})^\top \theta\big|$$

$$i \in [M], \phi_h \in \Phi_h, 0 \leq c \leq L, \max\{\|\theta\|_2, \|\theta'\|_2\} \leq 2H^2\sqrt{d}\}.$$

Let $\Theta$ be an $\ell_2$-cover of the set $\{\theta \in \mathbb{R}^d : \|\theta\|_2 \leq 2H^2\sqrt{d}\}$ at scale $\tilde{\varepsilon}$. Then we know $|\Theta| \leq \left(\frac{4H^2\sqrt{d}}{\tilde{\varepsilon}}\right)^d$. Let $\mathcal{W}$ be an $\ell_\infty$-cover of the set $[0, L]$ at scale $\tilde{\varepsilon}' := \frac{\tilde{\varepsilon}}{2H^2\sqrt{d}}$, we have $|\mathcal{W}| \leq \frac{2H^2L\sqrt{d}}{\tilde{\varepsilon}}$. Define the covering set by

$$\bar{\mathcal{F}}_h := \left\{ \bar{f}(s, \boldsymbol{a}) := r_{h,i}(s, \boldsymbol{a}) + \min\{\tilde{c}\phi_h(s, \boldsymbol{a})^\top \tilde{\theta}', H\} + \phi_h(s, \boldsymbol{a})^\top \tilde{\theta}\big| i \in [M], \phi_h \in \Phi_h, \tilde{c} \in \mathcal{W}, \tilde{\theta}, \tilde{\theta}' \in \Theta \right\}.$$

Then, for any $f \in \tilde{\mathcal{F}}_h$, by definition, suppose $f$ takes the following form:

$$f(s, \boldsymbol{a}) := r_{h,i}(s, \boldsymbol{a}) + \min\{c\phi_h(s, \boldsymbol{a})^\top \theta', H\} + \phi_h(s, \boldsymbol{a})^\top \theta, \quad 0 \leq c \leq L, \max\{\|\theta\|_2, \|\tilde{\theta}\|_2\} \leq 2H^2\sqrt{d}.$$

Then we can find $\tilde{\theta}, \tilde{\theta}' \in \Theta, \tilde{c} \in \mathcal{W}$ such that $\|\theta - \tilde{\theta}\|_2 \leq \tilde{\varepsilon}, \|\theta' - \tilde{\theta}'\|_2 \leq \tilde{\varepsilon}$ and $|c - \tilde{c}| \leq \tilde{\varepsilon}'$. Let

$$\bar{f}(s, \boldsymbol{a}) := r_{h,i}(s, \boldsymbol{a}) + \min\{\tilde{c}\phi_h(s, \boldsymbol{a})^\top \tilde{\theta}', H\} + \phi_h(s, \boldsymbol{a})^\top \tilde{\theta},$$

then we have

$$|f(s, \boldsymbol{a}) - \bar{f}(s, \boldsymbol{a})|$$
$$\leq \|\phi_h(s, \boldsymbol{a})\|_2 \left\|\theta - \tilde{\theta}\right\|_2 + \|\phi_h(s, \boldsymbol{a})\|_2 \left\|\tilde{c}\tilde{\theta}' - c\theta'\right\|_2$$
$$\leq \tilde{\varepsilon} + |\tilde{c} - c| \left\|\tilde{\theta}'\right\|_2 + c\left\|\theta' - \tilde{\theta}'\right\|_2$$
$$\leq \tilde{\varepsilon} + 2H^2\sqrt{d}\tilde{\varepsilon}' + L\tilde{\varepsilon}$$
$$\leq 3L\tilde{\varepsilon},$$

which implies $\bar{\mathcal{F}}_h$ is a $3L\tilde{\varepsilon}$-covering of $\tilde{F}'_h$ (therefore, is a $3L\tilde{\varepsilon}$-covering of $\tilde{F}_h$), and we have

$$\left|\bar{\mathcal{F}}_h\right| \leq M\left(\frac{4H^2Ld}{\tilde{\varepsilon}}\right)^{3d}|\Phi|.$$

Replacing $\tilde{\varepsilon}$ by $\frac{\tilde{\varepsilon}}{3L}$, we get an $\tilde{\varepsilon}$-covering of $\tilde{\mathcal{F}}_h$ whose size is no larger than $M\left(\frac{12H^2L^2d}{\tilde{\varepsilon}}\right)^{3d}|\Phi|$. For $\Pi_h$, since each policy is determined by $M$ members from $\mathcal{N}_h$, we have $|\Pi_h| \leq |\mathcal{N}_h|^M$, which has finished the proof. $\square$

**Lemma C.3** (Covering Number of $\mathcal{F}_h$). *When $\Phi_h$ is the set of one-hot vectors and $\lambda \geq 1$. The $\gamma$-covering number of $\mathcal{F}_h$ is at most $4M|\Pi_{h+1}|\left(\frac{6L^2d}{\gamma}\right)^{3d}|\Phi|^2$.*

*Proof.* We cover $\mathcal{F}_{1,h}, \mathcal{F}_{2,h}, \mathcal{F}_{3,h}, \mathcal{F}_{4,h}$ separately. For $\mathcal{F}_{1,h}$, let $\Theta$ be an $\ell_2$-cover of the set $\{\theta \in \mathbb{R}^d : \|\theta\|_2 \leq \sqrt{d}\}$ at scale $\gamma$. Then we know $|\Theta| \leq \left(\frac{2\sqrt{d}}{\gamma}\right)^d$. Define the covering set of $\mathcal{F}_{1,h}$ as

$$\tilde{\mathcal{F}}_{1,h} := \left\{ \tilde{f}(s) := \mathbb{E}_{\boldsymbol{a}\sim U(\mathcal{A})}\left[\left|\phi_h(s, \boldsymbol{a})^\top \tilde{\theta} - \phi'_h(s, \boldsymbol{a})^\top \tilde{\theta}'\right|\right] \Big| \phi_h, \phi'_h \in \Phi_h, \tilde{\theta}, \tilde{\theta}' \in \Theta \right\}.$$

For any $f \in \mathcal{F}_{1,h}$, suppose

$$f(s) = \mathbb{E}_{\boldsymbol{a} \sim U(\mathcal{A})} \left[ \left| \phi_h(s, \boldsymbol{a})^\top \theta - \phi_h'(s, \boldsymbol{a})^\top \theta' \right| \right], \quad \phi_h, \phi_h' \in \Phi_h, \max\{\|\theta\|_2, \|\theta'\|_2\} \leq \sqrt{d},$$

Then we can find $\tilde{\theta}, \tilde{\theta}' \in \Theta$ such that $\|\theta - \tilde{\theta}\|_2 \leq \gamma, \|\theta' - \tilde{\theta}'\|_2 \leq \gamma$. Let

$$\tilde{f}(s) := \mathbb{E}_{\boldsymbol{a} \sim U(\mathcal{A})} \left[ \left| \phi_h(s, \boldsymbol{a})^\top \tilde{\theta} - \phi_h'(s, \boldsymbol{a})^\top \tilde{\theta}' \right| \right].$$

Then we have

$$\begin{aligned}
|f(s) - \tilde{f}(s)| \leq &\frac{1}{A} \sum_{\boldsymbol{a} \in \mathcal{A}} \|\phi_h(s, \boldsymbol{a})\|_2 \left\| \theta - \tilde{\theta} \right\|_2 + \frac{1}{A} \sum_{\boldsymbol{a} \in \mathcal{A}} \|\phi_h'(s, \boldsymbol{a})\|_2 \left\| \theta' - \tilde{\theta}' \right\|_2 \\
\leq &2\gamma,
\end{aligned}$$

which implies $\tilde{\mathcal{F}}_{1,h}$ is a $2\gamma$ covering of $\mathcal{F}_{1,h}$. Furthermore, we have

$$|\tilde{\mathcal{F}}_{1,h}| \leq \left( \frac{2d}{\gamma} \right)^{2d} |\Phi|^2.$$

For $\mathcal{F}_{2,h}$ and $\mathcal{F}_{3,h}$, we construct

$$\tilde{\mathcal{F}}_{2,h} := \left\{ \tilde{f}(s) := \mathbb{E}_{\boldsymbol{a} \sim \pi_{h+1}(s)} \left[ \frac{r_{h+1,i}(s, \boldsymbol{a})}{H} + \phi_{h+1}(s, \boldsymbol{a})^\top \tilde{\theta} \right] \middle| i \in [M], \phi_{h+1} \in \Phi_{h+1}, \tilde{\theta} \in \Theta, \pi_{h+1} \in \Pi_{h+1} \right\}.$$

Similar to the proof of $\mathcal{F}_{1,h}$, we may verify $\tilde{\mathcal{F}}_{2,h}$ is a $\gamma$-covering of $\mathcal{F}_{2,h}$, and

$$|\tilde{\mathcal{F}}_{2,h}| \leq M |\Pi_{h+1}| \left( \frac{2d}{\gamma} \right)^d |\Phi|.$$

For $\mathcal{F}_{3,h}$, we only prove the case of NE or CCE, the case of CE can be proved in a similar manner. We construct

$$\begin{aligned}
\tilde{\mathcal{F}}_{3,h} := \Big\{ &\tilde{f}(s) := \max_{\tilde{\mu}_{h+1,i}} \mathbb{E}_{\boldsymbol{a} \sim (\tilde{\mu}_{h+1,i} \times \pi_{h+1,-i})(s)} \left[ \frac{r_{h+1,i}(s, \boldsymbol{a})}{H} + \phi_{h+1}(s, \boldsymbol{a})^\top \tilde{\theta} \right] \Big| \\
&i \in [M], \phi_{h+1} \in \Phi_{h+1}, \tilde{\theta} \in \Theta, \pi_{h+1} \in \Pi_{h+1} \Big\}.
\end{aligned}$$

For any $f \in \mathcal{F}_{3,h}$, suppose

$$f(s) = \max_{\tilde{\mu}_{h+1,i}} \mathbb{E}_{\boldsymbol{a} \sim (\tilde{\mu}_{h+1,i} \times \pi_{h+1,-i})(s)} \left[ \frac{r_{h+1,i}(s, \boldsymbol{a})}{H} + \phi_{h+1}(s, \boldsymbol{a})^\top \theta \right], \quad i \in [M], \pi_{h+1} \in \Pi_{h+1}, \phi_{h+1} \in \Phi_{h+1}, \|\theta\|_2 \leq \sqrt{d}.$$

Then we can find $\tilde{\theta} \in \Theta$ such that $\|\theta - \tilde{\theta}\|_2 \leq \gamma$. Let

$$\tilde{f}(s) = \max_{\tilde{\mu}_{h+1,i}} \mathbb{E}_{\boldsymbol{a} \sim (\tilde{\mu}_{h+1,i} \times \pi_{h+1,-i})(s)} \left[ \frac{r_{h+1,i}(s, \boldsymbol{a})}{H} + \phi_{h+1}(s, \boldsymbol{a})^\top \tilde{\theta} \right],$$

we have

$$\begin{aligned}
f(s) - \tilde{f}(s) = &\max_{\tilde{\mu}_{h+1,i}} \mathbb{E}_{\boldsymbol{a} \sim (\tilde{\mu}_{h+1,i} \times \pi_{h+1,-i})(s)} \left[ \frac{r_{h+1,i}(s, \boldsymbol{a})}{H} + \phi_{h+1}(s, \boldsymbol{a})^\top \theta \right] \\
&- \max_{\tilde{\mu}_{h+1,i}} \mathbb{E}_{\boldsymbol{a} \sim (\tilde{\mu}_{h+1,i} \times \pi_{h+1,-i})(s)} \left[ \frac{r_{h+1,i}(s, \boldsymbol{a})}{H} + \phi_{h+1}(s, \boldsymbol{a})^\top \tilde{\theta} \right] \\
\leq &\max_{\tilde{\mu}_{h+1,i}} \left( \mathbb{E}_{\boldsymbol{a} \sim (\tilde{\mu}_{h+1,i} \times \pi_{h+1,-i})(s)} \left[ \frac{r_{h+1,i}(s, \boldsymbol{a})}{H} + \phi_{h+1}(s, \boldsymbol{a})^\top \theta \right] \right. \\
&- \left. \mathbb{E}_{\boldsymbol{a} \sim (\tilde{\mu}_{h+1,i} \times \pi_{h+1,-i})(s)} \left[ \frac{r_{h+1,i}(s, \boldsymbol{a})}{H} + \phi_{h+1}(s, \boldsymbol{a})^\top \tilde{\theta} \right] \right) \\
= &\max_{\tilde{\mu}_{h+1,i}} \left( \mathbb{E}_{\boldsymbol{a} \sim (\tilde{\mu}_{h+1,i} \times \pi_{h+1,-i})(s)} \left[ \phi_{h+1}(s, \boldsymbol{a})^\top \theta - \phi_{h+1}(s, \boldsymbol{a})^\top \tilde{\theta} \right] \right) \\
\leq &\|\theta - \tilde{\theta}\|_2
\end{aligned}$$

$$\leq \gamma,$$

and

$$
\begin{aligned}
\tilde{f}(s) - f(s) &= \max_{\tilde{\mu}_{h+1,i}} \mathbb{E}_{\boldsymbol{a} \sim (\tilde{\mu}_{h+1,i} \times \pi_{h+1,-i})(s)} \left[ \frac{r_{h+1,i}(s, \boldsymbol{a})}{H} + \phi_{h+1}(s, \boldsymbol{a})^\top \tilde{\theta} \right] \\
&\quad - \max_{\tilde{\mu}_{h+1,i}} \mathbb{E}_{\boldsymbol{a} \sim (\tilde{\mu}_{h+1,i} \times \pi_{h+1,-i})(s)} \left[ \frac{r_{h+1,i}(s, \boldsymbol{a})}{H} + \phi_{h+1}(s, \boldsymbol{a})^\top \theta \right] \\
&\leq \max_{\tilde{\mu}_{h+1,i}} \left( \mathbb{E}_{\boldsymbol{a} \sim (\tilde{\mu}_{h+1,i} \times \pi_{h+1,-i})(s)} \left[ \frac{r_{h+1,i}(s, \boldsymbol{a})}{H} + \phi_{h+1}(s, \boldsymbol{a})^\top \tilde{\theta} \right] \right. \\
&\quad \left. - \mathbb{E}_{\boldsymbol{a} \sim (\tilde{\mu}_{h+1,i} \times \pi_{h+1,-i})(s)} \left[ \frac{r_{h+1,i}(s, \boldsymbol{a})}{H} + \phi_{h+1}(s, \boldsymbol{a})^\top \theta \right] \right) \\
&= \max_{\tilde{\mu}_{h+1,i}} \left( \mathbb{E}_{\boldsymbol{a} \sim (\tilde{\mu}_{h+1,i} \times \pi_{h+1,-i})(s)} \left[ \phi_{h+1}(s, \boldsymbol{a})^\top \tilde{\theta} - \phi_{h+1}(s, \boldsymbol{a})^\top \theta \right] \right) \\
&\leq \|\theta - \tilde{\theta}\|_2 \\
&\leq \gamma,
\end{aligned}
$$

which implies

$$
\left| \tilde{f}(s) - f(s) \right| \leq \gamma.
$$

Therefore, we conclude $\tilde{\mathcal{F}}_{3,h}$ is a $\gamma$-covering of $\mathcal{F}_{3,h}$, and

$$
|\tilde{\mathcal{F}}_{3,h}| \leq M |\Pi_{h+1}| \left( \frac{2d}{\gamma} \right)^d |\Phi|.
$$

For $\mathcal{F}_{4,h}$, note that when $\Phi_h$ is the set of one-hot vectors, $\Sigma_h$ will be a diagonal matrix. In this case, $\mathcal{F}_{4,h}$ is the subset of the following function class:

$$
\mathcal{F}'_{4,h} := \left\{ f(s) := \mathbb{E}_{\boldsymbol{a} \sim \pi_{h+1}(s)} \left[ \frac{\min\{c\phi_{h+1}(s, \boldsymbol{a})^\top \theta', H\}}{H^2} + \phi_{h+1}(s, \boldsymbol{a})^\top \theta \right] \right| \\
0 \leq c \leq L, \pi_{h+1} \in \Pi_{h+1}, \max\{\|\theta\|_2, \|\theta'\|_2\} \leq \sqrt{d}, \phi_{h+1} \in \Phi_{h+1} \right\}.
$$

In this case, let $\mathcal{W}$ be an $\ell_\infty$ cover of the set $[0, L]$ at scale $\tilde{\gamma} := \frac{\gamma}{\sqrt{d}}$, we have $|\mathcal{W}| \leq \frac{L\sqrt{d}}{\gamma}$. Let

$$
\tilde{\mathcal{F}}_{4,h} := \left\{ \tilde{f}(s) := \mathbb{E}_{\boldsymbol{a} \sim \pi_{h+1}(s)} \left[ \frac{\min\{\tilde{c}\phi_{h+1}(s, \boldsymbol{a})^\top \tilde{\theta}', H\}}{H^2} + \phi_{h+1}(s, \boldsymbol{a})^\top \tilde{\theta} \right] \right| \\
\tilde{c} \in \mathcal{W}, \pi_{h+1} \in \Pi_{h+1}, \tilde{\theta}, \tilde{\theta}' \in \Theta, \phi_{h+1} \in \Phi_{h+1} \right\}.
$$

Then, for any $f \in \mathcal{F}_{4,h}$, suppose

$$
f(s) := \mathbb{E}_{\boldsymbol{a} \sim \pi_{h+1}(s)} \left[ \frac{\min\{c\phi_{h+1}(s, \boldsymbol{a})^\top \theta', H\}}{H^2} + \phi_{h+1}(s, \boldsymbol{a})^\top \theta \right],
$$

$$
0 \leq c \leq L, \pi_{h+1} \in \Pi_{h+1}, \max\{\|\theta\|_2, \|\theta'\|_2\} \leq \sqrt{d}, \phi_{h+1} \in \Phi_{h+1}.
$$

Then we can find $\tilde{\theta}, \tilde{\theta}' \in \Theta, \tilde{c} \in \mathcal{W}$ such that $\|\theta - \tilde{\theta}\|_2 \leq \gamma, \|\theta' - \tilde{\theta}'\|_2 \leq \gamma$ and $|c - \tilde{c}| \leq \tilde{\gamma}$. Let

$$
\tilde{f}(s) := \mathbb{E}_{\boldsymbol{a} \sim \pi_{h+1}(s)} \left[ \frac{\min\{\tilde{c}\phi_{h+1}(s, \boldsymbol{a})^\top \tilde{\theta}', H\}}{H^2} + \phi_{h+1}(s, \boldsymbol{a})^\top \tilde{\theta} \right],
$$

then we have

$$
\begin{aligned}
&|f(s) - \tilde{f}(s)| \\
&\leq \mathbb{E}_{\boldsymbol{a} \sim \pi_{h+1}(s)} \left[ \|\phi_{h+1}(s, \boldsymbol{a})\|_2 \left\| \theta - \tilde{\theta} \right\|_2 \right] + \frac{1}{H^2} \mathbb{E}_{\boldsymbol{a} \sim \pi_{h+1}(s)} \left[ \|\phi_{h+1}(s, \boldsymbol{a})\|_2 \left\| \tilde{c}\tilde{\theta}' - c\theta' \right\|_2 \right]
\end{aligned}
$$

$$\leq \gamma + \frac{1}{H^2} \left( |\tilde{c} - c| \left\| \tilde{\theta}' \right\|_2 + c \left\| \theta' - \tilde{\theta}' \right\|_2 \right)$$

$$\leq \gamma + \frac{\sqrt{d}}{H^2} \tilde{\gamma} + \frac{L}{H^2} \gamma$$

$$\leq 3L\gamma,$$

which implies $\tilde{\mathcal{F}}_{4,h}$ is a $3L\gamma$-covering of $\mathcal{F}_{4,h}$, and we have

$$\left| \tilde{\mathcal{F}}_{4,h} \right| \leq |\Pi_{h+1}| \left( \frac{2Ld}{\gamma} \right)^{3d} |\Phi|.$$

In summary, we know $\tilde{\mathcal{F}}_h := \tilde{\mathcal{F}}_{1,h} \cup \tilde{\mathcal{F}}_{2,h} \cup \tilde{\mathcal{F}}_{3,h} \cup \tilde{\mathcal{F}}_{4,h}$ is a $3L\gamma$-covering of $\mathcal{F}_h$. And

$$|\mathcal{F}_h| \leq 4M|\Pi_{h+1}| \left( \frac{2Ld}{\gamma} \right)^{3d} |\Phi|^2.$$

Replacing $\gamma$ by $\frac{\gamma}{3L}$, we get an $\gamma$-covering of $\mathcal{F}_h$ whose size is no larger than $4M|\Pi_{h+1}| \left( \frac{6L^2 d}{\gamma} \right)^{3d} |\Phi|^2$, which has finished the proof. $\qquad \square$

Below we omit the superscript $n$ and subscript $h$ when clear from the context. Denote

$$\mathcal{L}_{\lambda,\mathcal{D}}(\phi, \theta, f) = \frac{1}{|\mathcal{D}|} \sum_{(s,\boldsymbol{a},s') \in \mathcal{D}} \left( \phi(s,\boldsymbol{a})^\top \theta - f(s') \right)^2 + \frac{\lambda}{|\mathcal{D}|} \|\theta\|_2^2 \tag{13}$$

$$\mathcal{L}_{\mathcal{D}}(\phi, \theta, f) = \frac{1}{|\mathcal{D}|} \sum_{(s,\boldsymbol{a},s') \in \mathcal{D}} \left( \phi(s,\boldsymbol{a})^\top \theta - f(s') \right)^2 \tag{14}$$

$$\mathcal{L}_\rho(\phi, \theta, f) = \mathbb{E}_{(s,\boldsymbol{a}) \sim \rho, s' \sim P^\star(s,\boldsymbol{a})} \left[ \left( \phi(s,\boldsymbol{a})^\top \theta - f(s') \right)^2 \right]. \tag{15}$$

**Lemma C.4** (Uniform Convergence for Square Loss). *Let there be a dataset $\mathcal{D} := \{(s_i, \boldsymbol{a}_i, s'_i)\}_{i=1}^n$ collected in $n$ episodes. Denote that the data generating distribution in iteration $i$ by $d_i$, and $\rho = \frac{1}{n} \sum_{i=1}^n d_i$. Note that $d_i$ can depend on the randomness in episodes $1, \ldots, i-1$. For a finite feature class $\Phi$ and a discriminator class $\mathcal{F} : \mathcal{S} \to [0,1]$ with $\gamma$-covering number $\|\mathcal{F}\|_\gamma$, we will show that, with probability at least $1 - \delta$:*

$$\left| \left[ \mathcal{L}_\rho(\phi, \theta, f) - \mathcal{L}_\rho(\phi^\star, \theta_f^\star, f) \right] - \left[ \mathcal{L}_{\mathcal{D}}(\phi, \theta, f) - \mathcal{L}_{\mathcal{D}}(\phi^\star, \theta_f^\star, f) \right] \right|$$

$$\leq \frac{1}{2} \left[ \mathcal{L}_\rho(\phi, \theta, f) - \mathcal{L}_\rho(\phi^\star, \theta_f^\star, f) \right] + \frac{64 \log(\frac{2(4n)^d \cdot |\Phi| \cdot \|\mathcal{F}\|_{1/2n}}{\delta})}{n}$$

*for all $\phi \in \Phi$, $\|\theta\|_\infty \leq 1$ and $f \in \mathcal{F}$, where recall that $\phi^\star$ is the true feature and $\theta_f^\star$ is defined as $\mathbb{E}_{s' \sim P^\star(s,\boldsymbol{a})}[f(s')] = \langle \phi^\star(s,\boldsymbol{a}), \theta_f^\star \rangle$.*

*Proof.* To start, we focus on a given $f \in \mathcal{F}$. We first give a high probability bound on the following deviation term:

$$\left| \mathcal{L}_\rho(\phi, \theta, f) - \mathcal{L}_\rho(\phi^*, \theta_f^*, f) - \left( \mathcal{L}_{\mathcal{D}}(\phi, \theta, f) - \mathcal{L}_{\mathcal{D}}(\phi^*, \theta_f^*, f) \right) \right|.$$

Denote $g(s_i, \boldsymbol{a}_i) = \phi(s_i, \boldsymbol{a}_i)^\top \theta$ and $g^\star(s_i, \boldsymbol{a}_i) = \phi^\star(s_i, \boldsymbol{a}_i)^\top \theta_f^\star$. At episode $i$, let $\mathcal{F}_{i-1}$ be the $\sigma$-field generated by all the random variables over the first $i-1$ episodes, for the random variable $Y_i := (g(s_i, \boldsymbol{a}_i) - f(s'_i))^2 - (g^\star(s_i, \boldsymbol{a}_i) - f(s'_i))^2$, we have

$$\mathbb{E}[Y_i | \mathcal{F}_{i-1}] = \mathbb{E} \left[ (g(s_i, \boldsymbol{a}_i) - f(s'_i))^2 - (g^\star(s_i, \boldsymbol{a}_i) - f(s'_i))^2 \right]$$

$$= \mathbb{E} \left[ (g(s_i, \boldsymbol{a}_i) + g^\star(s_i, \boldsymbol{a}_i) - 2f(s'_i)) (g(s_i, \boldsymbol{a}_i) - g^\star(s_i, \boldsymbol{a}_i)) \right]$$

$$= \mathbb{E} \left[ (g(s_i, \boldsymbol{a}_i) - g^\star(s_i, \boldsymbol{a}_i))^2 \right].$$

Here the conditional expectation is taken according to the distribution $d_i | \mathcal{F}_{i-1}$. The last equality is due to the fact that

$$\mathbb{E} \left[ (g^\star(s_i, \boldsymbol{a}_i) - f(s'_i)) (g(s_i, \boldsymbol{a}_i) - g^\star(s_i, \boldsymbol{a}_i)) \right]$$

$$= \mathbb{E}_{s_i, \boldsymbol{a}_i} \left[ \mathbb{E}_{s_i'} \left[ (g^\star(s_i, \boldsymbol{a}_i) - f(s_i')) (g(s_i, \boldsymbol{a}_i) - g^\star(s_i, \boldsymbol{a}_i)) | s_i, \boldsymbol{a}_i \right] \right]$$
$$= 0.$$

Next, for the conditional variance of the random variable, we have:

$$\mathbb{V}[Y_i | \mathcal{F}_{i-1}] \leq \mathbb{E} \left[ Y_i^2 | \mathcal{F}_{i-1} \right] = \mathbb{E} \left[ (g(s_i, \boldsymbol{a}_i) + g^\star(s_i, \boldsymbol{a}_i) - 2f(s_i'))^2 (g(s_i, \boldsymbol{a}_i) - g^\star(s_i, \boldsymbol{a}_i))^2 | \mathcal{F}_{i-1} \right]$$

$$\leq 16 \mathbb{E} \left[ (g(s_i, \boldsymbol{a}_i) - g^\star(s_i, \boldsymbol{a}_i))^2 | \mathcal{F}_{i-1} \right]$$

$$\leq 16 \mathbb{E}[Y_i | \mathcal{F}_{i-1}].$$

Noticing $Y_i \in [-4, 4]$. Applying Lemma 1 in (Foster and Rakhlin, 2020), we get with probability at least $1 - \delta'$, we can bound the deviation term above as:

$$\left| \mathcal{L}_\rho(\phi, \theta, f) - \mathcal{L}_\rho(\phi^\star, \theta_f^\star, f) - \left( \mathcal{L}_\mathcal{D}(\phi, \theta, f) - \mathcal{L}_\mathcal{D}(\phi^\star, \theta_f^\star, f) \right) \right|$$

$$\leq \sqrt{\frac{2 \sum_{i=1}^n \mathbb{V}[Y_i | \mathcal{F}_{i-1}] \log \frac{2}{\delta'}}{n^2}} + \frac{16 \log \frac{2}{\delta'}}{3n}$$

$$\leq \sqrt{\frac{32 \sum_{i=1}^n \mathbb{E}[Y_i | \mathcal{F}_{i-1}] \log \frac{2}{\delta'}}{n^2}} + \frac{16 \log \frac{2}{\delta'}}{3n},$$

Further, consider a finite point-wise cover of the function class $\mathcal{G} := \{g(s, \boldsymbol{a}) = \phi(s, \boldsymbol{a})^\top \theta : \phi \in \Phi, \|\theta\|_\infty \leq 1\}$. Note that, with a $\ell_\infty$-cover $\overline{\mathcal{W}}$ of $\mathcal{W} = \{\|\theta\|_\infty \leq 1\}$ at scale $\gamma$, we have for all $(s, \boldsymbol{a})$ and $\phi \in \Phi$, there exists $\bar{\theta} \in \overline{\mathcal{W}}$, $|\langle \phi(s, \boldsymbol{a}), \theta - \bar{\theta} \rangle| \leq \gamma$, and we have $|\mathcal{W}| = \left( \frac{2}{\gamma} \right)^d$. Let $\tilde{\mathcal{F}}$ be a $\gamma$-covering set of $\mathcal{F}$. For any $f \in \mathcal{F}$, there exists $\bar{f} \in \tilde{\mathcal{F}}$ such that $\|f - \bar{f}\|_\infty \leq \gamma$. Then, applying a union bound over elements in $\Phi \times \overline{\mathcal{W}} \times \tilde{\mathcal{F}}$, with probability $1 - |\Phi||\overline{\mathcal{W}}||\tilde{\mathcal{F}}|\delta'$, for all $\theta \in \mathcal{W}, f \in \mathcal{F}$, we have:

$$\left| \mathcal{L}_\rho(\phi, \theta, f) - \mathcal{L}_\rho(\phi^\star, \theta_f^\star, f) - \left( \mathcal{L}_\mathcal{D}(\phi, \theta, f) - \mathcal{L}_\mathcal{D}(\phi^\star, \theta_f^\star, f) \right) \right|$$

$$\leq \left| \mathcal{L}_\rho(\phi, \bar{\theta}, \bar{f}) - \mathcal{L}_\rho(\phi^\star, \theta_{\bar{f}}^\star, \bar{f}) - \left( \mathcal{L}_\mathcal{D}(\phi, \bar{\theta}, \bar{f}) - \mathcal{L}_\mathcal{D}(\phi^\star, \theta_{\bar{f}}^\star, \bar{f}) \right) \right| + 16\gamma$$

$$\leq \sqrt{\frac{32 \sum_{i=1}^n \mathbb{E}[\bar{Y}_i | \mathcal{F}_{i-1}] \log \frac{2}{\delta'}}{n^2}} + \frac{16 \log \frac{2}{\delta'}}{3n} + 16\gamma$$

$$\leq \frac{1}{2n} \sum_{i=1}^n \mathbb{E}[\bar{Y}_i | \mathcal{F}_{i-1}] + \frac{16 \log \frac{2}{\delta'}}{n} + \frac{16 \log \frac{2}{\delta'}}{3n} + 16\gamma$$

$$\leq \frac{1}{2n} \sum_{i=1}^n \mathbb{E}[Y_i | \mathcal{F}_{i-1}] + \frac{16 \log \frac{2}{\delta'}}{n} + \frac{16 \log \frac{2}{\delta'}}{3n} + 32\gamma$$

$$\leq \frac{1}{2} \left( \mathcal{L}_\rho(\phi, \theta, f) - \mathcal{L}_\rho(\phi^\star, \theta_f^\star, f) \right) + \frac{32 \log \frac{2}{\delta'}}{n} + 32\gamma$$

$$\leq \frac{1}{2} \left( \mathcal{L}_\rho(\phi, \theta, f) - \mathcal{L}_\rho(\phi^\star, \theta_f^\star, f) \right) + \frac{64 \log \frac{2}{\delta'}}{n} \qquad \text{(setting } \gamma = 1/n\text{)}$$

where $\bar{Y}_i := \left( \phi(s_i, \boldsymbol{a}_i)^\top \bar{\theta} - \bar{f}(s') \right)^2 - \left( \phi(s_i, \boldsymbol{a}_i)^\top \theta_{\bar{f}}^\star - \bar{f}(s') \right)^2$. Finally, setting $\delta = \delta' / \left( |\Phi||\overline{\mathcal{W}}||\tilde{\mathcal{F}}| \right)$, we get $\log \frac{2}{\delta'} \leq \log \frac{2(4n)^d |\Phi||\tilde{\mathcal{F}}|}{\delta}$. This completes the proof. $\qquad \square$

**Lemma C.5** (Deviation Bounds for Alg. 3). *Let* $\varepsilon' = \frac{128 \log \left( \frac{2(4n)^d \cdot |\Phi| \cdot \|\mathcal{F}\|_{1/2n}}{\delta} \right)}{n}$. *If Alg. 3 is called with a dataset $\mathcal{D}$ of size $n$, then with probability at least $1 - \delta$, for any $f \in \mathcal{F} \subset [0, 1]^\mathcal{S}$, we have*

$$\mathbb{E}_\rho \left[ \left( \hat{\phi}(s, \boldsymbol{a})^\top \hat{\theta}_f - \phi^\star(s, \boldsymbol{a})^\top \theta_f^\star \right)^2 \right] \leq \varepsilon' + \frac{2\lambda d}{n}.$$

*Proof.* We begin by using the result in Lemma C.4 such that, with probability at least $1 - \delta$, for all $\|\theta\|_\infty \leq 1, \phi \in \Phi$ and $f \in \mathcal{F}$, we have

$$\left| \left[ \mathcal{L}_\rho(\phi, \theta, f) - \mathcal{L}_\rho(\phi^\star, \theta_f^\star, f) \right] - \left[ \mathcal{L}_\mathcal{D}(\phi, \theta, f) - \mathcal{L}_\mathcal{D}(\phi^\star, \theta_f^\star, f) \right] \right| \leq \frac{1}{2} \left[ \mathcal{L}_\rho(\phi, \theta, f) - \mathcal{L}_\rho(\phi^\star, \theta_f^\star, f) \right] + \varepsilon'/2.$$

Thus, with probability at least $1 - \delta$ we have:

$$
\mathbb{E}_\rho \left[ \left( \hat\phi(s, \boldsymbol{a})^\top \hat\theta_f - \phi^\star(s, \boldsymbol{a})^\top \theta_f^\star \right)^2 \right]
$$

$$
= \mathcal{L}_\rho(\hat\phi, \hat\theta_f, f) - \mathcal{L}_\rho(\phi^\star, \theta_f^\star, f) \qquad \text{(since } \mathbb{E}_{s' \sim P^\star(s, \boldsymbol{a})}\left[f(s')\right] = \phi^\star(s, \boldsymbol{a})^\top \theta_f^\star\text{)}
$$

$$
\leq 2 \left( \mathcal{L}_\mathcal{D}(\hat\phi, \hat\theta_f, f) - \mathcal{L}_\mathcal{D}(\phi^\star, \theta_f^\star, f) \right) + \varepsilon'
$$

$$
\text{(Lemma C.4, and } \|\hat\theta_f\|_\infty \leq 1 \text{ according to the proof in Lemma C.1)}
$$

$$
\leq 2 \left( \mathcal{L}_{\lambda, \mathcal{D}}(\hat\phi, \hat\theta_f, f) - \mathcal{L}_{\lambda, \mathcal{D}}(\phi^\star, \theta_f^\star, f) + \frac{\lambda}{n} \|\theta_f^\star\|_2^2 \right) + \varepsilon'
$$

$$
\leq \varepsilon' + \frac{2\lambda d}{n}, \qquad \text{(by the optimality of } \hat\phi, \hat\theta_f \text{ under } \mathcal{L}_{\lambda, \mathcal{D}}(\cdot, \cdot, f)\text{)}
$$

which means the inequality in the lemma statement holds. Here, we use $\|\theta_f^\star\|_2^2 \leq d$. $\qquad \square$

**Lemma C.6.** *When $\hat{P}_h^{(n)}$ is computed using Alg. 3 and the Markov games is a block Markov game, if we set*

$$
\lambda = \Theta \left( d \log \frac{NH|\Phi|}{\delta} \right), \quad \zeta^{(n)} = \Theta \left( \frac{d^2 M \log \frac{dNHML|\Phi|}{\delta \tilde\varepsilon}}{n} \right).
$$

*then $\mathcal{E}$ holds with probability at least $1 - \delta$.*

*Proof.* Combining Lemma C.5 and Lemma C.3, we have that

$$
\max_{f \in \mathcal{F}_h} \mathbb{E}_\rho \left[ (\hat\phi(s, \boldsymbol{a})^\top \hat\theta_f - \phi^\star(s, \boldsymbol{a})^\top \theta_f^\star)^2 \right] \leq \varepsilon' + \frac{2\lambda d}{n} \leq \zeta^{(n)} := \Theta \left( d^2 M \frac{\log \left( \frac{dNHML|\Phi|}{\delta \tilde\varepsilon} \right)}{n} \right),
$$

which shows $\mathcal{E}_1$ holds with a high probability. Combining this result with Lemma E.1, we have proved Lemma C.6. $\qquad \square$

### C.3 STATISTICAL GUARANTEES

To ensure the algorithm is well-defined, we first prove the following lemma which implies the optimistic Q-value estimators always belong to the function class $\tilde{\mathcal{F}}_h$.

**Lemma C.7.** *When $\alpha^{(n)} \leq L$, we have $\overline{Q}_{h,i}^{(n)} \in \tilde{\mathcal{F}}_h, \forall h \in [H], i \in [M], n \in [N]$.*

*Proof.* Because $\hat\beta_h^{(n)}$ is upper bounded by $H$, by induction one can easily get $\overline{V}_{h+1,i}^{(n)} \leq 2H^2$. Then according to the result of Lemma C.1, we know $(\hat{P}_h^{(n)} \overline{V}_{h+1,i}^{(n)})(s, a) = \phi_h^{(n)}(s, \boldsymbol{a})^\top \theta$ with $\|\theta\|_2 \leq 2H^2 \sqrt{d}$. We conclude $\overline{Q}_{h,i}^{(n)} \in \tilde{\mathcal{F}}_h$. $\qquad \square$

We will show later that our choice of $\alpha^{(n)}$ and $L$ always satisfies the condition $\alpha^{(n)} \leq L$.

**Lemma C.8.** *We have*

- *For NE and CCE,*

$$
\max_{\pi_{h,i}} \left( \mathbb{D}_{\pi_{h,i}, \pi_{h,-i}^{(n)}} \overline{Q}_{h,i}^{(n)} \right)(s) \leq \left( \mathbb{D}_{\pi_h^{(n)}} \overline{Q}_{h,i}^{(n)} \right)(s) + 2\tilde\varepsilon;
$$

- *For CE,*

$$
\max_{\omega_{h,i} \in \Omega_{h,i}} \left( \mathbb{D}_{\omega_{h,i} \circ \pi_h^{(n)}} \overline{Q}_{h,i}^{(n)} \right)(s) \leq \left( \mathbb{D}_{\pi_h^{(n)}} \overline{Q}_{h,i}^{(n)} \right)(s) + 2\tilde\varepsilon.
$$

*Proof.* We only prove the case of NE and CCE, the case of CE can be proved similarly. Let $\tilde{Q}_{h,i}^{(n)}$ be the nearest neighbour of $\overline{Q}_{h,i}^{(n)}$ in $\mathcal{N}_h$, we have

$$\max_{\pi_{h,i}} \left( \mathbb{D}_{\pi_{h,i},\pi_{h,-i}^{(n)}} \overline{Q}_{h,i}^{(n)} \right)(s) \leq \max_{\pi_{h,i}} \left( \mathbb{D}_{\pi_{h,i},\pi_{h,-i}^{(n)}} \tilde{Q}_{h,i}^{(n)} \right)(s) + \tilde{\varepsilon}$$

$$\leq \left( \mathbb{D}_{\pi_h^{(n)}} \tilde{Q}_{h,i}^{(n)} \right)(s) + \tilde{\varepsilon} \qquad \text{(Definition of } \pi_h^{(n)})$$

$$\leq \left( \mathbb{D}_{\pi_h^{(n)}} \overline{Q}_{h,i}^{(n)} \right)(s) + 2\tilde{\varepsilon},$$

which has finished the proof. $\qquad\square$

**Lemma C.9** (One-step back inequality for the learned model). *Suppose the event $\mathcal{E}$ holds. Consider a set of functions $\{g_h\}_{h=1}^H$ that satisfies $g_h \in \mathcal{S} \times \mathcal{A} \to \mathbb{R}_+$, s.t. $\|g_h\|_\infty \leq B$. For a given policy $\pi$, suppose $\mathbb{E}_{\boldsymbol{a}\sim U(\mathcal{A})}[g_h(\cdot,\boldsymbol{a})] \in \mathcal{F}_{1,h}$, then we have*

$$\left| \mathbb{E}_{(s,\boldsymbol{a})\sim d_{\hat{P}^{(n)},h}^\pi} [g_h(s,\boldsymbol{a})] \right|$$

$$\leq \begin{cases} \sqrt{A\mathbb{E}_{(s,\boldsymbol{a})\sim\rho_1^{(n)}}[g_1^2(s,\boldsymbol{a})]}, & h=1 \\ \mathbb{E}_{(\tilde{s},\tilde{\boldsymbol{a}})\sim d_{\hat{P}^{(n)},h-1}^\pi} \left[ \min\left\{ \left\|\hat{\phi}_{h-1}^{(n)}(\tilde{s},\tilde{\boldsymbol{a}})\right\|_{\Sigma_{n,\rho_{h-1}^{(n)},\hat{\phi}_{h-1}^{(n)}}^{-1}} \sqrt{nA^2\mathbb{E}_{(s,\boldsymbol{a})\sim\tilde{\rho}_h^{(n)}}[g_h^2(s,\boldsymbol{a})] + B^2\lambda d + nA^2\zeta^{(n)}}, B \right\} \right], & h \geq 2 \end{cases}$$

Recall $\Sigma_{n,\rho_h^{(n)},\hat{\phi}_h^{(n)}} = n\mathbb{E}_{(s,\boldsymbol{a})\sim\rho_h^{(n)}}\left[\hat{\phi}_h^{(n)}(s,\boldsymbol{a})\hat{\phi}_h^{(n)}(s,\boldsymbol{a})^\top\right] + \lambda I_d$.

*Proof.* For step $h=1$, we have

$$\mathbb{E}_{(s,\boldsymbol{a})\sim d_{\hat{P}^{(n)},1}^\pi}[g_1(s,\boldsymbol{a})] = \mathbb{E}_{s\sim d_1,\boldsymbol{a}\sim\pi_1(s)}[g_1(s,\boldsymbol{a})]$$

$$\leq \sqrt{\max_{(s,\boldsymbol{a})} \frac{d_1(s)\pi_1(\boldsymbol{a}|s)}{\rho_1^{(n)}(s,\boldsymbol{a})} \mathbb{E}_{(s',\boldsymbol{a}')\sim\rho_1^{(n)}}[g_1^2(s',\boldsymbol{a}')]}$$

$$= \sqrt{\max_{(s,\boldsymbol{a})} \frac{d_1(s)\pi_1(\boldsymbol{a}|s)}{d_1(s)u_{\mathcal{A}}(\boldsymbol{a})} \mathbb{E}_{(s',\boldsymbol{a}')\sim\rho_1^{(n)}}[g_1^2(s',\boldsymbol{a}')]}$$

$$\leq \sqrt{A\mathbb{E}_{(s,\boldsymbol{a})\sim\rho_1^{(n)}}[g_1^2(s,\boldsymbol{a})]}.$$

For step $h=2,\ldots,H-1$, we observe the following one-step-back decomposition:

$$\mathbb{E}_{(s,\boldsymbol{a})\sim d_{\hat{P}^{(n)},h}^\pi}[g_h(s,\boldsymbol{a})]$$

$$= \mathbb{E}_{(\tilde{s},\tilde{\boldsymbol{a}})\sim d_{\hat{P}^{(n)},h-1}^\pi, s\sim\hat{P}_{h-1}^{(n)}(\tilde{s},\tilde{\boldsymbol{a}}),\boldsymbol{a}\sim\pi_h(s)}[g_h(s,\boldsymbol{a})]$$

$$= \mathbb{E}_{(\tilde{s},\tilde{\boldsymbol{a}})\sim d_{\hat{P}^{(n)},h-1}^\pi} \left[ \hat{\phi}_{h-1}^{(n)}(\tilde{s},\tilde{\boldsymbol{a}})^\top \int_{\mathcal{S}} \sum_{\boldsymbol{a}\in\mathcal{A}} \hat{w}_{h-1}^{(n)}(s)\pi_h(\boldsymbol{a}|s)g_h(s,\boldsymbol{a})\mathrm{d}s \right]$$

$$= \mathbb{E}_{(\tilde{s},\tilde{\boldsymbol{a}})\sim d_{\hat{P}^{(n)},h-1}^\pi} \left[ \min\left\{ \hat{\phi}_{h-1}^{(n)}(\tilde{s},\tilde{\boldsymbol{a}})^\top \int_{\mathcal{S}} \sum_{\boldsymbol{a}\in\mathcal{A}} \hat{w}_{h-1}^{(n)}(s)\pi_h(\boldsymbol{a}|s)g_h(s,\boldsymbol{a})\mathrm{d}s, B \right\} \right]$$

$$\leq \mathbb{E}_{(\tilde{s},\tilde{\boldsymbol{a}})\sim d_{\hat{P}^{(n)},h-1}^\pi} \left[ \min\left\{ \left\|\hat{\phi}_{h-1}^{(n)}(\tilde{s},\tilde{\boldsymbol{a}})\right\|_{\Sigma_{n,\rho_{h-1}^{(n)},\hat{\phi}_{h-1}^{(n)}}^{-1}} \left\| \int_{\mathcal{S}} \sum_{\boldsymbol{a}\in\mathcal{A}} \hat{w}_{h-1}^{(n)}(s)\pi_h(\boldsymbol{a}|s)g_h(s,\boldsymbol{a})\mathrm{d}s \right\|_{\Sigma_{n,\rho_{h-1}^{(n)},\hat{\phi}_{h-1}^{(n)}}}, B \right\} \right].$$

where we use the fact that $g_h$ is bounded by $B$. Then,

$$\left\| \int_{\mathcal{S}} \sum_{\boldsymbol{a}\in\mathcal{A}} \hat{w}_{h-1}^{(n)}(s)\pi_h(\boldsymbol{a}|s)g_h(s,\boldsymbol{a})\mathrm{d}s \right\|_{\Sigma_{n,\rho_{h-1}^{(n)},\hat{\phi}_{h-1}^{(n)}}}^2$$

$$\leq \left( \int_{\mathcal{S}} \sum_{\boldsymbol{a} \in \mathcal{A}} \hat{w}_{h-1}^{(n)}(s) \pi_h(\boldsymbol{a}|s) g_h(s, \boldsymbol{a}) \mathrm{d}s \right)^{\top} \left( n \mathbb{E}_{(s,\boldsymbol{a}) \sim \rho_{h-1}^{(n)}} \left[ \hat{\phi}_{h-1}^{(n)}(s, \boldsymbol{a}) \hat{\phi}_{h-1}^{(n)}(s, \boldsymbol{a})^{\top} \right] + \lambda I_d \right) \left( \int_{\mathcal{S}} \sum_{\boldsymbol{a} \in \mathcal{A}} \hat{w}_{h-1}^{(n)}(s) \pi_h(\boldsymbol{a}|s) g_h(s, \boldsymbol{a}) \mathrm{d}s \right)$$

$$\leq n \mathbb{E}_{(\tilde{s}, \tilde{\boldsymbol{a}}) \sim \rho_{h-1}^{(n)}} \left[ \left( \int_{\mathcal{S}} \sum_{\boldsymbol{a} \in \mathcal{A}} \hat{w}_{h-1}^{(n)}(s)^{\top} \hat{\phi}_{h-1}^{(n)}(\tilde{s}, \tilde{\boldsymbol{a}}) \pi_h(\boldsymbol{a}|s) g_h(s, \boldsymbol{a}) \mathrm{d}s \right)^2 \right] + B^2 \lambda d$$

$(\|\sum_{\boldsymbol{a} \in \mathcal{A}} \pi_h(\boldsymbol{a}|s) g_h(s, \boldsymbol{a})\|_{\infty} \leq B$ and by Lemma C.1 $\left\| \int_{\mathcal{S}} \hat{w}_{h-1}^{(n)}(s) l(s) \mathrm{d}s \right\|_2 \leq \sqrt{d}$ for any $l : \mathcal{S} \to [0, 1]$.)

$$= n \mathbb{E}_{(\tilde{s}, \tilde{\boldsymbol{a}}) \sim \rho_{h-1}^{(n)}} \left[ \left( \mathbb{E}_{s \sim \hat{P}_{h-1}^{(n)}(\tilde{s}, \tilde{\boldsymbol{a}}), \boldsymbol{a} \sim \pi_h(s)} [g_h(s, \boldsymbol{a})] \right)^2 \right] + B^2 \lambda d$$

$$\leq n A^2 \mathbb{E}_{(\tilde{s}, \tilde{\boldsymbol{a}}) \sim \rho_{h-1}^{(n)}} \left[ \left( \mathbb{E}_{s \sim \hat{P}_{h-1}^{(n)}(\tilde{s}, \tilde{\boldsymbol{a}}), \boldsymbol{a} \sim U(\mathcal{A})} [g_h(s, \boldsymbol{a})] \right)^2 \right] + B^2 \lambda d \qquad \text{(Importance sampling)}$$

$$\leq n A^2 \mathbb{E}_{(\tilde{s}, \tilde{\boldsymbol{a}}) \sim \rho_{h-1}^{(n)}} \left[ \left( \mathbb{E}_{s \sim P_{h-1}^{\star}(\tilde{s}, \tilde{\boldsymbol{a}}), \boldsymbol{a} \sim U(\mathcal{A})} [g_h(s, \boldsymbol{a})] \right)^2 \right] + B^2 \lambda d + n A^2 \xi^{(n)} \qquad \text{(Assumption on } g_h)$$

$$\leq n A^2 \mathbb{E}_{(\tilde{s}, \tilde{\boldsymbol{a}}) \sim \rho_{h-1}^{(n)}, s \sim P_{h-1}^{\star}(\tilde{s}, \tilde{\boldsymbol{a}}), \boldsymbol{a} \sim U(\mathcal{A})} [g_h^2(s, \boldsymbol{a})] + B^2 \lambda d + n A^2 \xi^{(n)}. \qquad \text{(Jensen)}$$

$$\leq n A^2 \mathbb{E}_{(s,\boldsymbol{a}) \sim \tilde{\rho}_h^{(n)}} [g_h^2(s, \boldsymbol{a})] + B^2 \lambda d + n A^2 \zeta^{(n)}. \qquad \text{(Definition of } \tilde{\rho}_h^{(n)})$$

Combing the above results together, we get

$$\mathbb{E}_{(s,\boldsymbol{a}) \sim d_{\hat{P}^{(n)}, h}^{\pi}} [g_h(s, \boldsymbol{a})]$$

$$\leq \mathbb{E}_{(\tilde{s}, \tilde{\boldsymbol{a}}) \sim d_{\hat{P}^{(n)}, h-1}^{\pi}} \left[ \min \left\{ \left\| \hat{\phi}_{h-1}^{(n)}(\tilde{s}, \tilde{\boldsymbol{a}}) \right\|_{\Sigma_{n, \rho_{h-1}^{(n)}, \hat{\phi}_{h-1}^{(n)}}^{-1}} \left\| \int_{\mathcal{S}} \sum_{\boldsymbol{a} \in \mathcal{A}} \hat{w}_{h-1}^{(n)}(s) \pi_h(\boldsymbol{a}|s) g_h(s, \boldsymbol{a}) \mathrm{d}s \right\|_{\Sigma_{n, \rho_{h-1}^{(n)}, \hat{\phi}_{h-1}^{(n)}}}, B \right\} \right]$$

$$\leq \mathbb{E}_{(\tilde{s}, \tilde{\boldsymbol{a}}) \sim d_{\hat{P}^{(n)}, h-1}^{\pi}} \left[ \min \left\{ \left\| \hat{\phi}_{h-1}^{(n)}(\tilde{s}, \tilde{\boldsymbol{a}}) \right\|_{\Sigma_{n, \rho_{h-1}^{(n)}, \hat{\phi}_{h-1}^{(n)}}^{-1}} \sqrt{n A^2 \mathbb{E}_{(s,\boldsymbol{a}) \sim \tilde{\rho}_h^{(n)}} [g_h^2(s, \boldsymbol{a})] + B^2 \lambda d + n A^2 \zeta^{(n)}}, B \right\} \right],$$

which has finished the proof. $\qquad\qquad \square$

The following lemma is an exact copy of Lemma B.4, and here we state it again just for completeness.

**Lemma C.10** (One-step back inequality for the true model). *Consider a set of functions* $\{g_h\}_{h=1}^H$ *that satisfies* $g_h \in \mathcal{S} \times \mathcal{A} \to \mathbb{R}_+$, *s.t.* $\|g_h\|_{\infty} \leq B$. *Then for any given policy* $\pi$, *we have*

$$\left| \mathbb{E}_{(s,\boldsymbol{a}) \sim d_{P^{\star}, h}^{\pi}} [g_h(s, \boldsymbol{a})] \right|$$

$$\leq \begin{cases} \sqrt{A \mathbb{E}_{(s,\boldsymbol{a}) \sim \rho_1^{(n)}} [g_1^2(s, \boldsymbol{a})]}, & h = 1 \\ \mathbb{E}_{(\tilde{s}, \tilde{\boldsymbol{a}}) \sim d_{P^{\star}, h-1}^{\pi}} \left[ \left\| \phi_{h-1}^{\star}(\tilde{s}, \tilde{\boldsymbol{a}}) \right\|_{\Sigma_{n, \gamma_{h-1}^{(n)}, \phi_{h-1}^{\star}}^{-1}} \right] \sqrt{n A \mathbb{E}_{(s,\boldsymbol{a}) \sim \tilde{\rho}_h^{(n)}} [g_h^2(s, \boldsymbol{a})] + B^2 \lambda d}, & h \geq 2 \end{cases}$$

Recall $\Sigma_{n, \gamma_h^{(n)}, \phi_h^{\star}} = n \mathbb{E}_{(s,\boldsymbol{a}) \sim \gamma_h^{(n)}} \left[ \phi_h^{\star}(s, \boldsymbol{a}) \phi_h^{\star}(s, \boldsymbol{a})^{\top} \right] + \lambda I_d$.

**Lemma C.11** (Optimism for NE and CCE). *Consider an episode* $n \in [N]$ *and set* $\alpha^{(n)} = \Theta \left( H \sqrt{n A^2 \zeta^{(n)} + d\lambda} \right)$. *When the event* $\mathcal{E}$ *holds and the policy* $\pi^{(n)}$ *is computed by solving NE or CCE, we have*

$$\overline{v}_i^{(n)}(s) - v_i^{\dagger, \pi_{-i}^{(n)}}(s) \geq -H \sqrt{A \zeta^{(n)}} - 2H \tilde{\varepsilon}, \quad \forall n \in [N], i \in [M].$$

*Proof.* Denote $\tilde{\mu}_{h,i}^{(n)}(\cdot|s) := \arg\max_{\mu} \left( \mathbb{D}_{\mu, \pi_{h,-i}^{(n)}} Q_{h,i}^{\dagger, \pi_{-i}^{(n)}} \right)(s)$ and let $\tilde{\pi}_h^{(n)} = \tilde{\mu}_{h,i}^{(n)} \times \pi_{h,-i}^{(n)}$. Let $f_h^{(n)}(s, \boldsymbol{a}) = \left| \frac{1}{H} \left( \hat{P}_h^{(n)} - P_h^{\star} \right) V_{h+1,i}^{\dagger, \pi_{-i}^{(n)}} \right| (s, \boldsymbol{a})$, note that by definition, we have $\frac{1}{H} V_{h+1,i}^{\dagger, \pi_{-i}^{(n)}}(s)$ is

bounded by 1, and

$$
\frac{1}{H} V_{h+1,i}^{\dagger,\pi_{-i}^{(n)}}(s) = \mathbb{E}_{\boldsymbol{a}\sim\tilde{\pi}_h^{(n)}(s)}\left[\frac{r_{h+1,i}(s,\boldsymbol{a})}{H} + \frac{1}{H}\left(P_{h+1}^\star V_{h+2,i}^{\dagger,\pi_{-i}^{(n)}}\right)(s,\boldsymbol{a})\right]
$$

$$
= \max_{\mu_{h+1,i}} \mathbb{E}_{\boldsymbol{a}\sim(\mu_{h+1,i}\times\pi_{h+1,-i}^{(n)})(s)}\left[\frac{r_{h+1,i}(s,\boldsymbol{a})}{H} + \frac{1}{H}\left(P_{h+1}^\star V_{h+2,i}^{\dagger,\pi_{-i}^{(n)}}\right)(s,\boldsymbol{a})\right] \in \mathcal{F}_{3,h}.
$$

where we use the result of Lemma C.1 and get $\frac{1}{H}\left(P_{h+1}^\star V_{h+2,i}^{\dagger,\pi_{-i}^{(n)}}\right)(s,\boldsymbol{a})$ is a linear function in $\phi_{h+1}^\star$ and the 2-norm of the weight is upper bounded by $\sqrt{d}$. Then according to the event $\mathcal{E}$, we have

$$
\mathbb{E}_{(s,\boldsymbol{a})\sim\rho_h^{(n)}}\left[\left(f_h^{(n)}(s,\boldsymbol{a})\right)^2\right] \le \zeta^{(n)}, \quad \mathbb{E}_{(s,\boldsymbol{a})\sim\tilde{\rho}_h^{(n)}}\left[\left(f_h^{(n)}(s,\boldsymbol{a})\right)^2\right] \le \zeta^{(n)}, \quad \forall n\in[N], h\in[H]
$$

$$
\|\phi_h(s,\boldsymbol{a})\|_{\left(\hat{\Sigma}_{h,\phi_h}^{(n)}\right)^{-1}} = \Theta\left(\|\phi_h(s,\boldsymbol{a})\|_{\Sigma_{n,\rho_h^{(n)},\phi_h}^{-1}}\right), \quad \forall n\in[N], h\in[H], \phi_h\in\Phi_h.
$$

A direct conclusion of the event $\mathcal{E}$ is we can find an absolute constant $c$, such that

$$
\beta_h^{(n)}(s,\boldsymbol{a}) = \min\left\{\alpha^{(n)}\left\|\hat{\phi}_h^{(n)}(\tilde{s},\tilde{\boldsymbol{a}})\right\|_{\left(\Sigma_{h,\hat{\phi}_h^{(n)}}^{(n)}\right)^{-1}}, H\right\}
$$

$$
\ge \min\left\{c\alpha^{(n)}\left\|\hat{\phi}_h^{(n)}(\tilde{s},\tilde{\boldsymbol{a}})\right\|_{\Sigma_{n,\rho_h^{(n)},\hat{\phi}_h^{(n)}}^{-1}}, H\right\}, \quad \forall n\in[N], h\in[H].
$$

Next, we prove by induction that

$$
\mathbb{E}_{s\sim d_{\hat{P}^{(n)},h}^{\tilde{\pi}^{(n)}}}\left[\overline{V}_{h,i}^{(n)}(s) - V_{h,i}^{\dagger,\pi_{-i}^{(n)}}(s)\right]
$$

$$
\ge \sum_{h'=h}^{H} \mathbb{E}_{(s,\boldsymbol{a})\sim d_{\hat{P}^{(n)},h'}^{\tilde{\pi}^{(n)}}}\left[\hat{\beta}_{h'}^{(n)}(s,\boldsymbol{a}) - Hf_{h'}^{(n)}(s,\boldsymbol{a})\right] - 2(H-h+1)\tilde{\varepsilon}, \quad \forall h\in[H]. \tag{16}
$$

First, notice that $\forall h\in[H]$,

$$
\mathbb{E}_{s\sim d_{\hat{P}^{(n)},h}^{\tilde{\pi}^{(n)}}}\left[\overline{V}_{h,i}^{(n)}(s) - V_{h,i}^{\dagger,\pi_{-i}^{(n)}}(s)\right] = \mathbb{E}_{s\sim d_{\hat{P}^{(n)},h}^{\tilde{\pi}^{(n)}}}\left[\left(\mathbb{D}_{\pi_h^{(n)}}\overline{Q}_{h,i}^{(n)}\right)(s) - \left(\mathbb{D}_{\tilde{\pi}_h^{(n)}}Q_{h,i}^{\dagger,\pi_{-i}^{(n)}}\right)(s)\right]
$$

$$
\ge \mathbb{E}_{s\sim d_{\hat{P}^{(n)},h}^{\tilde{\pi}^{(n)}}}\left[\left(\mathbb{D}_{\tilde{\pi}_h^{(n)}}\overline{Q}_{h,i}^{(n)}\right)(s) - \left(\mathbb{D}_{\tilde{\pi}_h^{(n)}}Q_{h,i}^{\dagger,\pi_{-i}^{(n)}}\right)(s)\right] - 2\tilde{\varepsilon}
$$

$$
= \mathbb{E}_{(s,\boldsymbol{a})\sim d_{\hat{P}^{(n)},h}^{\tilde{\pi}^{(n)}}}\left[\overline{Q}_{h,i}^{(n)}(s,\boldsymbol{a}) - Q_{h,i}^{\dagger,\pi_{-i}^{(n)}}(s,\boldsymbol{a})\right] - 2\tilde{\varepsilon},
$$

where the inequality uses the result of Lemma C.8. Now we are ready to prove equation 16,

- When $h = H$, we have

$$
\mathbb{E}_{s\sim d_{\hat{P}^{(n)},H}^{\tilde{\pi}^{(n)}}}\left[\overline{V}_{H,i}^{(n)}(s) - V_{H,i}^{\dagger,\pi_{-i}^{(n)}}(s)\right] \ge \mathbb{E}_{(s,\boldsymbol{a})\sim d_{\hat{P}^{(n)},H}^{\tilde{\pi}^{(n)}}}\left[\overline{Q}_{H,i}^{(n)}(s,\boldsymbol{a}) - Q_{H,i}^{\dagger,\pi_{-i}^{(n)}}(s,\boldsymbol{a})\right] - 2\tilde{\varepsilon}
$$

$$
= \mathbb{E}_{(s,\boldsymbol{a})\sim d_{\hat{P}^{(n)},H}^{\tilde{\pi}^{(n)}}}\left[\hat{\beta}_h^{(n)}(s,\boldsymbol{a})\right] - 2\tilde{\varepsilon}
$$

$$
\ge \mathbb{E}_{(s,\boldsymbol{a})\sim d_{\hat{P}^{(n)},H}^{\tilde{\pi}^{(n)}}}\left[\hat{\beta}_h^{(n)}(s,\boldsymbol{a}) - Hf_H^{(n)}(s,\boldsymbol{a})\right] - 2\tilde{\varepsilon}.
$$

- Suppose the statement is true for $h+1$, then for step $h$, we have

$$
\mathbb{E}_{s\sim d_{\hat{P}^{(n)},h}^{\tilde{\pi}^{(n)}}}\left[\overline{V}_{h,i}^{(n)}(s) - V_{h,i}^{\dagger,\pi_{-i}^{(n)}}(s)\right]
$$

$$\geq \mathbb{E}_{(s,\boldsymbol{a})\sim d_{\hat{P}^{(n)},h}^{\tilde{\pi}^{(n)}}} \left[ \overline{Q}_{h,i}^{(n)}(s,\boldsymbol{a}) - Q_{h,i}^{\dagger,\pi_{-i}^{(n)}}(s,\boldsymbol{a}) \right] - 2\tilde{\varepsilon}$$

$$= \mathbb{E}_{(s,\boldsymbol{a})\sim d_{\hat{P}^{(n)},h}^{\tilde{\pi}^{(n)}}} \left[ \hat{\beta}_h^{(n)}(s,\boldsymbol{a}) + \left( \hat{P}_h^{(n)} \overline{V}_{h+1,i}^{(n)} \right)(s,\boldsymbol{a}) - \left( P_h^{\star} V_{h+1,i}^{\dagger,\pi_{-i}^{(n)}} \right)(s,\boldsymbol{a}) \right] - 2\tilde{\varepsilon}$$

$$= \mathbb{E}_{(s,\boldsymbol{a})\sim d_{\hat{P}^{(n)},h}^{\tilde{\pi}^{(n)}}} \left[ \hat{\beta}_h^{(n)}(s,\boldsymbol{a}) + \left( \hat{P}_h^{(n)} \left( \overline{V}_{h+1,i}^{(n)} - V_{h+1,i}^{\dagger,\pi_{-i}^{(n)}} \right) \right)(s,\boldsymbol{a}) + \left( \left( \hat{P}_h^{(n)} - P_h^{\star} \right) V_{h+1,i}^{\dagger,\pi_{-i}^{(n)}} \right)(s,\boldsymbol{a}) \right] - 2\tilde{\varepsilon}$$

$$= \mathbb{E}_{(s,\boldsymbol{a})\sim d_{\hat{P}^{(n)},h}^{\tilde{\pi}^{(n)}}} \left[ \hat{\beta}_h^{(n)}(s,\boldsymbol{a}) + \left( \left( \hat{P}_h^{(n)} - P_h^{\star} \right) V_{h+1,i}^{\dagger,\pi_{-i}^{(n)}} \right)(s,\boldsymbol{a}) \right] + \mathbb{E}_{s\sim d_{\hat{P}^{(n)},h+1}^{\tilde{\pi}^{(n)}}} \left[ \overline{V}_{h+1,i}^{(n)}(s) - V_{h+1,i}^{\dagger,\pi_{-i}^{(n)}}(s) \right] - 2\tilde{\varepsilon}$$

$$\geq \mathbb{E}_{(s,\boldsymbol{a})\sim d_{\hat{P}^{(n)},h}^{\tilde{\pi}^{(n)}}} \left[ \hat{\beta}_h^{(n)}(s,\boldsymbol{a}) - H f_h^{(n)}(s,\boldsymbol{a}) \right] + \mathbb{E}_{s\sim d_{\hat{P}^{(n)},h+1}^{\tilde{\pi}^{(n)}}} \left[ \overline{V}_{h+1,i}^{(n)}(s) - V_{h+1,i}^{\dagger,\pi_{-i}^{(n)}}(s) \right] - 2\tilde{\varepsilon}$$

$$\geq \sum_{h'=h}^{H} \mathbb{E}_{(s,\boldsymbol{a})\sim d_{\hat{P}^{(n)},h'}^{\tilde{\pi}^{(n)}}} \left[ \hat{\beta}_{h'}^{(n)}(s,\boldsymbol{a}) - H f_{h'}^{(n)}(s,\boldsymbol{a}) \right] - 2(H-h+1)\tilde{\varepsilon},$$

where the last row uses the induction assumption.

Therefore, we have proved equation 16. We then apply $h = 1$ to equation 16, and get

$$\mathbb{E}_{s\sim d_1} \left[ \overline{V}_{1,i}^{(n)}(s) - V_{1,i}^{\dagger,\pi_{-i}^{(n)}}(s) \right]$$

$$= \mathbb{E}_{s\sim d_{\hat{P}^{(n)},1}^{\tilde{\pi}^{(n)}}} \left[ \overline{V}_{1,i}^{(n)}(s) - V_{1,i}^{\dagger,\pi_{-i}^{(n)}}(s) \right]$$

$$\geq \sum_{h=1}^{H} \mathbb{E}_{(s,\boldsymbol{a})\sim d_{\hat{P}^{(n)},h}^{\tilde{\pi}^{(n)}}} \left[ \hat{\beta}_h^{(n)}(s,\boldsymbol{a}) - H f_h^{(n)}(s,\boldsymbol{a}) \right] - 2H\tilde{\varepsilon}$$

$$= \sum_{h=1}^{H} \mathbb{E}_{(s,\boldsymbol{a})\sim d_{\hat{P}^{(n)},h}^{\tilde{\pi}^{(n)}}} \left[ \hat{\beta}_h^{(n)}(s,\boldsymbol{a}) \right] - H \sum_{h=1}^{H} \mathbb{E}_{(s,\boldsymbol{a})\sim d_{\hat{P}^{(n)},h}^{\tilde{\pi}^{(n)}}} \left[ f_h^{(n)}(s,\boldsymbol{a}) \right] - 2H\tilde{\varepsilon}.$$

For the second term, since $\frac{1}{H}\hat{P}_h^{(n)} V_{h+1,i}^{\dagger,\pi_{-i}^{(n)}}$ is linear in $\hat{\phi}_h^{(n)}$ and $\frac{1}{H} P_h^{\star} V_{h+1,i}^{\dagger,\pi_{-i}^{(n)}}$ is linear in $\phi_h^{\star}$, and according to the result of Lemma C.1, the 2-norm of their weights are both upper bounded by $\sqrt{d}$. Therefore, we have $\mathbb{E}_{\boldsymbol{a}\sim U(\mathcal{A})} \left[ f_h^{(n)}(\cdot,\boldsymbol{a}) \right] \in \mathcal{F}_{1,h}$. By Lemma C.9, we have for $h = 1$,

$$\mathbb{E}_{(s,\boldsymbol{a})\sim d_{\hat{P}^{(n)},1}^{\tilde{\pi}^{(n)}}} \left[ f_1^{(n)}(s,\boldsymbol{a}) \right] \leq \sqrt{A \mathbb{E}_{(s,\boldsymbol{a})\sim \rho_1^{(n)}} \left[ \left( f_1^{(n)}(s,\boldsymbol{a}) \right)^2 \right]} \leq \sqrt{A\zeta^{(n)}}.$$

And $\forall h \geq 2$, we have

$$\mathbb{E}_{(s,\boldsymbol{a})\sim d_{\hat{P}^{(n)},h}^{\tilde{\pi}^{(n)}}} \left[ f_h^{(n)}(s,\boldsymbol{a}) \right]$$

$$\leq \mathbb{E}_{(\tilde{s},\tilde{\boldsymbol{a}})\sim d_{\hat{P}^{(n)},h-1}^{\tilde{\pi}^{(n)}}} \left[ \min \left\{ \left\| \hat{\phi}_{h-1}^{(n)}(\tilde{s},\tilde{\boldsymbol{a}}) \right\|_{\Sigma_{n,\rho_{h-1}^{(n)},\hat{\phi}_{h-1}^{(n)}}^{-1}} \sqrt{nA^2 \mathbb{E}_{(s,\boldsymbol{a})\sim \tilde{\rho}_h^{(n)}} \left[ \left( f_h^{(n)}(s,\boldsymbol{a}) \right)^2 \right] + d\lambda + nA^2\zeta^{(n)}}, 1 \right\} \right]$$

$$\lesssim \mathbb{E}_{(\tilde{s},\tilde{\boldsymbol{a}})\sim d_{\hat{P}^{(n)},h-1}^{\tilde{\pi}^{(n)}}} \left[ \min \left\{ \left\| \hat{\phi}_{h-1}^{(n)}(\tilde{s},\tilde{\boldsymbol{a}}) \right\|_{\Sigma_{n,\rho_{h-1}^{(n)},\hat{\phi}_{h-1}^{(n)}}^{-1}} \sqrt{nA^2\zeta^{(n)} + d\lambda}, 1 \right\} \right].$$

Note that we here use $f_h^{(n)}(s,\boldsymbol{a}) \leq 1$, $\mathbb{E}_{(s,\boldsymbol{a})\sim \rho_h^{(n)}} \left[ \left( f_h^{(n)}(s,\boldsymbol{a}) \right)^2 \right] \leq \zeta^{(n)}$ and $\mathbb{E}_{(s,\boldsymbol{a})\sim \tilde{\rho}_h^{(n)}} \left[ \left( f_h^{(n)}(s,\boldsymbol{a}) \right)^2 \right] \leq \zeta^{(n)}$. Then according to our choice of $\alpha^{(n)}$, we get

$$\mathbb{E}_{(s,\boldsymbol{a})\sim d_{\hat{P}^{(n)},h}^{\tilde{\pi}^{(n)}}} \left[ f_h^{(n)}(s,\boldsymbol{a}) \right] \leq \mathbb{E}_{(\tilde{s},\tilde{\boldsymbol{a}})\sim d_{\hat{P}^{(n)},h-1}^{\tilde{\pi}^{(n)}}} \left[ \min \left\{ \frac{c\alpha^{(n)}}{H} \left\| \hat{\phi}_{h-1}^{(n)}(\tilde{s},\tilde{\boldsymbol{a}}) \right\|_{\Sigma_{n,\rho_{h-1}^{(n)},\hat{\phi}_{h-1}^{(n)}}^{-1}}, 1 \right\} \right].$$

Combining all things together,

$$
\begin{aligned}
\overline{v}_i^{(n)} - v_i^{\dagger,\pi_{-i}^{(n)}} &= \mathbb{E}_{s\sim d_1}\left[\overline{V}_{1,i}^{(n)}(s) - V_{1,i}^{\dagger,\pi_{-i}^{(n)}}(s)\right] \\
&\geq \sum_{h=1}^{H} \mathbb{E}_{(s,\boldsymbol{a})\sim d_{\hat{P}^{(n)},h}^{\tilde{\pi}^{(n)}}}\left[\hat{\beta}_h^{(n)}(s,\boldsymbol{a})\right] - H\sum_{h=1}^{H}\mathbb{E}_{(s,\boldsymbol{a})\sim d_{\hat{P}^{(n)},h}^{\tilde{\pi}^{(n)}}}\left[f_h^{(n)}(s,\boldsymbol{a})\right] - 2H\tilde{\varepsilon} \\
&\geq \sum_{h=1}^{H-1}\mathbb{E}_{(\tilde{s},\tilde{\boldsymbol{a}})\sim d_{\hat{P}^{(n)},h}^{\tilde{\pi}^{(n)}}}\left[\hat{\beta}_h^{(n)}(s,\boldsymbol{a}) - \min\left\{c\alpha^{(n)}\left\|\hat{\phi}_h^{(n)}(\tilde{s},\tilde{\boldsymbol{a}})\right\|_{\Sigma_{n,\rho_h^{(n)},\hat{\phi}_h^{(n)}}^{-1}}, H\right\}\right] - H\sqrt{A\zeta^{(n)}} - 2H\tilde{\varepsilon} \\
&= -H\sqrt{A\zeta^{(n)}} - 2H\tilde{\varepsilon},
\end{aligned}
$$

which proves the inequality. $\qquad\square$

**Lemma C.12** (Optimism for CE). *Consider an episode* $n \in [N]$ *and set* $\alpha^{(n)} = \Theta\left(H\sqrt{nA^2\zeta^{(n)}+d\lambda}\right)$*. When the event* $\mathcal{E}$ *holds, we have*

$$
\overline{v}_i^{(n)}(s) - \max_{\omega\in\Omega_i} v_i^{\omega\circ\pi^{(n)}}(s) \geq -H\sqrt{A\zeta^{(n)}} - 2H\tilde{\varepsilon}, \quad \forall n\in[N], i\in[M].
$$

*Proof.* Denote $\tilde{\omega}_{h,i}^{(n)} = \arg\max_{\omega_h\in\Omega_{h,i}}\left(\mathbb{D}_{\omega_h\circ\pi_h^{(n)}}\max_{\omega\in\Omega_i}Q_{h,i}^{\omega\circ\pi^{(n)}}\right)(s)$ and let $\tilde{\pi}_h^{(n)} = \tilde{\omega}_{h,i} \circ \pi_h^{(n)}$. Let $f_h^{(n)}(s,\boldsymbol{a}) = \left|\frac{1}{H}\left(\hat{P}_h^{(n)} - P_h^{\star}\right)\max_{\omega\in\Omega_i}V_{h+1,i}^{\omega\circ\pi^{(n)}}\right|(s,\boldsymbol{a})$, note that by definition, we have $\frac{1}{H}\max_{\omega\in\Omega_i}V_{h+1,i}^{\omega\circ\pi^{(n)}}(s)$ is bounded by 1, and

$$
\frac{1}{H}\max_{\omega\in\Omega_i}V_{h+1,i}^{\omega\circ\pi^{(n)}}(s) = \max_{\omega_{h+1,i}\in\Omega_{h+1,i}}\mathbb{E}_{\boldsymbol{a}\sim(\omega_{h+1,i}\circ\pi_h)(s)}\left[\frac{r_{h+1,i}(s,\boldsymbol{a})}{H} + \frac{1}{H}\left(P_{h+1}^{\star}\max_{\omega\in\Omega_i}V_{h+2,i}^{\omega\circ\pi^{(n)}}\right)(s,\boldsymbol{a})\right] \in \mathcal{F}_{3,h}.
$$

where we use the result of Lemma C.1 and get $\frac{1}{H}\left(P_{h+1}^{\star}\max_{\omega\in\Omega_i}V_{h+2,i}^{\omega\circ\pi^{(n)}}\right)(s,\boldsymbol{a})$ is a linear function in $\phi_h^{\star}$ and the 2-norm of the weight is upper bounded by $\sqrt{d}$. Then according to the event $\mathcal{E}$, we have

$$
\mathbb{E}_{(s,\boldsymbol{a})\sim\rho_h^{(n)}}\left[\left(f_h^{(n)}(s,\boldsymbol{a})\right)^2\right] \leq \zeta^{(n)}, \quad \mathbb{E}_{(s,\boldsymbol{a})\sim\tilde{\rho}_h^{(n)}}\left[\left(f_h^{(n)}(s,\boldsymbol{a})\right)^2\right] \leq \zeta^{(n)}, \quad \forall n\in[N], h\in[H]
$$

$$
\|\phi_h(s,\boldsymbol{a})\|_{\left(\hat{\Sigma}_{h,\phi_h}^{(n)}\right)^{-1}} = \Theta\left(\|\phi_h(s,\boldsymbol{a})\|_{\Sigma_{n,\rho_h^{(n)},\phi_h}^{-1}}\right), \quad \forall n\in[N], h\in[H], \phi_h\in\Phi_h.
$$

A direct conclusion of the event $\mathcal{E}$ is we can find an absolute constant $c$, such that

$$
\begin{aligned}
\beta_h^{(n)}(s,\boldsymbol{a}) &= \min\left\{\alpha^{(n)}\left\|\hat{\phi}_h^{(n)}(\tilde{s},\tilde{\boldsymbol{a}})\right\|_{\left(\Sigma_{h,\hat{\phi}_h^{(n)}}^{(n)}\right)^{-1}}, H\right\} \\
&\geq \min\left\{c\alpha^{(n)}\left\|\hat{\phi}_h^{(n)}(\tilde{s},\tilde{\boldsymbol{a}})\right\|_{\Sigma_{n,\rho_h^{(n)},\hat{\phi}_h^{(n)}}^{-1}}, H\right\}, \quad \forall n\in[N], h\in[H].
\end{aligned}
$$

Next, we prove by induction that

$$
\begin{aligned}
&\mathbb{E}_{s\sim d_{\hat{P}^{(n)},h}^{\tilde{\pi}^{(n)}}}\left[\overline{V}_{h,i}^{(n)}(s) - \max_{\omega\in\Omega_i}V_{h,i}^{\omega\circ\pi^{(n)}}(s)\right] \\
&\geq \sum_{h'=h}^{H}\mathbb{E}_{(s,\boldsymbol{a})\sim d_{\hat{P}^{(n)},h'}^{\tilde{\pi}^{(n)}}}\left[\hat{\beta}_{h'}^{(n)}(s,\boldsymbol{a}) - Hf_{h'}^{(n)}(s,\boldsymbol{a})\right] - 2(H-h+1)\tilde{\varepsilon}, \quad \forall h\in[H]. \quad (17)
\end{aligned}
$$

First, notice that $\forall h\in[H]$,

$$
\mathbb{E}_{s\sim d_{\hat{P}^{(n)},h}^{\tilde{\pi}^{(n)}}}\left[\overline{V}_{h,i}^{(n)}(s) - \max_{\omega\in\Omega_i}V_{h,i}^{\omega\circ\pi^{(n)}}(s)\right] = \mathbb{E}_{s\sim d_{\hat{P}^{(n)},h}^{\tilde{\pi}^{(n)}}}\left[\left(\mathbb{D}_{\pi_h^{(n)}}\overline{Q}_{h,i}^{(n)}\right)(s) - \left(\mathbb{D}_{\tilde{\pi}_h^{(n)}}\max_{\omega\in\Omega_i}Q_{h,i}^{\omega\circ\pi^{(n)}}\right)(s)\right]
$$

$$\geq \mathbb{E}_{s \sim d^{\tilde{\pi}^{(n)}}_{\hat{P}^{(n)},h}} \left[ \left( \mathbb{D}_{\tilde{\pi}^{(n)}_h} \overline{Q}^{(n)}_{h,i} \right)(s) - \left( \mathbb{D}_{\tilde{\pi}^{(n)}_h} \max_{\omega \in \Omega_i} Q^{\omega \circ \pi^{(n)}}_{h,i} \right)(s) \right] - 2\tilde{\varepsilon}$$

$$= \mathbb{E}_{(s,\boldsymbol{a}) \sim d^{\tilde{\pi}^{(n)}}_{\hat{P}^{(n)},h}} \left[ \overline{Q}^{(n)}_{h,i}(s,\boldsymbol{a}) - \max_{\omega \in \Omega_i} Q^{\omega \circ \pi^{(n)}}_{h,i}(s,\boldsymbol{a}) \right] - 2\tilde{\varepsilon}.$$

where the inequality uses the result of Lemma C.8. Now we are ready to prove equation 17,

- When $h = H$, we have

$$\mathbb{E}_{s \sim d^{\tilde{\pi}^{(n)}}_{\hat{P}^{(n)},H}} \left[ \overline{V}^{(n)}_{H,i}(s) - \max_{\omega \in \Omega_i} V^{\omega \circ \pi^{(n)}}_{H,i}(s) \right] \geq \mathbb{E}_{(s,\boldsymbol{a}) \sim d^{\tilde{\pi}^{(n)}}_{\hat{P}^{(n)},H}} \left[ \overline{Q}^{(n)}_{H,i}(s,\boldsymbol{a}) - \max_{\omega \in \Omega_i} Q^{\omega \circ \pi^{(n)}}_{H,i}(s,\boldsymbol{a}) \right]$$

$$= \mathbb{E}_{(s,\boldsymbol{a}) \sim d^{\tilde{\pi}^{(n)}}_{\hat{P}^{(n)},H}} \left[ \hat{\beta}^{(n)}_h(s,\boldsymbol{a}) \right]$$

$$\geq \mathbb{E}_{(s,\boldsymbol{a}) \sim d^{\tilde{\pi}^{(n)}}_{\hat{P}^{(n)},H}} \left[ \hat{\beta}^{(n)}_h(s,\boldsymbol{a}) - H f^{(n)}_H(s,\boldsymbol{a}) \right] - 2\tilde{\varepsilon}.$$

- Suppose the statement is true for $h + 1$, then for step $h$, we have

$$\mathbb{E}_{s \sim d^{\tilde{\pi}^{(n)}}_{\hat{P}^{(n)},h}} \left[ \overline{V}^{(n)}_{h,i}(s) - \max_{\omega \in \Omega_i} V^{\omega \circ \pi^{(n)}}_{h,i}(s) \right]$$

$$\geq \mathbb{E}_{(s,\boldsymbol{a}) \sim d^{\tilde{\pi}^{(n)}}_{\hat{P}^{(n)},h}} \left[ \overline{Q}^{(n)}_{h,i}(s,\boldsymbol{a}) - \max_{\omega \in \Omega_i} Q^{\omega \circ \pi^{(n)}}_{h,i}(s,\boldsymbol{a}) \right] - 2\tilde{\varepsilon}$$

$$= \mathbb{E}_{(s,\boldsymbol{a}) \sim d^{\tilde{\pi}^{(n)}}_{\hat{P}^{(n)},h}} \left[ \hat{\beta}^{(n)}_h(s,\boldsymbol{a}) + \left( \hat{P}^{(n)}_h \overline{V}^{(n)}_{h+1,i} \right)(s,\boldsymbol{a}) - \left( P^\star_h \max_{\omega \in \Omega_i} V^{\omega \circ \pi^{(n)}_{-i}}_{h+1,i} \right)(s,\boldsymbol{a}) \right] - 2\tilde{\varepsilon}$$

$$= \mathbb{E}_{(s,\boldsymbol{a}) \sim d^{\tilde{\pi}^{(n)}}_{\hat{P}^{(n)},h}} \left[ \hat{\beta}^{(n)}_h(s,\boldsymbol{a}) + \left( \hat{P}^{(n)}_h \left( \overline{V}^{(n)}_{h+1,i} - \max_{\omega \in \Omega_i} V^{\omega \circ \pi^{(n)}}_{h+1,i} \right) \right)(s,\boldsymbol{a}) - \left( \left( \hat{P}^{(n)}_h - P^\star_h \right) \max_{\omega \in \Omega_i} V^{\omega \circ \pi^{(n)}}_{h+1,i} \right)(s,\boldsymbol{a}) \right]$$
$$- 2\tilde{\varepsilon}$$

$$= \mathbb{E}_{(s,\boldsymbol{a}) \sim d^{\tilde{\pi}^{(n)}}_{\hat{P}^{(n)},h}} \left[ \hat{\beta}^{(n)}_h(s,\boldsymbol{a}) - \left( \left( \hat{P}^{(n)}_h - P^\star_h \right) \max_{\omega \in \Omega_i} V^{\omega \circ \pi^{(n)}_{-i}}_{h+1,i} \right)(s,\boldsymbol{a}) \right]$$

$$+ \mathbb{E}_{s \sim d^{\tilde{\pi}^{(n)}}_{\hat{P}^{(n)},h+1}} \left[ \overline{V}^{(n)}_{h+1,i}(s) - \max_{\omega \in \Omega_i} V^{\omega \circ \pi^{(n)}}_{h+1,i}(s) \right] - 2\tilde{\varepsilon}$$

$$\geq \mathbb{E}_{(s,\boldsymbol{a}) \sim d^{\tilde{\pi}^{(n)}}_{\hat{P}^{(n)},h}} \left[ \hat{\beta}^{(n)}_h(s,\boldsymbol{a}) - H f^{(n)}_h(s,\boldsymbol{a}) \right] + \mathbb{E}_{s \sim d^{\tilde{\pi}^{(n)}}_{\hat{P}^{(n)},h+1}} \left[ \overline{V}^{(n)}_{h+1,i}(s) - \max_{\omega \in \Omega_i} V^{\omega \circ \pi^{(n)}}_{h+1,i}(s) \right] - 2\tilde{\varepsilon}$$

$$\geq \sum_{h'=h}^{H} \mathbb{E}_{(s,\boldsymbol{a}) \sim d^{\tilde{\pi}^{(n)}}_{\hat{P}^{(n)},h'}} \left[ \hat{\beta}^{(n)}_{h'}(s,\boldsymbol{a}) - H f^{(n)}_{h'}(s,\boldsymbol{a}) \right] - 2(H - h + 1)\tilde{\varepsilon},$$

where the last row uses the induction assumption.

Therefore, we have proved equation 17. We then apply $h = 1$ to equation 17, and get

$$\mathbb{E}_{s \sim d_1} \left[ \overline{V}^{(n)}_{1,i}(s) - \max_{\omega \in \Omega_i} V^{\omega \circ \pi^{(n)}}_{1,i}(s) \right]$$

$$= \mathbb{E}_{s \sim d^{\tilde{\pi}^{(n)}}_{\hat{P}^{(n)},1}} \left[ \overline{V}^{(n)}_{1,i}(s) - \max_{\omega \in \Omega_i} V^{\omega \circ \pi^{(n)}}_{1,i}(s) \right]$$

$$\geq \sum_{h=1}^{H} \mathbb{E}_{(s,\boldsymbol{a}) \sim d^{\tilde{\pi}^{(n)}}_{\hat{P}^{(n)},h}} \left[ \hat{\beta}^{(n)}_h(s,\boldsymbol{a}) - H f^{(n)}_h(s,\boldsymbol{a}) \right] - 2H\tilde{\varepsilon}$$

$$= \sum_{h=1}^{H} \mathbb{E}_{(s,\boldsymbol{a}) \sim d^{\tilde{\pi}^{(n)}}_{\hat{P}^{(n)},h}} \left[ \hat{\beta}^{(n)}_h(s,\boldsymbol{a}) \right] - H \sum_{h=1}^{H} \mathbb{E}_{(s,\boldsymbol{a}) \sim d^{\tilde{\pi}^{(n)}}_{\hat{P}^{(n)},h}} \left[ f^{(n)}_h(s,\boldsymbol{a}) \right] - 2H\tilde{\varepsilon}.$$

For the second term, since $\frac{1}{H} \hat{P}^{(n)}_h \max_{\omega \in \Omega_i} V^{\omega \circ \pi^{(n)}}_{h+1,i}$ is linear in $\hat{\phi}^{(n)}_h$ and $\frac{1}{H} P^\star_h \max_{\omega \in \Omega_i} V^{\omega \circ \pi^{(n)}}_{h+1,i}$ is linear in $\phi^\star_h$, and according to the result of Lemma C.1, the 2-norm of their weights are both upper

bounded by $\sqrt{d}$. Therefore, we have $\mathbb{E}_{\boldsymbol{a}\sim U(\mathcal{A})}\left[f_h^{(n)}(\cdot,\boldsymbol{a})\right]\in\mathcal{F}_{1,h}$. By Lemma C.9, we have for $h=1$,

$$\mathbb{E}_{(s,\boldsymbol{a})\sim d_{\hat{P}^{(n)},1}^{\tilde{\pi}^{(n)}}}\left[f_1^{(n)}(s,\boldsymbol{a})\right]\leq\sqrt{A\mathbb{E}_{(s,\boldsymbol{a})\sim\rho_1^{(n)}}\left[\left(f_1^{(n)}(s,\boldsymbol{a})\right)^2\right]}\leq\sqrt{A\zeta^{(n)}}.$$

And $\forall h\geq 2$, we have

$$\mathbb{E}_{(s,\boldsymbol{a})\sim d_{\hat{P}^{(n)},h}^{\tilde{\pi}^{(n)}}}\left[f_h^{(n)}(s,\boldsymbol{a})\right]$$

$$\leq\mathbb{E}_{(\tilde{s},\tilde{\boldsymbol{a}})\sim d_{\hat{P}^{(n)},h-1}^{\tilde{\pi}^{(n)}}}\left[\min\left\{\left\|\hat{\phi}_{h-1}^{(n)}(\tilde{s},\tilde{\boldsymbol{a}})\right\|_{\Sigma_{n,\rho_{h-1}^{(n)},\hat{\phi}_{h-1}}^{-1}}\sqrt{nA^2\mathbb{E}_{(s,\boldsymbol{a})\sim\tilde{\rho}_h^{(n)}}\left[\left(f_h^{(n)}(s,\boldsymbol{a})\right)^2\right]+d\lambda+nA^2\zeta^{(n)}},1\right\}\right]$$

$$\lesssim\mathbb{E}_{(\tilde{s},\tilde{\boldsymbol{a}})\sim d_{\hat{P}^{(n)},h-1}^{\tilde{\pi}^{(n)}}}\left[\min\left\{\left\|\hat{\phi}_{h-1}^{(n)}(\tilde{s},\tilde{\boldsymbol{a}})\right\|_{\Sigma_{n,\rho_{h-1}^{(n)},\hat{\phi}_{h-1}}^{-1}}\sqrt{nA^2\zeta^{(n)}+d\lambda},1\right\}\right].$$

Note that we here use $f_h^{(n)}(s,\boldsymbol{a})\leq 1$, $\mathbb{E}_{(s,\boldsymbol{a})\sim\rho_h^{(n)}}\left[\left(f_h^{(n)}(s,\boldsymbol{a})\right)^2\right]\leq\zeta^{(n)}$ and $\mathbb{E}_{(s,\boldsymbol{a})\sim\tilde{\rho}_h^{(n)}}\left[\left(f_h^{(n)}(s,\boldsymbol{a})\right)^2\right]\leq\zeta^{(n)}$. Then according to our choice of $\alpha^{(n)}$, we get

$$\mathbb{E}_{(s,\boldsymbol{a})\sim d_{\hat{P}^{(n)},h}^{\tilde{\pi}^{(n)}}}\left[f_h^{(n)}(s,\boldsymbol{a})\right]\leq\mathbb{E}_{(\tilde{s},\tilde{\boldsymbol{a}})\sim d_{\hat{P}^{(n)},h-1}^{\tilde{\pi}^{(n)}}}\left[\min\left\{\frac{c\alpha^{(n)}}{H}\left\|\hat{\phi}_{h-1}^{(n)}(\tilde{s},\tilde{\boldsymbol{a}})\right\|_{\Sigma_{n,\rho_{h-1}^{(n)},\hat{\phi}_{h-1}}^{-1}},1\right\}\right].$$

Combining all things together,

$$\overline{v}_i^{(n)}-\max_{\omega\in\Omega_i}v_i^{\omega\circ\pi^{(n)}}$$

$$=\mathbb{E}_{s\sim d_1}\left[\overline{V}_{1,i}^{(n)}(s)-\max_{\omega\in\Omega_i}V_{1,i}^{\omega\circ\pi^{(n)}}(s)\right]$$

$$\geq\sum_{h=1}^H\mathbb{E}_{(s,\boldsymbol{a})\sim d_{\hat{P}^{(n)},h}^{\tilde{\pi}^{(n)}}}\left[\hat{\beta}_h^{(n)}(s,\boldsymbol{a})\right]-H\sum_{h=1}^H\mathbb{E}_{(s,\boldsymbol{a})\sim d_{\hat{P}^{(n)},h}^{\tilde{\pi}^{(n)}}}\left[f_h^{(n)}(s,\boldsymbol{a})\right]-2H\tilde{\varepsilon}$$

$$\geq\sum_{h=1}^{H-1}\mathbb{E}_{(\tilde{s},\tilde{\boldsymbol{a}})\sim d_{\hat{P}^{(n)},h}^{\tilde{\pi}^{(n)}}}\left[\hat{\beta}_h^{(n)}(s,\boldsymbol{a})-\min\left\{c\alpha^{(n)}\left\|\hat{\phi}_h^{(n)}(\tilde{s},\tilde{\boldsymbol{a}})\right\|_{\Sigma_{n,\rho_h^{(n)},\hat{\phi}_h}^{-1}},H\right\}\right]-H\sqrt{A\zeta^{(n)}}-2H\tilde{\varepsilon}$$

$$=-H\sqrt{A\zeta^{(n)}}-2H\tilde{\varepsilon},$$

which proves the inequality. $\qquad\square$

**Lemma C.13** (pessimism). *Consider an episode $n\in[N]$ and set $\alpha^{(n)}=\Theta\left(H\sqrt{nA^2\zeta^{(n)}+d\lambda}\right)$. When the event $\mathcal{E}$ holds, we have*

$$\underline{v}_i^{(n)}(s)-v_i^{\pi^{(n)}}(s)\leq H\sqrt{A\zeta^{(n)}},\quad\forall n\in[N],i\in[M].$$

*Proof.* Let $\tilde{f}_h^{(n)}(s,\boldsymbol{a})=\left|\frac{1}{H}\left(\hat{P}_h^{(n)}-P_h^\star\right)V_{h+1,i}^{\pi^{(n)}}\right|(s,\boldsymbol{a})$, note that by definition, we have $\frac{1}{H}V_{h+1,i}^{\pi^{(n)}}(s)$ is bounded by 1, and

$$\frac{1}{H}V_{h+1,i}^{\pi^{(n)}}(s)=\mathbb{E}_{\boldsymbol{a}\sim\pi_h^{(n)}(s)}\left[\frac{r_{h+1,i}(s,\boldsymbol{a})}{H}+\frac{1}{H}\left(P_{h+1}^\star V_{h+2,i}^{\pi^{(n)}}\right)(s,\boldsymbol{a})\right]\in\mathcal{F}_{2,h}.$$

where we use the result of Lemma C.1 and get $\frac{1}{H}\left(P_{h+1}^\star V_{h+2,i}^{\pi^{(n)}}\right)(s,\boldsymbol{a})$ is a linear function in $\phi_{h+1}^\star$ and the 2-norm of the weight is upper bounded by $\sqrt{d}$. Then according to the event $\mathcal{E}$, we have

$$\mathbb{E}_{(s,\boldsymbol{a})\sim\rho_h^{(n)}}\left[\left(f_h^{(n)}(s,\boldsymbol{a})\right)^2\right]\leq\zeta^{(n)},\quad\mathbb{E}_{(s,\boldsymbol{a})\sim\tilde{\rho}_h^{(n)}}\left[\left(f_h^{(n)}(s,\boldsymbol{a})\right)^2\right]\leq\zeta^{(n)},\quad\forall n\in[N],h\in[H]$$

$$\|\phi_h(s, \boldsymbol{a})\|_{\left(\hat{\Sigma}_{h,\phi_h}^{(n)}\right)^{-1}} = \Theta\left(\|\phi_h(s, \boldsymbol{a})\|_{\Sigma_{n,\rho_h^{(n)},\phi_h}^{-1}}\right), \quad \forall n \in [N], h \in [H], \phi_h \in \Phi_h.$$

A direct conclusion of the event $\mathcal{E}$ is we can find an absolute constant $c$, such that

$$\beta_h^{(n)}(s, \boldsymbol{a}) = \min\left\{\alpha^{(n)} \left\|\hat{\phi}_h^{(n)}(\tilde{s}, \tilde{\boldsymbol{a}})\right\|_{\left(\Sigma_{h,\hat{\phi}_h^{(n)}}^{(n)}\right)^{-1}}, H\right\}$$

$$\geq \min\left\{c\alpha^{(n)} \left\|\hat{\phi}_h^{(n)}(\tilde{s}, \tilde{\boldsymbol{a}})\right\|_{\Sigma_{n,\rho_h^{(n)},\hat{\phi}_h^{(n)}}^{-1}}, H\right\}, \quad \forall n \in [N], h \in [H].$$

Again, we prove the following inequality by induction:

$$\mathbb{E}_{s \sim d_{\hat{P}^{(n)},h}^{\pi^{(n)}}}\left[\underline{V}_{h,i}^{(n)}(s) - V_{h,i}^{\pi^{(n)}}(s)\right] \leq \sum_{h'=h}^{H} \mathbb{E}_{(s,\boldsymbol{a}) \sim d_{\hat{P}^{(n)},h'}^{\pi^{(n)}}}\left[-\hat{\beta}_{h'}^{(n)}(s, \boldsymbol{a}) + Hf_{h'}^{(n)}(s, \boldsymbol{a})\right], \quad \forall h \in [H]. \tag{18}$$

- When $h = H$, we have

$$\mathbb{E}_{s \sim d_{\hat{P}^{(n)},H}^{\pi^{(n)}}}\left[\underline{V}_{H,i}^{(n)}(s) - V_{H,i}^{\pi^{(n)}}(s)\right] = \mathbb{E}_{(s,\boldsymbol{a}) \sim d_{\hat{P}^{(n)},H}^{\pi^{(n)}}}\left[\underline{Q}_{H,i}^{(n)}(s, \boldsymbol{a}) - Q_{H,i}^{\pi^{(n)}}(s, \boldsymbol{a})\right]$$

$$= \mathbb{E}_{(s,\boldsymbol{a}) \sim d_{\hat{P}^{(n)},H}^{\pi^{(n)}}}\left[-\hat{\beta}_H^{(n)}(s, \boldsymbol{a})\right]$$

$$\leq \mathbb{E}_{(s,\boldsymbol{a}) \sim d_{\hat{P}^{(n)},H}^{\pi^{(n)}}}\left[-\hat{\beta}_H^{(n)}(s, \boldsymbol{a}) + Hf_H^{(n)}(s, \boldsymbol{a})\right]$$

- Suppose the statement is true for $h + 1$, then for step $h$, we have

$$\mathbb{E}_{s \sim d_{\hat{P}^{(n)},h}^{\pi^{(n)}}}\left[\underline{V}_{h,i}^{(n)}(s) - V_{h,i}^{\pi^{(n)}}(s)\right]$$

$$= \mathbb{E}_{(s,\boldsymbol{a}) \sim d_{\hat{P}^{(n)},h}^{\pi^{(n)}}}\left[\underline{Q}_{h,i}^{(n)}(s, \boldsymbol{a}) - Q_{h,i}^{\pi^{(n)}}(s, \boldsymbol{a})\right]$$

$$= \mathbb{E}_{(s,\boldsymbol{a}) \sim d_{\hat{P}^{(n)},h}^{\pi^{(n)}}}\left[-\hat{\beta}_h^{(n)}(s, \boldsymbol{a}) + \left(\hat{P}_h^{(n)} \underline{V}_{h+1,i}^{(n)}\right)(s, \boldsymbol{a}) - \left(P_h^{\star} V_{h+1,i}^{\pi^{(n)}}\right)(s, \boldsymbol{a})\right]$$

$$= \mathbb{E}_{(s,\boldsymbol{a}) \sim d_{\hat{P}^{(n)},h}^{\pi^{(n)}}}\left[-\hat{\beta}_h^{(n)}(s, \boldsymbol{a}) + \left(\hat{P}_h^{(n)} \left(\underline{V}_{h+1,i}^{(n)} - V_{h+1,i}^{\pi^{(n)}}\right)\right)(s, \boldsymbol{a}) + \left(\left(\hat{P}_h^{(n)} - P_h^{\star}\right) V_{h+1,i}^{\pi^{(n)}}\right)(s, \boldsymbol{a})\right]$$

$$= \mathbb{E}_{(s,\boldsymbol{a}) \sim d_{\hat{P}^{(n)},h}^{\pi^{(n)}}}\left[-\hat{\beta}_h^{(n)}(s, \boldsymbol{a}) + \left(\left(\hat{P}_h^{(n)} - P_h^{\star}\right) V_{h+1,i}^{\pi^{(n)}}\right)(s, \boldsymbol{a})\right] + \mathbb{E}_{s \sim d_{\hat{P}^{(n)},h+1}^{\pi^{(n)}}}\left[\left(\underline{V}_{h+1,i}^{(n)} - V_{h+1,i}^{\pi^{(n)}}\right)(s)\right]$$

$$\leq \mathbb{E}_{(s,\boldsymbol{a}) \sim d_{\hat{P}^{(n)},h}^{\pi^{(n)}}}\left[-\hat{\beta}_h^{(n)}(s, \boldsymbol{a}) + Hf_h^{(n)}(s, \boldsymbol{a})\right] + \mathbb{E}_{s \sim d_{\hat{P}^{(n)},h+1}^{\pi^{(n)}}}\left[\left(\underline{V}_{h+1,i}^{(n)} - V_{h+1,i}^{\pi^{(n)}}\right)(s)\right]$$

$$\leq \sum_{h'=h}^{H} \mathbb{E}_{(s,\boldsymbol{a}) \sim d_{\hat{P}^{(n)},h'}^{\pi^{(n)}}}\left[-\hat{\beta}_{h'}^{(n)}(s, \boldsymbol{a}) + Hf_{h'}^{(n)}(s, \boldsymbol{a})\right].$$

where the last row uses the induction assumption.

The remaining steps are exactly the same as the proof in Lemma C.11 or Lemma C.12, we may prove

$$\mathbb{E}_{(s,\boldsymbol{a}) \sim d_{\hat{P}^{(n)},1}^{\pi^{(n)}}}\left[\min\left\{f_1^{(n)}(s, \boldsymbol{a}), 1\right\}\right] \leq \sqrt{A\zeta^{(n)}},$$

and

$$\mathbb{E}_{(s,\boldsymbol{a}) \sim d_{\hat{P}^{(n)},h}^{\pi^{(n)}}}\left[f_h^{(n)}(s, \boldsymbol{a})\right] \leq \mathbb{E}_{(\tilde{s},\tilde{\boldsymbol{a}}) \sim d_{\hat{P}^{(n)},h-1}^{\pi^{(n)}}}\left[\min\left\{\frac{c\alpha^{(n)}}{H} \left\|\hat{\phi}_{h-1}^{(n)}(\tilde{s}, \tilde{\boldsymbol{a}})\right\|_{\Sigma_{n,\rho_{h-1}^{(n)},\hat{\phi}_{h-1}^{(n)}}^{-1}}, 1\right\}\right], \quad \forall h \geq 2.$$

Combining all things together, we get

$$
\begin{aligned}
\underline{v}_i^{(n)} - v_i^{\pi^{(n)}} =& \mathbb{E}_{s \sim d_1} \left[ \underline{V}_{1,i}^{(n)}(s) - V_{1,i}^{\pi^{(n)}}(s) \right] \\
\leq& \sum_{h=1}^{H} \mathbb{E}_{(s,\boldsymbol{a}) \sim d_{\hat{P}^{(n)},h}^{\pi^{(n)}}} \left[ -\hat{\beta}_h^{(n)}(s,\boldsymbol{a}) + H f_h^{(n)}(s,\boldsymbol{a}) \right] \\
\leq& \sum_{h=1}^{H-1} \mathbb{E}_{(s,\boldsymbol{a}) \sim d_{\hat{P}^{(n)},h}^{\pi^{(n)}}} \left[ -\hat{\beta}_h^{(n)}(s,\boldsymbol{a}) + \min\left\{ c\alpha^{(n)} \left\| \hat{\phi}_h^{(n)}(\tilde{s},\tilde{\boldsymbol{a}}) \right\|_{\Sigma_{n,\rho_h^{(n)},\hat{\phi}_h^{(n)}}^{-1}}, H \right\} \right] + H\sqrt{A\zeta^{(n)}} \\
\leq& H\sqrt{A\zeta^{(n)}},
\end{aligned}
$$

which has finished the proof. $\qquad\square$

**Lemma C.14.** *For the model-free algorithm, suppose $N$ is large enough, when we pick $\lambda = \Theta\left(d \log \frac{NH|\Phi|}{\delta}\right)$, $\zeta^{(n)} = \Theta\left(\frac{d^2 M}{n} \log \frac{dNHML|\Phi|}{\tilde{\varepsilon}\delta}\right)$, $L = \Theta(NHAMd)$, $\tilde{\varepsilon} = \frac{1}{2HN}$ and $\alpha^{(n)} = \Theta\left(H\sqrt{nA^2\zeta^{(n)} + d\lambda}\right)$, with probability $1 - \delta$, we have*

$$
\sum_{n=1}^{N} \Delta^{(n)} \lesssim H^3 d^2 A^{\frac{3}{2}} N^{\frac{1}{2}} M^{\frac{1}{2}} \log \frac{dNHAM|\Phi|}{\delta}.
$$

*Proof.* With our choice of $\lambda$ and $\zeta^{(n)}$, according to Lemma C.6, we know $\mathcal{E}$ holds with probability $1 - \delta$. Furthermore, with a proper choice of the absolute constants, we have

$$
\begin{aligned}
\alpha^{(n)} =& \Theta\left( H\sqrt{d^2 A^2 M \log \frac{dNHML|\Phi|}{\delta} + d^2 \log \frac{NH|\Phi|}{\delta}} \right) \\
\leq& O\left( HdA\sqrt{M \log \frac{dNHMA|\Phi|}{\delta}} \right) \\
\leq& O\left( NHAMd \right) \leq L.
\end{aligned}
$$

Let $f_h^{(n)}(s,\boldsymbol{a}) = \frac{1}{2H^2} \left| \left( \hat{P}_h^{(n)} - P_h^\star \right) \left( \overline{V}_{h+1,i}^{(n)} - \underline{V}_{h+1,i}^{(n)} \right) \right| (s,\boldsymbol{a})$. We first verify $\frac{1}{2H^2}\left( \overline{V}_{h+1,i}^{(n)} - \underline{V}_{h+1,i}^{(n)} \right) \in \mathcal{F}_{4,h}$. By definition, we have

$$
\frac{1}{2H^2}\left( \overline{V}_{h+1,i}^{(n)} - \underline{V}_{h+1,i}^{(n)} \right) = \mathbb{E}_{\boldsymbol{a} \sim \pi_h^{(n)}(s)} \left[ \frac{1}{H^2} \hat{\beta}_{h+1}^{(n)}(s,\boldsymbol{a}) + \frac{1}{2H^2} P_{h+1}^\star \left( \overline{V}_{h+2,i}^{(n)} - \underline{V}_{h+2,i}^{(n)} \right)(s,\boldsymbol{a}) \right]
$$

The first term is equal to $\frac{1}{H^2} \min\left( \alpha^{(n)} \sqrt{\hat{\phi}_h^{(n)}(s,\boldsymbol{a})^\top \left( \hat{\Sigma}_h^{(n)} \right)^{-1} \hat{\phi}_h^{(n)}(s,\boldsymbol{a})}, H \right)$, which is exactly the same as that in the definition of $\mathcal{F}_{4,h}$ (note that we use the property $\alpha^{(n)} \leq L, \forall n \in [N]$). For the second term, note that we have $0 \leq \frac{1}{2H^2}\left( \overline{V}_{h,i}^{(n)} - \underline{V}_{h,i}^{(n)} \right) \leq 1, \forall h$. Therefore, by Lemma C.1, $\frac{1}{2H^2} P_{h+1}^\star \left( \overline{V}_{h+2,i}^{(n)} - \underline{V}_{h+2,i}^{(n)} \right)(s,\boldsymbol{a})$ is a linear function in $\phi_{h+1}^\star$ whose weight's 2-norm is upper bounded by $\sqrt{d}$. Combing the above arguments, we conclude $\frac{1}{2H^2}\left( \overline{V}_{h+1,i}^{(n)} - \underline{V}_{h+1,i}^{(n)} \right) \in \mathcal{F}_{4,h}$. According to the definition of the event $\mathcal{E}$, we have

$$
\mathbb{E}_{s \sim \rho_h^{(n)}} \left[ \left( f_h^{(n)}(s,\boldsymbol{a}) \right)^2 \right] \leq \zeta^{(n)}, \quad \|\phi_h(s,\boldsymbol{a})\|_{\left( \hat{\Sigma}_{h,\phi_h}^{(n)} \right)^{-1}} = \Theta\left( \|\phi_h(s,\boldsymbol{a})\|_{\Sigma_{n,\rho_h^{(n)},\phi_h}^{-1}} \right), \quad \forall n \in [N], \phi_h \in \Phi_h, h \in [H]. \tag{19}
$$

By definition, we have

$$
\Delta^{(n)} = \max_{i \in [M]} \left\{ \overline{v}_i^{(n)} - \underline{v}_i^{(n)} \right\} + 2H\sqrt{A\zeta^{(n)}}.
$$

For each fixed $i \in [M], h \in [H]$ and $n \in [N]$, we have

$$\mathbb{E}_{s \sim d^{\pi^{(n)}}_{P^\star, h}} \left[ \overline{V}^{(n)}_{h,i}(s) - \underline{V}^{(n)}_{h,i}(s) \right]$$

$$= \mathbb{E}_{s \sim d^{\pi^{(n)}}_{P^\star, h}} \left[ \left( \mathbb{D}_{\pi^{(n)}_h} \overline{Q}^{(n)}_{h,i} \right)(s) - \left( \mathbb{D}_{\pi^{(n)}_h} \underline{Q}^{(n)}_{h,i} \right)(s) \right]$$

$$= \mathbb{E}_{(s,\boldsymbol{a}) \sim d^{\pi^{(n)}}_{P^\star, h}} \left[ \overline{Q}^{(n)}_{h,i}(s, \boldsymbol{a}) - \underline{Q}^{(n)}_{h,i}(s, \boldsymbol{a}) \right]$$

$$= \mathbb{E}_{(s,\boldsymbol{a}) \sim d^{\pi^{(n)}}_{P^\star, h}} \left[ 2\hat{\beta}^{(n)}_h + \left( \hat{P}^{(n)}_h \left( \overline{V}^{(n)}_{h+1,i} - \underline{V}^{(n)}_{h+1,i} \right) \right)(s, \boldsymbol{a}) \right]$$

$$= \mathbb{E}_{(s,\boldsymbol{a}) \sim d^{\pi^{(n)}}_{P^\star, h}} \left[ 2\hat{\beta}^{(n)}_h + \left( \left( \hat{P}^{(n)}_h - P^\star_h \right) \left( \overline{V}^{(n)}_{h+1,i} - \underline{V}^{(n)}_{h+1,i} \right) \right)(s, \boldsymbol{a}) \right] + \mathbb{E}_{s \sim d^{\pi^{(n)}}_{P^\star, h+1}} \left[ \overline{V}^{(n)}_{h+1,i}(s) - \underline{V}^{(n)}_{h+1,i}(s) \right]$$

$$\leq \mathbb{E}_{(s,\boldsymbol{a}) \sim d^{\pi^{(n)}}_{P^\star, h}} \left[ 2\hat{\beta}^{(n)}_h(s, \boldsymbol{a}) + 2H^2 f^{(n)}_h(s, \boldsymbol{a}) \right] + \mathbb{E}_{s \sim d^{\pi^{(n)}}_{P^\star, h+1}} \left[ \overline{V}^{(n)}_{h+1,i}(s) - \underline{V}^{(n)}_{h+1,i}(s) \right]$$

$$\leq \cdots$$

$$\leq 2 \sum_{h'=h}^{H} \mathbb{E}_{(s,\boldsymbol{a}) \sim d^{\pi^{(n)}}_{P^\star, h'}} \left[ \hat{\beta}^{(n)}_{h'}(s, \boldsymbol{a}) + H^2 f^{(n)}_{h'}(s, \boldsymbol{a}) \right],$$

where the last inequality is calculated using induction. In particular,

$$\mathbb{E}_{s \sim d^{\pi^{(n)}}_{P^\star, 1}} \left[ \overline{V}^{(n)}_{1,i}(s) - \underline{V}^{(n)}_{1,i}(s) \right] \leq 2 \underbrace{\sum_{h=1}^{H} \mathbb{E}_{(s,\boldsymbol{a}) \sim d^{\pi^{(n)}}_{P^\star, h}} \left[ \hat{\beta}^{(n)}_h(s, \boldsymbol{a}) \right]}_{(a)} + 2H^2 \underbrace{\sum_{h=1}^{H} \mathbb{E}_{(s,\boldsymbol{a}) \sim d^{\pi^{(n)}}_{P^\star, h}} \left[ f^{(n)}_h(s, \boldsymbol{a}) \right]}_{(b)}. \tag{20}$$

First, we calculate the first term (a) in Inequality equation 20. Following Lemma C.10 and noting the bonus $\hat{\beta}^{(n)}_h$ is $O(H)$, we have

$$\sum_{h=1}^{H} \mathbb{E}_{(s,\boldsymbol{a}) \sim d^{\pi^{(n)}}_{P^\star, h}} \left[ \hat{\beta}^{(n)}_h(s, \boldsymbol{a}) \right]$$

$$\lesssim \sum_{h=1}^{H} \mathbb{E}_{(s,\boldsymbol{a}) \sim d^{\pi^{(n)}}_{P^\star, h}} \left[ \min \left( \alpha^{(n)} \left\| \hat{\phi}^{(n)}_h(s, \boldsymbol{a}) \right\|_{\Sigma^{-1}_{n, \rho^{(n)}_h, \hat{\phi}^{(n)}_h}}, H \right) \right] \qquad \text{(From equation 19)}$$

$$\lesssim \sum_{h=1}^{H-1} \mathbb{E}_{(\tilde{s},\tilde{\boldsymbol{a}}) \sim d^{\pi^{(n)}}_{P^\star, h}} \left[ \| \phi^\star_h(\tilde{s}, \tilde{\boldsymbol{a}}) \|_{\Sigma^{-1}_{n, \gamma^{(n)}_h, \phi^\star_h}} \right] \sqrt{nA \left( \alpha^{(n)} \right)^2 \mathbb{E}_{(s,\boldsymbol{a}) \sim \rho^{(n)}_h} \left[ \left\| \hat{\phi}^{(n)}_h(s, \boldsymbol{a}) \right\|^2_{\Sigma^{-1}_{n, \rho^{(n)}_h, \hat{\phi}^{(n)}_h}} \right] + H^2 d\lambda}$$

$$+ \sqrt{A \left( \alpha^{(n)} \right)^2 \mathbb{E}_{(s,\boldsymbol{a}) \sim \rho^{(n)}_1} \left[ \left\| \hat{\phi}^{(n)}_1(s, \boldsymbol{a}) \right\|^2_{\Sigma^{-1}_{n, \rho^{(n)}_1, \hat{\phi}^{(n)}_1}} \right]}.$$

Note that we use the fact that $B = H$ when applying Lemma D.3. In addition, we have

$$n \mathbb{E}_{(s,\boldsymbol{a}) \sim \rho^{(n)}_h} \left[ \left\| \hat{\phi}^{(n)}_h(s, \boldsymbol{a}) \right\|^2_{\Sigma^{-1}_{n, \rho^{(n)}_h, \hat{\phi}^{(n)}_h}} \right]$$

$$= n \text{Tr} \left( \mathbb{E}_{(s,\boldsymbol{a}) \sim \rho^{(n)}_h} \left[ \hat{\phi}^{(n)}_h(s, \boldsymbol{a}) \hat{\phi}^{(n)}_h(s, \boldsymbol{a})^\top \right] \left( n \mathbb{E}_{(s,\boldsymbol{a}) \sim \rho^{(n)}_h} \left[ \hat{\phi}^{(n)}_h(s, \boldsymbol{a}) \hat{\phi}^{(n)}_h(s, \boldsymbol{a})^\top \right] + \lambda I_d \right)^{-1} \right)$$

$$\leq d.$$

Then,

$$\sum_{h=1}^{H} \mathbb{E}_{(s,\boldsymbol{a}) \sim d^{\pi^{(n)}}_{P^\star, h}} \left[ \hat{\beta}^{(n)}_h(s, \boldsymbol{a}) \right] \leq \mathbb{E}_{(\tilde{s},\tilde{\boldsymbol{a}}) \sim d^{\pi^{(n)}}_{P^\star, h}} \left[ \| \phi^\star_h(\tilde{s}, \tilde{\boldsymbol{a}}) \|_{\Sigma^{-1}_{n, \gamma^{(n)}_h, \phi^\star_h}} \right] \sqrt{dA \left( \alpha^{(n)} \right)^2 + H^2 d\lambda} + \sqrt{dA \left( \alpha^{(n)} \right)^2 / n}.$$

Second, we calculate the term (b) in inequality equation 23. Following Lemma D.3 and noting $\left(f_h^{(n)}(s, \boldsymbol{a})\right)^2$ is upper-bounded by 1 (i.e., $B = 1$ in Lemma D.3), we have

$$\sum_{h=1}^{H} \mathbb{E}_{(s,\boldsymbol{a}) \sim d_{P^\star,h}^{\pi^{(n)}}} [f_h^{(n)}(s,\boldsymbol{a})]$$

$$\leq \sum_{h=1}^{H-1} \mathbb{E}_{(\tilde{s},\tilde{\boldsymbol{a}}) \sim d_{P^\star,h}^{\pi^{(n)}}} \left[ \|\phi_h^\star(\tilde{s},\tilde{\boldsymbol{a}})\|_{\Sigma_{n,\gamma_h^{(n)},\phi_h^\star}^{-1}} \right] \sqrt{nA\mathbb{E}_{(s,\boldsymbol{a}) \sim \rho_h^{(n)}} \left[ \left(f_h^{(n)}(s,\boldsymbol{a})\right)^2 \right] + d\lambda} + \sqrt{A\mathbb{E}_{(s,\boldsymbol{a}) \sim \rho_h^{(n)}} \left[ \left(f_1^{(n)}(s,\boldsymbol{a})\right)^2 \right]}$$

$$\leq \sum_{h=1}^{H-1} \mathbb{E}_{(\tilde{s},\tilde{\boldsymbol{a}}) \sim d_{P^\star,h}^{\pi^{(n)}}} \left[ \|\phi_h^\star(\tilde{s},\tilde{\boldsymbol{a}})\|_{\Sigma_{n,\gamma_h^{(n)},\phi_h^\star}^{-1}} \right] \sqrt{nA\zeta^{(n)} + d\lambda} + \sqrt{A\zeta^{(n)}}$$

$$\lesssim \frac{\alpha^{(n)}}{H} \sum_{h=1}^{H-1} \mathbb{E}_{(\tilde{s},\tilde{\boldsymbol{a}}) \sim d_{P^\star,h}^{\pi_n}} \left[ \|\phi_h^\star(\tilde{s},\tilde{\boldsymbol{a}})\|_{\Sigma_{n,\gamma_h^{(n)},\phi_h^\star}^{-1}} \right] + \sqrt{A\zeta^{(n)}},$$

where in the second inequality, we use $\mathbb{E}_{(s,\boldsymbol{a}) \sim \rho_h^{(n)}} \left[ \left(f_h^{(n)}(s,\boldsymbol{a})\right)^2 \right] \leq \zeta^{(n)}$, and in the last line, recall $\sqrt{nA\zeta^{(n)} + d\lambda} \lesssim \alpha^{(n)}/H$. Then, by combining the above calculation of the term (a) and term (b) in inequality equation 23, we have:

$$\overline{v}_i^{(n)} - \underline{v}_i^{(n)} = \mathbb{E}_{s \sim d_{P^\star,1}^{\pi^{(n)}}} \left[ \overline{V}_{1,i}^{(n)}(s) - \underline{V}_{1,i}^{(n)}(s) \right]$$

$$\lesssim \sum_{h=1}^{H-1} \left( \mathbb{E}_{(\tilde{s},\tilde{\boldsymbol{a}}) \sim d_{P^\star,h}^{\pi^{(n)}}} \left[ \|\phi_h^\star(\tilde{s},\tilde{\boldsymbol{a}})\|_{\Sigma_{n,\gamma_h^{(n)},\phi_h^\star}^{-1}} \right] \sqrt{dA\left(\alpha^{(n)}\right)^2 + H^2 d\lambda} + \sqrt{\frac{dA\left(\alpha^{(n)}\right)^2}{n}} \right)$$

$$+ H^2 \sum_{h=1}^{H-1} \left( \frac{\alpha^{(n)}}{H} \mathbb{E}_{(\tilde{s},\tilde{\boldsymbol{a}}) \sim d_{P^\star,h}^{\pi^{(n)}}} \left[ \|\phi_h^\star(\tilde{s},\tilde{\boldsymbol{a}})\|_{\Sigma_{n,\gamma_h^{(n)},\phi_h^\star}^{-1}} \right] + \sqrt{A\zeta^{(n)}} \right).$$

Taking maximum over $i$ on both sides and use the definition of $\Delta^{(n)}$, we get

$$\Delta^{(n)} = \max_{i \in [M]} \left\{ \overline{v}_i^{(n)} - \underline{v}_i^{(n)} \right\} + 2H\sqrt{A\zeta^{(n)}}$$

$$\lesssim \sum_{h=1}^{H-1} \left( \mathbb{E}_{(\tilde{s},\tilde{\boldsymbol{a}}) \sim d_{P^\star,h}^{\pi^{(n)}}} \left[ \|\phi_h^\star(\tilde{s},\tilde{\boldsymbol{a}})\|_{\Sigma_{n,\gamma_h^{(n)},\phi_h^\star}^{-1}} \right] \sqrt{dA\left(\alpha^{(n)}\right)^2 + H^2 d\lambda} + \sqrt{\frac{dA\left(\alpha^{(n)}\right)^2}{n}} \right)$$

$$+ H^2 \sum_{h=1}^{H-1} \left( \frac{\alpha^{(n)}}{H} \mathbb{E}_{(\tilde{s},\tilde{\boldsymbol{a}}) \sim d_{P^\star,h}^{\pi^{(n)}}} \left[ \|\phi_h^\star(\tilde{s},\tilde{\boldsymbol{a}})\|_{\Sigma_{n,\gamma_h^{(n)},\phi_h^\star}^{-1}} \right] + \sqrt{A\zeta^{(n)}} \right).$$

Hereafter, we take the dominating term out. Note that

$$\sum_{n=1}^{N} \mathbb{E}_{(\tilde{s},\tilde{\boldsymbol{a}}) \sim d_{P^\star,h}^{\pi^{(n)}}} \left[ \|\phi_h^\star(\tilde{s},\tilde{\boldsymbol{a}})\|_{\Sigma_{n,\gamma_h^{(n)},\phi_h^\star}^{-1}} \right] \leq \sqrt{N \sum_{n=1}^{N} \mathbb{E}_{(\tilde{s},\tilde{\boldsymbol{a}}) \sim d_{P^\star,h}^{\pi^{(n)}}} \left[ \phi_h^\star(\tilde{s},\tilde{\boldsymbol{a}})^\top \Sigma_{n,\gamma_h^{(n)},\phi_h^\star}^{-1} \phi_h^\star(\tilde{s},\tilde{\boldsymbol{a}}) \right]}$$

$$\text{(CS inequality)}$$

$$\lesssim \sqrt{N \left( \log \det \left( \sum_{n=1}^{N} \mathbb{E}_{(\tilde{s},\tilde{\boldsymbol{a}}) \sim d_{P^\star,h}^{\pi^{(n)}}} [\phi_h^\star(\tilde{s},\tilde{\boldsymbol{a}})\phi_h^\star(\tilde{s},\tilde{\boldsymbol{a}})^\top] \right) - \log \det(\lambda I_d) \right)} \qquad \text{(Lemma E.2)}$$

$$\leq \sqrt{dN \log \left( 1 + \frac{N}{d\lambda} \right)}.$$

$$\text{(Potential function bound, Lemma E.3 noting } \|\phi_h^\star(s,\boldsymbol{a})\|_2 \leq 1 \text{ for any } (s,\boldsymbol{a}).\text{)}$$

Finally,

$$\sum_{n=1}^{N} \Delta^{(n)} \lesssim H \left( \sqrt{dN \log \left( 1 + \frac{N}{d} \right)} \sqrt{dA\left(\alpha^{(N)}\right)^2 + H^2 d\lambda} + \sum_{n=1}^{N} \sqrt{\frac{dA\left(\alpha^{(n)}\right)^2}{n}} \right)$$

$$+ H^3 \left( \frac{1}{H} \sqrt{dN \log \left( 1 + \frac{N}{d\lambda} \right)} \alpha^{(N)} + \sum_{n=1}^{N} \sqrt{A\zeta^{(n)}} \right) + 2HN\tilde{\varepsilon}$$

$$\lesssim H^2 d \sqrt{NA \log \left( 1 + \frac{N}{d\lambda} \right)} \alpha^{(N)}$$

(Some algebra. We take the dominating term out. Note that $\alpha^{(n)}$ is increasing in $n$)

$$\lesssim H^3 d^2 A^{\frac{3}{2}} N^{\frac{1}{2}} M^{\frac{1}{2}} \log \frac{dNHAM|\Phi|}{\delta}.$$

This concludes the proof. $\qquad \square$

**Proof of Theorem 4.2**

*Proof.* For any fixed episode $n$ and agent $i$, by Lemma C.11, Lemma C.12 and Lemma C.13, we have

$$v_i^{\dagger, \pi_{-i}^{(n)}} - v_i^{\pi^{(n)}} \left( \text{or } \max_{\omega \in \Omega_i} v_i^{\omega \circ \pi^{(n)}} - v_i^{\pi^{(n)}} \right) \leq \bar{v}_i^{(n)} - \underline{v}_i^{(n)} + 2\sqrt{A\zeta^{(n)}} + 2H\tilde{\varepsilon} \leq \Delta^{(n)} + 2H\tilde{\varepsilon}.$$

Taking maximum over $i$ on both sides, we have

$$\max_{i \in [M]} \left\{ v_i^{\dagger, \pi_{-i}^{(n)}} - v_i^{\pi^{(n)}} \right\} \left( \text{or } \max_{i \in [M]} \left\{ \max_{\omega \in \Omega_i} v_i^{\omega \circ \pi^{(n)}} - v_i^{\pi^{(n)}} \right\} \right) \leq \Delta^{(n)} + 2H\tilde{\varepsilon}. \qquad (21)$$

From Lemma C.14, with probability $1 - \delta$, we can ensure

$$\sum_{n=1}^{N} (\Delta^{(n)} + 2H\tilde{\varepsilon}) \lesssim H^3 d^2 A^{\frac{3}{2}} N^{\frac{1}{2}} M^{\frac{1}{2}} \log \frac{dNHAM|\Phi|}{\delta}.$$

Therefore, according to Lemma E.4, when we pick $N$ to be

$$O \left( \frac{H^6 d^4 A^3 M}{\varepsilon^2} \log^2 \left( \frac{HdAM|\Phi|}{\delta\varepsilon} \right) \right),$$

we have

$$\frac{1}{N} \sum_{n=1}^{N} (\Delta^{(n)} + 2H\tilde{\varepsilon}) \leq \varepsilon.$$

On the other hand, from equation 21, we have

$$\max_{i \in [M]} \left\{ v_i^{\dagger, \hat{\pi}_{-i}} - v_i^{\hat{\pi}} \right\} \left( \text{or } \max_{i \in [M]} \left\{ \max_{\omega \in \Omega_i} v_i^{\omega \circ \hat{\pi}} - v_i^{\hat{\pi}} \right\} \right)$$

$$= \max_{i \in [M]} \left\{ v_i^{\dagger, \pi_{-i}^{(n^\star)}} - v_i^{\pi^{(n^\star)}} \right\} \left( \text{or } \max_{i \in [M]} \left\{ \max_{\omega \in \Omega_i} v_i^{\omega \circ \pi^{(n^\star)}} - v_i^{\pi^{(n^\star)}} \right\} \right)$$

$$\leq \Delta^{(n^\star)} + 2H\tilde{\varepsilon} = \min_{n \in [N]} \Delta^{(n)} + 2H\tilde{\varepsilon} \leq \frac{1}{N} \sum_{n=1}^{N} (\Delta^{(n)} + 2H\tilde{\varepsilon}) \leq \varepsilon,$$

which has finished the proof. $\qquad \square$

## D ANALYSIS OF THE FACTORED MARKOV GAMES

### D.1 HIGH PROBABILITY EVENTS

Define the set $\bar{\Phi}_{h,i} = \{ \bar{\phi}_{h,i}(s, \boldsymbol{a}) := \bigotimes_{j \in Z_i} \phi_{h,j}(s[Z_j], \boldsymbol{a}_j) | \phi_{h,j} \in \Phi_{h,j} \}$. Let $|\Phi| = \max_{h,j} |\Phi_{h,j}|$ and $|\bar{\Phi}| = \max_{h,i} |\bar{\Phi}_{h,i}|$. Clearly, we have $|\bar{\Phi}| \leq |\Phi|^L$. Define the following event

$$\mathcal{E}_1 : \forall n \in [N], h \in [H], i \in [M], \rho \in \left\{ \rho_h^{(n)}, \tilde{\rho}_h^{(n)} \right\}, \quad \mathbb{E}_\rho \left[ \left\| \hat{P}_{h,i}^{(n)}(\cdot | s[Z_i], \boldsymbol{a}_i) - P_{h,i}^\star(\cdot | s[Z_i], \boldsymbol{a}_i) \right\|_1^2 \right] \leq \zeta^{(n)},$$

$$\mathcal{E}_2 : \ \forall n \in [N], h \in [H], i \in [M], \bar{\phi}_{h,i} \in \bar{\Phi}_{h,i}, \quad \|\bar{\phi}_{h,i}(s,\boldsymbol{a})\|_{\left(\hat{\Sigma}_{h,\bar{\phi}_{h,i}}^{(n)}\right)^{-1}} = \Theta\left(\|\bar{\phi}_{h,i}(s,\boldsymbol{a})\|_{\Sigma_{n,\rho_h^{(n)},\bar{\phi}_{h,i}}^{-1}}\right)$$

$$\mathcal{E} := \mathcal{E}_1 \cap \mathcal{E}_2.$$

The following lemma shows that the event $\mathcal{E}$ holds with a high probability with proper choices of the parameters.

**Lemma D.1.** *When $\hat{P}_h^{(n)}$ is computed using Alg. 2, if we set*

$$\lambda = \Theta\left(Ld^L \log \frac{NHM|\Phi|}{\delta}\right), \ \zeta^{(n)} = \Theta\left(\frac{1}{n}\log \frac{|\mathcal{M}|HNM}{\delta}\right),$$

*then $\mathcal{E}$ holds with probability at least $1 - \delta$.*

The proof of Lemma D.1 is follows a similar procedure as that of Lemma B.2, with minor changes on the notations as well as some modifications on the union bound.

## D.2 STATISTICAL GUARANTEES

**Lemma D.2** (One-step back inequality for the learned model). *Suppose the event $\mathcal{E}$ holds. Consider a set of functions $\{g_h\}_{h=1}^H$ that satisfies $g_h \in \mathcal{S}[Z_i] \times \mathcal{A}_i \to \mathbb{R}_+$, s.t. $\|g_h\|_\infty \leq B$. For a given policy $\pi$, we have*

$$\left|\mathbb{E}_{(s,\boldsymbol{a})\sim d_{\hat{P}^{(n)},h}^\pi}[g_h(s[Z_i],\boldsymbol{a}_i)]\right|$$

$$\leq \begin{cases} \sqrt{\tilde{A}\mathbb{E}_{(s,\boldsymbol{a})\sim\rho_1^{(n)}}[g_1^2(s[Z_i],\boldsymbol{a}_i)]}, & h = 1 \\ \mathbb{E}_{(\tilde{s},\tilde{\boldsymbol{a}})\sim d_{\hat{P}^{(n)},h-1}^\pi}\left[\min\left\{\tilde{A}\left\|\bar{\phi}_{h-1,i}^{(n)}(\tilde{s},\tilde{\boldsymbol{a}})\right\|_{\Sigma_{n,\rho_{h-1}^{(n)},\bar{\phi}_{h-1,i}^{(n)}}^{-1}}\sqrt{n\mathbb{E}_{(s,\boldsymbol{a})\sim\tilde{\rho}_h^{(n)}}[g_h^2(s[Z_i],\boldsymbol{a}_i)]+B^2\lambda d^L+nB^2\zeta^{(n)}}, B\right\}\right], & h \geq 2 \end{cases}$$

*where* $\bar{\phi}_{h,i}^{(n)}(s,\boldsymbol{a}) := \bigotimes_{j\in Z_i}\hat{\phi}_{h-1,j}^{(n)}(s[Z_j],\boldsymbol{a}_j),$ *and* $\Sigma_{n,\rho_h^{(n)},\bar{\phi}_{h,i}^{(n)}} = n\mathbb{E}_{(s,\boldsymbol{a})\sim\rho_h^{(n)}}\left[\bar{\phi}_{h,i}^{(n)}(s,\boldsymbol{a})\bar{\phi}_{h,i}^{(n)}(s,\boldsymbol{a})^\top\right] + \lambda I_{d^{|Z_i|}}.$

*Proof.* For step $h = 1$, we have

$$\mathbb{E}_{(s,\boldsymbol{a}_i)\sim d_{\hat{P}^{(n)},1}^\pi}[g_1(s[Z_i],\boldsymbol{a}_i)] = \mathbb{E}_{s\sim d_1,\boldsymbol{a}_i\sim\pi_1(s)}[g_1(s[Z_i],\boldsymbol{a}_i)]$$

$$\leq \sqrt{\max_{(s,\boldsymbol{a}_i)}\frac{d_1(s)\pi_1(\boldsymbol{a}_i|s)}{\rho_1^{(n)}(s,\boldsymbol{a}_i)}\mathbb{E}_{(s',\boldsymbol{a}_i')\sim\rho_1^{(n)}}[g_1^2(s'[Z_i],\boldsymbol{a}_i')]}$$

$$= \sqrt{\max_{(s,\boldsymbol{a}_i)}\frac{d_1(s)\pi_1(\boldsymbol{a}_i|s)}{d_1(s)u_{\mathcal{A}}(\boldsymbol{a}_i)}\mathbb{E}_{(s',\boldsymbol{a}_i')\sim\rho_1^{(n)}}[g_1^2(s'[Z_i],\boldsymbol{a}_i')]}$$

$$\leq \sqrt{\tilde{A}\mathbb{E}_{(s,\boldsymbol{a}_i)\sim\rho_1^{(n)}}[g_1^2(s[Z_i],\boldsymbol{a}_i)]}.$$

For $h \geq 2$, we observe the following one-step-back decomposition:

$$\mathbb{E}_{(\tilde{s},\tilde{\boldsymbol{a}}_i)\sim d_{\hat{P}^{(n)},h}^\pi}[g_h(s[Z_i],\boldsymbol{a}_i)]$$

$$= \mathbb{E}_{(\tilde{s},\tilde{\boldsymbol{a}})\sim d_{\hat{P}^{(n)},h-1}^\pi,s\sim\hat{P}_{h-1}^{(n)}(\tilde{s},\tilde{\boldsymbol{a}}),\boldsymbol{a}_i\sim\pi_h(s)}[g_h(s[Z_i],\boldsymbol{a}_i)]$$

$$= \mathbb{E}_{(\tilde{s},\tilde{\boldsymbol{a}})\sim d_{\hat{P}^{(n)},h-1}^\pi}\left[\int_{\mathcal{S}}\prod_{j=1}^M\left[\hat{\phi}_{h-1,j}^{(n)}(\tilde{s}[Z_j],\tilde{\boldsymbol{a}}_j)^\top\hat{w}_{h-1,j}^{(n)}(s_j)\right]\sum_{\boldsymbol{a}_i\in\mathcal{A}_i}\pi_h(\boldsymbol{a}_i|s)g_h(s[Z_i],\boldsymbol{a}_i)\mathrm{d}s\right]$$

$$= \mathbb{E}_{(\tilde{s},\tilde{\boldsymbol{a}})\sim d_{\hat{P}^{(n)},h-1}^\pi}\left[\min\left\{\int_{\mathcal{S}}\prod_{j=1}^M\left[\hat{\phi}_{h-1,j}^{(n)}(\tilde{s}[Z_j],\tilde{\boldsymbol{a}}_j)^\top\hat{w}_{h-1,j}^{(n)}(s_j)\right]\sum_{\boldsymbol{a}_i\in\mathcal{A}_i}\pi_h(\boldsymbol{a}_i|s)g_h(s[Z_i],\boldsymbol{a}_i)\mathrm{d}s, B\right\}\right]$$

$$\leq \mathbb{E}_{(\tilde{s},\tilde{a})\sim d^{\pi}_{\hat{P}^{(n)},h-1}}\left[\min\left\{\tilde{A}\int_{\mathcal{S}}\prod_{j=1}^{M}\left[\hat{\phi}^{(n)}_{h-1,j}(\tilde{s}[Z_j],\tilde{a}_j)^{\top}\hat{w}^{(n)}_{h-1,j}(s_j)\right]\frac{1}{|\mathcal{A}_i|}\sum_{\boldsymbol{a}_i\in\mathcal{A}_i}g_h(s[Z_i],\boldsymbol{a}_i)\mathrm{d}s,B\right\}\right]$$

$$= \mathbb{E}_{(\tilde{s},\tilde{a})\sim d^{\pi}_{\hat{P}^{(n)},h-1}}\left[\min\left\{\tilde{A}\int_{\mathcal{S}[Z_i]}\prod_{j\in Z_i}\left[\hat{\phi}^{(n)}_{h-1,j}(\tilde{s}[Z_j],\tilde{a}_j)^{\top}\hat{w}^{(n)}_{h-1,j}(s_j)\right]\frac{1}{|\mathcal{A}_i|}\sum_{\boldsymbol{a}_i\in\mathcal{A}_i}g_h(s[Z_i],\boldsymbol{a}_i)\mathrm{d}s[Z_i],B\right\}\right]$$

$$= \mathbb{E}_{(\tilde{s},\tilde{a})\sim d^{\pi}_{\hat{P}^{(n)},h-1}}\left[\min\left\{\tilde{A}\int_{\mathcal{S}[Z_i]}\left[\bigotimes_{j\in Z_i}\hat{\phi}^{(n)}_{h-1,j}(\tilde{s}[Z_j],\tilde{a}_j)\right]^{\top}\left[\bigotimes_{j\in Z_i}\hat{w}^{(n)}_{h-1,j}(s_j)\right]\frac{1}{|\mathcal{A}_i|}\sum_{\boldsymbol{a}_i\in\mathcal{A}_i}g_h(s[Z_i],\boldsymbol{a}_i)\mathrm{d}s[Z_i],B\right\}\right]$$

$$= \mathbb{E}_{(\tilde{s},\tilde{a})\sim d^{\pi}_{\hat{P}^{(n)},h-1}}\left[\min\left\{\tilde{A}\int_{\mathcal{S}[Z_i]}\bar{\phi}^{(n)}_{h-1,i}(\tilde{s},\tilde{a})^{\top}\left[\bigotimes_{j\in Z_i}\hat{w}^{(n)}_{h-1,j}(s_j)\right]\frac{1}{|\mathcal{A}_i|}\sum_{\boldsymbol{a}_i\in\mathcal{A}_i}g_h(s[Z_i],\boldsymbol{a}_i)\mathrm{d}s[Z_i],B\right\}\right]$$

$$\leq \mathbb{E}_{(\tilde{s},\tilde{a})\sim d^{\pi}_{\hat{P}^{(n)},h-1}}\left[\min\left\{\tilde{A}\left\|\bar{\phi}^{(n)}_{h-1,i}(\tilde{s},\tilde{a})\right\|_{\Sigma^{-1}_{n,\rho^{(n)}_{h-1},\bar{\phi}^{(n)}_{h-1,i}}}\left\|\int_{\mathcal{S}[Z_i]}\frac{1}{|\mathcal{A}_i|}\sum_{\boldsymbol{a}_i\in\mathcal{A}_i}\left(\bigotimes_{j\in Z_i}\hat{w}^{(n)}_{h-1,j}(s_j)\right)g_h(s[Z_i],\boldsymbol{a}_i)\mathrm{d}s[Z_i]\right\|_{\Sigma_{n,\rho^{(n)}_{h-1},\bar{\phi}^{(n)}_{h-1,i}}}\right.\right.$$

$$\left.\left.,B\right\}\right].$$

Then,

$$\left\|\int_{\mathcal{S}[Z_i]}\frac{1}{|\mathcal{A}_i|}\sum_{\boldsymbol{a}_i\in\mathcal{A}}\left(\bigotimes_{j\in Z_i}\hat{w}^{(n)}_{h-1,j}(s_j)\right)g_h(s[Z_i],\boldsymbol{a}_i)\mathrm{d}s[Z_i]\right\|^2_{\Sigma_{n,\rho^{(n)}_{h-1},\hat{\phi}^{(n)}_{h-1},i}}$$

$$\leq n\mathbb{E}_{(\tilde{s},\tilde{a})\sim\rho^{(n)}_{h-1}}\left[\left(\int_{\mathcal{S}[Z_i]}\frac{1}{|\mathcal{A}_i|}\sum_{\boldsymbol{a}_i\in\mathcal{A}}\prod_{j\in Z_i}\left(\hat{w}^{(n)}_{h-1,j}(s_j)^{\top}\hat{\phi}^{(n)}_{h-1,j}(\tilde{s},\tilde{a}_j)\right)g_h(s[Z_i],\boldsymbol{a}_i)\mathrm{d}s[Z_i]\right)^2\right]+B^2\lambda d^L$$

$$\left(\left\|\tfrac{1}{|\mathcal{A}_i|}\sum_{\boldsymbol{a}_i\in\mathcal{A}_i}g_h(s[Z_i],\boldsymbol{a}_i)\right\|_{\infty}\leq B\text{ and }\left\|\hat{w}^{(n)}_{h-1,i}(s_i)\right\|_2\leq\sqrt{d}.\right)$$

$$= n\mathbb{E}_{(\tilde{s},\tilde{a})\sim\rho^{(n)}_{h-1}}\left[\left(\mathbb{E}_{s\sim\hat{P}^{(n)}_{h-1}(\tilde{s},\tilde{a}),\boldsymbol{a}_i\sim U(\mathcal{A}_i)}\left[g_h(s[Z_i],\boldsymbol{a}_i)\right]\right)^2\right]+B^2\lambda d^L$$

$$\leq n\mathbb{E}_{(\tilde{s},\tilde{a})\sim\rho^{(n)}_{h-1}}\left[\left(\mathbb{E}_{s\sim P^{\star}_{h-1}(\tilde{s},\tilde{a}),\boldsymbol{a}_i\sim U(\mathcal{A}_i)}\left[g_h(s[Z_i],\boldsymbol{a}_i)\right]\right)^2\right]+B^2\lambda d^L+nB^2\xi^{(n)} \qquad (\text{Event }\mathcal{E})$$

$$\leq n\mathbb{E}_{(\tilde{s},\tilde{a})\sim\rho^{(n)}_{h-1},s\sim P^{\star}_{h-1}(\tilde{s},\tilde{a}),\boldsymbol{a}_i\sim U(\mathcal{A}_i)}\left[g^2_h(s[Z_i],\boldsymbol{a}_i)\right]+B^2\lambda d^L+B^2n\xi^{(n)}. \qquad (\text{Jensen})$$

$$= n\mathbb{E}_{(s,\boldsymbol{a}_i)\sim\tilde{\rho}^{(n)}_h}\left[g^2_h(s[Z_i],\boldsymbol{a}_i)\right]+B^2\lambda d^L+B^2n\zeta^{(n)}. \qquad (\text{Definition of }\tilde{\rho}^{(n)}_h)$$

Combing the above results together, we get

$$\mathbb{E}_{(\tilde{s},\tilde{a}_i)\sim d^{\pi}_{\hat{P}^{(n)},h}}\left[g_h(s[Z_i],\boldsymbol{a}_i)\right]$$

$$\leq \mathbb{E}_{(\tilde{s},\tilde{a})\sim d^{\pi}_{\hat{P}^{(n)},h-1}}\left[\min\left\{\tilde{A}\left\|\bar{\phi}^{(n)}_{h-1,i}(\tilde{s},\tilde{a})\right\|_{\Sigma^{-1}_{n,\rho^{(n)}_{h-1},\bar{\phi}^{(n)}_{h-1,i}}}\left\|\int_{\mathcal{S}[Z_i]}\frac{1}{|\mathcal{A}_i|}\sum_{\boldsymbol{a}_i\in\mathcal{A}_i}\left(\bigotimes_{j\in Z_i}\hat{w}^{(n)}_{h-1,j}(s_j)\right)g_h(s[Z_i],\boldsymbol{a}_i)\mathrm{d}s[Z_i]\right\|_{\Sigma_{n,\rho^{(n)}_{h-1},\bar{\phi}^{(n)}_{h-1,i}}}\right.\right.$$

$$\left.\left.,B\right\}\right]$$

$$\leq \mathbb{E}_{(\tilde{s},\tilde{a})\sim d^{\pi}_{\hat{P}^{(n)},h-1}}\left[\min\left\{\tilde{A}\left\|\bar{\phi}^{(n)}_{h-1,i}(\tilde{s},\tilde{a})\right\|_{\Sigma^{-1}_{n,\rho^{(n)}_{h-1},\bar{\phi}^{(n)}_{h-1,i}}}\sqrt{n\mathbb{E}_{(s,\boldsymbol{a}_i)\sim\tilde{\rho}^{(n)}_h}\left[g^2_h(s[Z_i],\boldsymbol{a}_i)\right]+B^2\lambda d^L+B^2n\zeta^{(n)}},B\right\}\right],$$

which has finished the proof. $\qquad\square$

**Lemma D.3** (One-step back inequality for the true model). *Consider a set of functions $\{g_h\}_{h=1}^H$ that satisfies $g_h\in\mathcal{S}[Z_i]\times\mathcal{A}_i\to\mathbb{R}_+$, s.t. $\|g_h\|_{\infty}\leq B$. Then for any policy $\pi$, we have*

$$\left|\mathbb{E}_{(s,\boldsymbol{a}_i)\sim d^{\pi}_{P^{\star},h}}\left[g_h(s[Z_i],\boldsymbol{a}_i)\right]\right|$$

$$\leq \begin{cases} \sqrt{\tilde{A}\mathbb{E}_{(s,\boldsymbol{a}_i)\sim\rho_1^{(n)}}[g_1^2(s[Z_i],\boldsymbol{a}_i)]}, & h=1, \\ \tilde{A}\mathbb{E}_{(\tilde{s},\tilde{\boldsymbol{a}})\sim d_{P^\star,h-1}^\pi}\left[\left\|\bar{\phi}_{h-1,i}^\star(\tilde{s},\tilde{\boldsymbol{a}})\right\|_{\Sigma_{n,\gamma_{h-1}^{(n)},\bar{\phi}_{h-1,i}^\star}^{-1}}\right]\sqrt{n\mathbb{E}_{(s,\boldsymbol{a})\sim\rho_h^{(n)}}[g_h^2(s[Z_i],\boldsymbol{a}_i)]+B^2\lambda d^L}, & h\geq 2, \end{cases}$$

where $\quad \bar{\phi}_{h,i}^\star(s,\boldsymbol{a}) \quad := \quad \bigotimes_{j\in Z_i}\phi_{h-1,j}^\star(s[Z_j],\boldsymbol{a}_j), \quad$ and $\quad \Sigma_{n,\gamma_h^{(n)},\bar{\phi}_{h,i}^\star} \quad =$
$n\mathbb{E}_{(s,\boldsymbol{a})\sim\gamma_h^{(n)}}\left[\bar{\phi}_{h,i}^\star(s,\boldsymbol{a})\bar{\phi}_{h,i}^\star(s,\boldsymbol{a})^\top\right]+\lambda I_{d^{|Z_i|}}.$

*Proof.* For step $h=1$, we have

$$\mathbb{E}_{(s,\boldsymbol{a})\sim d_{P^\star,1}^\pi}[g_1(s[Z_i],\boldsymbol{a}_i)]=\mathbb{E}_{s\sim d_1,\boldsymbol{a}_i\sim\pi_1(s)}[g_1(s[Z_i],\boldsymbol{a}_i)]$$

$$\leq\sqrt{\max_{(s,\boldsymbol{a}_i)}\frac{d_1(s)\pi_1(\boldsymbol{a}_i|s)}{\rho_1^{(n)}(s,\boldsymbol{a}_i)}\mathbb{E}_{(s',\boldsymbol{a}_i')\sim\rho_1^{(n)}}[g_1^2(s'[Z_i],\boldsymbol{a}_i')]}$$

$$=\sqrt{\max_{(s,\boldsymbol{a}_i)}\frac{d_1(s)\pi_1(\boldsymbol{a}_i|s)}{d_1(s)u_{\mathcal{A}_i}(\boldsymbol{a}_i)}\mathbb{E}_{(s',\boldsymbol{a}_i')\sim\rho_1^{(n)}}[g_1^2(s'[Z_i],\boldsymbol{a}_i')]}$$

$$\leq\sqrt{\tilde{A}\mathbb{E}_{(s,\boldsymbol{a}_i)\sim\rho_1^{(n)}}[g_1^2(s[Z_i],\boldsymbol{a}_i)]}.$$

For step $h=2,\ldots,H-1$, we observe the following one-step-back decomposition:

$$\mathbb{E}_{(\tilde{s},\tilde{\boldsymbol{a}}_i)\sim d_{P^\star,h}^\pi}[g_h(s[Z_i],\boldsymbol{a}_i)]$$

$$=\mathbb{E}_{(\tilde{s},\tilde{\boldsymbol{a}})\sim d_{P^\star,h-1}^\pi,s\sim P_{h-1}^\star(\tilde{s},\tilde{\boldsymbol{a}}),\boldsymbol{a}_i\sim\pi_h(s)}[g_h(s[Z_i],\boldsymbol{a}_i)]$$

$$=\mathbb{E}_{(\tilde{s},\tilde{\boldsymbol{a}})\sim d_{P^\star,h-1}^\pi}\left[\left(\bigotimes_{j\in Z_i}\phi_{h-1,j}^\star(\tilde{s}[Z_j],\tilde{\boldsymbol{a}}_j)\right)^\top\int_{\mathcal{S}}\sum_{\boldsymbol{a}_i\in\mathcal{A}_i}\left(\bigotimes_{j\in Z_i}w_{h-1,j}^\star(s_j)\right)\pi_h(\boldsymbol{a}_i|s)g_h(s[Z_i],\boldsymbol{a}_i)\mathrm{d}s\right]$$

$$\leq\tilde{A}\mathbb{E}_{(\tilde{s},\tilde{\boldsymbol{a}})\sim d_{P^\star,h-1}^\pi}\left[\left(\bigotimes_{j\in Z_i}\phi_{h-1,j}^\star(\tilde{s}[Z_j],\tilde{\boldsymbol{a}}_j)\right)^\top\int_{\mathcal{S}}\sum_{\boldsymbol{a}_i\in\mathcal{A}_i}\frac{1}{|\mathcal{A}_i|}\left(\bigotimes_{j\in Z_i}w_{h-1,j}^\star(s_j)\right)g_h(s[Z_i],\boldsymbol{a}_i)\mathrm{d}s[Z_i]\right]$$

$$\leq\tilde{A}\mathbb{E}_{(\tilde{s},\tilde{\boldsymbol{a}})\sim d_{P^\star,h-1}^\pi}\left[\left\|\bigotimes_{j\in Z_i}\phi_{h-1,j}^\star(\tilde{s}[Z_j],\tilde{\boldsymbol{a}}_j)\right\|_{\Sigma_{n,\gamma_{h-1}^{(n)},\bar{\phi}_{h-1,i}^\star}^{-1}}\right]$$

$$\cdot\left\|\int_{\mathcal{S}}\sum_{\boldsymbol{a}_i\in\mathcal{A}_i}\frac{1}{|\mathcal{A}_i|}\left(\bigotimes_{j\in Z_i}w_{h-1,j}^\star(s_j)\right)g_h(s[Z_i],\boldsymbol{a}_i)\mathrm{d}s[Z_i]\right\|_{\Sigma_{n,\gamma_{h-1}^{(n)},\bar{\phi}_{h-1},i}}.$$

Then,

$$\left\|\int_{\mathcal{S}}\sum_{\boldsymbol{a}_i\in\mathcal{A}_i}\frac{1}{|\mathcal{A}_i|}\left(\bigotimes_{j\in Z_i}w_{h-1,j}^\star(s_j)\right)g_h(s[Z_i],\boldsymbol{a}_i)\mathrm{d}s[Z_i]\right\|_{\Sigma_{n,\gamma_{h-1}^{(n)},\bar{\phi}_{h-1},i}}^2$$

$$\leq n\mathbb{E}_{(\tilde{s},\tilde{\boldsymbol{a}})\sim\gamma_{h-1}^{(n)}}\left[\left(\int_{\mathcal{S}}\sum_{\boldsymbol{a}_i\in\mathcal{A}_i}\frac{1}{|\mathcal{A}_i|}\left(\bigotimes_{j\in Z_i}w_{h-1,j}^\star(s_j)\right)^\top\left(\bigotimes_{j\in Z_i}\phi_{h-1,j}^\star(\tilde{s}[Z_j],\tilde{\boldsymbol{a}}_j)\right)g_h(s[Z_i],\boldsymbol{a}_i)\mathrm{d}s[Z_i]\right)^2\right]+B^2\lambda d^L$$

(Use the assumption $\left\|\sum_{\boldsymbol{a}_i\in\mathcal{A}_i}\frac{1}{|\mathcal{A}_i|}g_h(s[Z_i],\boldsymbol{a}_i)\right\|_\infty\leq B$ and $\left\|w_{h-1,i}^\star(s_i)\right\|_2\leq\sqrt{d}$.)

$$=n\mathbb{E}_{(\tilde{s},\tilde{\boldsymbol{a}})\sim\gamma_{h-1}^{(n)}}\left[\left(\mathbb{E}_{s\sim P_{h-1}^\star(\tilde{s},\tilde{\boldsymbol{a}}),\boldsymbol{a}_i\sim U(\mathcal{A}_i)}[g_h(s[Z_i],\boldsymbol{a}_i)]\right)^2\right]+B^2\lambda d^L$$

$$\leq n\mathbb{E}_{(\tilde{s},\tilde{\boldsymbol{a}})\sim\gamma_{h-1}^{(n)},s\sim P_{h-1}^\star(\tilde{s},\tilde{\boldsymbol{a}}),\boldsymbol{a}_i\sim U(\mathcal{A}_i)}[g_h^2(s[Z_i],\boldsymbol{a}_i)]+B^2\lambda d^L \qquad\text{(Jensen)}$$

$$\leq n\mathbb{E}_{(s,\boldsymbol{a}_i)\sim\rho_h^{(n)}}\left[g_h^2(s[Z_i],\boldsymbol{a}_i)\right] + B^2\lambda d^L, \qquad\qquad\qquad\text{(Definition of }\rho_h^{(n)}\text{)}$$

which has finished the proof. $\qquad\qquad\square$

**Lemma D.4** (One-step back inequality for the true model). *Consider a set of functions $\{g_h\}_{h=1}^H$ that satisfies $g_h \in \mathcal{S}[\cup_{j\in Z_i}Z_j] \times \mathcal{A}[Z_i] \to \mathbb{R}$, s.t. $\|g_h\|_\infty \leq B$. Then for any policy $\pi$, we have*

$$\left|\mathbb{E}_{(s,\boldsymbol{a})\sim d_{P^\star,h}^\pi}\left[g_h(s[\cup_{j\in Z_i}Z_j],\boldsymbol{a}[Z_i])\right]\right|$$

$$\leq \begin{cases} \sqrt{\tilde{A}^L\mathbb{E}_{(s,\boldsymbol{a})\sim\rho_1^{(n)}}\left[g_1^2(s[\cup_{j\in Z_i}Z_j],\boldsymbol{a}[Z_i])\right]}, & h=1, \\[2ex] \tilde{A}^L\mathbb{E}_{(\tilde{s},\tilde{\boldsymbol{a}})\sim d_{P^\star,h-1}^\pi}\left[\left\|\tilde{\phi}_{h-1,i}^\star(\tilde{s},\tilde{\boldsymbol{a}})\right\|_{\Sigma_{n,\gamma_{h-1}^{(n)},\tilde{\phi}_{h-1,i}^\star}^{-1}}\right]\sqrt{n\mathbb{E}_{(s,\boldsymbol{a})\sim\rho_h^{(n)}}\left[g_h^2(s[\cup_{j\in Z_i}Z_j],\boldsymbol{a}[Z_i])\right] + B^2\lambda d^{L^2}}, & h\geq 2, \end{cases}$$

*where* $\quad\tilde{\phi}_{h,i}^\star(s,\boldsymbol{a}) \quad := \quad \bigotimes_{k\in\cup_{j\in Z_i}Z_j}\phi_{h-1,j}^\star(s[Z_k],\boldsymbol{a}_k), \quad$ *and* $\quad\Sigma_{n,\gamma_h^{(n)},\tilde{\phi}_{h,i}^\star} \quad =$ $n\mathbb{E}_{(s,\boldsymbol{a})\sim\gamma_h^{(n)}}\left[\tilde{\phi}_{h,i}^\star(s,\boldsymbol{a})\tilde{\phi}_{h,i}^\star(s,\boldsymbol{a})^\top\right] + \lambda I_{d^{|\cup_{j\in Z_i}Z_j|}}.$

*Proof.* This Lemma can be proved using similar steps as those in the proof of Lemma D.3, noting that in this case the dimension of $\tilde{\phi}_{h,i}^\star$ is at most $L^2$. $\qquad\square$

**Lemma D.5** (Optimism for NE and CCE). *Consider an episode $n \in [N]$ and set $\alpha^{(n)} = \Theta\left(H\tilde{A}\sqrt{n\zeta^{(n)} + d^L\lambda}\right)$. When the event $\mathcal{E}$ holds and the policy $\pi^{(n)}$ is computed by solving NE or CCE, we have*

$$\overline{v}_i^{(n)}(s) - v_i^{\dagger,\pi_{-i}^{(n)}}(s) \geq -HM\sqrt{\tilde{A}\zeta^{(n)}}, \quad \forall n \in [N], i \in [M].$$

*Proof.* Denote $\quad\tilde{\mu}_{h,i}^{(n)}(\cdot|s) \quad := \quad \arg\max_\mu\left(\mathbb{D}_{\mu,\pi_{h,-i}^{(n)}}Q_{h,i}^{\dagger,\pi_{-i}^{(n)}}\right)(s) \quad$ and let $\quad\tilde{\pi}_h^{(n)} \quad =$ $\tilde{\mu}_{h,i}^{(n)} \times \pi_{h,-i}^{(n)}.$ Let $f_h^{(n)}(s,\boldsymbol{a}) = \left\|\hat{P}_h^{(n)}(\cdot|s,\boldsymbol{a}) - P_h^\star(\cdot|s,\boldsymbol{a})\right\|_1$ and $f_{h,i}^{(n)}(s[Z_i],\boldsymbol{a}_i) = \left\|\hat{P}_{h,i}^{(n)}(\cdot|s[Z_i],\boldsymbol{a}_i) - P_{h,i}^\star(\cdot|s[Z_i],\boldsymbol{a}_i)\right\|_1.$ Then according to the event $\mathcal{E}$, we have

$$\mathbb{E}_{(s,\boldsymbol{a})\sim\rho_h^{(n)}}\left[\left(f_{h,i}^{(n)}(s[Z_i],\boldsymbol{a}_i)\right)^2\right] \leq \zeta^{(n)}, \quad \mathbb{E}_{(s,\boldsymbol{a})\sim\tilde{\rho}_h^{(n)}}\left[\left(f_{h,i}^{(n)}(s[Z_i],\boldsymbol{a}_i)\right)^2\right] \leq \zeta^{(n)}, \quad \forall n \in [N], h \in [H], i \in [M]$$

$$\left\|\bar{\phi}_{h,i}(s,\boldsymbol{a})\right\|_{\left(\hat{\Sigma}_{h,\bar{\phi}_{h,i}}^{(n)}\right)^{-1}} = \Theta\left(\left\|\bar{\phi}_{h,i}(s,\boldsymbol{a})\right\|_{\Sigma_{n,\rho_h^{(n)},\bar{\phi}_{h,i}}^{-1}}\right), \quad \forall n \in [N], h \in [H], \bar{\phi}_{h,i} \in \bar{\Phi}_{h,i}, i \in [M].$$

A direct conclusion of the event $\mathcal{E}$ is we can find an absolute constant $c$, such that

$$\beta_h^{(n)}(s,\boldsymbol{a}) = \sum_{i=1}^M \min\left\{\alpha^{(n)}\left\|\bar{\phi}_{h,i}^{(n)}(\tilde{s},\tilde{\boldsymbol{a}})\right\|_{\left(\Sigma_{h,\bar{\phi}_{h,i}^{(n)}}^{(n)}\right)^{-1}}, H\right\}$$

$$\geq \sum_{i=1}^M \min\left\{c\alpha^{(n)}\left\|\bar{\phi}_{h,i}^{(n)}(\tilde{s},\tilde{\boldsymbol{a}})\right\|_{\Sigma_{n,\rho_h^{(n)},\bar{\phi}_{h,i}^{(n)}}^{-1}}, H\right\}, \quad \forall n \in [N], h \in [H].$$

Next, similar to the proof in Lemma B.5, we may prove

$$\mathbb{E}_{s\sim d_1}\left[\overline{V}_{1,i}^{(n)}(s) - V_{1,i}^{\dagger,\pi_{-i}^{(n)}}(s)\right] \geq \sum_{h=1}^H \mathbb{E}_{(s,\boldsymbol{a})\sim d_{\hat{P}^{(n)},h}^{\tilde{\pi}^{(n)}}}\left[\hat{\beta}_h^{(n)}(s,\boldsymbol{a})\right] - H\sum_{h=1}^H \mathbb{E}_{(s,\boldsymbol{a})\sim d_{\hat{P}^{(n)},h}^{\tilde{\pi}^{(n)}}}\left[\min\left\{f_h^{(n)}(s,\boldsymbol{a}),1\right\}\right].$$

For the second term, note that we have the relation $\min\{f_h^{(n)}(s,\boldsymbol{a}),1\} \leq \sum_{i=1}^M \min\{f_{h,i}^{(n)}(s[Z_i],\boldsymbol{a}_i),1\}$. By Lemma D.2, we have for $h=1$,

$$\mathbb{E}_{(s,\boldsymbol{a})\sim d_{\hat{P}^{(n)},1}^{\tilde{\pi}^{(n)}}}\left[\min\left\{f_{1,i}^{(n)}(s[Z_i],\boldsymbol{a}_i),1\right\}\right] \leq \sqrt{A\mathbb{E}_{(s,\boldsymbol{a})\sim\rho_1^{(n)}}\left[\left(f_{1,i}^{(n)}(s[Z_i],\boldsymbol{a}_i)\right)^2\right]} \leq \sqrt{\tilde{A}\zeta^{(n)}}.$$

And $\forall h \geq 2$, we have

$$\mathbb{E}_{(s,\boldsymbol{a})\sim d^{\tilde{\pi}^{(n)}}_{\hat{P}^{(n)},h}} \left[ \min\left\{ f^{(n)}_{h,i}(s[Z_i],\boldsymbol{a}_i),1 \right\} \right]$$

$$\lesssim \mathbb{E}_{(\tilde{s},\tilde{\boldsymbol{a}})\sim d^{\tilde{\pi}^{(n)}}_{\hat{P}^{(n)},h-1}} \left[ \min\left\{ \tilde{A} \left\| \bar{\phi}^{(n)}_{h-1,i}(\tilde{s},\tilde{\boldsymbol{a}}) \right\|_{\Sigma^{-1}_{n,\rho^{(n)}_{h-1},\bar{\phi}^{(n)}_{h-1,i}}} \sqrt{n\mathbb{E}_{(s,\boldsymbol{a})\sim\tilde{\rho}^{(n)}_h}\left[ \left( f^{(n)}_{h,i}(s[Z_i],\boldsymbol{a}_i) \right)^2 \right] + d^L\lambda + n\zeta^{(n)}}, 1 \right\} \right]$$

$$\lesssim \mathbb{E}_{(\tilde{s},\tilde{\boldsymbol{a}})\sim d^{\tilde{\pi}^{(n)}}_{\hat{P}^{(n)},h-1}} \left[ \min\left\{ \tilde{A} \left\| \bar{\phi}^{(n)}_{h-1,i}(\tilde{s},\tilde{\boldsymbol{a}}) \right\|_{\Sigma^{-1}_{n,\rho^{(n)}_{h-1},\bar{\phi}^{(n)}_{h-1,i}}} \sqrt{n\zeta^{(n)} + d^L\lambda}, 1 \right\} \right]$$

Note that we here use $\min\{f^{(n)}_{h,i}(s[Z_i],\boldsymbol{a}_i),1\} \leq 1$, $\mathbb{E}_{(s,\boldsymbol{a})\sim\rho^{(n)}_h}\left[ \left( f^{(n)}_{h,i}(s[Z_i],\boldsymbol{a}_i) \right)^2 \right] \leq \zeta^{(n)}$ and

$\mathbb{E}_{(s,\boldsymbol{a})\sim\tilde{\rho}^{(n)}_h}\left[ \left( f^{(n)}_{h,i}(s[Z_i],\boldsymbol{a}_i) \right)^2 \right] \leq \zeta^{(n)}$. Then according to our choice of $\alpha^{(n)}$, we get

$$\mathbb{E}_{(s,\boldsymbol{a})\sim d^{\tilde{\pi}^{(n)}}_{\hat{P}^{(n)},h}} \left[ \min\left\{ f^{(n)}_{h,i}(s[Z_i],\boldsymbol{a}_i),1 \right\} \right] \leq \mathbb{E}_{(\tilde{s},\tilde{\boldsymbol{a}})\sim d^{\tilde{\pi}^{(n)}}_{\hat{P}^{(n)},h-1}} \left[ \min\left\{ \frac{c\alpha^{(n)}}{H} \left\| \bar{\phi}^{(n)}_{h-1,i}(\tilde{s},\tilde{\boldsymbol{a}}) \right\|_{\Sigma^{-1}_{n,\rho^{(n)}_{h-1},\bar{\phi}^{(n)}_{h-1,i}}}, 1 \right\} \right].$$

Combining all things together,

$$\bar{v}^{(n)}_i - v^{\dagger,\pi^{(n)}_{-i}}_i = \mathbb{E}_{s\sim d_1} \left[ \overline{V}^{(n)}_{1,i}(s) - V^{\dagger,\pi^{(n)}_{-i}}_{1,i}(s) \right]$$

$$\geq \sum_{h=1}^{H} \mathbb{E}_{(s,\boldsymbol{a})\sim d^{\tilde{\pi}^{(n)}}_{\hat{P}^{(n)},h}} \left[ \hat{\beta}^{(n)}_h(s,\boldsymbol{a}) \right] - H\sum_{h=1}^{H} \mathbb{E}_{(s,\boldsymbol{a})\sim d^{\tilde{\pi}^{(n)}}_{\hat{P}^{(n)},h}} \left[ \min\left\{ f^{(n)}_h(s,\boldsymbol{a}),1 \right\} \right]$$

$$\geq \sum_{h=1}^{H-1} \mathbb{E}_{(s,\boldsymbol{a})\sim d^{\tilde{\pi}^{(n)}}_{\hat{P}^{(n)},h}} \left[ \hat{\beta}^{(n)}_h(s,\boldsymbol{a}) - \sum_{j=1}^{M} \min\left\{ c\alpha^{(n)} \left\| \bar{\phi}^{(n)}_{h,j}(s,\boldsymbol{a}) \right\|_{\Sigma^{-1}_{\rho^{(n)}_h,\bar{\phi}^{(n)}_{h,j}}}, H \right\} \right] - HM\sqrt{\tilde{A}\zeta^{(n)}}$$

$$\geq -HM\sqrt{\tilde{A}\zeta^{(n)}},$$

which proves the inequality. $\qquad\square$

**Lemma D.6** (Optimism for CE). *Consider an episode $n \in [N]$ and set $\alpha^{(n)} = \Theta\left( H\tilde{A}\sqrt{n\zeta^{(n)} + d^L\lambda} \right)$. When the event $\mathcal{E}$ holds, we have*

$$\bar{v}^{(n)}_i(s) - \max_{\omega\in\Omega_i} v^{\omega\circ\pi^{(n)}}_i(s) \geq -HM\sqrt{A\zeta^{(n)}}, \quad \forall n\in[N], i\in[M].$$

*Proof.* Denote $\tilde{\omega}^{(n)}_{h,i} = \arg\max_{\omega_h\in\Omega_{h,i}} \left( \mathbb{D}_{\omega_h\circ\pi^{(n)}_h} \max_{\omega\in\Omega_i} Q^{\omega\circ\pi^{(n)}}_{h,i} \right)(s)$ and let $\tilde{\pi}^{(n)}_h = \tilde{\omega}_{h,i} \circ \pi^{(n)}_h$. Let $f^{(n)}_h(s,\boldsymbol{a}) = \left\| \hat{P}^{(n)}_h(\cdot|s,\boldsymbol{a}) - P^\star_h(\cdot|s,\boldsymbol{a}) \right\|_1$ and $f^{(n)}_{h,i}(s[Z_i],\boldsymbol{a}_i) = \left\| \hat{P}^{(n)}_{h,i}(\cdot|s[Z_i],\boldsymbol{a}_i) - P^\star_{h,i}(\cdot|s[Z_i],\boldsymbol{a}_i) \right\|_1$. Then according to the event $\mathcal{E}$, we have

$$\mathbb{E}_{(s,\boldsymbol{a})\sim\rho^{(n)}_h}\left[ \left( f^{(n)}_{h,i}(s[Z_i],\boldsymbol{a}_i) \right)^2 \right] \leq \zeta^{(n)}, \quad \mathbb{E}_{(s,\boldsymbol{a})\sim\tilde{\rho}^{(n)}_h}\left[ \left( f^{(n)}_{h,i}(s[Z_i],\boldsymbol{a}_i) \right)^2 \right] \leq \zeta^{(n)}, \quad \forall n\in[N], h\in[H], i\in[M]$$

$$\left\| \bar{\phi}_{h,i}(s,\boldsymbol{a}) \right\|_{\left(\hat{\Sigma}^{(n)}_{h,\bar{\phi}_{h,i}}\right)^{-1}} = \Theta\left( \left\| \bar{\phi}_{h,i}(s,\boldsymbol{a}) \right\|_{\Sigma^{-1}_{n,\rho^{(n)}_h,\bar{\phi}_{h,i}}} \right), \quad \forall n\in[N], h\in[H], \bar{\phi}_{h,i}\in\bar{\Phi}_{h,i}, i\in[M].$$

A direct conclusion of the event $\mathcal{E}$ is we can find an absolute constant $c$, such that

$$\beta^{(n)}_h(s,\boldsymbol{a}) = \min\left\{ \alpha^{(n)} \sum_{i=1}^{M} \left\| \bar{\phi}^{(n)}_{h,i}(\tilde{s},\tilde{\boldsymbol{a}}) \right\|_{\left(\Sigma^{(n)}_{h,\bar{\phi}^{(n)}_{h,i}}\right)^{-1}}, H \right\}$$

$$\geq c \min \left\{ \alpha^{(n)} \sum_{i=1}^{M} \left\| \bar{\phi}_{h,i}^{(n)}(\tilde{s}, \tilde{\boldsymbol{a}}) \right\|_{\Sigma_{n,\rho_h^{(n)},\bar{\phi}_{h,i}^{(n)}}^{-1}} , H \right\}, \quad \forall n \in [N], h \in [H].$$

Next, similar to the proof in Lemma B.6, we may prove

$$\mathbb{E}_{s \sim d_1} \left[ \overline{V}_{1,i}^{(n)}(s) - \max_{\omega \in \Omega_i} V_{1,i}^{\omega \circ \pi^{(n)}}(s) \right] \geq \sum_{h=1}^{H} \mathbb{E}_{(s,\boldsymbol{a}) \sim d_{\hat{P}^{(n)},h}^{\tilde{\pi}^{(n)}}} \left[ \hat{\beta}_h^{(n)}(s, \boldsymbol{a}) \right] - H \sum_{h=1}^{H} \mathbb{E}_{(s,\boldsymbol{a}) \sim d_{\hat{P}^{(n)},h}^{\tilde{\pi}^{(n)}}} \left[ \min \left\{ f_h^{(n)}(s, \boldsymbol{a}), 1 \right\} \right].$$

Note that we can use exactly the same steps in the proof of Lemma D.5 to bound the second term, and we get for $h = 1$,

$$\mathbb{E}_{(s,\boldsymbol{a}) \sim d_{\hat{P}^{(n)},1}^{\tilde{\pi}^{(n)}}} \left[ \min \left\{ f_{1,i}^{(n)}(s[Z_i], \boldsymbol{a}_i), 1 \right\} \right] \leq \sqrt{\tilde{A} \zeta^{(n)}}.$$

And $\forall h \geq 2$,

$$\mathbb{E}_{(s,\boldsymbol{a}) \sim d_{\hat{P}^{(n)},h}^{\tilde{\pi}^{(n)}}} \left[ \min \left\{ f_{h,i}^{(n)}(s[Z_i], \boldsymbol{a}_i), 1 \right\} \right] \leq \frac{c \alpha^{(n)}}{H} \mathbb{E}_{(\tilde{s},\tilde{\boldsymbol{a}}) \sim d_{\hat{P}^{(n)},h-1}^{\tilde{\pi}^{(n)}}} \left[ \left\| \bar{\phi}_{h-1,i}^{(n)}(\tilde{s}, \tilde{\boldsymbol{a}}) \right\|_{\Sigma_{n,\rho_{h-1}^{(n)},\bar{\phi}_{h-1,i}^{(n)}}^{-1}} \right].$$

Combining all things together,

$$\overline{v}_i^{(n)} - \max_{\omega \in \Omega_i} v_i^{\omega \circ \pi^{(n)}}$$

$$= \mathbb{E}_{s \sim d_1} \left[ \overline{V}_{1,i}^{(n)}(s) - \max_{\omega \in \Omega_i} V_{1,i}^{\omega \circ \pi^{(n)}}(s) \right]$$

$$\geq \sum_{h=1}^{H} \mathbb{E}_{(s,\boldsymbol{a}) \sim d_{\hat{P}^{(n)},h}^{\tilde{\pi}^{(n)}}} \left[ \hat{\beta}_h^{(n)}(s, \boldsymbol{a}) \right] - H \sum_{h=1}^{H} \mathbb{E}_{(s,\boldsymbol{a}) \sim d_{\hat{P}^{(n)},h}^{\tilde{\pi}^{(n)}}} \left[ \min \left\{ f_h^{(n)}(s, \boldsymbol{a}), 1 \right\} \right]$$

$$\geq \sum_{h=1}^{H-1} \mathbb{E}_{(s,\boldsymbol{a}) \sim d_{\hat{P}^{(n)},h}^{\tilde{\pi}^{(n)}}} \left[ \hat{\beta}_h^{(n)}(s, \boldsymbol{a}) - \sum_{j=1}^{M} \min \left( c \alpha^{(n)} \left\| \bar{\phi}_{h,j}^{(n)}(s, \boldsymbol{a}) \right\|_{\Sigma_{\rho_h^{(n)},\bar{\phi}_{h,j}^{(n)}}^{-1}} , H \right) \right] - HM \sqrt{\tilde{A} \zeta^{(n)}}$$

$$\geq - HM \sqrt{\tilde{A} \zeta^{(n)}},$$

which proves the inequality. $\qquad \square$

**Lemma D.7** (pessimism). *Consider an episode $n \in [N]$ and set $\alpha^{(n)} = \Theta\left( H \tilde{A} \sqrt{n \zeta^{(n)} + d^L \lambda} \right)$. When the event $\mathcal{E}$ holds, we have*

$$\underline{v}_i^{(n)}(s) - v_i^{\pi^{(n)}}(s) \leq HM \sqrt{\tilde{A} \zeta^{(n)}}, \quad \forall n \in [N], i \in [M].$$

*Proof.* Let $f_h^{(n)}(s, \boldsymbol{a}) = \left\| \hat{P}_h^{(n)}(\cdot | s, \boldsymbol{a}) - P_h^\star(\cdot | s, \boldsymbol{a}) \right\|_1$ and $f_{h,i}^{(n)}(s[Z_i], \boldsymbol{a}_i) = \left\| \hat{P}_{h,i}^{(n)}(\cdot | s[Z_i], \boldsymbol{a}_i) - P_{h,i}^\star(\cdot | s[Z_i], \boldsymbol{a}_i) \right\|_1$. Then according to the event $\mathcal{E}$, we have

$$\mathbb{E}_{(s,\boldsymbol{a}) \sim \rho_h^{(n)}} \left[ \left( f_{h,i}^{(n)}(s[Z_i], \boldsymbol{a}_i) \right)^2 \right] \leq \zeta^{(n)}, \quad \mathbb{E}_{(s,\boldsymbol{a}) \sim \tilde{\rho}_h^{(n)}} \left[ \left( f_{h,i}^{(n)}(s[Z_i], \boldsymbol{a}_i) \right)^2 \right] \leq \zeta^{(n)}, \quad \forall n \in [N], h \in [H], i \in [M]$$

$$\left\| \bar{\phi}_{h,i}(s, \boldsymbol{a}) \right\|_{\left( \hat{\Sigma}_{h,\bar{\phi}_{h,i}}^{(n)} \right)^{-1}} = \Theta\left( \left\| \bar{\phi}_{h,i}(s, \boldsymbol{a}) \right\|_{\Sigma_{n,\rho_h^{(n)},\bar{\phi}_{h,i}}^{-1}} \right), \quad \forall n \in [N], h \in [H], \bar{\phi}_{h,i} \in \bar{\Phi}_{h,i}, i \in [M].$$

A direct conclusion of the event $\mathcal{E}$ is we can find an absolute constant $c$, such that

$$\beta_h^{(n)}(s, \boldsymbol{a}) = \sum_{i=1}^{M} \min \left\{ \alpha^{(n)} \left\| \bar{\phi}_{h,i}^{(n)}(\tilde{s}, \tilde{\boldsymbol{a}}) \right\|_{\left( \Sigma_{h,\bar{\phi}_{h,i}^{(n)}}^{(n)} \right)^{-1}} , H \right\}$$

$$\geq \sum_{i=1}^{M} \min \left\{ c\alpha^{(n)} \left\| \bar{\phi}_{h,i}^{(n)}(\tilde{s}, \tilde{\boldsymbol{a}}) \right\|_{\Sigma_{n,\rho_h^{(n)},\bar{\phi}_{h,i}^{(n)}}^{-1}}, H \right\}, \quad \forall n \in [N], h \in [H].$$

Next, similar to the proof in Lemma B.7, we may prove

$$\mathbb{E}_{s \sim d_{\hat{P}^{(n)},h}^{\pi^{(n)}}} \left[ \underline{V}_{h,i}^{(n)}(s) - V_{h,i}^{\pi^{(n)}}(s) \right] \leq \sum_{h'=h}^{H} \mathbb{E}_{(s,\boldsymbol{a}) \sim d_{\hat{P}^{(n)},h'}^{\pi^{(n)}}} \left[ -\hat{\beta}_{h'}^{(n)}(s, \boldsymbol{a}) + H \min \left\{ f_{h'}^{(n)}(s, \boldsymbol{a}), 1 \right\} \right], \quad \forall h \in [H].$$

$$(22)$$

and we get for $h = 1$,

$$\mathbb{E}_{(s,\boldsymbol{a}) \sim d_{\hat{P}^{(n)},1}^{\tilde{\pi}^{(n)}}} \left[ \min \left\{ f_{1,i}^{(n)}(s[Z_i], \boldsymbol{a}_i), 1 \right\} \right] \leq \sqrt{\tilde{A}\zeta^{(n)}}.$$

And $\forall h \geq 2$,

$$\mathbb{E}_{(s,\boldsymbol{a}) \sim d_{\hat{P}^{(n)},h}^{\tilde{\pi}^{(n)}}} \left[ \min \left\{ f_{h,i}^{(n)}(s[Z_i], \boldsymbol{a}_i), 1 \right\} \right] \leq \frac{c\alpha^{(n)}}{H} \mathbb{E}_{(\tilde{s},\tilde{\boldsymbol{a}}) \sim d_{\hat{P}^{(n)},h-1}^{\tilde{\pi}^{(n)}}} \left[ \left\| \bar{\phi}_{h-1,i}^{(n)}(\tilde{s}, \tilde{\boldsymbol{a}}) \right\|_{\Sigma_{n,\rho_{h-1}^{(n)},\bar{\phi}_{h-1,i}^{(n)}}^{-1}} \right].$$

Finally, we get

$$\begin{aligned}
\underline{v}_i^{(n)} - v_i^{\pi^{(n)}} &= \mathbb{E}_{s \sim d_1} \left[ \underline{V}_{1,i}^{(n)}(s) - V_{1,i}^{\pi^{(n)}}(s) \right] \\
&\leq \sum_{h=1}^{H-1} \mathbb{E}_{(s,\boldsymbol{a}) \sim d_{\hat{P}^{(n)},h}^{\pi^{(n)}}} \left[ -\hat{\beta}_h^{(n)}(s, \boldsymbol{a}) + \sum_{j=1}^{M} \min \left( c\alpha^{(n)} \left\| \bar{\phi}_{h,j}^{(n)}(s, \boldsymbol{a}) \right\|_{\Sigma_{\rho_h^{(n)},\bar{\phi}_{h,j}^{(n)}}^{-1}}, H \right) \right] + HM\sqrt{\tilde{A}\zeta^{(n)}} \\
&\leq HM\sqrt{\tilde{A}\zeta^{(n)}},
\end{aligned}$$

which has finished the proof. $\qquad \square$

**Lemma D.8.** *When the event $\mathcal{E}$ holds and $\alpha^{(n)} = \Theta\left(H\tilde{A}\sqrt{n\zeta^{(n)} + d^L\lambda}\right)$ satisfies $\alpha^{(1)} \leq \alpha^{(2)} \leq \ldots \leq \alpha^{(N)}$, we have*

$$\sum_{n=1}^{N} \Delta^{(n)} \lesssim H^2 M d^{L^2} A^L \sqrt{N \log\left(1 + \frac{N}{d\lambda}\right)} \alpha^{(N)}.$$

*Proof.* Let $f_h^{(n)}(s, \boldsymbol{a}) = \left\| \hat{P}_h^{(n)}(\cdot|s, \boldsymbol{a}) - P_h^\star(\cdot|s, \boldsymbol{a}) \right\|_1$ and $f_{h,i}^{(n)}(s[Z_i], \boldsymbol{a}_i) = \left\| \hat{P}_{h,i}^{(n)}(\cdot|s[Z_i], \boldsymbol{a}_i) - P_{h,i}^\star(\cdot|s[Z_i], \boldsymbol{a}_i) \right\|_1$. Then according to the event $\mathcal{E}$, we have

$$\mathbb{E}_{(s,\boldsymbol{a}) \sim \rho_h^{(n)}} \left[ \left( f_{h,i}^{(n)}(s[Z_i], \boldsymbol{a}_i) \right)^2 \right] \leq \zeta^{(n)}, \quad \mathbb{E}_{(s,\boldsymbol{a}) \sim \tilde{\rho}_h^{(n)}} \left[ \left( f_{h,i}^{(n)}(s[Z_i], \boldsymbol{a}_i) \right)^2 \right] \leq \zeta^{(n)}, \quad \forall n \in [N], h \in [H], i \in [M]$$

$$\left\| \bar{\phi}_{h,i}(s, \boldsymbol{a}) \right\|_{\left(\hat{\Sigma}_{h,\bar{\phi}_{h,i}}^{(n)}\right)^{-1}} = \Theta\left( \left\| \bar{\phi}_{h,i}(s, \boldsymbol{a}) \right\|_{\Sigma_{n,\rho_h^{(n)},\bar{\phi}_{h,i}}^{-1}} \right), \quad \forall n \in [N], h \in [H], \bar{\phi}_{h,i} \in \bar{\Phi}_{h,i}, i \in [M].$$

By definition, we have

$$\Delta^{(n)} = \max_{i \in [M]} \left\{ \bar{v}_i^{(n)} - \underline{v}_i^{(n)} \right\} + 2HM\sqrt{\tilde{A}\zeta^{(n)}}.$$

With similar steps as those in the proof of Lemma B.8 (note that $\overline{V}_{h,i}^{(n)}(s) - \underline{V}_{h,i}^{(n)}(s)$ is upper bounded by $2H^2M$), we have

$$\mathbb{E}_{s \sim d_{P^\star,1}^{\pi^{(n)}}} \left[ \overline{V}_{1,i}^{(n)}(s) - \underline{V}_{1,i}^{(n)}(s) \right] \leq 2 \underbrace{\sum_{h=1}^{H} \mathbb{E}_{(s,\boldsymbol{a}) \sim d_{P^\star,h}^{\pi^{(n)}}} \left[ \hat{\beta}_h^{(n)}(s, \boldsymbol{a}) \right]}_{(a)} + 2H^2M \underbrace{\sum_{h=1}^{H} \mathbb{E}_{(s,\boldsymbol{a}) \sim d_{P^\star,h}^{\pi^{(n)}}} \left[ f_h^{(n)}(s, \boldsymbol{a}) \right]}_{(b)}.$$

$$(23)$$

First, we calculate the first term (a) in Inequality equation 23. Following Lemma D.4, we have

$$
\sum_{h=1}^{H} \mathbb{E}_{(s,\boldsymbol{a}) \sim d_{P^{\star},h}^{\pi^{(n)}}} \left[ \hat{\beta}_h^{(n)}(s,\boldsymbol{a}) \right]
$$

$$
\lesssim \sum_{h=1}^{H} \sum_{i=1}^{M} \mathbb{E}_{(s,\boldsymbol{a}) \sim d_{P^{\star},h}^{\pi^{(n)}}} \left[ \min\left( \alpha^{(n)} \left\| \bar{\phi}_{h,i}^{(n)}(s,\boldsymbol{a}) \right\|_{\Sigma_{n,\rho_h^{(n)},\bar{\phi}_{h,i}^{(n)}}^{-1}}, H \right) \right]
$$

$$
\lesssim \sum_{h=1}^{H-1} \sum_{i=1}^{M} \tilde{A}^L \mathbb{E}_{(\tilde{s},\tilde{\boldsymbol{a}}) \sim d_{P^{\star},h}^{\pi^{(n)}}} \left[ \left\| \tilde{\phi}_{h,i}^{\star}(\tilde{s},\tilde{\boldsymbol{a}}) \right\|_{\Sigma_{n,\gamma_h^{(n)},\bar{\phi}_{h,i}^{\star}}^{-1}} \right]
$$

$$
\cdot \sqrt{ n\left(\alpha^{(n)}\right)^2 \mathbb{E}_{(s,\boldsymbol{a}) \sim \rho_h^{(n)}} \left[ \left\| \bar{\phi}_{h,i}^{(n)}(s,\boldsymbol{a}) \right\|_{\Sigma_{n,\rho_h^{(n)},\bar{\phi}_{h,i}^{(n)}}^{-1}}^2 \right] + H^2 d^{L^2} \lambda }
$$

$$
+ \sqrt{ \tilde{A}^L \left(\alpha^{(n)}\right)^2 \mathbb{E}_{(s,\boldsymbol{a}) \sim \rho_1^{(n)}} \left[ \left\| \bar{\phi}_{1,i}^{(n)}(s,\boldsymbol{a}) \right\|_{\Sigma_{n,\rho_1^{(n)},\bar{\phi}_{1,i}^{(n)}}^{-1}}^2 \right] } .
$$

Note that we use the fact that $B = H$ when applying Lemma D.3. In addition, we have

$$
n\mathbb{E}_{(s,\boldsymbol{a}) \sim \rho_h^{(n)}} \left[ \left\| \bar{\phi}_{h,i}^{(n)}(s,\boldsymbol{a}) \right\|_{\Sigma_{n,\rho_h^{(n)},\bar{\phi}_{h,i}^{(n)}}^{-1}}^2 \right]
$$

$$
= n\mathrm{Tr}\left( \mathbb{E}_{(s,\boldsymbol{a}) \sim \rho_h^{(n)}} \left[ \bar{\phi}_{h,i}^{(n)}(s,\boldsymbol{a}) \bar{\phi}_{h,i}^{(n)}(s,\boldsymbol{a})^{\top} \right] \left( n\mathbb{E}_{(s,\boldsymbol{a}) \sim \rho_h^{(n)}} \left[ \bar{\phi}_{h,i}^{(n)}(s,\boldsymbol{a}) \bar{\phi}_{h,i}^{(n)}(s,\boldsymbol{a})^{\top} \right] + \lambda I_{d^{|z_i|}} \right)^{-1} \right)
$$

$$
\leq d^L .
$$

Then,

$$
\sum_{h=1}^{H} \mathbb{E}_{(s,\boldsymbol{a}) \sim d_{P^{\star},h}^{\pi^{(n)}}} \left[ \hat{\beta}_h^{(n)}(s,\boldsymbol{a}) \right]
$$

$$
\leq \sum_{h=1}^{H-1} \sum_{i=1}^{M} \tilde{A}^L \mathbb{E}_{(\tilde{s},\tilde{\boldsymbol{a}}) \sim d_{P^{\star},h}^{\pi^{(n)}}} \left[ \left\| \tilde{\phi}_{h,i}^{\star}(\tilde{s},\tilde{\boldsymbol{a}}) \right\|_{\tilde{\Sigma}_{n,\gamma_h^{(n)},\bar{\phi}_{h,i}^{\star}}^{-1}} \right] \sqrt{ d^L \left(\alpha^{(n)}\right)^2 + H^2 d^{L^2} \lambda }
$$

$$
+ \sqrt{ d^L \tilde{A}^L \left(\alpha^{(n)}\right)^2 / n } .
$$

Second, we calculate the term (b) in inequality equation 23. Following Lemma D.3 and noting $f_{h,i}^{(n)}(s[Z_i], \boldsymbol{a}_i)$ is upper-bounded by 2 (i.e., $B = 2$ in Lemma D.3), we have

$$
\sum_{h=1}^{H} \mathbb{E}_{(s,\boldsymbol{a}) \sim d_{P^{\star},h}^{\pi^{(n)}}} \left[ f_h^{(n)}(s,\boldsymbol{a}) \right]
$$

$$
\leq \sum_{i=1}^{M} \sum_{h=1}^{H} \mathbb{E}_{(s,\boldsymbol{a}) \sim d_{P^{\star},h}^{\pi^{(n)}}} \left[ f_{h,i}^{(n)}(s[Z_i], \boldsymbol{a}_i) \right]
$$

$$
\leq \sum_{i=1}^{M} \sum_{h=1}^{H-1} \tilde{A} \mathbb{E}_{(\tilde{s},\tilde{\boldsymbol{a}}) \sim d_{P^{\star},h}^{\pi^{(n)}}} \left[ \left\| \bar{\phi}_{h,i}^{\star}(\tilde{s},\tilde{\boldsymbol{a}}) \right\|_{\Sigma_{n,\gamma_h^{(n)},\bar{\phi}_{h,i}^{\star}}^{-1}} \right] \sqrt{ n\mathbb{E}_{(s,\boldsymbol{a}) \sim \rho_h^{(n)}} \left[ \left( f_{h,i}^{(n)}(s[Z_i], \boldsymbol{a}_i) \right)^2 \right] + d^L \lambda }
$$

$$
+ \sqrt{ \tilde{A} \mathbb{E}_{(s,\boldsymbol{a}) \sim \rho_h^{(n)}} \left[ \left( f_1^{(n)}(s[Z_j], \boldsymbol{a}_j) \right)^2 \right] }
$$

$$\leq \sum_{i=1}^{M}\sum_{h=1}^{H-1}\tilde{A}\,\mathbb{E}_{(\tilde{s},\tilde{\boldsymbol{a}})\sim d_{P^\star,h}^{\pi^{(n)}}}\left[\left\|\bar{\phi}_{h,i}^{\star}(\tilde{s},\tilde{\boldsymbol{a}})\right\|_{\Sigma_{n,\gamma_h^{(n)},\bar{\phi}_{h,i}^{\star}}^{-1}}\right]\sqrt{n\zeta^{(n)}+d^L\lambda}+\sqrt{\tilde{A}\zeta^{(n)}}$$

$$\lesssim \frac{\alpha^{(n)}}{H}\sum_{i=1}^{M}\sum_{h=1}^{H-1}\mathbb{E}_{(\tilde{s},\tilde{\boldsymbol{a}})\sim d_{P^\star,h}^{\pi^{(n)}}}\left[\left\|\bar{\phi}_{h,i}^{\star}(\tilde{s},\tilde{\boldsymbol{a}})\right\|_{\Sigma_{n,\gamma_h^{(n)},\bar{\phi}_{h,i}^{\star}}^{-1}}\right]+\sqrt{\tilde{A}\zeta^{(n)}},$$

where in the second inequality, we use $\mathbb{E}_{(s,\boldsymbol{a})\sim\rho_h^{(n)}}\left[\left(f_{h,i}^{(n)}(s[Z_i],\boldsymbol{a}_i)\right)^2\right]\leq\zeta^{(n)}$, and in the last line,

recall $\tilde{A}\sqrt{n\zeta^{(n)}+d^L\lambda}\lesssim\alpha^{(n)}/H$. Then, by combining the above calculation of the term (a) and term (b) in inequality equation 23, we have:

$$\overline{v}_i^{(n)}-\underline{v}_i^{(n)}$$

$$=\mathbb{E}_{s\sim d_{P^\star,1}^{\pi^{(n)}}}\left[\overline{V}_{1,i}^{(n)}(s)-\underline{V}_{1,i}^{(n)}(s)\right]$$

$$\lesssim\sum_{i=1}^{M}\sum_{h=1}^{H-1}\left(\tilde{A}^L\mathbb{E}_{(\tilde{s},\tilde{\boldsymbol{a}})\sim d_{P^\star,h}^{\pi^{(n)}}}\left[\left\|\tilde{\phi}_{h,i}^{\star}(\tilde{s},\tilde{\boldsymbol{a}})\right\|_{\Sigma_{n,\gamma_h^{(n)},\tilde{\phi}_{h,i}^{\star}}^{-1}}\right]\sqrt{d^L\left(\alpha^{(n)}\right)^2+H^2d^{L^2}\lambda}+\sqrt{\frac{d^L\tilde{A}^L\left(\alpha^{(n)}\right)^2}{n}}\right)$$

$$+H^2M\sum_{i=1}^{M}\sum_{h=1}^{H-1}\left(\frac{\alpha^{(n)}}{H}\mathbb{E}_{(\tilde{s},\tilde{\boldsymbol{a}})\sim d_{P^\star,h}^{\pi^{(n)}}}\left[\left\|\bar{\phi}_{h,i}^{\star}(\tilde{s},\tilde{\boldsymbol{a}})\right\|_{\Sigma_{n,\gamma_h^{(n)},\bar{\phi}_{h,i}^{\star}}^{-1}}\right]+\sqrt{\tilde{A}\zeta^{(n)}}\right).$$

Taking maximum over $i$ on both sides and use the definition of $\Delta^{(n)}$, we get

$$\Delta^{(n)}=\max_{i\in[M]}\left\{\overline{v}_i^{(n)}-\underline{v}_i^{(n)}\right\}+2HM\sqrt{\tilde{A}\zeta^{(n)}}$$

$$\lesssim\sum_{i=1}^{M}\sum_{h=1}^{H-1}\left(\tilde{A}^L\mathbb{E}_{(\tilde{s},\tilde{\boldsymbol{a}})\sim d_{P^\star,h}^{\pi^{(n)}}}\left[\left\|\tilde{\phi}_{h,i}^{\star}(\tilde{s},\tilde{\boldsymbol{a}})\right\|_{\tilde{\Sigma}_{n,\gamma_h^{(n)},\tilde{\phi}_{h,i}^{\star}}^{-1}}\right]\sqrt{d^L\left(\alpha^{(n)}\right)^2+H^2d^{L^2}\lambda}+\sqrt{\frac{d^L\tilde{A}^L\left(\alpha^{(n)}\right)^2}{n}}\right)$$

$$+H^2M\sum_{i=1}^{M}\sum_{h=1}^{H-1}\left(\frac{\alpha^{(n)}}{H}\mathbb{E}_{(\tilde{s},\tilde{\boldsymbol{a}})\sim d_{P^\star,h}^{\pi^{(n)}}}\left[\left\|\bar{\phi}_{h,i}^{\star}(\tilde{s},\tilde{\boldsymbol{a}})\right\|_{\Sigma_{n,\gamma_h^{(n)},\bar{\phi}_{h,i}^{\star}}^{-1}}\right]+\sqrt{\tilde{A}\zeta^{(n)}}\right).$$

Hereafter, we take the dominating term out. Note that

$$\sum_{n=1}^{N}\mathbb{E}_{(\tilde{s},\tilde{\boldsymbol{a}})\sim d_{P^\star,h}^{\pi^{(n)}}}\left[\left\|\tilde{\phi}_{h,i}^{\star}(\tilde{s},\tilde{\boldsymbol{a}})\right\|_{\Sigma_{n,\gamma_h^{(n)},\tilde{\phi}_{h,i}^{\star}}^{-1}}\right]$$

$$\leq\sqrt{N\sum_{n=1}^{N}\mathbb{E}_{(\tilde{s},\tilde{\boldsymbol{a}})\sim d_{P^\star,h}^{\pi^{(n)}}}\left[\tilde{\phi}_{h,i}^{\star}(\tilde{s},\tilde{\boldsymbol{a}})^\top\Sigma_{n,\gamma_h^{(n)},\tilde{\phi}_{h,i}^{\star}}^{-1}\tilde{\phi}_{h,i}^{\star}(\tilde{s},\tilde{\boldsymbol{a}})\right]}\qquad\text{(CS inequality)}$$

$$\lesssim\sqrt{N\left(\log\det\left(\lambda I_{d^{|\cup_{j\in Z_i}z_j|}}+\sum_{n=1}^{N}\mathbb{E}_{(\tilde{s},\tilde{\boldsymbol{a}})\sim d_{P^\star,h}^{\pi^{(n)}}}\left[\tilde{\phi}_{h,i}^{\star}(\tilde{s},\tilde{\boldsymbol{a}})\tilde{\phi}_{h,i}^{\star}(\tilde{s},\tilde{\boldsymbol{a}})^\top\right]\right)-\log\det(\lambda I_{d^{|\cup_{j\in Z_i}z_j|}})\right)}$$

$$\text{(Lemma E.2)}$$

$$\leq\sqrt{d^{L^2}N\log\left(1+\frac{N}{d\lambda}\right)}.$$

(Potential function bound, Lemma E.3 noting $\|\phi_{h,i}^{\star}(s[Z_i],\boldsymbol{a}_i)\|_2\leq1$ for any $(s,\boldsymbol{a})$.)

Similarly, we have

$$\sum_{n=1}^{N}\mathbb{E}_{(\tilde{s},\tilde{\boldsymbol{a}})\sim d_{P^\star,h}^{\pi^{(n)}}}\left[\left\|\bar{\phi}_{h,i}^{\star}(\tilde{s},\tilde{\boldsymbol{a}})\right\|_{\Sigma_{n,\gamma_h^{(n)},\bar{\phi}_{h,i}^{\star}}^{-1}}\right]\leq\sqrt{d^L N\log\left(1+\frac{N}{d\lambda}\right)}.$$

Finally,

$$\sum_{n=1}^{N}\Delta^{(n)}\lesssim HM\left(\sqrt{d^{L^2}N\log\left(1+\frac{N}{d\lambda}\right)}\tilde{A}^L\sqrt{d^L\left(\alpha^{(N)}\right)^2+H^2d^{L^2}\lambda}+\sum_{n=1}^{N}\sqrt{\frac{d^L\tilde{A}^L\left(\alpha^{(n)}\right)^2}{n}}\right)$$

$$+ H^3 M^2 \left( \frac{1}{H} \sqrt{d^L N \log\left(1 + \frac{N}{d\lambda}\right)} \alpha^{(N)} + \sum_{n=1}^{N} \sqrt{\tilde{A}\zeta^{(n)}} \right)$$

$$\lesssim H^2 M^2 d^{L^2} \tilde{A}^L \sqrt{N \log\left(1 + \frac{N}{d\lambda}\right)} \alpha^{(N)}.$$

(Some algebra. We take the dominating term out. Note that $\alpha^{(n)}$ is increasing in $n$)

This concludes the proof. $\qquad\square$

### D.3 Proof of the Main Theorems

**Lemma D.9.** *For the model-based algorithm, when we pick $\lambda = \Theta\left(L d^L \log \frac{NHM|\Phi|}{\delta}\right)$, $\alpha^{(n)} = \Theta\left(H\tilde{A}\sqrt{n\zeta^{(n)} + d^L\lambda}\right)$ and $\zeta^{(n)} = \Theta\left(\frac{1}{n}\log\frac{|\mathcal{M}|HNM}{\delta}\right)$, with probability $1 - \delta$, we have*

$$\sum_{n=1}^{N} \Delta^{(n)} \lesssim H^3 M^2 d^{(L+1)^2} A^{\frac{L+1}{2}} N^{\frac{1}{2}} \log \frac{|\mathcal{M}|HNM}{\delta}.$$

*Proof.* The result of Lemma D.1 implies with our choice of $\lambda$ and $\zeta^{(n)}$, the event $\mathcal{E}$ holds with probability at least $1 - \delta$. In this case, we have

$$\alpha^{(n)} = \Theta\left( H\tilde{A}\sqrt{\log \frac{|\mathcal{M}|HNM}{\delta} + L d^{2L} \log \frac{NHM|\Phi|}{\delta}} \right), \tag{24}$$

which is a constant unrelated with $n$. Therefore, using the result of Lemma D.8, we get

$$\sum_{n=1}^{N} \Delta^{(n)} \lesssim H^2 d^{L^2} \tilde{A}^L M^2 \sqrt{N \log\left(1 + \frac{N}{d\lambda}\right)} \alpha^{(N)} \lesssim H^3 M^2 d^{(L+1)^2} A^{L+1} L^{\frac{1}{2}} N^{\frac{1}{2}} \log \frac{|\mathcal{M}|HNM}{\delta},$$

which has finished the proof. $\qquad\square$

**Proof of Theorem 4.1**

*Proof.* For any fixed episode $n$ and agent $i$, by Lemma D.5, Lemma D.6 and Lemma D.7, we have

$$v_i^{\dagger,\pi_{-i}^{(n)}} - v_i^{\pi^{(n)}} \left( \text{or } \max_{\omega \in \Omega_i} v_i^{\omega \circ \pi^{(n)}} - v_i^{\pi^{(n)}} \right) \leq \overline{v}_i^{(n)} - \underline{v}_i^{(n)} + 2HM\sqrt{\tilde{A}\zeta^{(n)}} \leq \Delta^{(n)}.$$

Taking maximum over $i$ on both sides, we have

$$\max_{i \in [M]} \left\{ v_i^{\dagger,\pi_{-i}^{(n)}} - v_i^{\pi^{(n)}} \right\} \left( \text{or } \max_{i \in [M]} \left\{ \max_{\omega \in \Omega_i} v_i^{\omega \circ \pi^{(n)}} - v_i^{\pi^{(n)}} \right\} \right) \leq \Delta^{(n)}. \tag{25}$$

From Lemma B.8, with probability $1 - \delta$, we can ensure

$$\sum_{n=1}^{N} \Delta^{(n)} \lesssim H^3 M^2 d^{(L+1)^2} A^{L+1} L^{\frac{1}{2}} N^{\frac{1}{2}} \log \frac{|\mathcal{M}|HNM}{\delta}.$$

Therefore, according to Lemma E.4, when we pick $N$ to be

$$O\left( \frac{L^5 M^4 H^6 d^{2(L+1)^2} \tilde{A}^{2(L+1)}}{\varepsilon^2} \log^2\left( \frac{HdALM|\mathcal{M}|}{\delta\varepsilon} \right) \right),$$

we have

$$\frac{1}{N} \sum_{n=1}^{N} \Delta^{(n)} \leq \varepsilon.$$

On the other hand, from equation 25, we have

$$\max_{i \in [M]} \left\{ v_i^{\dagger, \hat{\pi}_{-i}} - v_i^{\hat{\pi}} \right\} \left( \text{or} \max_{i \in [M]} \left\{ \max_{\omega \in \Omega_i} v_i^{\omega \circ \hat{\pi}} - v_i^{\hat{\pi}} \right\} \right)$$

$$= \max_{i \in [M]} \left\{ v_i^{\dagger, \pi_{-i}^{(n^\star)}} - v_i^{\pi^{(n^\star)}} \right\} \left( \text{or} \max_{i \in [M]} \left\{ \max_{\omega \in \Omega_i} v_i^{\omega \circ \pi^{(n^\star)}} - v_i^{\pi^{(n^\star)}} \right\} \right)$$

$$\leq \Delta^{(n^\star)} = \min_{n \in [N]} \Delta^{(n)} \leq \frac{1}{N} \sum_{n=1}^{N} \Delta^{(n)} \leq \varepsilon,$$

which has finished the proof, noting our assumption that $L = O(1)$. $\square$

## E  AUXILIARY LEMMAS

**Lemma E.1** (Concentration of the bonus term (Zanette et al. (2021), Lemma 39)). *Set* $\lambda^{(n)} \geq \Theta(d \log(nH|\Phi|/\delta))$ *for any* $n$*. Define*

$$\Sigma_{n,\rho_h^{(n)},\phi} = n\mathbb{E}_{(s,\boldsymbol{a}) \sim \rho_h^{(n)}}[\phi(s,\boldsymbol{a})\phi^\top(s,\boldsymbol{a})] + \lambda^{(n)} I_d, \quad \hat{\Sigma}_{h,\phi}^{(n)} = \sum_{i=1}^{n} \phi(s_h^{(i)}, \boldsymbol{a}_h^{(i)})\phi^\top(s_h^{(i)}, \boldsymbol{a}_h^{(i)}) + \lambda^{(n)} I_d.$$

*With probability* $1 - \delta$*, we have*

$$\forall n \in \mathbb{N}^+, \forall h \in [H], \forall \phi \in \Phi, \quad c_1 \|\phi(s,\boldsymbol{a})\|_{\Sigma_{\rho_h^{(n)},\phi}^{-1}} \leq \|\phi(s,\boldsymbol{a})\|_{\left(\hat{\Sigma}_{h,\phi}^{(n)}\right)^{-1}} \leq c_2 \|\phi(s,\boldsymbol{a})\|_{\Sigma_{\rho_h^{(n)},\phi}^{-1}}.$$

**Lemma E.2** (Agarwal et al. (2020a), Lemma G.2). *Consider the following process. For* $n = 1, \ldots, N$*,* $M_n = M_{n-1} + G_n$ *with* $M_0 = \lambda_0 I$ *and* $G_n$ *being a positive semidefinite matrix with eigenvalues upper bounded by* $1$*. We have*

$$2 \log \det(M_N) - 2 \log \det(\lambda_0 I) \geq \sum_{n=1}^{N} \mathrm{Tr}(G_n M_{n-1}^{-1}).$$

**Lemma E.3** (Potential function lemma). *Suppose* $\mathrm{Tr}(G_n) \leq B^2$*.*

$$2 \log \det(M_N) - 2 \log \det(\lambda_0 I) \leq d \log \left( 1 + \frac{NB^2}{d\lambda_0} \right)$$

*Proof.* Let $\sigma_1, \cdots, \sigma_d$ be the set of singular values of $M_N$ recalling $M_N$ is a positive semidefinite matrix. Then, by the AM-GM inequality,

$$\log \det(M_N) / \det(\lambda_0 I) = \log \prod_{i=1}^{d} (\sigma_i/\lambda_0) \leq \log d \left( \frac{1}{d} \sum_{i=1}^{d} (\sigma_i/\lambda_0) \right)$$

Since we have $\sum_i \sigma_i = \mathrm{Tr}(M_N) \leq d\lambda_0 + NB^2$, the statement is concluded. $\square$

**Lemma E.4.** *For parameters* $A, B, \varepsilon$ *such that* $\frac{A^2 B}{\varepsilon^2}$ *is larger than some absolute constant, when we pick* $N = \frac{A^2}{\varepsilon^2} \log^2 \frac{A^4 B^2}{\varepsilon^4} = O\left( \frac{A^2}{\varepsilon^2} \log^2 \frac{AB}{\varepsilon} \right)$*, we have*

$$\frac{A}{\sqrt{N}} \log(BN) \leq \varepsilon.$$

*Proof.* We have

$$\frac{A}{\sqrt{N}} \log(BN) = \varepsilon \frac{\log \left( \frac{A^2 B}{\varepsilon^2} \log^2 \frac{A^4 B^2}{\varepsilon^4} \right)}{\log \frac{A^4 B^2}{\varepsilon^4}}$$

Note that

$$\frac{A^2 B}{\varepsilon^2} \log^2 \frac{A^4 B^2}{\varepsilon^4} \leq \frac{A^4 B^2}{\varepsilon^4} \Leftrightarrow \log^2 \frac{A^4 B^2}{\varepsilon^4} \leq \frac{A^2 B}{\varepsilon^2}$$

where the right hand side is always true whenever $\frac{A^2 B}{\varepsilon^2}$ is larger than some given constant. Therefore, we get

$$\frac{A}{\sqrt{N}} \log(BN) \leq \varepsilon.$$

$\square$

# F  EXPERIMENT DETAILS

## F.1  DETAILED ENVIRONMENT SETUP

In this section we introduce the details of the environment construction of the Block Markov games. For completeness we repeat certain details already introduced in the main text. We design our Block Markov game by first randomly generating a tabular Markov game with horizon $H$, 3 states, 2 players each with 3 actions, and random reward matrix $R_h \in (0, 1)^{3 \times 3^2 \times H}$ and random transition matrix $T_h(s_h, a_h) \in \Delta(\mathcal{S}_{h+1})$. For the reward generalization, for each $r(s, a, s')$ entry in the reward matrix, we assign it with a random number sampled from a uniform distribution from -1 to 1. For the probability matrix generation, for each conditional distribution $T(\cdot|s, a)$, we randomly sample 3 numbers from a uniform distribution from -1 to 1 and form the probability simplex by normalization. For the generation of rich observation (emission distribution), we follow the experiment design of (Misra et al., 2020): the dimension of the observation is $2^{\lceil \log(H + |\mathcal{S}| + 1) \rceil}$. For an observation $o$ that emitted from state $s$ and time step $h$, we concatenate the one-hot vector of $s$ and $h$, adding i.i.d. Gaussian noise $\mathcal{N}(0, 0.1)$ on each entry, pend zero at the end if necessary, and finally multiply with a Hadamard matrix. In our setting, we have variants with different horizons $H$.

## F.2  IMPLEMENTATION DETAILS

For the implementation of GERL_MG2, we break down the introduction into two parts: the implementation of Alg. 3 and the implementation of game solving algorithm with current features (line. 10 and line. 11 of Algorithm. 1). For the implementation of Algorithm. 3, we follow the same function approximation as (Zhang et al. (2022)) and adapt their open-sourced code at `https://github.com/yudasong/briee`. We include an overview of the function class for completeness: we adopt a two layer neural network with tanh non-linearity as the function class as the discriminator class. For the decoder, we let $\psi(o) = \text{softmax}(A^\top o)$, where $A \in \mathbb{R}^{|\mathcal{O}| \times 3}$, and we let $\phi(o, \mathbf{a}) = \psi(o) \otimes \mathbf{a}$. Here $\mathbf{a}$ denotes the one-hot encoding in the joint action space.

Different from Zhang et al. (2022), we solve the optimization problem by directly solving the min-max-min problem instead of using an iterative method. We show the implementation in Algorithm. 5. We first perform minibatch stochastic gradient descent aggressively on the discriminator selection step (line. 5, on $\hat{\phi}$ and $f$) and the feature selection step (line. 6, on $\phi$), where in each step we first compute the linear weight $w$ and $\hat{w}$ closed-formly and then perform gradient descent/ascend on the features and discriminators. Note that here the number of iteration $T$ is very small.

For solving the Markov games, in addition to following Algorithm. 1, to solve line.10 (i.e., solving equation 2 or equation 3 or equation 4), we implement the NE/CCE solvers based on the public repository: `https://github.com/quantumiracle/MARS`. Note that the essential difference lies in that (Xie et al., 2020) assumes that the algorithm has the access to the ground-truth feature but our algorithm needs to utilize the different features we learn for each iteration. We also adopt the Deep RL baseline from the same public repository.

## F.3  ZERO-SUM EXPERIMENT TRAINING CURVES

In this section we provide the training curves of GERL_MG2 and Deep RL baseline in the zero-sum setting in Figure. 1.

## F.4  GENERAL-SUM EXPERIMENT DETAILS

In this section we complete the remaining details for the general-sum experiment. We include the training curve in Fig. 2.

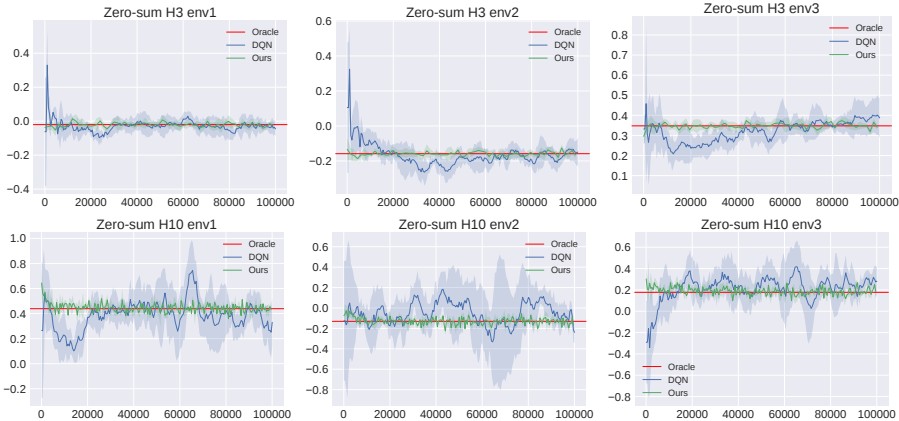

Figure 1: Training curve in the zero-sum setting. We evaluate each method over 5 random seeds and report the mean and standard deviation of the moving average of evaluation returns, wherein for each evaluation we perform 1000 runs. We use "Oracle" to denote the ground truth NE values of the Markov game. The x-axis denotes the number of episodes and the y-axis denotes the value of returns.

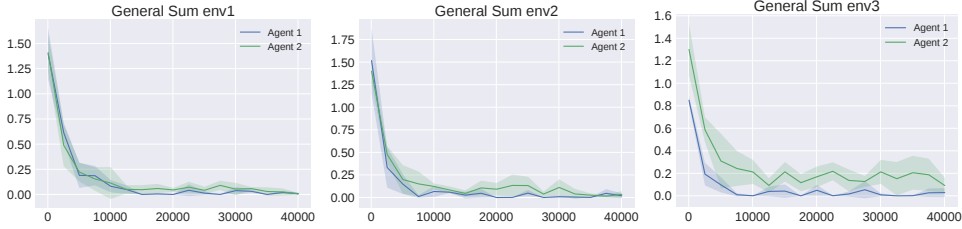

Figure 2: Training curve of GERL_MG2 in the general sum setting. In this setting, the y-axis denotes exploitability instead of raw returns.

## F.5 HYPERPARAMETERS

In this section, we include the hyperparameter for GERL_MG2 in Table. 2, and the hyperparameter for DQN in Table. 3 and Table. 4.

Table 2: Hyperparameters for GERL_MG2.

|  | Value Considered | Final Value |
|---|---|---|
| Decoder $\phi$ learning rate | {1e-2} | 1e-2 |
| Discriminator $f$ learning rate | {1e-} | 1e-2 |
| Discriminator $f$ hidden layer size | {128,256,512} | 256 |
| RepLearn Iteration $T$ | {10,20,30,50} | 10 |
| Decoder $\phi$ number of gradient steps | {64,128,256} | 256 |
| Discriminator $f$ number of gradient steps | {64,128,256} | 256 |
| Decoder $\phi$ batch size | {128,256,512} | 512 |
| Discriminator $f$ batch size | {128,256,512} | 512 |
| RepLearn regularization coefficient $\lambda$ | {0.01} | 0.01 |
| Decoder $\phi$ softmax temperature | {1,0.5,0.1} | 1 |
| LSVI bonus coefficient $\beta$ | {0.1,0.5,1} | 0.1 |
| LSVI regularization coefficient $\lambda$ | {1} | 1 |
| Warm up samples | {0,200} | 0 |

---

**Algorithm 5** Model-free Representation Learning in Practice

1: **Input:** Dataset $\mathcal{D}$, step $h$, regularization $\lambda$, iterations $T$.
2: Denote least squares loss: $\mathcal{L}_{\lambda,\mathcal{D}}(\phi,\theta,f) := \mathbb{E}_{\mathcal{D}}\left[\left(\phi(s,\boldsymbol{a})^\top\theta - f(s)\right)^2\right] + \lambda\|\theta\|_2^2$.
3: Initialize $\phi_0 \in \Phi_h$ arbitrarily;
4: **for** $t = 0, 1, \ldots, T$ **do**
5:     Discriminator selection: $f_t = \arg\max_{f \in \mathcal{F}_h}[\min_\theta \mathcal{L}_{\lambda,\mathcal{D}}(\phi_t, \theta, f) - \min_{\tilde{\phi}\in\Phi,\tilde{\theta}} \mathcal{L}_{\lambda,\mathcal{D}}(\tilde{\phi},\tilde{\theta},f)]$
6:     Feature selection: $\phi_{t+1} = \arg\min_{\phi\in\Phi_h} \sum_{i=1}^t \min_{\theta_i} \mathcal{L}_{\lambda,\mathcal{D}}(\phi,\theta_i,f_i)$, $\hat{\phi} \leftarrow \phi_{t+1}$
7: **end for**
8: **Return** $\hat{\phi}$, $\hat{P}$ where $\hat{P}$ is calculated from equation 1.

---

Table 3: Hyperparameters for DQN in short horizon environment.

|  | Value considered | Final Value |
|---|---|---|
| Target update interval | {1000} | 1000 |
| $\epsilon_0$ | {1} | 1 |
| $\epsilon_N$ | {0.01} | 0.01 |
| $\epsilon$ decay frequency | {8000} | 8000 |
| Batch size | {8000} | 8000 |
| Optimizer | {Adam} | Adam |
| Learning Rate | {0.0001} | 0.0001 |
| Hidden layer | {[32,32,32]} | [32,32,32] |
| Self-play $\delta$ | {1.5} | 1.5 |

Table 4: Hyperparameters for DQN in long horizon environment.

|  | Value considered | Final Value |
|---|---|---|
| Target update interval | {1000} | 1000 |
| $\epsilon_0$ | {1} | 1 |
| $\epsilon_N$ | {0.01} | 0.01 |
| $\epsilon$ decay frequency | {8000} | 8000 |
| Batch size | {8000} | 8000 |
| Optimizer | {Adam} | Adam |
| Learning Rate | {0.0001} | 0.0001 |
| Hidden layer | {[32,32,32]} | [32,32,32] |
| Self-play $\delta$ | {1.5,2} | 2 |

