# OpenReview forum: "Representation Learning for Low-rank General-sum Markov Games"
_ICLR.cc/2023/Conference — ICLR 2023 poster_

### Official Review · Reviewer_oMoF · 2022-10-26

**Confidence:** 4
**Correctness:** 4
**Technical Novelty And Significance:** 3
**Empirical Novelty And Significance:** Not applicable
**Recommendation:** 6

**Clarity, Quality, Novelty And Reproducibility:**

Overall, the writing of this paper is clear to readers. The authors provide detailed proofs and codes in the Appendix which will help to reproduce the theoretical results and the experiment results. Part of the proof follows the techniques developed in the work [Uehara et al., 2021]. But the model-free algorithm and the associated analysis are novel to the topic of low-rank representation learning in RL.

**Strength And Weaknesses:**

Strength:

1. This paper provides novel theoretical results for representation learning in general-sum Markov games with a low-rank transition structure, which extends the previous single-agent MDP setting to the multi-agent general-sum Markov game setting.

2. This paper presents a new result for low-rank representation learning in RL with a model-free learning algorithm, which is different from the existing model-based algorithm.

2. This paper also provides an extension from the general transition case to the factored transition setting.

4. The theoretical analysis is clear and this paper is well structured.

Weaknesses:

1. I feel that the proof of this paper largely depends on the recent work [Uehara et al., 2021]. Most of the proof can be directly modified from [Uehara et al., 2021] by setting the action a to $(a_1, a_2,...)$. To handle the proof with CCE, CE, and NE in the general sum game setting, one can also use the proof techniques in recent advances in general-sum Markov games.

2. In Theorem 5.1, there is a $L^2$ in the exponential term, which does not appear in Theorem 4.1. In my understanding, this can be a problem in the upper bound. Can the authors provide a more detailed discussion on this? What is the major technical reason for obtaining this factor?

3. The authors may need to further provide a detailed comparison between the results in Theorem 4.1 and Theorem 5.1, and show whether there are advantages of using the factored transition modeling.

**Summary Of The Paper:**

This paper studies multi-agent general-sum Markov games with nonlinear function approximation. They focus on low-rank Markov games whose transition matrix admits a hidden low-rank structure. The authors provide a model-based and a model-free approach to learning the Nash equilibrium under such a transition model. The proposed algorithm achieves a sample complexity of $poly(H, d, A, 1/\epsilon)$. They further consider Markov Games with a factorized transition structure.  The authors present the empirical results to verify their theoretical results.

**Summary Of The Review:**

See the section of Strength And Weaknesses.

---

> ### Author Response · Authors · 2022-11-12
> **Thank you for your review!**
>
> Thank you for your review! Below we provide a detailed response to each of your feedback and we hope our response could address your concerns.
>
> 1. (**Proof Idea.**) Our framework is based on [Uehara et al., 2021], but we also solved some new problems that aren’t covered in the previous works. (1) For the model-free representation learning module, we designed the discriminator function class $\mathcal{F}_h$ carefully to solve the extra degrees of freedom problem. The major concern is the value function of player $i$ not only depends on the policy of player $i$ itself, but also the policies of other players. These extra degrees of freedom make the covering number of a naively constructed $\mathcal{F}_h$ too large. We try to reduce the size of $\mathcal{F}_h$ by restricting the set of the possible policies, which is achieved by using $\tilde{Q}$ instead of the optimistic estimator $\overline{Q}$ when doing the planning. (2) We propose a factored structure that is particularly designed to solve the exponential dependency on the number of players. Significantly different analysis is needed when dealing with this new structure.
>
> 2. (**Exponential Term & Comparison between Theorem 4.1 and 5.1.**) The appearance of the $L^2$ exponential term is mainly due to a technical reason that we hope to solve in future. It comes from bounding the bonus term. Note that the bonus $\beta$ is constructed using $\bar{\phi}$, which is a function of ~$L^2$ states. A key step in our proof is (here the description is informal) we showed if a function $g(s,a)$ is only related to $l$ entries of the state vector, then its expectation can be bounded at an order ~$d^l$, in our case $g=\beta$ and $l=L^2$. This is the main cause of the $L^2$ appearance in our final results. Note that here we assume $L$ is a pretty small quantity compared with the number of players $M$. If $L$ is large, replacing $L^2$ by $M$ can be a better option since the state itself has at most $M$ entries. Another remark is if we want to convert the factor model into a linear model by combing the features of each component into a long feature vector, its dimension would be $d^M$. Therefore, when comparing the results of Theorem 4.1 and Theorem 5.1, the $d$ term in Theorem 4.1 should be replaced by $d^M$. And therefore the result of Theorem 5.1 is always better than that of Theorem 4.1 because we have $d^M \geq d^{L^2}$ and $A\geq \tilde{A}^L$.

---

### Official Review · Reviewer_xksf · 2022-10-29

**Confidence:** 3
**Correctness:** 3
**Technical Novelty And Significance:** 2
**Empirical Novelty And Significance:** 2
**Recommendation:** 6

**Clarity, Quality, Novelty And Reproducibility:**

The main results are clear except for some ambiguity. The motivation from real-world games seems to be a bit vague. It is important to point out real-world scenarios that fail existing provably sample-efficient algorithms for Markov games. In my view, the biggest reason is that we are mostly limited to the fully observable case, which could be a useful motivation for applying representation learning. The authors claim 'more computationally efficient solutions', which are not convincing without calibrating the computational complexity of proposed methods. In theory, it is less discussed the novelty in the statistical analysis of representation learning and the policy improvement analysis of Markov games. Some claims could be problematic, e.g., 'first algorithm that solves general-sum Markov games under function approximation', which can be controversial since the literature on Markov games with function approximation is evolving due to many recent works.

Overall, the authors have shown the promising use of representation learning for Markov games. Viewing existing model-based MARL algorithms, representation learning methods are more or less expected to be applicable. Hence, the novelty of the proposed methods is questionable. In theory, sub-optimal PAC bounds seem to indicate a loose statistical analysis of representation learning, which I didn't check all proofs. However, it is worthy making more efforts to either tight bounds or demonstrate lower bounds. The authors also demonstrate the scalability of proposed methods without detailed evidence provided in experiments. Last but not least, computational efficiency is not discussed in the paper, which is a key factor when we consider how to apply them to real-world games.

Here are some other questions for consideration:

- What is MEL oracle?

- What is 'more computationally efficient solutions' in Remark 2.1?

- How do you solve policy planning oracles in Eqs. (2-4)?

- Can you summarize the main idea of analysis of model-based and model-free cases? Why different dependence on problem parameters?

- In experiments, can you increase the number of players to show the scalability in the number of players?


**Strength And Weaknesses:**

Strengths

- The studied setting considers a general nonlinear representation of transition dynamics of Markov games. This setting is interesting and useful since it is often the case in practice that the model features are unknown, which brings the approach of representation learning to Markov games.

- Under some assumptions on the model and the planning oracle, the provided representation learning methods enjoy polynomial sample complexities for learning equilibria, although it has an exponential dependence on the number of players. In the case of low-rank factored Markov games, the proposed method enjoys a polynomial dependence on the number of players.

Weaknesses

- The application of representation learning to Markov games is more straightforward under assumptions on the realizability of model class or feature mapping and access to policy planning oracles. These assumptions could be too strong to hold, for example, exact solutions from policy planning oracles and real games do not satisfy your realizability assumptions.

- The established PAC bounds are sub-optimal in terms of state/action space sizes, especially when we specialize it to the tabular case. It is important to compare it with some model-based MARL algorithms in the literature. Recovering the optimal sample complexity in the tabular case should be a sanity check of the efficiency of the proposed methods.

 -  The proposed methods are not decentralized, which prevents them to work in many real-world games that prohibit information sharing, especially policies.

- The provided computational experiments only consider small-scale Markov games, which are not very sufficient to demonstrate the scalability of proposed methods under different low-rank structures.


**Summary Of The Paper:**

The paper studies representation learning for Markov games with low-rank structures. Under the low-rank assumption, the authors develop two representation learning methods to learn the underlying transitions: model-based and model-free representation learning and add the UCB bonus into the estimated Q-value functions for computing explorative policies. In theory, the authors show sample complexity PAC guarantees for two methods for obtaining near-optimal equilibria. Moreover, the authors study a special case of factored Markov games by showing a similar PAC guarantee while improving the exponential dependence on the number of players. Finally, the authors provide a series of experiments to compare the performance of the proposed method with DQN.

**Summary Of The Review:**

The paper studies a class of Markov games with low-rank structure and provides representation learning methods for learning equilibria with PAC guarantees. Although the setting is useful, many assumptions in algorithm design and analysis need to be further justified. The novelty in the analysis needs to be discussed in detail. The experiments only consider small-scale Markov games that are not very sufficient to demonstrate the effectiveness of the proposed methods.


============================

POST-REBUTTAL. Thank you for your response. I have read other reviews. I am inclined to recommend acceptance. The score has been increased.

---

> ### Author Response · Authors · 2022-11-12
> **Thank you for your review! (1/2)**
>
> Thank you for your review! Below we provide a detailed response to each of your feedback and we hope our response could address your concerns.
>
> 1. (**The Assumption of Realizability and Planning Oracle.**) Realizability is a standard assumption that is made in most RL theory papers with function approximation. Compared to the vast majority of works on linear function approximation, in which it is assumed that the exact feature map $\phi^\star$ is given, our assumption is already significantly weaker, by allowing an exponentially large function class that captures $\phi^\star$. In practice, one usually uses neural networks for such function approximation tasks, and neural networks are known to be universal function approximators and will essentially satisfy realizability.
> For planning, we want to clarify that we do not assume access to a planning oracle that solves any Markov Game. In fact, we designed our own planning algorithm (cf. alg 1, line 9-12) that is able to solve the particular type of Markov Games constructed using $\hat P$. Within this algorithm, we make a polynomial number of calls to NE/CE/CCE solvers for normal-form games. Particularly for CE and CCE, there exist well-known algorithms that formulate them as solutions of linear programs which can be solved in polynomial time (See e.g., [1]). Therefore, overall, for CE and CCE, we indeed presented an end-to-end planning algorithm with polynomial computational complexity. For NE, since in general it’s known to be PPAD-hard to solve, assuming we are given a planning oracle for the normal-form game is acceptable when we are focused to study the statistical complexity of the algorithm.
>
> [1]. Papadimitriou, C. H. and Roughgarden, T. (2008). Computing correlated equilibria in multi-player games. Journal of the ACM (JACM), 55(3):1–29.
>
> 2. (**How tight is the established PAC bound.**) We also believe that the presented PAC bound might be improvable, though it will likely require significantly different techniques that we are not aware of. While the tabular setting offers good intuitions to check the optimality of the bound, we want to emphasize that in some situations it could be unfair to directly compare the results with those in the tabular setting. In our case because the tabular setting automatically provides a known feature function i.e., the one-hot vectors, it bypasses the main difficulty of the representation learning framework. We believe it is acceptable that an algorithmic framework introduces a certain degree of redundancy in order to solve a larger class of problems.
>
> 3. (**Comparing with Prior Works.**) Thanks for your comments. We could feel the reviewer is concerned about the comparisons between our work and some related works on MARL, e.g. “other model-based MARL algorithms in the literature”, “existing provably sample-efficient algorithms for Markov games”, “the literature on Markov games with function approximation is evolving due to many recent works” and “Viewing existing model-based MARL algorithms, representation learning methods are more or less expected to be applicable”. Our work is motivated from the rather recent advances on the provable efficient algorithms for Markov games. **To the best of our knowledge, prior to us, the only papers that provide a PAC guarantee for equilibrium learning in multi-player general-sum markov games are the these two [2,3], both of which only work in the tabular setting.** We are not aware of any other work that provides polynomial PAC bound for the Markov Games with more than two agents and allows function approximation. It could be possible that we missed some important works in the literature, and we are always happy to make further clarifications if the reviewer has any particular works that he/she wants us to compare to.
>
> [2]. Chi Jin, Qinghua Liu, Yuanhao Wang, and Tiancheng Yu. V-learning–a simple, efficient, decentralized algorithm for multiagent RL. 2021.
>
> [3]. Qinghua Liu, Tiancheng Yu, Yu Bai, and Chi Jin. A sharp analysis of model-based reinforcement learning with self-play. 2021.
>
> 4. (**Decentralized Algorithm.**) Indeed, we also believe that decentralized algorithm is an important direction for future research. For the current framework, in particular, it seems hard to extend to the decentralized setting because computing the NE/CCE/CE always requires joint planning. Utilizing adversarial bandit subroutines as in the V-learning algorithm could be one potential way, but how to incorporate it with function approximation remains unknown.

---

> > ### Author Response · Authors · 2022-11-12
> > **Thank you for your review! (2/2)**
> >
> > 5. (**Computational experiments only consider small-scale Markov games**) Yes, we agree that the current version only considers small-scale Markov games. And we are making an ongoing effort to design a scalable implementation of our algorithm by combining deep learning tricks and tools. Yet, we believe the main contribution of this paper is on the theoretical results, and the computational results serve mostly as a proof of concept. We leave designing scalable implementation of our algorithm as future work.
> >
> > 6. (**What is an MLE oracle?**) An MLE oracle solves the maximum likelihood estimation problem in supervised learning. For a neural function class with a softmax final layer, for example, an MLE oracle corresponds to training the network using gradient descent with cross entropy loss.
> >
> > 7. (**Remark 2.1 & planning oracles.**) The remark is to say that while it’s in general computationally hard (PPAD-hard) to solve for NE, computing the CE/CCE can be much easier and can be solved in polynomial time. See response point 1.
> >
> > 8. (**Analysis idea.**) Both cases are based on similar proof ideas. We first prove the representation learning module which ensures the model estimation error $(\hat{P}-P)f$ is small, where $f$ is from a function class $\mathcal{F}$ that includes all possible value functions that we care about. Then we do some regret decomposition procedure to reduce the policy suboptimality into some term related to the model estimation error. The difference between the model-based and model-free results is due to the difference in the choice of $\mathcal{F}$. The function class for the model-free case is much more restricted compared to the model-based case, and therefore we need to add some extra steps (by importance sampling) to transform a function into a member of $\mathcal{F}$. This is the main cause of the additional $A$ in the final result.
> >
> > 9. (**Scalability of Experiments.**) If we understand correctly, by “scale” the reviewer means to  scale up the number of players instead of the state dimension. We note the latter is easy to scale even in our proof-of-concept experiments. Note that there are two difficulties in our experiment setting on scaling the number of players: one is that the number of latent state scales exponentially and the other is that the number of constraints in the LP for solving CCE also scales exponentially, both with respect to the number of players. However, our current implementation can directly apply to any number of players but we would expect it to be slow. That being said, our current experiment contains proof-of-concept results, but it is indeed applicable to a larger scale. We also remark that the difficulties to scale to a large number of players indeed align with our theory findings.

---

### Official Review · Reviewer_H4wD · 2022-11-01

**Confidence:** 4
**Correctness:** 4
**Technical Novelty And Significance:** 3
**Empirical Novelty And Significance:** 3
**Recommendation:** 8

**Clarity, Quality, Novelty And Reproducibility:**

- The authors provide three definitions of optimal policy (policy we aim to evaluate), namely NE, CCE, and CE. Although the authors mention that NE \subset CE \subset CCE, the sample complexity of the algorithms to reach any of the optimal policies in the same. Can the authors comment on this? Is this sample complexity improvable?
- What is the intuition behind the definition of CE? I would appreciate it if the authors could explain why such policy is of interest.
- Can we reduce the memory requirement of the algorithm in the Low-Rank Factored Markov Game setting?
- There are works such as"Scalable Reinforcement Learning for Multiagent Networked Systems" by Qu et al. where the correlation between the agents are controlled by a geometric factor. In particular, the agents which are far away from one another, have low correlation on each other. Can we adopt such notion instead of the restrictive Low-Rank Factored Markov Game?

**Strength And Weaknesses:**

Strength:
- The authors study a general setting of Markov games fir linear MDPs. The problem formulation is novel and interesting, and the sample complexities are tight.
- paper is well-written and easy to follow.
Weakness:
- In order to run the algorithm 1 and 2, we need to assume access to optimization oracles for calculating max and min-max-min. I was wondering if the authors can comment on the hardness of access to these oracles.
- The definition of Low-Rank Factored Markov Game is slightly strong. I believe according to this definition, you are assuming that only L number of agents around each agent us can affect an agent.
- Although the sample complexity is improved under Low-Rank Factored Markov Game setting, but the authors still require to save a large vector of parameters, in the size exponential in the number of agents.

**Summary Of The Paper:**

This paper studies the low raw rank General sum Markov games, and provides two algorithms, model based and model free for finding the Nash equilibrium. In the model based algorithm, we sample state-action-next state tuples from the environment, and we estimate the transition probability directly from these tuples. In the model-free setting, however, we first estimate the feature vectors, and then we estimate the transition probability from these feature vectors. After estimating the transition probability, the algorithm performs the planning step, which is  conducted with a UCB style approach.
At the end, the authors provide the sample complexity of their algorithm in both model based and model free settings.
Furthermore, the authors argue that their sample complexity for general MDP settings can be exponentially bad in terms of the number of agents. Hence, the authors provide the sample complexity of their algorithm for the Low-Rank Factored Markov Game. At the end the authors provide experimental results for proof-of-concept.

**Summary Of The Review:**

The paper is interesting and well written. It is comprehensive and studies a broad range of settings with providing tight sample complexity bounds for all of them. Beside some minor issues, I believe it is a good paper.

---

> ### Author Response · Authors · 2022-11-12
> **Thank you for your review!**
>
> Thank you for your review! Below we provide a detailed response to each of your feedback and we hope our response could address your concerns.
>
> 1. (**The hardness of access to these oracles.**) The MLE problem is a standard supervised learning problem for which there exist many empirical tools. Even though the problem is non-concave, empirically optimizing over neural function approximation can often find global optimal solutions with good generalization performance. The optimization planning oracle in Alg. 2 can be solved efficiently using approximation algorithms. We give such an approximate algorithm in the appendix (See Alg. 5 in Appendix F), which is implementable and is actually adopted in our experiments.
>
> 2. (**Low-Rank Factored Markov Game.**) We want to clarify that the factored Markov Game assumption does not imply that each agent can only be affected by its neighbors. For a connected factor graph, any agent can make a delayed impact on any other agent with the time of delay upper bounded by the degree of the graph, where such impact propagates in the state space. In terms of space complexity, each agent will maintain a vector of length $d^L$, instead of being exponential in the total number of agents.
>
> 3. (**Difference among NE/CE/CCE**) The difference between NE/CE/CCE is two-fold: they differ by the computational complexity as well as the statistical complexity. For normal-form games, CCE is the most efficiently solvable, then CE and then NE, but this difference is not revealed in our result because we use weaker oracles to solve weaker notions in the planning phase. In terms of their statistical complexity, the current proof analyzes these different notions in a unified style since all the three notions face the same representation learning problem, which then becomes the major contributor to the error bound. We believe the results for all these settings could be improvable, up to different limits according to the properties of different notions, but the algorithm design and analysis are still unclear and could be much more sophisticated. Deriving sharper bounds for the current problem setting and finding out their statistical limits is a future direction that we are super interested to work on, and we hope our results could lead to more attention and good works on this problem in the future.
>
> 4. (**Can we reduce the memory requirement of the algorithm in the Low-Rank Factored Markov Game setting?**) The main memory requirement for our work and any value iteration based methods is the need to save the whole history of the collected samples. Designing streaming algorithms that avoid extensive data reuse is an active area of research in RL theory in general, but currently there are very limited results even in the single agent setting.
>
> 5. (**What is the intuition behind the definition of CE.**) Both CE and CCE are relaxations of NE. The key characteristic of CE and CCE is that they consider the correlated policies instead of the product policy used in NE. The difference between CE and CCE is that the CE setting allows the player to observe its own sampled action and modify it if she wants to (i.e., strategy modification). Therefore, the notion of CE means that no player can increase her value by using a strategy modification. Such notions are highly desirable in cooperative games like soccer, where you want each player to act while accounting for and depending on the actions of the other agents, rather than being independent when conditioned on the state.
>
> 6. (**Qu et al.**) Thanks for your suggestion. Their model setting does have a connection with ours. We are interested in learning a good representation using the factored structure, while the work of Qu et al. tries to develop a better approximation to the value function. The similar idea may be applicable to the representation learning task but is not immediate, which requires further work.

---

### Official Review · Reviewer_tHaH · 2022-11-02

**Confidence:** 2
**Correctness:** 4
**Technical Novelty And Significance:** 3
**Empirical Novelty And Significance:** 2
**Recommendation:** 8

**Clarity, Quality, Novelty And Reproducibility:**

This work is novel and original, and the results are solid. Overall this paper is well-written, while some notations should be clarified as mentioned above.

**Strength And Weaknesses:**

Strength:

- Overall this paper is well-written and easy to follow.
- The authors have clearly explained the goal of this paper and their contributions. The difference between the problem setting in the current paper and those in the existing literature is properly discussed.
- The description of the proposed algorithm is clear and easy to understand.
- The theoretical analysis is solid, and numerical simulations are provided in complement to the theoretical results.

Weaknesses:

- This is mainly a theory paper, and in the main context, the authors should discuss more the technical challenges and novelties compared with the existing works. It seems not clear to me what are the difficulties caused by the more general setting considered in this paper. A high-level description of the proof idea in Section 4 would also be appreciated. It would also be helpful to compare the results in Theorem 4.1 and 4.2 with existing results for Markov games when restricting the setting to two-player.
- Some notations should be clarified in Section 2. For example, the definition of the function classes for the features should be provided. In line 6 of Algorithm 1, I guess $d_{P^*,h}^{\pi^{(n-1)}}$ means the induced distribution on the state space and $U(\mathcal{A})$ is the uniform distribution on the action space. Please define these notations (or let me know if I've missed anything).
- When describing the components of the main algorithms, please provide references if a certain approach has been proposed in some related work.
- It is remarked that the proposed algorithm is oracle-efficient, but it is unclear to me how to implement certain steps in Algorithm 1 and what are the computational costs. For example, line 6 of Algorithm 1 seems to require a certain simulator of the environment, instead of executing the policy to get the trajectory of an episode. I'd like the authors to provide more discussion about what oracles are needed and the corresponding implementations, especially since it is claimed that the proposed algorithm is scalable.
- The organization of the experiment section is a bit weird. It seems that the description of the three different environments should be grouped together. Also, it should be clarified what Environment 1, 2 and 3 refer to.

**Summary Of The Paper:**

This paper studies general-sum Markov games where the probability transition admits a low-rank structure, and proposes algorithms that exploit the underlying structure via representation learning to efficiently learn the (approximate) equilibrium policy. Both model-based and model-free methods are provided and theoretically analyzed in a unified way. Quantitative regret bounds are provided for the general case and also the special case of factored Markov game, where for the latter case, the dependence on the number of players is shown to avoid the exponential scaling. Numerical experiments are also provided to show the advantage of the proposed approach over the DQN baseline.

**Summary Of The Review:**

I think this paper studies an important setting of multi-agent general-sum Markov games and the results are solid. This is a good submission as a complete work with both theoretical analysis and numerical simulations. Some aspects of the writing can be improved as I've discussed above.

---
Other minor problems:

- In the fourth line under Algorithm 3, what's the meaning of 'M induced'?
- In the last line of Remark 5.1, 'up to'.

---

> ### Author Response · Authors · 2022-11-12
> **Thank you for your review!**
>
> Thank you for your review! Below we provide a detailed response to each of your feedback and we hope our response could address your concerns.
>
> 1. (**Technical challenges and novelties compared with the existing works.**) When considering the problem of representation learning, there is no existing work related with multiplayer Markov games (MG), even for the two-player zero-sum setting. Prior  work on the two-player zero-sum setting applies to either tabular MGs or MGs with linear function approximations, assuming the state-action feature map is known. Our work, though sharing similar procedures in the planning phase compared with some of the prior works, indeed has a very different analysis since the major technical contribution lies in the representation learning part. Another line of comparison is the representation learning algorithms for single player MDP. Compared with existing work, the Markov Game setting requires the analysis of several new estimators that don’t exist in the single player setting, including the pessimistic estimation, optimality gap and so on. Furthermore, for the model-free setting, one major problem that doesn’t exist in the single player setting is that the value function has extra degrees of freedom, as the value function of player $i$ not only depends on the policy of player $i$ itself, but also the policies of other players. These extra degrees of freedom make the covering number of the set of possible value functions (which is directly related to the final bound) much bigger than $\textrm{poly}(d)$. To overcome the problem we have to restrict the outcome policy to a smaller set, which is achieved by using $\tilde{Q}$ instead of the optimistic estimator $\overline{Q}$ when doing the planning. So our work is non-trivial compared to either MG with known features, or the single agent MDP setting.
>
> 2. (**Notation Clarification.**) Thanks for your suggestions. We uploaded a revision that adds these definitions.
>
> 3. (**When describing the components of the main algorithms, please provide references if a certain approach has been proposed in some related work.**) Thanks for your suggestions. We added some sources of the papers on the single agent MDP at the beginning of section 3, from which we drew inspiration to develop our algorithm.
>
> 4. (**It is unclear to me how to implement certain steps in Algorithm 1 and what are the computational costs.**) The sampling step in line 6 of alg. 1 can in fact be done online. We can run the policy $\pi$ till step $h$, then do uniform sampling on the action space, and collect the transition sample $(s_h, a_h, s_{h+1})$. Such a procedure only generates one state-action-state tuple from each trajectory, and therefore may result in additional $H$ factors in the final bound when considering the total number of interactions to the environment. In terms of the computational oracles, in theory, we require an MLE oracle for model-based representation learning, and NE/CE/CCE oracle for normal form game. In experiment, however, we provide implementations for most of oracles ourselves (the CCE solver is adapted from some open source repo, but we include all references and our own implementation details in the appendix), so indeed these are realistic computational oracles (for example, we use LP solvers for solving CCE) that one can have access to.
>
> 5. (**The organization of the experiment section.**) In fact the experiments are performed on two model settings (zero-sum and general-sum) instead of three, and all environments are set to be block MGs. In the main text we only present the result of the zero-sum game, while more results are deferred to the appendix, due to the space limit of the paper. Environment 1,2,3 in the experiment refers to 3 instances of a random Markov Games, with different transitions and reward functions (all randomly selected, and we include detailed constructions in the Appendix).

---

> > ### Comment · Reviewer_tHaH · 2022-11-20
> > **Thank you for the response!**
> >
> > I appreciate the clarifications and I don't have further questions. Please consider incorporating some of these into the main context for future revision.

---

### Decision · Program_Chairs · 2023-01-20

**Decision:**

Accept: poster

**Justification For Why Not Higher Score:**

Although the results are novel in general-sum Markov games, the theoretical proofs rely on the existing techniques, which is not novel.

**Justification For Why Not Lower Score:**

The results are novel and significant in general-sum Markov games.

The proposed algorithms have been justified empirically.

**Metareview: Summary, Strengths And Weaknesses:**


In this paper, the authors investigated the general-sum Markov games with representation learning. Under the low-rank assumption, the authors provided model-free and model-based representation learning with planning algorithms for achieving NE/CCE/CE, and proved polynomial sample complexity. The authors further extended the algorithms and proofs for factorized MG and tested the algorithm empirically.

The paper is well-organized and easy-to-follow. The theoretical results and empirical performances are solid and interesting.


Although the technique is largely relying on the existing results from [Uehera et al., 2021], I believe the result extends the community understanding and is deserved to be published.

**Note From Pc:**

if the above contains the word "oral" or "spotlight" please see: "oral" presentation means -> notable-top-5% and "spotlight" means -> notable-top-25%. As stated in our emails, we are disassociating presentation type from AC recommendations